# kdotpy: k · p theory on a lattice for simulating semiconductor band structures

Wouter Beugeling[1,2,⋆,†], Florian Bayer[1,2], Christian Berger[1,2], Jan Böttcher[3], Leonid Bovkun[1,2], Christopher Fuchs[1,2], Maximilian Hofer[1,2], Saquib Shamim[1,2], Moritz Siebert[1,2], Li-Xian Wang[1,2], Ewelina M. Hankiewicz[3], Tobias Kießling[1,2], Hartmut Buhmann[1,2] and Laurens W. Molenkamp[1,2]

**1** Physikalisches Institut (EP3), Universität Würzburg, Am Hubland, 97074 Würzburg, Germany
**2** Institute for Topological Insulators, Am Hubland, 97074 Würzburg, Germany
**3** Institut für Theoretische Physik und Astrophysik (TP4), Universität Würzburg, Am Hubland, 97074 Würzburg, Germany

⋆ kdotpy@uni-wuerzburg.de ,   † wouter.beugeling@physik.uni-wuerzburg.de

## Abstract

The software project `kdotpy` provides a Python application for simulating electronic band structures of semiconductor devices with k · p theory on a lattice. The application implements the widely used Kane model, capable of reliable predictions of transport and optical properties for a large variety of topological and non-topological materials with a zincblende crystal structure. The application automates the tedious steps of simulating band structures. The user inputs the relevant physical parameters on the command line, for example materials and dimensions of the device, magnetic field, and temperature. The program constructs the appropriate matrix Hamiltonian on a discretized lattice of spatial coordinates and diagonalizes it. The physical observables are extracted from the eigenvalues and eigenvectors and saved as output. The program is highly customizable with a large set of configuration options and material parameters.

The project is released as free open source software under the GNU General Public License, version 3. The code that accompanies this article is available from our Gitlab repository at https://git.physik.uni-wuerzburg.de/kdotpy/kdotpy/-/tags/v1.0.0.

# 1 Introduction

In the field of solid-state physics, particularly in the branch of semiconductor physics, the theory of electronic band structures plays a central role. Semiconductor materials are inherently very complex systems due to the typically large number of atomic orbitals of their constituent atoms. A reduction to the essential degrees of freedom (namely, the electronic bands close to the Fermi level at charge neutrality) can be achieved by effective models, such as the perturbative method known as $\mathbf{k} \cdot \mathbf{p}$ theory. The Kane model [1], a $\mathbf{k} \cdot \mathbf{p}$ theory that was originally proposed for describing the electronic band structure of indium antimonide (InSb), has since become an established model for the large category of binary semiconductor compounds with zincblende crystal structure, and in particular for narrow gap semiconductor compounds. The Kane model has been widely used in the semiconductor community, because the resulting band structures are well suited for predicting transport phenomena as well as optical (spectroscopic) properties. It is therefore not only a valuable asset upon analyzing data, but also a very powerful tool for designing device functionality.

The minimal number of the degrees of freedom in the Kane model is still too large to permit exact analytic solutions of the Schrödinger equation, that defines the band structure. The problem can be approached numerically, but setting up numerical simulations and extracting physical observables can be a tedious task, for example due to the quantity and complexity of the matrix elements in the Hamiltonian. These issues mandate a software solution where the tedious tasks are automated, i.e., an application which takes physical parameters like geometry and materials of a device as input, and which produces physical observables as its output.

Here, we present the Python application `kdotpy` as a means to deal with these challenges. The motivation for developing `kdotpy` has been the need to analyze the experimental results of the research group of L. W. Molenkamp in Würzburg. In many regards the situation that triggered the development of `kdotpy` was archetypical for an experimental group engaged in the study of sophisticated semiconductor heterostructures. The complexity of the collected data called for a description beyond simplified effective models. It was well understood that $\mathbf{k} \cdot \mathbf{p}$ theory provided a suitable framework for a realistic description of the experimental findings and a Fortran program set up by Pfeuffer-Jeschke and Novik [2,3] was already used for band structure simulations for many years. However, the modelling used in this program proved to be inadequate for simulating edge channels and surface states. Moreover, it lacks an intuitive user interface, and the program structure and the programming style have proven to be major obstacles towards maintainability and towards future extension. In particular, the fact that the initial developers had moved on to other positions and a general lack of detail in the documentation of the code constitute major entrance thresholds for inclusion of novel functionalities.

In the `kdotpy` project, we attempt to steer clear of these problems by applying modern concepts regarding project design and development workflow. The development strategy is guided by the philosophy that `kdotpy`'s primary goal is to support current (experimental and theoretical) research, for which feedback from the community is an important aspect. While `kdotpy` has started out with the specific task of analyzing transport phenomena in Mn-doped HgTe quantum wells [4–6], it has since been extended to tackle a much broader variety of problems [7–10]. As $\mathbf{k} \cdot \mathbf{p}$ theory is much more widely applicable than the study of topological insulators or HgTe in particular, we have equipped `kdotpy` with the infrastructure for simulating any material whose band structure is described by the Kane model. This includes other II-VI materials as well as III-V semiconductors like GaAs and InSb. We expect that `kdotpy` will thus be beneficial for a broad community spanning various disciplines in semiconductor physics. We intend to create a diverse and international user base and encourage active discussion among users and developers, in the spirit of open science and open software development. The community driven nature facilitates sustainable maintenance and future project development.

In line with modern standards in scientific research, we strive towards transparency by publishing this project as open source software. We believe that the choice of Python as a widely known programming language facilitates inspection of our code and thus fits the ideas of open science. We aim for a high level of reproducibility and for compliance with the FAIR principles for research data [11] by providing rich metadata in the output files and by avoiding the use of proprietary file formats.

The purpose of this article is to provide a comprehensive overview of all important components of `kdotpy` together with relevant physical background. We emphasize that reading the complete manuscript is not required for being able to use the application: We encourage the reader to focus on the parts of personal interest and/or those needed to address a specific physical problem. The article is structured as follows: In Sec. 2, we provide an in-depth review of the physical concepts and theories

essential for understanding the implementation. In Sec. 3, we discuss all important components of the implementation, making the connection between the physics and the project's source code. Installation instructions are provided in Sec. 4. We illustrate the usage of kdotpy with several detailed examples in Sec. 5. The Appendices contain technical details of the implementation as well as a comprehensive reference of configuration options, material parameters, command-line arguments, and several other aspects.

## 2 Physics background

The Kane model [1] implemented in `kdotpy` was originally proposed as a theory for the electronic band structure of InSb. It has proven to be a suitable model for a broad class of binary semiconductor compounds with a zincblende crystal structure, to which InSb also belongs. The Kane model defines an **8 × 8** Hamiltonian depending on the momentum coordinate **k**. The basis of the Hilbert space is formed by two **s** orbitals and six **p** orbitals. These states are treated exactly at the **Γ** point (**k = 0**), while couplings to other (so-called *remote*) bands are treated perturbatively in the framework of **k · p** theory.

Importantly, the Kane model takes into account spin-orbit interaction, which is essential for explaining the energetic positions of the electronic bands at the **Γ** point [1]. In compounds of heavy elements, spin-orbit coupling and other relativistic effects become strong enough to result in an inversion of the band ordering. The latter is the key ingredient for the formation of topological phases, which constitute a major research branch at the Department of Physics in Würzburg since the first experimental realizations of such materials in mercury telluride (HgTe) quantum wells [12]. **k · p** theory is well suited towards accessing topological properties and accordingly this was a main focus upon setting up the `kdotpy` project.

The numerical approach also admits analysis of more complicated configurations than bulk materials. In particular, experiments are typically performed with devices with several layers of zincblende-type materials stacked on top of a substrate. The latter constellation is described by taking the bulk Hamiltonian and substituting the momentum coordinate $k_z$ by its spatial representation $-i\hbar\partial_z$, a derivative with respect to the spatial coordinate $z$. For simulating the band structure, the $z$ coordinate is discretized to a finite set $\{z_j\}$, so that the Hilbert space dimension (and the size of the Hamiltonian matrix) becomes $8n_z$. The different layers in the system are modelled by making the coefficients in the Kane model $z$-dependent. This approach goes back to Burt [13, 14] and has since been used by many others, see, e.g., Refs. [2, 3, 15]. This discretization method distinguishes `kdotpy` from the aforementioned Fortran program [2, 3], which uses an expansion into Legendre polynomials in order to convert the Hamiltonian to matrix form. The present approach has the benefits that the modelling is more intuitive and that it can more reliably resolve essential features like edge channels [4] and surface states [7].

The bulk Kane model and the reduction to lower dimensional geometries are discussed Secs. 2.1 and 2.2, respectively. The remainder of Sec. 2 is dedicated to the effects of magnetic fields, bulk inversion asymmetry, and strain, as well as to topology.

### 2.1 k · p theory and the Kane model in a nutshell

#### 2.1.1 k · p theory

The **k · p** method aims at finding an effective model for the band structure near a high-symmetry point. The method is based on perturbation theory on the Bloch wave functions, where eigenenergies and eigenstates at **k = 0** are treated as exact and the contributions for **k ≠ 0** as perturbation. The Schrödinger equation $[\hat{p}^2/2m + V(r)]\psi_{n,k}(r) = E_n(k)\psi_{n,k}(r)$ acting on the Bloch wave functions $\psi_{n,k}(r) = e^{ik\cdot r}u_{n,k}(r)$ can be written in terms of the Bloch functions $u_{n,k}(r)$ as

$$\left[ \frac{\hbar^2 k^2}{2m} + \frac{2\hbar k \cdot \hat{p}}{2m} + \frac{\hat{p}^2}{2m} + V(r) \right] u_{n,k}(r) = E_n(k)u_{n,k}(r), \tag{1}$$

where $\hat{p} = -i\hbar\nabla$ is the momentum operator, **k** is a vector of real values, and the index $n$ labels the bands. The Hamiltonian $H_0 = \hat{p}/2m + V(r)$ at **k = 0** is treated as exact, whereas $H'_k = \hbar^2 k^2/2m + \hbar k \cdot \hat{p}/m$ is treated as perturbation. The second term in $H'_k$ gives **k · p** theory its name. The unperturbed eigenenergies $E_n(0)$ and eigenfunctions $|u_{n,0}\rangle$, that solve $H_0|u_{n,0}\rangle = E_n(0)|u_{n,0}\rangle$, are assumed to be known. The first order perturbation to the energy is

$$E_n^{(1)}(k) = \langle u_{n,0}|H'_k|u_{n,0}\rangle = \frac{\hbar^2 k^2}{2m}. \tag{2}$$

An additional term proportional to $\langle u_{n,0}|\mathbf{k}\cdot\hat{\mathbf{p}}|u_{n,0}\rangle$ vanishes if we assume that the dispersions attain a maximum or minimum at $\mathbf{k} = \mathbf{0}$. The first order perturbation to the eigenfunctions is

$$|u_{n,\mathbf{k}}^{(1)}\rangle = \frac{\hbar}{m}\sum_{n':E_{n'}\neq E_n}\frac{\mathbf{k}\cdot\mathbf{P}_{n'n}}{E_n(0)-E_{n'}(0)}|u_{n',0}\rangle, \tag{3}$$

where $\mathbf{P}_{n'n}\equiv\langle u_{n',0}|\hat{\mathbf{p}}|u_{n,0}\rangle$ and the sum runs over all bands $n'$ for which $E_{n'}(0)\neq E_n(0)$. From Eq. (3), we obtain the second order perturbation of the energy as

$$E_n^{(2)}(\mathbf{k}) = \langle u_{n,0}|H_{\mathbf{k}}'|u_{n,\mathbf{k}}^{(1)}\rangle = \frac{\hbar^2}{m^2}\sum_{n':E_{n'}\neq E_n}\frac{|\mathbf{k}\cdot\mathbf{P}_{n'n}|^2}{E_n(0)-E_{n'}(0)}+\mathcal{O}(\mathbf{k}^3), \tag{4}$$

noting that $\mathbf{P}_{nn'}=\mathbf{P}_{n'n}^*$ and discarding terms of cubic order in momentum. This observation completes the perturbation theory up to lowest nontrivial order in momentum, as

$$E_n(\mathbf{k}) = E_n(0) + \frac{\hbar^2\mathbf{k}^2}{2m} + \frac{\hbar^2}{m^2}\sum_{n':E_{n'}\neq E_n}\frac{|\mathbf{k}\cdot\mathbf{P}_{n'n}|^2}{E_n(0)-E_{n'}(0)}+\mathcal{O}(\mathbf{k}^3), \tag{5}$$

$$|u_{n,\mathbf{k}}^{(1)}\rangle = \frac{\hbar}{m}\sum_{n':E_{n'}\neq E_n}\frac{\mathbf{k}\cdot\mathbf{P}_{n'n}}{E_n(0)-E_{n'}(0)}|u_{n,0}\rangle+\mathcal{O}(\mathbf{k}^2). \tag{6}$$

Importantly, it is possible to consider only those bands $n$ with energies $E_n(0)$ sufficiently close to the charge neutrality point, or in other words, to reduce the number of degrees of freedom by choosing a restricted selection of the $|u_{n,0}\rangle$ as basis. However, the perturbations involve a sum over all bands $n'$, also those outside of the basis. These bands are known as *remote bands* in the context of $\mathbf{k}\cdot\mathbf{p}$ theory.

### 2.1.2 The Kane model

In the seminal paper by Kane [1], $\mathbf{k}\cdot\mathbf{p}$ theory was used to calculate the band structure of InSb. As a matter of fact, Kane's model applies to many more semiconductor materials with a zincblende lattice structure. The important extra ingredient in Kane's analysis is spin-orbit coupling, one of the relativistic corrections that affect the atomic orbitals. The Hamiltonian $H_0$ (cf. Eq. (1)) is modified by adding the spin-orbit term [1]

$$H_{SO} = \frac{\hbar}{4m^2c^2}(\nabla V\times\hat{\mathbf{p}})\cdot\boldsymbol{\sigma} \tag{7}$$

where $\boldsymbol{\sigma}=(\sigma_x,\sigma_y,\sigma_z)$ is the vector of Pauli matrices acting on the spin degree of freedom and $V$ is the potential near the atomic core.

In InSb, the relevant atomic orbitals are the **5s** orbitals of In and the **5p** orbitals of Sb. Together with the spin degree of freedom ($S=\frac{1}{2}$), this yields the eight component basis,

$$|S,\uparrow\rangle,-\tfrac{1}{\sqrt{2}}|X+iY,\downarrow\rangle,|Z,\uparrow\rangle,\tfrac{1}{\sqrt{2}}|X-iY,\downarrow\rangle,|S,\downarrow\rangle,+\tfrac{1}{\sqrt{2}}|X-iY,\uparrow\rangle,|Z,\downarrow\rangle,\tfrac{1}{\sqrt{2}}|X+iY,\uparrow\rangle, \tag{8}$$

where $|S\rangle$ labels the $s$ orbital and $|X\rangle,|Y\rangle,|Z\rangle$ the $p$ orbitals. In this basis, the Hamiltonian at $\mathbf{k}=\mathbf{0}$ is represented by the diagonal matrix $H_0=\mathbf{diag}(E_s,E_p,E_p,E_p,E_s,E_p,E_p,E_p)$, and $H_{SO}$ can be written as

$$H_{SO}=\begin{pmatrix}h_{SO} & 0 \\ 0 & h_{SO}\end{pmatrix} \qquad\text{with}\qquad h_{SO}=\begin{pmatrix}0 & 0 & 0 & 0 \\ 0 & -\frac{1}{3}\Delta_{SO} & \frac{\sqrt{2}}{3}\Delta_{SO} & 0 \\ 0 & \frac{\sqrt{2}}{3}\Delta_{SO} & 0 & 0 \\ 0 & 0 & 0 & \frac{1}{3}\Delta_{SO}\end{pmatrix} \tag{9}$$

where $\Delta_{SO}=(3i\hbar/4m^2c^2)\langle X|(\partial_x V\hat{p}_y-\partial_y V\hat{p}_x)|Y\rangle$ is the spin-orbit splitting; the eigenvalues of $h_{SO}$ are $0,\frac{1}{3}\Delta_{SO},\frac{1}{3}\Delta_{SO},-\frac{2}{3}\Delta_{SO}$. Thus, $h_{SO}$ partially lifts the degeneracy between the $p$ orbital states (six states at energy $E_p$) to four states at $E_p+\frac{1}{3}\Delta_{SO}$ and two states at $E_p-\frac{2}{3}\Delta_{SO}$.

The eigenstates of $H_0 + H_{SO}$ are formed by the basis

$$|1\rangle = |\Gamma_6, +\tfrac{1}{2}\rangle = |S, \uparrow\rangle,$$

$$|2\rangle = |\Gamma_6, -\tfrac{1}{2}\rangle = |S, \downarrow\rangle,$$

$$|3\rangle = |\Gamma_8, +\tfrac{3}{2}\rangle = \frac{1}{\sqrt{2}}|X + iY, \uparrow\rangle,$$

$$|4\rangle = |\Gamma_8, +\tfrac{1}{2}\rangle = \frac{1}{\sqrt{6}}\left[|X + iY, \downarrow\rangle - 2|Z, \uparrow\rangle\right],$$

$$|5\rangle = |\Gamma_8, -\tfrac{1}{2}\rangle = -\frac{1}{\sqrt{6}}\left[|X - iY, \uparrow\rangle + 2|Z, \downarrow\rangle\right], \tag{10}$$

$$|6\rangle = |\Gamma_8, -\tfrac{3}{2}\rangle = -\frac{1}{\sqrt{2}}|X - iY, \downarrow\rangle,$$

$$|7\rangle = |\Gamma_7, +\tfrac{1}{2}\rangle = \frac{1}{\sqrt{3}}\left[|X + iY, \downarrow\rangle + |Z, \uparrow\rangle\right],$$

$$|8\rangle = |\Gamma_7, -\tfrac{1}{2}\rangle = \frac{1}{\sqrt{3}}\left[|X - iY, \uparrow\rangle - |Z, \downarrow\rangle\right],$$

where the notation $|\Gamma_r, m_j\rangle$ refers to the irreducible representation $\Gamma_r$ ($r = 6, 7, 8$) of the point group $T_d$ under which the states transform as well as the total angular momentum quantum number ($J_z$ eigenvalue) $m_j$. In the basis of Eq. (10), the Hamiltonian $H_0 + H_{SO}$ can be diagonalized as

$$H_0 + H_{SO} = \begin{pmatrix} E_c & 0 & & & & & & \\ 0 & E_c & & & & & & \\ & & E_v & 0 & 0 & 0 & & \\ & & 0 & E_v & 0 & 0 & & \\ & & 0 & 0 & E_v & 0 & & \\ & & 0 & 0 & 0 & E_v & & \\ & & & & & & E_v - \Delta_{SO} & 0 \\ & & & & & & 0 & E_v - \Delta_{SO} \end{pmatrix}, \tag{11}$$

with $E_c \equiv E_s$, $E_v \equiv E_p + \tfrac{1}{3}\Delta$, and $E_v - \Delta_{SO} \equiv E_p - \tfrac{2}{3}\Delta$ referring to the 'conduction band', 'valence band' and split-off band energies at $\Gamma$, respectively. Note that in the notation we use the labels for 'conduction' and 'valence' band for the $\Gamma_6$ and $\Gamma_8$ bands, respectively, without regard to the actual band ordering. For inverted materials, $E_c$ lies in the valence band and $E_v$ in the conduction band.

We note that between references (cf. Refs. [1–3, 16], for example), the basis may differ slightly by complex phases and order. Here, we group the multiplets belonging to each representation ($\Gamma_6$, $\Gamma_8$, $\Gamma_7$) together. The Hamiltonian thus has the block structure

$$H = \left( \begin{array}{c|c|c} H_{66} & H_{68} & H_{67} \\ \hline H_{86} & H_{88} & H_{87} \\ \hline H_{76} & H_{78} & H_{77} \end{array} \right). \tag{12}$$

Where necessary due to space restriction, we will write terms of the full $8 \times 8$ Hamiltonian in terms of these smaller blocks.

The $\mathbf{k} \cdot \hat{\mathbf{p}}$ term in the Hamiltonian introduces matrix elements proportional to $k_x \langle S|p_x|X\rangle$, $k_y \langle S|p_y|Y\rangle$ and $k_z \langle S|p_z|Z\rangle$. To write these terms, one defines the *Kane parameter*

$$P = \frac{\hbar}{m}|\langle S|\hat{p}_x|X\rangle| = \frac{\hbar}{m}|\langle S|\hat{p}_y|Y\rangle| = \frac{\hbar}{m}|\langle S|\hat{p}_z|Z\rangle|. \tag{13}$$

The ambiguity in the definitions of $P$ in the literature (again, cf. Refs. [1–3, 16]) is resolved by choosing a real and positive value. The value of the Kane parameter is of similar magnitude for many materials due to the similarity of the orbital wave functions.

The next step towards the formulation of the full Kane model is to identify all possible inversion symmetric terms up to quadratic order. (We will discuss inversion asymmetric terms in Sec. 2.6.) Using the conventions of Ref. [3], we write

$$H_{Kane} = H_0 + H_{SO} + H_k \tag{14}$$

in terms of the constant part $H_0 + H_{\text{SO}}$ [Eq. (11)] and the momentum-dependent part

$$
H_{\text{k}} =
\begin{pmatrix}
T_{\text{k}} & 0 & -\sqrt{\tfrac{1}{2}}Pk_+ & \sqrt{\tfrac{2}{3}}Pk_z & \sqrt{\tfrac{1}{6}}Pk_- & 0 & -\sqrt{\tfrac{1}{3}}Pk_z & -\sqrt{\tfrac{1}{3}}Pk_- \\
0 & T_{\text{k}} & 0 & -\sqrt{\tfrac{1}{6}}Pk_+ & \sqrt{\tfrac{2}{3}}Pk_z & \sqrt{\tfrac{1}{2}}Pk_- & -\sqrt{\tfrac{1}{3}}Pk_+ & \sqrt{\tfrac{1}{3}}Pk_z \\
-\sqrt{\tfrac{1}{2}}Pk_- & 0 & U_{\text{k}}+V_{\text{k}} & -S_{\text{k}}^- & R_{\text{k}} & 0 & \tfrac{1}{\sqrt{2}}S_{\text{k}}^- & -\sqrt{2}R_{\text{k}} \\
\sqrt{\tfrac{2}{3}}Pk_z & -\sqrt{\tfrac{1}{6}}Pk_- & -S_{\text{k}}^{-\dagger} & U_{\text{k}}-V_{\text{k}} & C_{\text{k}} & R_{\text{k}} & \sqrt{2}V_{\text{k}} & -\sqrt{\tfrac{3}{2}}\tilde{S}_{\text{k}}^- \\
\sqrt{\tfrac{1}{6}}Pk_+ & \sqrt{\tfrac{2}{3}}Pk_z & R_{\text{k}}^\dagger & C_{\text{k}}^\dagger & U_{\text{k}}-V_{\text{k}} & S_{\text{k}}^{+\dagger} & -\sqrt{\tfrac{3}{2}}\tilde{S}_{\text{k}}^+ & -\sqrt{2}V_{\text{k}} \\
0 & \sqrt{\tfrac{1}{2}}Pk_+ & 0 & R_{\text{k}}^\dagger & S_{\text{k}}^+ & U_{\text{k}}+V_{\text{k}} & \sqrt{2}R_{\text{k}}^\dagger & \tfrac{1}{\sqrt{2}}S_{\text{k}}^+ \\
-\sqrt{\tfrac{1}{3}}Pk_z & -\sqrt{\tfrac{1}{3}}Pk_- & \tfrac{1}{\sqrt{2}}S_{\text{k}}^{-\dagger} & \sqrt{2}V_{\text{k}} & -\sqrt{\tfrac{3}{2}}\tilde{S}_{\text{k}}^{+\dagger} & \sqrt{2}R_{\text{k}} & U_{\text{k}} & C_{\text{k}} \\
-\sqrt{\tfrac{1}{3}}Pk_+ & \sqrt{\tfrac{1}{3}}Pk_z & -\sqrt{2}R_{\text{k}}^\dagger & -\sqrt{\tfrac{3}{2}}\tilde{S}_{\text{k}}^{-\dagger} & -\sqrt{2}V_{\text{k}} & \tfrac{1}{\sqrt{2}}S_{\text{k}}^{+\dagger} & C_{\text{k}}^\dagger & U_{\text{k}}
\end{pmatrix}
\tag{15}
$$

with

$$
T_{\text{k}} = \frac{\hbar^2}{2m}(2F+1)|\mathbf{k}|^2
$$

$$
U_{\text{k}} = -\frac{\hbar^2}{2m}\gamma_1|\mathbf{k}|^2 \qquad\qquad V_{\text{k}} = -\frac{\hbar^2}{2m}\gamma_2\left(k_x^2+k_y^2-2k_z^2\right)
$$

$$
R_{\text{k}} = \frac{\hbar^2}{2m}\sqrt{3}\left(\gamma_2\left(k_x^2-k_y^2\right)-2\mathrm{i}\gamma_3 k_x k_y\right)
\tag{16}
$$

$$
S_{\text{k}}^\pm = -\frac{\hbar^2}{2m}\sqrt{3}\left(k_\pm\{\gamma_3,k_z\}+k_\pm[\kappa,k_z]\right) \qquad \tilde{S}_{\text{k}}^\pm = -\frac{\hbar^2}{2m}\sqrt{3}\left(k_\pm\{\gamma_3,k_z\}-\frac{1}{3}k_\pm[\kappa,k_z]\right)
$$

$$
C_{\text{k}} = \frac{\hbar^2}{2m}2k_-[\kappa,k_z].
$$

where $|\mathbf{k}|^2 = k_x^2+k_y^2+k_z^2$ and $k_\pm = k_x \pm \mathrm{i}k_y$. The band energies $E_{\text{c}}$, $E_{\text{v}}$ and $E_{\text{v}}-\Delta_{\text{SO}}$, the Kane parameter $P$, and the band structure parameters $F$, $\gamma_{1,2,3}$, $\kappa$ are material properties, that contain contributions from couplings with remote bands.

### 2.1.3 Axial symmetry

Most of the matrix elements of Eq. (16) preserve axial symmetry, i.e., they are invariant under a rotation around the $z$ axis. The only exception is $R_{\text{k}}$, which can be expanded into an axial and nonaxial part, $R_{\text{k}} = R_{\text{k}}^{\text{ax}} + R_{\text{k}}^{\text{nonax}}$ with

$$
R_{\text{k}}^{\text{ax}} = \frac{\hbar^2}{2m}\frac{\sqrt{3}}{2}(\gamma_2+\gamma_3)k_-^2, \qquad R_{\text{k}}^{\text{nonax}} = \frac{\hbar^2}{2m}\frac{\sqrt{3}}{2}(\gamma_2-\gamma_3)k_+^2.
\tag{17}
$$

In some calculations, we use the *axial approximation*, where we approximate $R_{\text{k}} \approx R_{\text{k}}^{\text{ax}}$ and neglect the non-axial part $R_{\text{k}}^{\text{nonax}}$. We note that the non-axial contributions are significant in many cases, so that the axial approximation should be used with care. This issue will be discussed in more detail in the context of Landau levels in Sec. 2.4.

### 2.1.4 Band structure parameters

For the simulations in kdotpy, we use the material-dependent band structure parameters established by Refs. [2, 3], summarized in Table 1. Since kdotpy has started out as a simulation program for HgTe and CdTe, we use these materials as an example here, noting that the modelling can be applied for a much wider variety of materials. We define the band energy of the $\Gamma_8$ orbitals of unstrained HgTe as the reference energy $E = 0$. Thus, $E_{\text{v}} = 0\,\text{meV}$ for HgTe by definition. The valence band energy for CdTe is determined by the valence band offset $E_{\text{VBO}} = -570\,\text{meV}$ of CdTe with respect to HgTe. (All band energies, including the valence band offset, are treated as a material parameters in this model.)

All parameters come with substantial error bars. Especially for the Luttinger parameters $\gamma_{1,2,3}$ one finds substantial variations among various literature sources [16, 17]. In order to allow for adjustments of the material parameters, we have equipped kdotpy with an interface where the values can be

| | $E_c$ meV | $E_v$ meV | $E_v - \Delta_{SO}$ meV | $P$ meV nm | $2F+1$ | $\gamma_1$ | $\gamma_2$ | $\gamma_3$ | $\kappa$ |
|---|---|---|---|---|---|---|---|---|---|
| HgTe | −303 | 0 | −1080 | 846 | 1 | 4.10 | 0.50 | 1.30 | −0.40 |
| CdTe | 1036 | −570 | −1480 | 846 | 0.82 | 1.47 | −0.28 | 0.03 | −1.31 |

Table 1: Coefficients of the inversion-symmetric $\mathbf{k} \cdot \mathbf{p}$ Hamiltonian $H_k$ at zero temperature. The values are adapted from Refs. [2,3]. We have taken into account the valence band offset $E_{VBO} = -570\,\text{meV}$ for CdTe, taking $E_v = 0\,\text{meV}$ for HgTe as the reference energy. These are also the default parameters implemented in kdotpy.

changed in the form of a configuration file (see Sec. 3.2.4 and Appendix B.2). We have supplied the default parameters as listed in Table 1.

For ternary compounds that are formed by alloying two binary compounds for example $\text{Hg}_{1-x}\text{Cd}_x\text{Te}$ from HgTe and CdTe, we determine the parameters by suitable interpolation. These interpolations are linear to first approximation, but higher-order corrections are typically also taken into consideration. For example, the gap energy $E_g = E_c - E_v$ for $\text{Hg}_{1-x}\text{Cd}_x\text{Te}$ is

$$E_g(\text{Hg}_{1-x}\text{Cd}_x\text{Te}) = -303(1-x) + 1606x - 132x(1-x) \tag{18}$$

in units of meV, at zero temperature [18,19]. This expression is a quadratic perturbation to the linear interpolation, with the coefficient of the latter term being known as the *bowing parameter*. The Luttinger parameters $\gamma_{1,2,3}$ and $\kappa$ are approximated by cubic polynomials, approximating the result of the interpolation scheme used in Ref. [2]. In general, material parameters like $E_g$ are temperature-dependent [18,19].

### 2.1.5 Kane models with different number of orbitals

The Kane model can also be formulated with a different number of orbitals. For example, a simpler **6 × 6** version can be used, where the $\Gamma_7$ orbitals are omitted. Another version contains **14** orbitals, which adds a quadruplet of $\Gamma_8$ states and a doublet of $\Gamma_7$ states in the conduction band. We will neither consider the **14** orbital in this work nor have implemented it into kdotpy, because the six additional orbitals add complexity that is not necessary for most purposes for which we use kdotpy. The so-called Luttinger model with only the four $\Gamma_8$ orbitals [20] is also left out of consideration because it cannot capture the essential physics of topological insulators, where $\Gamma_6$ and $\Gamma_8$ are inverted.

If one changes perspective between the **6** and **8** orbital Kane model, one needs to take into account renormalizations of the coefficients. The reason for this renormalization is that in the **6** orbital model, the $\Gamma_7$ orbitals become remote bands, and thus one must consider their additional perturbative contributions $\mathbf{P}_{n'n}$ (cf. Eqs. (5) and (6)). This perturbation contributes to (for example) the band masses of the $\Gamma_6$ and $\Gamma_8$ orbitals. In the **8** orbital model, the $\Gamma_7$ orbitals are treated explicitly and thus its perturbative contribution on the other bands is absent. To ascertain that both models yield the same dispersion, the band masses of the $\Gamma_6$ and $\Gamma_8$ orbitals need to be renormalized. Similar renormalizations apply to other coefficients and between different pairs of models. Details are provided, for example, in the tables of Winkler [15].

## 2.2 Lower dimensional geometries

So far, we have considered the Hamiltonian in terms of the momentum coordinates $(k_x, k_y, k_z)$. These momenta are good quantum numbers by virtue of Bloch's theorem. For Bloch's theorem to be valid, the system must have (discrete) translational symmetry in three dimensions. Thus, for a bulk crystal, this description is appropriate.

However, experimentally relevant systems are of finite size; thus the translational symmetry is broken. Nevertheless, if the dimensions of the system are sufficiently large, the system may still be treated as approximately infinite. The decisive criterion is the size of the system compared to the de Broglie wavelength of the particles, or equivalently, the (quantum mechanical) confinement energy compared to other energy scales in the system.

Typically, we have a 'hybrid' situation, where the system is infinite in some and finite in other directions. For semiconductor devices simulated by kdotpy, we typically consider '2D' and '1D' geometries,

where $n$D refers to the number $n$ of dimensions in which the system has translational symmetry, or, in other words, the number of momentum coordinates.

A prominent example of a 2D geometry is a quantum well system. In the 2D geometry, translational symmetry is broken in $z$ direction whereas it is preserved in the $x$ and $y$ direction. The appropriate coordinates in this geometry are thus $(k_x, k_y, z)$. This system may be thought of as an infinite slab of material. More generically, any stack of layers of different materials, called a 'layer stack', is described as a 2D geometry. In this case, the material parameters (like $E_c$, $E_v$, $\gamma_{1,2,3}$, etc.) are treated as a function of $z$ [1]. The material parameters are effectively constant in the bulk of each layer and transition smoothly between them at the interfaces between layers. Many important topological aspects, like surface states, can exist only at the interface between a topological and a trivial material and thus need to be simulated in the 2D geometry.

The 1D geometry is equivalent to an infinite wire, with translational symmetry only in $x$ direction and a finite size both in $y$ and $z$ direction, with coordinates $(k_x, y, z)$. The extent in $y$ and $z$ direction must not necessarily be equal or even similar in size. Typically, in order to simulate the edge states of the quantum spin Hall effect, one considers a 'ribbon' or 'strip' of material, where the thickness is $\sim 10\,\mathrm{nm}$ and the width is $\sim 500\,\mathrm{nm}$ or larger [4].

### 2.2.1 Dimensional reduction

The reduction of the bulk Hamiltonian (3D) to a lower dimensionality is a two step process for each of the confined dimensions. Firstly, the momentum coordinate in the Hamiltonian is substituted by its representation in spatial coordinates. For example, $k_z$ is substituted by the derivative $-i\partial_z$ (where $\partial_z = \partial/\partial z$). Likewise, $k_z^2$ is substituted by the second-order derivative $-\partial_z^2$. These substitutions essentially encode an inverse Fourier transform. Secondly, we choose a finite basis for each spatial direction. This step is necessary, because the computational Hilbert space must be of finite dimension. Here, we specifically choose the basis defined by a finite set of points at which the wave functions are evaluated. We consider uniformly spaced grids of the form $\{z_j\}_{j=j_{\min},\dots,j_{\max}}$, with $z_j = j\Delta z$, where $\Delta z$ is the grid resolution and $j$ is an integer index taken from the finite range $[j_{\min}, j_{\max}]$. We note that our choice differs from the Fortran code of Refs. [2,3], which uses a set of envelope functions based on Legendre polynomials.

In the chosen basis, the discretization of the first derivatives is given by

$$\partial_z\psi(z) = \lim_{dz\to 0}\frac{\psi(z+dz)-\psi(z-dz)}{2dz} \approx \frac{\psi(z+\Delta z)-\psi(z-\Delta z)}{2\Delta z} \to \partial_z\psi_j = \frac{\psi_{j+1}-\psi_{j-1}}{2\Delta z}, \quad (19)$$

where in the final step, we substitute $z = z_j$ and write $\psi_j = \psi(z_j)$ and $\psi_{j\pm 1} = \psi(z_{j\pm 1}) = \psi(z_j \pm \Delta z)$. The second derivative is obtained by applying the same principle twice, but substituting $dz \to \Delta z/2$,

$$\begin{aligned}\partial_z^2\psi(z) &= \lim_{dz\to 0}\frac{\psi(z+2dz)-2\psi(z)+\psi(z-2dz)}{4dz^2} \\ &\approx \frac{\psi(z+\Delta z)-2\psi(z)+\psi(z-\Delta z)}{\Delta z^2} \to \partial_z^2\psi_j = \frac{\psi_{j+1}-2\psi_j+\psi_{j-1}}{\Delta z^2}.\end{aligned} \quad (20)$$

The inverse-Fourier and discretization steps combined lead to the substitution rules

$$k_z\psi \to \hat{k}_z\psi_j = \frac{-i}{2\Delta z}(\psi_{j+1}-\psi_{j-1}), \quad (21)$$

$$k_z^2\psi \to \hat{k}_z^2\psi_j = \frac{-1}{\Delta z^2}(\psi_{j+1}-2\psi_j+\psi_{j-1}), \quad (22)$$

where we use the notation $\hat{k}_z$ to emphasize that this object is an operator. The substitution rules for $k_y$ and $k_y^2$ are analogous to those for $k_z$ and $k_z^2$, respectively.

### 2.2.2 Hermitian discretization

In a layered system (2D geometry), where the material parameters are functions of $z$, special care needs to be taken that these functions generally do not commute with $\hat{k}_z = -i\partial_z$. For this reason, we find

---

[1] We treat the Kane parameter $P$ on equal footing as the other material parameters. In some works (see, e.g., Ref. [2]), it is argued that $P$ must be equal in all layers, as a result of how it is defined in terms of the atomic orbitals, cf. Eq. (13). Since the model yields physical result also if we relax this restriction, we allow $P$ to have different values between layers.

anticommutators and commutators of the form $\{Q, k_z\}$ and $[Q, k_z]$, respectively, in the off-diagonal matrix elements of the Hamiltonian, where $Q$ denotes a generic $z$-dependent material parameter.

The operator $\hat{k}_z$ being a derivative, this leads to contributions involving $\partial_z Q$. To clarify this statement, it is useful to expand the matrix element $\langle \phi | \{Q, \hat{k}_z\} | \psi \rangle$ in terms of its spatial representation as an integral over $z$,

$$\langle \phi | \{Q, \hat{k}_z\} | \psi \rangle = -i \int dz \, \phi^*(z) (Q(z) \partial_z + \partial_z Q(z)) \psi(z)$$

$$= -i \int dz \, (2\phi^*(z) Q(z) \psi'(z) + \psi^*(z) Q'(z) \psi(z)), \tag{23}$$

where a prime denotes the $z$ derivative. In the same representation, invoking integration by parts and assuming that the states vanish at the boundaries of the integration domain, one can also write

$$\langle \phi | \{Q, \hat{k}_z\} | \psi \rangle = -i \int dz \left( \phi^*(z) Q(z) \partial_z \psi(z) + \phi^*(z) \partial_z (Q(z) \psi(z)) \right) = -i \left( \langle \phi | Q | \partial_z \psi \rangle - \langle \partial_z \phi | Q | \psi \rangle \right). \tag{24}$$

For the matrix element $\langle \phi | [Q, \hat{k}_z] | \psi \rangle$ involving a commutator, we find that

$$\langle \phi | [Q, \hat{k}_z] | \psi \rangle = -i \int dz \, \phi^*(z) (Q(z) \partial_z - \partial_z Q(z)) \psi(z) = -i \int dz \, \psi^*(z) Q'(z) \psi(z) = \langle \phi | Q' | \psi \rangle. \tag{25}$$

In other words, the commutator $[Q, \hat{k}_z]$ only contributes where $Q'(z) \equiv \partial_z Q(z)$ is nonzero, which is only near the interfaces between layers, as the material parameters are constant in the bulk of each layer. The commutator terms can thus be interpreted as interface terms.

On the diagonal of the Hamiltonian, the matrix elements $T_k$ and $U_k$ contain effective-mass terms, quadratic in the momentum $k_z$. The correct substitution to operator form is given by a symmetrized triple product, in general

$$q k_i k_j \rightarrow \{k_i q k_j\}_S \equiv \tfrac{1}{2} (\hat{k}_i \hat{q} \hat{k}_j + \hat{k}_j \hat{q} \hat{k}_i), \tag{26}$$

where $i, j = x, y, z$ and $\hat{q}$ is a hermitian operator [2, 21, 22]. For the relevant diagonal matrix elements with effective mass terms of the form $Q k_z^2$, the substitution rule reads $Q k_z^2 \rightarrow \hat{k}_z Q(z) \hat{k}_z$. In the spatial representation, these terms can be written as

$$\langle \phi | \hat{k}_z Q \hat{k}_z | \psi \rangle = -\int dz \, \phi^*(z) \partial_z (Q(z) \partial_z \psi(z)) = \int dz \, \phi'^*(z) Q(z) \psi'(z) = \langle \phi' | Q | \psi' \rangle, \tag{27}$$

where $\psi'(z) \equiv \partial_z \psi(z)$.

The first-order terms are discretized as follows. We take the form of Eq. (24) and substitute $\psi(z) \rightarrow (\psi(z + \tfrac{1}{2}\Delta z) + \psi(z - \tfrac{1}{2}\Delta z))/2$ and $\partial_z \psi(z) \rightarrow (\psi(z + \tfrac{1}{2}\Delta z) - \psi(z - \tfrac{1}{2}\Delta z))/\Delta z$ and analogous rules for $\phi^*(z)$ and $\partial_z \phi^*(z)$. We thus obtain

$$\frac{-i}{2\Delta z} \sum_z \Big[ \Big( \phi^*(z + \tfrac{1}{2}\Delta z) + \phi^*(z - \tfrac{1}{2}\Delta z) \Big) Q(z) \Big( \psi(z + \tfrac{1}{2}\Delta z) - \psi(z - \tfrac{1}{2}\Delta z) \Big)$$

$$- \Big( \phi^*(z + \tfrac{1}{2}\Delta z) - \phi^*(z - \tfrac{1}{2}\Delta z) \Big) Q(z) \Big( \psi(z + \tfrac{1}{2}\Delta z) + \psi(z - \tfrac{1}{2}\Delta z) \Big) \Big] \tag{28}$$

By expanding and erasing the terms that cancel, we find

$$\frac{-i}{\Delta z} \sum_z \Big( \phi^*(z - \tfrac{1}{2}\Delta z) Q(z) \psi(z + \tfrac{1}{2}\Delta z) - \phi^*(z + \tfrac{1}{2}\Delta z) Q(z) \psi(z - \tfrac{1}{2}\Delta z) \Big) \tag{29}$$

In our computational basis, the Hilbert space is defined as the wave functions on the discrete coordinates $z_j$, so we must align $z \pm \tfrac{1}{2}\Delta z$ with these values. This implies that $Q(z)$ is evaluated at intermediate points $z_j \pm \tfrac{1}{2}\Delta z$, which is not a problem since $Q$ is a function which we can evaluate anywhere. The resulting expression can be written in three equivalent forms, related by shifts of the 'dummy variable'

$j$,

$$\langle\phi|\{Q,\hat{k}_z\}|\psi\rangle \rightarrow \frac{-\mathrm{i}}{\Delta z}\sum_j \left(\phi_j^* Q(z_j+\tfrac{1}{2}\Delta z)\psi_{j+1} - \phi_j^* Q(z_j-\tfrac{1}{2}\Delta z)\psi_{j-1}\right) \tag{30}$$

$$= \frac{-\mathrm{i}}{\Delta z}\sum_j Q(z_j-\tfrac{1}{2}\Delta z)\left(\phi_{j-1}^*\psi_j - \phi_j^*\psi_{j-1}\right) \tag{31}$$

$$= \frac{-\mathrm{i}}{\Delta z}\sum_j Q(z_j+\tfrac{1}{2}\Delta z)\left(\phi_j^*\psi_{j+1} - \phi_{j+1}^*\psi_j\right). \tag{32}$$

We use the first form [Eq. (30)] to extract the action of the Hamiltonian matrix, $(H\psi)_i = \sum_j H_{ij}\psi_j$. From the second and third forms [Eqs. (31) and (32)], we find that the resulting Hamiltonian matrix is hermitian.

For the matrix elements of commutator form $[Q,\hat{k}_z]$, we simply substitute the derivative $Q'$ into Eq. (25),

$$\langle\phi|[Q,\hat{k}_z]|\psi\rangle \rightarrow \sum_j \phi_j^* Q'(z_j)\psi_j, \tag{33}$$

where $Q'(z_j)$ is the derivative of $Q$ evaluated at the grid point $z_j$. In kdotpy, we use the discrete derivative $Q'(z_j) = (Q'(z_j+\Delta z)-Q'(z_j-\Delta z))/2\Delta z$.

The discretization of the quadratic terms follows from Eq. (27), where we substitute $\psi'(z) = \partial_z\psi(z) \rightarrow (\psi(z+\tfrac{1}{2}\Delta z)-\psi(z-\tfrac{1}{2}\Delta z))/\Delta z$. We thus obtain

$$-\frac{1}{\Delta z^2}\sum_z\left(\phi^*(z+\tfrac{1}{2}\Delta z)-\phi^*(z-\tfrac{1}{2}\Delta z)\right)Q(z)\left(\psi(z+\tfrac{1}{2}\Delta z)-\psi(z-\tfrac{1}{2}\Delta z)\right). \tag{34}$$

We expand and align $z\pm\tfrac{1}{2}\Delta z$ with the discrete coordinates $z_j$, and obtain

$$\langle\phi|\hat{k}_z Q\hat{k}_z|\psi\rangle \rightarrow \frac{1}{\Delta z^2}\sum_j\left(\phi_j^* Q(z_j+\tfrac{1}{2}\Delta z)(\psi_{j+1}-\psi_j) + \phi_j^* Q(z_j-\tfrac{1}{2}\Delta z)(\psi_{j-1}-\psi_j)\right) \tag{35}$$

$$= -\frac{1}{\Delta z^2}\sum_j(\phi_j^*-\phi_{j-1}^*)Q(z_j-\tfrac{1}{2}\Delta z)(\psi_j-\psi_{j-1}) \tag{36}$$

$$= -\frac{1}{\Delta z^2}\sum_j(\phi_{j+1}^*-\phi_j^*)Q(z_j+\tfrac{1}{2}\Delta z)(\psi_{j+1}-\psi_j). \tag{37}$$

The first form [Eq. (35)] again defines the action of the Hamiltonian matrix $(H\psi)_i = \sum_j H_{ij}\psi_j$ and the second and third forms [Eqs. (36) and (37)] show that the expression is hermitian if $\psi = \phi$.

## 2.3 Magnetic fields

### 2.3.1 Peierls substitution

In (classical) Hamiltonian mechanics, the motion of a charged particle with dispersion $p^2/2m$ and charge $-e$ in a magnetic field $\mathbf{B}$ is given by

$$H = \frac{1}{2m}\left(\mathbf{p}+e\mathbf{A}\right)^2, \tag{38}$$

in terms of the magnetic gauge field $\mathbf{A}$ that satisfies $\mathbf{B} = \nabla\times\mathbf{A}$. In this expression, $\mathbf{p}$ acts as the canonical momentum, while $\mathbf{\Pi} = \mathbf{p}+e\mathbf{A}$ is the kinetic momentum, equal to mass times velocity. In order to obtain the Hamiltonian for the motion of a particle in a magnetic field from a generic zero-field Hamiltonian, one applies the Peierls substitution

$$\mathbf{p} \rightarrow \mathbf{\Pi} = \mathbf{p}+e\mathbf{A}, \tag{39}$$

or equivalently, $\mathbf{k} \rightarrow \mathbf{k}+(e/\hbar)\mathbf{A}$.

The electromagnetic gauge field $\mathbf{A}$ is subject to gauge invariance by the transformation $\mathbf{A} \rightarrow \mathbf{A}+\nabla\Lambda$ where $\Lambda$ is a function depending on the spatial coordinates. This gauge transformation does not alter the relation $\mathbf{B} = \nabla\times\mathbf{A}$. (Here, and in the remainder of the work, we assume that $\mathbf{B}$ and $\mathbf{A}$ are not

time-dependent.) This leaves us with a freedom to choose the gauge conveniently. For example, for a perpendicular magnetic field $\mathbf{B} = (0, 0, B_z)$, two common gauge choices are the symmetric gauge $\mathbf{A} = B_z(-y/2, x/2, 0)$ and the Lorentz gauge $\mathbf{A} = B_z(-y, 0, 0)$. For the numerical simulations, one should notice that the symmetric gauge breaks translational invariance in $x$ and $y$ direction, while the Lorentz gauge breaks it in $y$ direction only. Thus, for simulations of a device in a magnetic field, kdotpy uses the Lorentz gauge so that the translational symmetry in $x$ direction is left intact. This simulation is done in the strip geometry, i.e., the 1D geometry with momentum coordinates $k_x$ and spatial coordinates $y$ and $z$.

For in-plane fields $\mathbf{B} = (B_x, B_y, 0)$, on similar grounds we choose a gauge that depends only on the $z$ coordinate, leaving translational symmetries in $x$ and $y$ directions intact (if they are not yet broken by some other means). The gauge that satisfies this property is $\mathbf{A} = (B_y z, -B_x z, 0)$. For generic magnetic fields $\mathbf{B} = (B_x, B_y, B_z)$ we simply take the sum of in-plane and out-of-plane, and use the gauge

$$\mathbf{A} = (B_y z - B_z y, -B_x z, 0). \tag{40}$$

The appropriate geometry for the simulations depends on the out-of-plane component: If $B_z \neq 0$, a 1D geometry is needed, while for a purely in-plane field, the 2D geometry is typically the appropriate one.

In quantum mechanics, the momentum and gauge fields discussed above must be replaced by the appropriate operators. Importantly, the momentum $\hat{k}$ and gauge field $\hat{A}$ operators (which is a function of spatial coordinates) do not commute in general. We quantize the quadratic terms in a symmetric way

$$\left(k_i + \frac{e}{\hbar} A_i\right)\left(k_j + \frac{e}{\hbar} A_j\right) \rightarrow \hat{k}_i \hat{k}_j + \frac{e}{\hbar}(\hat{k}_i A_j + \hat{k}_j A_i) + \left(\frac{e}{\hbar}\right)^2 A_i A_j \tag{41}$$

$$= -\partial_i \partial_j - i\frac{e}{\hbar}\left(\partial_i A_j + \partial_j A_i + A_i \partial_j + A_j \partial_i\right) + \left(\frac{e}{\hbar}\right)^2 A_i A_j \tag{42}$$

with $i, j = x, y, z$, because this form assures that the Hamiltonian is hermitian [23]. The Hamiltonian is gauge invariant under the gauge transformation $\mathbf{A} \rightarrow \mathbf{A} + \nabla \Lambda$ and $|\psi\rangle \rightarrow \exp(-ie\Lambda/\hbar)|\psi\rangle$, where $\Lambda$ is a function of the spatial coordinates [23].

Importantly, the kinetic momentum operators do not commute,

$$\left[k_i + \frac{e}{\hbar} A_i, k_j + \frac{e}{\hbar} A_j\right] = -i\frac{e}{\hbar}(\partial_i A_j - \partial_j A_i) = -i\frac{e}{\hbar} \sum_k \epsilon_{ijk} B_k, \tag{43}$$

which raises the question how terms of the form $(k_i + (e/\hbar)A_i)(k_j + (e/\hbar)A_j)$ should be quantized. Obviously, if $i = j$, there is no ambiguity. For the off-diagonal terms, we assume the following quantization rules

$$k_\pm^2 \rightarrow (\hat{k}_x')^2 - (\hat{k}_y')^2 \pm i(\hat{k}_x' \hat{k}_y' + \hat{k}_y' \hat{k}_x')$$

$$= -\partial_x^2 + \partial_y^2 \mp 2i\partial_x \partial_y + 2\frac{e}{\hbar}(A_x \pm iA_y)(-i\partial_x \pm \partial_y) \mp \frac{e}{\hbar} B_z + \left(\frac{e}{\hbar}\right)^2 (A_x \pm iA_y)^2 \tag{44}$$

$$k_\pm k_z \rightarrow \frac{1}{2}\{\hat{k}_x' \pm i\hat{k}_y', \hat{k}_z'\}$$

$$= -i(\partial_x \pm i\partial_y)\partial_z - i\frac{e}{\hbar}(A_x \pm iA_y)\partial_z \mp \frac{e}{2\hbar}(B_x \pm iB_y) \tag{45}$$

where we have defined $\hat{k}_i' = \hat{k} + (e/\hbar)A_i$ and we have assumed the gauge defined by Eq. (40). The terms $eB_z/\hbar$ and $e(B_x \pm iB_y)/2\hbar$ in Eqs. (44) and (45), respectively, can be interpreted as originating from the non-commutative character of the kinetic momentum operators. For the sake of doing the numerics, we further substitute $A_x \pm iA_y = (B_y \mp iB_x)z - B_z y = \mp i(B_x \pm iB_y)z - B_z y$.

### 2.3.2 Out-of-plane magnetic field in strip geometry

The case of a pure out-of-plane field, $\mathbf{B} = (0, 0, B_z)$, can be treated in the strip geometry, i.e., the 1D case with coordinates $(k_x, y, z)$. We choose the gauge $\mathbf{A} = (A_x, 0, 0)$ with $A_x = -B_z(y - y_0)$, where the *gauge origin* $y_0$ can be freely chosen.

In kdotpy, we choose $y_0 = 0$ to be the centre of the strip, so that $A_x$ is antisymmetric under reflection in $y$. The momentum coordinate is simply shifted according to the Peierls substitution, $k_x \rightarrow$

$k_x - (e/\hbar)B_z y$. The operators $\hat{k}_\pm^2$ are calculated along similar lines as Eq. (44), but with $\hat{k}'_x = k_x - (eB_z/\hbar)y$,

$$k_\pm^2 \rightarrow \left(k_x - \frac{eB_z}{\hbar}y\right)^2 \pm 2\mathrm{i}\left(k_x - \frac{eB_z}{\hbar}y\right)(-\mathrm{i}\partial_y) + \partial_y^2 \mp \frac{eB_z}{\hbar}. \tag{46}$$

We describe its action in the Hilbert space defined by the discrete coordinates $(y_i, z_j)$ on a grid with resolutions $(\Delta y, \Delta z)$ as the matrix element

$$\langle\phi|\hat{k}_\pm^2|\psi\rangle = \sum_{i,j}\phi_{i,j}^*\left[\left(k_x - \frac{eB_z}{\hbar}y\right)^2\psi_{i,j} \pm 2\left(k_x - \frac{eB_z}{\hbar}y\right)\left(\frac{\psi_{i+1,j}-\psi_{i-1,j}}{2\Delta y}\right)\right.$$
$$\left. -\frac{\psi_{i+1,j}-2\psi_{i,j}+\psi_{i-1,j}}{(\Delta y)^2} \mp \frac{eB_z}{2\hbar}\left(\psi_{i+1,j}+\psi_{i-1,j}\right)\right] \tag{47}$$

$$= \sum_{i,j}\left[\phi_{i,j}^*\left(k_x - \frac{eB_z}{\hbar}y\right)^2 \mp 2\frac{\phi_{i+1,j}^*-\phi_{i-1,j}^*}{2\Delta y}\left(k_x - \frac{eB_z}{\hbar}y\right)\right.$$
$$\left. -\frac{\phi_{i+1,j}^*-2\phi_{i,j}^*+\phi_{i-1,j}^*}{(\Delta y)^2} \pm \frac{eB_z}{2\hbar}\left(\phi_{i+1,j}^*+\phi_{i-1,j}^*\right)\right]\psi_{i,j} \tag{48}$$

where $\psi_{i,j} \equiv \psi(y_i, z_j)$. Equation (47) is implemented as matrix element in kdotpy. Equation (48) is the conjugate form obtained from Eq. (47) by shifting the dummy variables $i \rightarrow i \pm 1$ and $y \rightarrow y \pm \Delta y$. Note that the variable shift yields a contribution $(eB_z/\hbar)(\phi_{i+1,j}^* + \phi_{i-1,j}^*)\psi_{i,j}$ from the second term, which has the same structure as the final term. For this reason, we use the non-diagonal form $\mp(eB_z/2\hbar)\phi_{i,j}^*(\psi_{i+1,j}+\psi_{i-1,j})$ for the matrix element of the final term, and not the diagonal form $\mp(eB_z/\hbar)\phi_{i,j}^*\psi_{i,j}$ as one could have expected naively.

### 2.3.3 In-plane magnetic field in slab and strip geometries

Let us consider a pure in-plane field, $\mathbf{B} = (B_x, B_y, 0)$. The gauge choice is $\mathbf{A} = (A_x, A_y, 0)$ with $A_x = B_y(z-z_0)$ and $A_y = -B_x(z-z_0)$. For kdotpy we choose $z_0 = 0$, i.e., the centre of the layer stack.

In the slab (2D) geometry, $k_\pm^2$ simply evaluates as $(k'_x)^2 - (k'_y)^2 \pm 2\mathrm{i}k'_x k'_y$ with $k'_x = k_x + (eB_y/\hbar)z$ and $k'_y = k_y - (eB_x/\hbar)z$. With $k'_\pm = k'_x \pm \mathrm{i}k'_y$ and $B_\pm = B_x \pm \mathrm{i}B_y$, we find $k'_\pm = k_\pm \mp \mathrm{i}(eB_\pm/\hbar)z$. For the $S_\mathbf{k}^\pm$ matrix elements in the Hamiltonian, we consider $k'_\pm\{\gamma_3, \hat{k}_z\}$. Combining Eqs. (30) and (45), we find the matrix element

$$\langle\phi|k'_\pm\{\gamma_3, \hat{k}_z\}|\psi\rangle = \sum_j \phi_j^*\left[-\frac{\mathrm{i}}{\Delta z}\left(k_\pm \mp \mathrm{i}\frac{eB_\pm}{\hbar}z\right)\left(\gamma_3(z+\tfrac{1}{2}\Delta z)\psi_{j+1} - \gamma_3(z-\tfrac{1}{2}\Delta z)\psi_{j-1}\right)\right.$$
$$\left. \mp\frac{eB_\pm}{2\hbar}\left(\gamma_3(z+\tfrac{1}{2}\Delta z)\psi_{j+1} + \gamma_3(z-\tfrac{1}{2}\Delta z)\psi_{j-1}\right)\right] \tag{49}$$

$$= \sum_j\left[\frac{\mathrm{i}}{\Delta z}\left(k_\pm \mp \mathrm{i}\frac{eB_\pm}{\hbar}z\right)\left(\phi_{j+1}^*\gamma_3(z+\tfrac{1}{2}\Delta z) - \phi_{j-1}^*\gamma_3(z-\tfrac{1}{2}\Delta z)\right)\right.$$
$$\left. \pm\frac{eB_\pm}{2\hbar}\phi_{j+1}^*\left(\gamma_3(z+\tfrac{1}{2}\Delta z) + \phi_{j-1}^*\gamma_3(z-\tfrac{1}{2}\Delta z)\right)\right]\psi_j \tag{50}$$

where $k'_\pm = k_\pm \mp \mathrm{i}(eB_\pm/\hbar)z$ and $\psi_j \equiv \psi(z_j)$. Again, these two expressions are related by a shift in variables $j \rightarrow j \pm 1$ and $z \rightarrow z \pm \Delta z$.

In the strip geometry, we obtain $\langle\phi|k'_\pm\{\gamma_3, \hat{k}_z\}|\psi\rangle$ by simply substituting $k_y \rightarrow \hat{k}_y$, where $\hat{k}_y\psi_{i,j} = -\mathrm{i}(\psi_{i+1,j}-\psi_{i-1,j})/\Delta y$. The bracketed parts in Eqs. (49) and (50) do not contain $y$-dependent terms other than the wave functions $\phi$ and $\psi$; $\hat{k}_y$ commutes with the other factors.

### 2.3.4 Generic magnetic fields

For a generic magnetic field $\mathbf{B} = (B_x, B_y, B_z)$, we can straightforwardly adapt the equations above. For the $k_\pm^2$ terms, we simply make the substitutions $\hat{k}'_x = k_x - (eB_z/\hbar)y + (eB_y/\hbar)z$ and $\hat{k}'_y = -\mathrm{i}\partial_y - (eB_x/\hbar)z$, adding the in-plane component of the field to the gauge compared to Eq. (44). The derivative $\partial_y$ commutes with the terms proportional with $z$, hence no extra terms of the form $eB_x/\hbar$ or $eB_y/\hbar$ (constant in space) appear. Likewise, for $k'_\pm\{\gamma_3, \hat{k}_z\}$, the out-of-plane component appears as an extra term $(eB_z/\hbar)y$ in $k_x + (e/\hbar)A_x$ compared to Eqs. (49) and (50). This term commutes with $\partial_z$.

## 2.4 Landau level formalism

### 2.4.1 Out-of-plane field in the axial approximation

If the magnetic field is purely out-of-plane, we have the commutator relation $[\hat{k}'_-, \hat{k}'_+] = 2eB_z/\hbar$ while $\hat{k}'_z$ commutes with both $\hat{k}'_x$ and $\hat{k}'_y$. This structure is formally equivalent to the ladder operators of the harmonic oscillator with commutation relation $[a, a^\dagger] = 1$, if we define

$$a = \sqrt{\frac{\hbar}{2eB_z}}\hat{k}'_- = \frac{1}{\sqrt{2}}l_B\hat{k}'_-, \qquad a^\dagger = \sqrt{\frac{\hbar}{2eB_z}}k'_+ = \frac{1}{\sqrt{2}}l_B\hat{k}'_+, \tag{51}$$

where $l_B = \sqrt{\hbar/eB_z}$ is the magnetic length. (Here, we have tacitly assumed $B_z > 0$.) The eigenstates of the number operator $a^\dagger a$ are denoted $|n\rangle$, where $n$ is the eigenvalue, i.e., $a^\dagger a|n\rangle = n|n\rangle$. The raising and lowering operator act as $a^\dagger|n\rangle = \sqrt{n+1}|n+1\rangle$ and $a|n\rangle = \sqrt{n-1}|n-1\rangle$, respectively.

In this context, these eigenstates are called Landau level states (see, e.g., Refs. [2,3]). The Hamiltonian can be reformulated in terms of ladder operators by substituting $k_+$ and $k_-$ by $a^\dagger$ and $a$, respectively. The combination $k_x^2 + k_y^2 = \frac{1}{2}(k_+k_- + k_-k_+)$ is substituted by a term proportional to $a^\dagger a + aa^\dagger = 2n + 1$.

It can be shown [2,3] that in the axial approximation, the eigenstates of the Hamiltonian can be written in the form

$$|\Psi^{(n)}\rangle = \begin{pmatrix} f_1^{(n)}(z)|n\rangle \\ f_2^{(n)}(z)|n+1\rangle \\ f_3^{(n)}(z)|n-1\rangle \\ f_4^{(n)}(z)|n\rangle \\ f_5^{(n)}(z)|n+1\rangle \\ f_6^{(n)}(z)|n+2\rangle \\ f_7^{(n)}(z)|n\rangle \\ f_8^{(n)}(z)|n+1\rangle \end{pmatrix}, \tag{52}$$

where $n = -2, -1, 0, 1, \ldots$ is called the Landau level index (LL index). For $n = -2, -1, 0$, some of the indices $n'$ in $|n'\rangle$ on the right-hand side are negative; these components are understood to be zero. For $n \geq 1$, the dimension of the subspace spanned by $|\Psi^{(n)}\rangle$ is $8n_z$, where $n_z$ is the number of degrees of freedom in the $z$ direction. (For discrete coordinates $z_j$, as in the kdotpy calculation, $n_z$ is equal to the number of grid points.) For $n = -2, -1, 0$, the dimensionality of the subspace is reduced to $1n_z$, $4n_z$, and $7n_z$, respectively, where $1$, $4$, and $7$ refer to the number of nonzero components in Eq. (52). This observation is important for the implementation, as we will discuss in detail later.

The Landau level index $n$ is a conserved quantity in axial approximation [where we neglect $R_k^{nonax}$, Eq. (17)], meaning that the Hamiltonian is block-diagonal in the basis $\{|\Psi^{(n)}\rangle\}_{n=-2,-1,0,1,\ldots}$, i.e., $\langle\Phi^{(n')}|H^{ax}|\Psi^{(n)}\rangle = 0$ if $n' \neq n$. This property allows to calculate Landau level spectra separately for each Landau level $n = -2, -1, 0, \ldots, n_{max}$, where $n_{max}$ is the desired maximal Landau level index. The result is a set of magnetic-field dependent energy eigenvalues $E_j^{(n)}(B_z)$ for each Landau index $n$, where $j$ runs over all eigenstates within each Landau level. Due to the fact that the basis is finite, by choosing a Landau level cutoff $n_{max}$, the spectrum is inherently incomplete. This is typically a problem only at small magnetic fields, where the energy spacing between Landau levels is small. This energy spacing is equivalent to the cyclotron energy $\hbar\omega_c = \hbar eB_z/m$ known from the theory of the quantum Hall effect.

### 2.4.2 Landau level formalism with axial symmetry breaking

If the nonaxial term $R_k^{nonax}$ [Eq. (17)] is added, the Landau level index is no longer a conserved quantum number. In order to demonstrate this, let us define

$$H^{nonax} = R_k^{nonax}(|3\rangle\langle5| + |4\rangle\langle6|) + R_k^{nonax\dagger}(|5\rangle\langle3| + |6\rangle\langle4|) \tag{53}$$

to be the nonaxial part of the Hamiltonian, expressed in the orbital basis of Eq. (10) (with bold-face numbers indicating the orbitals). In the ladder operator formalism, $R_k^{\text{nonax}} \sim a^\dagger a^\dagger$, so that

$$
H^{\text{nonax}}|\Psi^{(n)}\rangle \sim [(|3\rangle\langle 5| + |4\rangle\langle 6|)a^\dagger a^\dagger + (|5\rangle\langle 3| + |6\rangle\langle 4|)aa]|\Psi^{(n)}\rangle \sim
\begin{pmatrix}
0 \\
0 \\
f_5^{(n)}(z)\sqrt{n+2}\sqrt{n+3}|n+3\rangle \\
f_6^{(n)}(z)\sqrt{n+2}\sqrt{n+3}|n+4\rangle \\
f_3^{(n)}(z)\sqrt{n-1}\sqrt{n-2}|n-3\rangle \\
f_4^{(n)}(z)\sqrt{n}\sqrt{n-1}|n-2\rangle \\
0 \\
0
\end{pmatrix},
$$
(54)

where $\sim$ indicates that we have suppressed the prefactors in the notation. Examining Eqs. (52) and (54), we observe that $\langle\Phi^{(n')}|H^{\text{nonax}}|\Psi^{(n)}\rangle$ is generally nonzero if $n' = n \pm 4$ and zero otherwise. Thus, the nonaxial terms couple Landau levels with indices differing by 4. In the basis of $|\Psi^{(n)}\rangle$, the total Hamiltonian (axial and nonaxial terms) has matrix elements between the $n$ and $n'$ blocks for $n' - n = -4, 0, 4$. We note that the Landau index modulo 4 remains conserved, because $\langle\Phi^{(n')}|H^{\text{nonax}}|\Psi^{(n)}\rangle = 0$ if $n' - n$ is not divisable by 4.

The block off-diagonal ($n' \neq n$) nonaxial terms are typically much weaker than dominant axial terms on the block diagonal ($n' = n$), so that the former can be viewed as perturbation to the latter. For this reason, basis of Landau level states is still useful even if the Landau index is not conserved. In typical Landau level spectra, the Landau index is often almost conserved, unless two levels of indices $n, n'$ with $n' - n = \pm 4$ come close in energy: In the latter case, hybridization between the states occurs and the spectrum exhibits an anticrossing.

In kdotpy, we always use the Landau level basis in the Landau level mode (kdotpy ll). In the axial approximation where the Hamiltonian matrix is block diagonal, the diagonalization can thus be performed block by block, which gives a performance bonus in view of the smaller matrix size. If nonaxial terms are considered, the Hamiltonian is written as one large matrix with all $|\Psi^{(n)}\rangle$ with $n = -2, -1, 0, \dots, n_{\max}$. This Hamiltonian is diagonalized as a whole. In kdotpy, we do not perform perturbation theory explicitly, but always diagonalize the full matrix. Due to the upper limit $n_{\max}$, the levels with $n = n_{\max}-3, n_{\max}-2, n_{\max}-1, n_{\max}$ are not coupled to their counterpart with $n' = n+4$; as a result, there is a slight inaccuracy in the energies of the highest levels. This numerical error can often be estimated by raising $n_{\max}$ and analyzing how well the energies have converged to definite value.

## 2.5 Other magnetic couplings

### 2.5.1 Zeeman effect

The electrons are subject to additional magnetic coupling, for example the Zeeman effect $H_Z = g\mu_B \mathbf{B}\cdot\mathbf{S}$, where $g$ is the gyromagnetic ratio or "$g$-factor" ($g \approx 2$ in vacuum), $\mu_B$ is the Bohr magneton, and $\mathbf{S}$ is the vector of spin operators ($\hat{S}_x, \hat{S}_y, \hat{S}_z$). The nonzero blocks (cf. Eq. (12)) of the Zeeman Hamiltonian $H_Z$ are

$$
H_{Z,66} = g_e\mu_B \begin{pmatrix} \frac{1}{2}B_z & \frac{1}{2}B_- \\ \frac{1}{2}B_+ & -\frac{1}{2}B_z \end{pmatrix}, \qquad
H_{Z,88} = 2\kappa\mu_B \begin{pmatrix} -\frac{3}{2}B_z & -\frac{1}{2}\sqrt{3}B_- & 0 & 0 \\ -\frac{1}{2}\sqrt{3}B_+ & -\frac{1}{2}B_z & -B_- & 0 \\ 0 & -B_+ & \frac{1}{2}B_z & -\frac{1}{2}\sqrt{3}B_- \\ 0 & 0 & -\frac{1}{2}\sqrt{3}B_+ & \frac{3}{2}B_z \end{pmatrix}
$$

$$
H_{Z,77} = 2(\kappa+\tfrac{1}{2})\mu_B \begin{pmatrix} -\frac{1}{2}B_z & -\frac{1}{2}B_- \\ -\frac{1}{2}B_+ & \frac{1}{2}B_z \end{pmatrix}, \quad
H_{Z,87} = 2(\kappa+1)\mu_B \begin{pmatrix} \sqrt{3/8}B_- & 0 \\ -\sqrt{1/2}B_z & \sqrt{1/8}B_- \\ -\sqrt{1/8}B_+ & -\sqrt{1/2}B_z \\ 0 & -\sqrt{3/8}B_+ \end{pmatrix}
$$
(55)

where $g_e$ is the $g$-factor for the $\Gamma_6$ block and $-2\kappa$ acts as the $g$-factor of the combined $\Gamma_8, \Gamma_7$ block [2, 3, 15].

| | $g_e$ | $\kappa$ | $N_0\alpha$ meV | $N_0\beta$ meV | $g_{\mathrm{Mn}}$ | $T_0$ K |
|---|---|---|---|---|---|---|
| HgTe | 2 | −0.4 | – | – | – | – |
| CdTe | 2 | −1.31 | – | – | – | – |
| $\mathrm{Hg_{1-y}Mn_yTe}$ | 2 | −0.4 | 400 | −600 | 2 | 2.6 |

Table 2: Coefficients of the magnetic couplings (Zeeman and paramagnetic exchange). The values of $\kappa$ also appear in Table 1. The values relevant for the paramagnetic exchange in $\mathrm{Hg_{1-y}Mn_yTe}$ are based on Ref. [3]. The value for $T_0$ is appropriate only for $y \sim 0.02$. (However, kdotpy implements it as this constant value, to remain consistent with past models.)

### 2.5.2 Exchange coupling in Mn-doped materials

In the dilute magnetic semiconductor $\mathrm{Hg_{1-y}Mn_yTe}$, the Mn atoms carry a finite magnetic moment, such that the material behaves paramagnetically if the Mn content $y$ does not exceed a few percent [24]. The coupling between the Mn magnetic moments and the carrier spins has a similar matrix structure as the Zeeman effect [Eq. (55)], but the response to the external magnetic field $\mathbf{B}$ is nonlinear. This coupling is modelled by the (paramagnetic) exchange Hamiltonian [3, 4, 24],

$$H_{\mathrm{ex}} = \sum_l C^{(l)} \langle \mathbf{m} \rangle \cdot \mathbf{S}^{(l)}, \tag{56}$$

where $l$ labels the blocks, (the $\Gamma_6$ block and the $\Gamma_8, \Gamma_7$ block), $\langle \mathbf{m} \rangle$ is the average Mn spin and $\mathbf{S}^{(l)}$ the spin operators of the respective block. The coupling constants $C^{(l)}$ (cf. $g_e$ and $2\kappa$ in Eq. (55)) are phenomenologically determined material parameters. For $\mathrm{Hg_{1-y}Mn_yTe}$, we assume that they are proportional to the Mn content $y$,

$$C^{\Gamma_6} = -y N_0\alpha \qquad \text{and} \qquad C^{\Gamma_8,\Gamma_7} = -y N_0\beta, \tag{57}$$

with $N_0\alpha = 400\,\mathrm{meV}$ and $N_0\beta = -600\,\mathrm{meV}$ [3, 24].

The paramagnetic response of the Mn magnetic moments to the external field is modelled by the empirical law [3, 24, 25]

$$\langle \mathbf{m} \rangle = -S_0 \frac{\mathbf{B}}{|\mathbf{B}|} B_{5/2}\left( \frac{\frac{5}{2} g_{\mathrm{Mn}} \mu_B |\mathbf{B}|}{k_B(T + T_0)} \right), \tag{58}$$

where $B_{5/2}$ is the Brillouin function for spin $J = \frac{5}{2}$,

$$B_J(x) = \frac{2J+1}{2J} \coth\left( \frac{2J+1}{2J} x \right) - \frac{1}{2J} \coth\left( \frac{1}{2J} x \right). \tag{59}$$

The effective total spin $S_0 = -\frac{5}{2}$ and the temperature offset $T_0$ are material parameters. For historical reasons, we have used the value $T_0 = 2.6\,\mathrm{K}$, but recent experiments have proved that $T_0$ depends on the Mn content $y$ and is typically smaller than this value, especially for smaller $y$ [26]. In kdotpy, the values $g_{\mathrm{Mn}}$ and $T_0$ are treated as material parameter and need not be constant in the Mn content, though the default material definitions for $\mathrm{Hg_{1-y}Mn_yTe}$ contain the $y$-independent values for historical reasons.

## 2.6 Bulk inversion asymmetry

The terms in Hamiltonian $H_{\mathrm{Kane}}$ [Eq. (14)] are symmetric under spatial inversion, i.e., the transformation given by $(x, y, z) \to (-x, -y, -z)$ and $(k_x, k_y, k_z) \to (-k_x, -k_y, -k_z)$. For $H_{\mathrm{Kane}}$, the relevant point group is $O_h$. However, the zincblende crystal structure is not symmetric under inversion; it has point group $T_d$. This means that terms breaking spatial inversion symmetry are allowed to appear in the Hamiltonian. Indeed, *bulk-inversion asymmetry* (BIA) is known to affect the band structure, although the inversion symmetric terms remain dominant. In other words, bulk-inversion asymmetry can be treated as a perturbation to the inversion symmetric Hamiltonian $H_{\mathrm{Kane}}$.

From representation theory of the point group $T_d$, it can be derived which BIA terms are permitted. Like before, we consider only those terms up to quadratic order in momentum. There is a single

|      | $C$ meV nm | $B_{8+}$ meV nm² | $B_{8-}$ meV nm² | $B_7$ meV nm² |
|------|-----------|------------------|------------------|---------------|
| HgTe | −7.4      | −106.46          | −13.77           | −100          |
| CdTe | −2.34     | −224.1           | −6.347           | −204.7        |

Table 3: Coefficients of the BIA Hamiltonian $H_{\text{BIA}}$ up to quadratic order in momentum. The values for CdTe are taken from Ref. [15], which in turn cites Ref. [27] for $C$. The values for $C$, $B_{8+}$ and $B_{8-}$ for HgTe are based on the result of a calculation by Di Sante and Sangiovanni based on density-functional theory (DFT) [28]. The value for $B_7$ for HgTe is an educated guess approximately equal to $B_{7,\text{CdTe}}B_{8+,\text{HgTe}}/B_{8+,\text{CdTe}}$. The values in this table are included as material parameters in kdotpy.

independent term linear in momentum, given by the Hamiltonian blocks [15]

$$
H_{\text{BIA},88} = C \begin{pmatrix} 0 & -\frac{1}{2}k_+ & k_z & -\frac{1}{2}\sqrt{3}k_- \\ -\frac{1}{2}k_- & 0 & \frac{1}{2}\sqrt{3}k_+ & -k_z \\ k_z & \frac{1}{2}\sqrt{3}k_- & 0 & -\frac{1}{2}k_+ \\ -\frac{1}{2}\sqrt{3}k_+ & -k_z & -\frac{1}{2}k_- & 0 \end{pmatrix}, \qquad H_{\text{BIA},87} = \frac{1}{2\sqrt{2}}C \begin{pmatrix} k_+ & 2k_z \\ 0 & -\sqrt{3}k_+ \\ \sqrt{3}k_- & 0 \\ 2k_z & -k_- \end{pmatrix},
$$

$$(60)$$

and $H_{\text{BIA},78} = H_{\text{BIA},87}^\dagger$. The coefficient $C$ is the material parameter for the strength of linear BIA. There are three independent quadratic terms, given by

$$
H_{\text{BIA},68} = \frac{1}{\sqrt{6}}B_{8+} \begin{pmatrix} \sqrt{3}k_-k_z & 2ik_xk_y & k_+k_z & 0 \\ 0 & k_-k_z & 2ik_xk_y & \sqrt{3}k_+k_z \end{pmatrix} + \frac{1}{3\sqrt{2}}B_{8-} \begin{pmatrix} 0 & \sqrt{3}K_4 & 0 & K_5 \\ -K_5 & 0 & -\sqrt{3}K_4 & 0 \end{pmatrix}
$$

$$
H_{\text{BIA},67} = \frac{1}{\sqrt{3}}B_7 \begin{pmatrix} -ik_xk_y & -k_+k_z \\ k_-k_z & ik_xk_y \end{pmatrix},
$$

$$(61)$$

where $K_4 \equiv k_x^2 - k_y^2$ and $K_5 \equiv k_x^2 + k_y^2 - 2k_z^2$, and the coefficients $B_{8+}$, $B_{8-}$, and $B_7$ are material parameters. The blocks $H_{\text{BIA},76}$ and $H_{\text{BIA},86}$ are given by the respective hermitian conjugates. The blocks $H_{66}$ and $H_{77}$ do not contain BIA terms up to quadratic order. Representative values of the material parameters for CdTe [15, 27] and HgTe [28] are listed in Table 3. These values are also included as material parameters in kdotpy.

For nonzero magnetic fields, the BIA terms also lead to extra contributions from the Peierls substitution. For a strip geometry, this leads to extra contributions involving the gauge field **A** (see Sec. 2.3.1). In the Landau level formalism (see Sec. 2.4), additional ladder operators appear. For growth direction (001), this leads to terms coupling Landau levels with indices $n, n'$ with $n' - n = \pm 2$. Whereas for the inversion symmetric LL Hamiltonian $n \mod 4$ is a conserved quantum number, the BIA lowers this symmetry such that only $n \mod 2$ remains conserved.

## 2.7 Strain

### 2.7.1 Strain and stress

The materials used in a heterostructure typically have slight differences in their equilibrium lattice constant. Two adjacent epitaxially grown layers can be strained as to match their in-plane lattice constants, if the mismatch between the equilibrium lattice constants is sufficiently small. This is known as pseudomorphic growth (see Ref. [29] and references therein). In practice, the in-plane lattice constant is generally determined by that of the substrate, which is usually much thicker than the epitaxial layers on top. If the epitaxial layers are too thick (more than a few hundred nm for HgTe and CdTe), relaxation occurs towards the equilibrium lattice constant.

The physical quantity *strain* is the deviation of the lattice constant of the strained layer $a_s$ relative to the equilibrium lattice constant $a_0$; in one dimension, strain is the dimensionless quantity

$$
\epsilon = \frac{a_s - a_0}{a_0}. \tag{62}
$$

In a three-dimensional crystal, strain leads to a displacement of each of the three lattice vectors $\mathbf{a}_i$ ($i = x, y, z$ in a crystal with cubic symmetry). The displacement can be written in terms of a tensor

|      | $C_1$ $10^3$ meV | $D_d$ $10^3$ meV | $D_u$ $10^3$ meV | $D_u'$ $10^3$ meV |
|------|------|------|------|------|
| HgTe | −3.83 | 0 | 2.25 | $\frac{1}{2}\sqrt{3} \times 2.08$ |
| CdTe | −4.06 | −0.7 | 1.755 | $\frac{1}{2}\sqrt{3} \times 3.2$ |

Table 4: Deformation potentials for HgTe and CdTe, see Refs. [2] and references therein. Note that some references use the alternative notation $C = C_1$, $a = D_d$, $b = -\frac{2}{3}D_u$ and $d = -\frac{2}{\sqrt{3}}D_u'$ [2,31]. The values in this table are included as material parameters in kdotpy.

$A_{ij}$, which satisfies [30]

$$\mathbf{a}_{i,s} = \sum_{j=1}^{3}(\delta_{ij} + A_{ij})\mathbf{a}_{j,0}, \tag{63}$$

where $\delta_{ij}$ is the Kronecker delta. The non-rotational (i.e., symmetrical) part of $A_{ij}$ is the strain tensor

$$\epsilon_{ij} = \frac{1}{2}(A_{ij} + A_{ji}). \tag{64}$$

The strain tensor $\epsilon_{ij}$ is symmetric by definition. The diagonal terms $\epsilon_{ii}$ represent *linear strain* and the off-diagonal terms $\epsilon_{xy}, \epsilon_{yz}, \epsilon_{zx}$ represent *shear strain*.

A strained crystal is not in equilibrium and thus experiences forces that pushes the crystal back to its equilibrium lattice constant, or conversely, external forces are needed to bring a crystal into a strained state. Here, we consider the linear response regime, where Hooke's law is valid, i.e., the force $F$ and the displacement $u$ are linearly proportional, $F = ku$, with the stiffness constant $k$. The generalization of Hooke's law to a crystal in three dimensions relates the *stress* tensor $\sigma_{ij}$ to the strain tensor $\epsilon_{kl}$ as

$$\sigma_{ij} = \sum_{k,l=1}^{3} S_{ijkl}\epsilon kl, \tag{65}$$

in terms of the stiffness tensor $C_{ijkl}$. The stiffness tensor is a rank-4 tensor with $81 = 3 \times 3 \times 3 \times 3$ components (also known as elasticity modules). The stress tensor and stiffness tensor (elasticity modules) both carry the units of pressure, typically **GPa** in this context.

For a crystal with cubic symmetry, the stiffness tensor has only **3** independent components [30]. If the strain and stress tensors are vectorized as $\tilde{\epsilon} = (\epsilon_{xx}, \epsilon_{yy}, \epsilon_{zz}, 2\epsilon_{yz}, 2\epsilon_{zx}, 2\epsilon_{xy})$ and $\tilde{\sigma} = (\sigma_{xx}, \sigma_{yy}, \sigma_{zz}, \sigma_{yz}, \sigma_{zx}, \sigma_{xy})$, respectively, Eq. (65) can be written as $\tilde{\sigma} = \tilde{C}\tilde{\epsilon}$ in terms of the elasticity matrix

$$\tilde{C} = \begin{pmatrix} C_{11} & C_{12} & C_{12} & 0 & 0 & 0 \\ C_{12} & C_{11} & C_{12} & 0 & 0 & 0 \\ C_{12} & C_{12} & C_{11} & 0 & 0 & 0 \\ 0 & 0 & 0 & C_{44} & 0 & 0 \\ 0 & 0 & 0 & 0 & C_{44} & 0 \\ 0 & 0 & 0 & 0 & 0 & C_{44} \end{pmatrix}. \tag{66}$$

The three independent components $C_{11}$, $C_{12}$, and $C_{44}$ are material parameters that can be found in the literature.

### 2.7.2  Strain Hamiltonian

The effect of strain on the band structure is formalized in the Bir-Pikus formalism [31]: The strain Hamiltonian is obtained from the $\mathbf{k} \cdot \mathbf{p}$ Hamiltonian by substitution of all terms quadratic in momentum as

$$k_i k_j \rightarrow \epsilon_{ij} \tag{67}$$

and changing the band mass parameters $F$, $\gamma_{1,2,3}$ to the *deformation potentials* (i.e., strain coefficients) $C_1$, $D_d$, $D_u$, and $D'_u$ [2,3,15]. Taking Eq. (15) as a starting point, we find the strain Hamiltonian

$$
H_s = \begin{pmatrix}
T_s & 0 & 0 & 0 & 0 & 0 & 0 & 0 \\
0 & T_s & 0 & 0 & 0 & 0 & 0 & 0 \\
0 & 0 & U_s + V_s & S_s & R_s & 0 & -\frac{1}{\sqrt{2}}S_s & -\sqrt{2}R_s \\
0 & 0 & S_s^* & U_s - V_s & 0 & R_s & \sqrt{2}V_s & \sqrt{\frac{3}{2}}S_s \\
0 & 0 & R_s^* & 0 & U_s - V_s & -S_s & \sqrt{\frac{3}{2}}S_s^* & -\sqrt{2}V_s \\
0 & 0 & 0 & R_s^* & -S_s^* & U_s + V_s & \sqrt{2}R_s^* & -\frac{1}{\sqrt{2}}S_s^* \\
0 & 0 & -\frac{1}{\sqrt{2}}S_s^* & \sqrt{2}V_s & \sqrt{\frac{3}{2}}S_s & \sqrt{2}R_s & U_s & 0 \\
0 & 0 & -\sqrt{2}R_s^* & \sqrt{\frac{3}{2}}S_s^* & -\sqrt{2}V_s & -\frac{1}{\sqrt{2}}S_s & 0 & U_s
\end{pmatrix}
\tag{68}
$$

with

$$
T_s = C_1 \operatorname{tr} \epsilon
$$

$$
U_s = D_d \operatorname{tr} \epsilon, \qquad\qquad\qquad R_s = \frac{1}{\sqrt{3}}D_u(\epsilon_{xx} - \epsilon_{yy}) - \frac{2}{\sqrt{3}}D'_u i\epsilon_{xy}, \tag{69}
$$

$$
V_s = -\frac{1}{3}D_u(\epsilon_{xx} + \epsilon_{yy} - 2\epsilon_{zz}), \qquad S_s = \frac{2}{\sqrt{3}}D'_u(\epsilon_{xz} - i\epsilon_{yz}),
$$

where $\operatorname{tr}\epsilon = \epsilon_{xx} + \epsilon_{yy} + \epsilon_{zz}$. The values of the deformation potentials $C_1$, $D_d$, $D_u$, and $D'_u$ for HgTe and CdTe that have been provided with kdotpy, are listed in Table 4 and have been taken from Ref. [2]. We note that this and other literature sources (e.g., Ref. [31]) use the alternative set of deformation potentials $C = C_1$, $a = D_d$, $b = -\frac{2}{3}D_u$ and $d = -\frac{2}{\sqrt{3}}D'_u$.

Symmetry allows further terms in the strain Hamiltonian. For example, application of the Bir-Pikus substitution to cubic momentum terms leads to strain terms linear in strain and momentum, i.e., involving linear combinations of $\epsilon_{ij}k_l$. (See the tables in Ref. [15] for examples of these terms.) Here, we neglect these 'higher-order' terms, as they are negligible for these materials [2]. For a systematic study of these higher-order strain terms, we refer to Ref. [32].

## 2.8  Topology

### 2.8.1  Berry connection, Berry curvature, Chern number

The topological character of a material is typically understood theoretically in terms of the Chern number of the bands. In a nutshell, band inversion can cause the Chern numbers to become non-trivial, which leads to measurable signatures in the Hall conductance.

A common picture to understand the concept of a Chern number is the analogy with a winding number of the spin of a spin-$\frac{1}{2}$ particle [33]. The spin eigenstate $|\psi(\mathbf{k})\rangle$ can be represented by a momentum dependent vector $\mathbf{d}(\mathbf{k})$ on the Bloch sphere. Then one can consider the area swept out by $\mathbf{d}(\mathbf{k})$ where the momenta $\mathbf{k}$ span the complete Brillouin zone. The signed area is $4\pi C$, i.e., the area $4\pi$ of the unit sphere times the Chern number $C$. If the Brillouin zone is finite, the Chern number is integer, whereas for the continuum limit of infinitely large Brillouin zones, it may also be half-integer.

The above picture can be formalized in terms of concepts from differential geometry. If we consider an eigenstate $|\psi(\mathbf{k})\rangle$ that is differentiable in the Brillouin zone, we can define the Berry connection [33]

$$
\mathbf{A}^{(\psi)}(\mathbf{k}) = i\langle\psi(\mathbf{k})|\nabla_{\mathbf{k}}|\psi(\mathbf{k})\rangle \tag{70}
$$

at each point $\mathbf{k}$. This quantity is analogous to the magnetic vector potential, hence it also goes by the name Berry vector potential [33]. The Berry phase is an integral of the Berry curvature over the closed contour $\mathcal{C}$ that surrounds the Brillouin zone,

$$
\gamma^{(\psi)} = \int_{\mathcal{C}} d\mathbf{k} \cdot \mathbf{A}^{(\psi)}(\mathbf{k}). \tag{71}
$$

The Berry phase is an integer number times $2\pi$, which defines the Chern number $C^{(\psi)}$ as $\gamma^{(\psi)} = 2\pi C^{(\psi)}$ [33].

The integral of Eq. (71) can be rewritten by virtue of Stokes' theorem as an integral over the interior $\mathcal{S}$ of the Brillouin zone,

$$\gamma^{(\psi)} = \int_{\mathcal{S}} d\mathbf{S} \cdot \nabla_\mathbf{k} \times \mathbf{A}^{(\psi)}(\mathbf{k}) \equiv \int_{\mathcal{S}} d\mathbf{S} \cdot \mathbf{\Omega}^{(\psi)}(\mathbf{k}), \tag{72}$$

where $d\mathbf{S}$ is a surface element and $\mathbf{\Omega}^{(\psi)}(\mathbf{k})$ is the Berry curvature, defined by $\mathbf{\Omega}^{(\psi)}(\mathbf{k}) = \nabla_\mathbf{k} \times \mathbf{A}^{(\psi)}$, or in component notation ($i, l, m = 1, 2, 3$),

$$\Omega_i^{(\psi)}(\mathbf{k}) = \sum_{l,m} \epsilon_{ilm} F_{lm}^{(\psi)}(\mathbf{k}) \tag{73}$$

with

$$F_{lm}^{(\psi)}(\mathbf{k}) = \partial_{k_l} A_m^{(\psi)}(\mathbf{k}) - \partial_{k_m} A_l^{(\psi)}(\mathbf{k}) = i\langle \partial_{k_l}\psi(\mathbf{k})|\partial_{k_m}\psi(\mathbf{k})\rangle - i\langle \partial_{k_m}\psi(\mathbf{k})|\partial_{k_l}\psi(\mathbf{k})\rangle. \tag{74}$$

The analogue of the Berry curvature in electrodynamics is the magnetic field $\mathbf{B} = \nabla \times \mathbf{A}$.

In practice, the Berry connection is challenging to calculate numerically: The derivative is discretized, where we evaluate the eigenstates at $\mathbf{k}$ and $\mathbf{k} + d\mathbf{k}$. However, the result of the numerical evaluation has an indefinite overall phase factor ($U(1)$ gauge factor) which can vary randomly even between two points $\mathbf{k}$ and $\mathbf{k} + d\mathbf{k}$ close in momentum space. In Ref. [33], the Berry curvature $\Omega_i^{(\psi)}(\mathbf{k})$ [Eqs. (73) and (74)] is rewritten by 'insertion of one', $\sum_{\psi'} |\psi'\rangle\langle\psi'|$. This manipulation eventually yields a gauge invariant form of the Berry curvature

$$\Omega_i^{(\psi)}(\mathbf{k}) = -\mathrm{Im} \sum_{\psi' \neq \psi} \sum_{l,m} \frac{\epsilon_{ilm} \langle \psi'(\mathbf{k})|(\partial_{k_l} H)|\psi(\mathbf{k})\rangle \langle \psi(\mathbf{k})|(\partial_{k_m} H)|\psi'(\mathbf{k})\rangle}{(E_{\psi'} - E_\psi)^2}, \tag{75}$$

where the summation is over all states in the spectrum other than $|\psi\rangle$ itself, and $E_\psi$ and $E_{\psi'}$ are the energy eigenvalues. This expression is manifestly $U(1)$ invariant, as all phase factors from the eigenstates come in conjugate pairs. The momentum derivatives of the Hamiltonian $\partial_{k_l} H$ ($l = 1, 2, 3$) can be evaluated analytically or numerically; even for the numerical derivative, there are no indefinite phase factors to be taken care of.

### 2.8.2   Hall conductance

The bulk-boundary theorem (see, e.g., Ref. [33]) connects the Chern numbers to the Hall conductance $\sigma_\mathrm{H} = \sigma_{xy}$ of a device in a magnetic field, as

$$\sigma_\mathrm{H} = \frac{e^2}{h} \sum_{\text{occupied states } \psi} C^{(\psi)}. \tag{76}$$

That is, the sum of the Chern numbers of the occupied bands yields the Hall conductance in units of the conductance quantum $e^2/h$. This result can be used to find the Hall conductance $\sigma_\mathrm{H}$ in a Landau fan: For each additional occupied Landau level, the Hall conductance is increased by $C^{(\psi)} e^2/h$. Typically, $C^{(\psi)} = 1$ for all Landau levels, so that finding the Hall conductance simplifies to a mere counting problem of occupied states. In kdotpy, the Hall conductance can be determined either with Chern numbers calculated with Eq. (75) or with simulated Chern numbers based on the assumption that $C^{(\psi)} = 1$. The implementation is discussed in more detail in Sec. 3.5.3.

# 3 Implementation

## 3.1 Overview of the program

### 3.1.1 General remarks

Like its name suggests, `kdotpy` implements $\mathbf{k} \cdot \mathbf{p}$ theory in a Python program. The program is designed as a command-line interface (CLI): The operation of the program is determined primarily by a sequence of command line arguments entered by the user on the shell (e.g., bash). Moreover, the behaviour and output can be adjusted with configuration settings.

### 3.1.2 Package structure

The `kdotpy` application is actually a collection of several subprograms, that do the actual work. Most of these subprograms are Python scripts themselves. The subprograms are referred to by the first two arguments on the command line, for example `kdotpy 2d`. The command `kdotpy` invokes the main script `main.py` (provided that the `kdotpy` module has been installed successfully with PIP, see Section 4). In this example, `main.py` imports `kdotpy-2d.py` and runs its `main()` function. The latter contains the 'recipe' of this subprogram, which we will discuss in greater detail in Sec. 3.1.3.

The `kdotpy` module defines five subprograms that we classify as *calculation subprograms*, because they do the actual computational work: constructing a Hamiltonian, diagonalizing it, and processing the eigenvalues and eigenstates.

- `kdotpy 2d`: Calculates a dispersion of a structure with two translational degrees of freedom. The $z$ direction is kept spatial and is appropriately discretized. A typical configuration for this case is a quantum well.

- `kdotpy 1d`: Calculates a dispersion of a structure with one translational degree of freedom, for example a Hall bar. The $y$ and $z$ direction are kept spatial and are discretized. Due to the very large size of the problem, it usually makes sense to run this subprogram on a cluster that supports 'jobs' with large memory requirements.

- `kdotpy ll`: Calculates a Landau level spectrum for a configuration similar to the '2D' mode, using the Landau level formalism of Sec. 2.4.

- `kdotpy bulk`: Calculates the dispersion for the 'bulk', i.e., with translational degrees of freedom in all **3** directions.

- `kdotpy bulk-ll`: Calculates a Landau level spectrum for a geometry with translational symmetry in the $z$ direction.

The package also contains several auxiliary subprograms for several other tasks other than doing the actual calculations:

- `kdotpy merge`: This subprogram is used for re-plotting data from earlier runs of `kdotpy`. It takes as input one or more XML files. The name `kdotpy merge` derives from the fact that it can be used to merge several data files, and show all results in a single plot.

- `kdotpy compare`: Similar to `kdotpy merge`, but shows data files (or sets of data files) using different colours/symbols in a single plot.

- `kdotpy batch`: A simple tool for running batch calculations with `kdotpy`, for example iterating a calculation while varying one of the input parameters. This subprogram serves a similar purpose as a shell script, but has a convenient monitor that shows the estimated time at which the calculations will finish.

- `kdotpy test`: Test suite. This tool runs all standardized tests or a selection of them, as to test whether `kdotpy` runs properly.

- `kdotpy config`: For viewing and modifying the configuration settings.

- `kdotpy help`: Show the built-in help file (with command line and configuration option reference) in a terminal viewer. With `kdotpy help item`, can search for a specific `item` in the help file.

- `kdotpy doc`: With `kdotpy doc object`, one gets developer information on the given `object`, e.g., a function or class. This tool shows the relevant docstring on the terminal.

- `kdotpy version`: Show the version of `kdotpy`.

The final three are not separate Python scripts, but functions called directly from the main program. For detailed instructions on how to use `kdotpy` from the command line, we refer to Sec. 5.

The Python code is structured as follows. The root directory of contains the metadata, such as the `pyproject.toml` file, the `README.md` file and the license text. The code itself is found in the subdirectory `src/kdotpy`. This directory contains the main module (in `main.py`), the scripts for the subprograms, and several other source files that are imported by the subprograms. Inside `src/kdotpy`, the following subdirectories collect several larger components of the program:

- `bandalign`: For band alignment, see Sec. 3.6.

- `cmdargs`: Functions that parse the command line arguments, see Sec. 3.2.1.

- `density`: Functions for calculating carrier density, density of states, etc., see Sec. 3.7.

- `diagonalization`: The infrastructure for diagonalization of the Hamiltonians, see Sec. 3.4.

- `hamiltonian`: For construction of the Hamiltonians, see Sec. 3.3.

- `materials`: For parsing the files that contain the material parameters, see Sec. 3.2.4.

- `ploto`: For plot output, see Sec. 3.10.4.

- `tableo`: For table output, see Sec. 3.10.3.

- `xmlio`: For XML input and output, see Sec. 3.10.2.

The `materials` subdirectory also contains the default materials file. Finally, the built-in helpfile is located at `docs/helpfile.txt` inside `src/kdotpy`. The components will be discussed below in the remainder of Sec. 3, roughly in the order in which they are executed in a `kdotpy` calculation.

### 3.1.3   Calculation subprogram workflow

The five calculation subprograms differ substantially in the details, but they all follow a similar workflow, going through the same sequence of stages. Figure 1 illustrates a schematic flow diagram with the important stages. The sequence can be summarized as follows.

- Preprocessing:

  - Parse command line arguments, read configuration values and material parameters
  - Define electrostatic potential (read from file and/or calculated with the self-consistent Hartree method)
  - Prepare for diagonalization (determine band characters, charge neutrality point (CNP))

- Diagonalization (iteration over $k$ or $B$ points):

  - Construct Hamiltonian
  - Diagonalize using *diagsolver* (e.g., `eigsh`)
  - Calculate observables, transitions, Berry curvature

- Band alignment

- Postprocessing and output:

  - Optional extras: Extrema, DOS, BHZ, wave functions
  - Output to files: `csv`, `pdf`, `xml`, `hdf5`, etc.

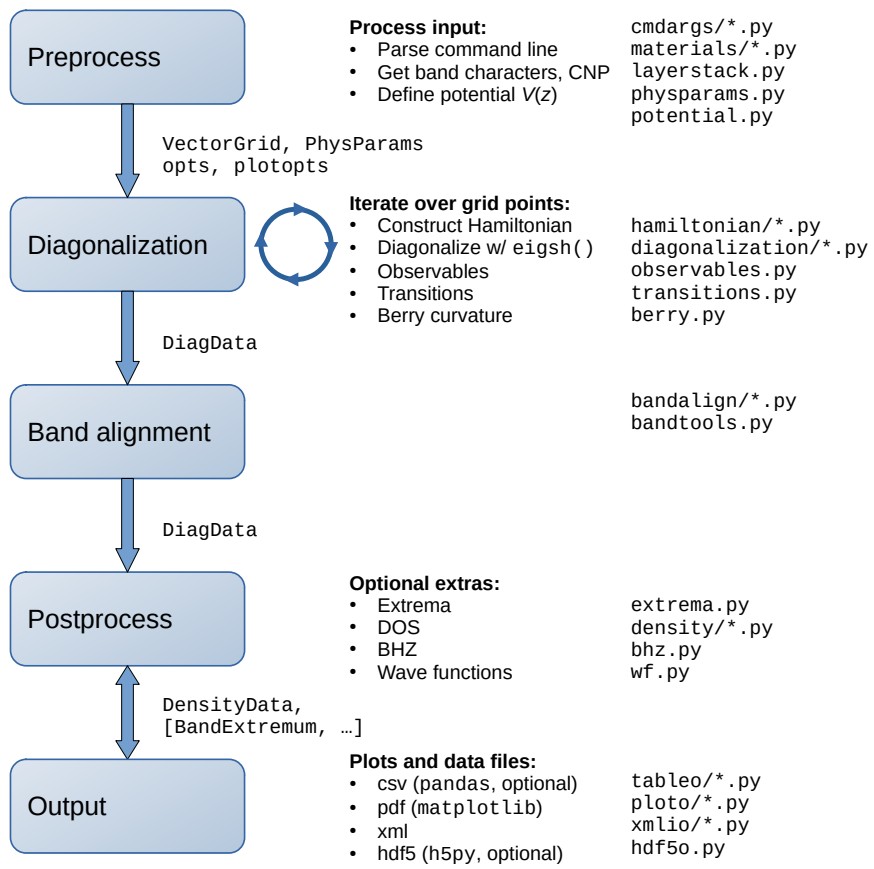

Figure 1: Flow diagram of `kdotpy`, that illustrates the workflow for the calculation sub-programs `kdotpy 1d`, `kdotpy 2d`, `kdotpy bulk`, `kdotpy ll`, and `kdotpy bulk-ll`. The diagram shows the most important components of the program and some relevant data classes and source files (not a complete list).

### 3.1.4 Data structures and data flow

The different components of the program 'communicate' via classes which contain the important data. In the following list, we provide the most important ones:

- `VectorGrid`: The class `VectorGrid` contains the $k$ and/or $B$ values on which the diagonalization is performed. The important class attributes are the vector type (`vtype`) and one, two or three arrays of values. The class represents a sequence of vector values (if the dimensionality is one) or a cartesian product of two or three sequences of values.

- `PhysParams`: The class `PhysParams` contains the physical parameters and coefficients related to the simulated structure, such as its size, the resolution of the spatial coordinates, material parameters, and strain values.

- `DiagDataPoint`: Each instance of `DiagDataPoint` contains the eigenvalues and (optionally) the eigenvectors at a single value $(k, B)$. The eigenvectors are calculated and stored temporarily, and usually deleted after the observables have been calculated in order to reduce the memory requirement. The observables are stored in the `DiagDataPoint` instance, together with other eigenstate data such as Landau level index, band index, and subband character. The `DiagDataPoint` class defines several member functions for retrieving and manipulating the eigenstate data.

- `DiagData`: The class `DiagData` contains the result of the diagonalization as an array of `DiagDataPoint` instances. Optionally, it also contains a `VectorGrid` instance to give a multidimensional view of the data. (The data is stored as a flat list of `DiagDataPoint` instances.) The `DiagData` class contains several member functions for retrieving eigenstate data for the complete data set, for example for the output functions. Importantly, after successful band alignment, these member functions can access the eigenstates 'by band', e.g., the eigenvalues of all eigenstates with the same band index. This is essential for making the plots and for some of the postprocessing functions.

- `DensityData`: The result of the calculation of the density of states is stored in a `DensityData` instance. The key quantities inside this class are the integrated density of states (IDOS) and the corresponding energy values. The class can store an IDOS integrated over all $k$ as well as separate values per $k$ or $B$. In the latter case, also a grid of $k$ or $B$ values is stored. The member functions allow retrieval of the IDOS and the ordinary density of states (as energy derivative of the IDOS), where the values can be scaled to the desired set of units (inverse units of length, in terms of nm, cm, or m). Broadening can also be applied.

Option values taken from command-line input are stored in the `dict` instances `opts`, `plotopts`, `modelopts`, etc.

## 3.2 Input and preprocessing

### 3.2.1 Command line parsing

The kdotpy program is designed as a non-interactive command-line interface (CLI). The behaviour of the program is determined by the string of arguments provided by the user in the shell command that starts kdotpy. The first argument after the program name kdotpy determines the subprogram that is used, for example

```
kdotpy 2d 8o noax msubst CdZnTe 4% mlayer HgCdTe 68% HgTe HgCdTe 68%
llayer 10 7 10 zres 0.25 k -0.6 0.6 / 120 kphi 45 split 0.01
```

runs the script `kdotpy-2d.py` with further arguments being passed to this script. (For a detailed explanation of all arguments in this example, please refer to the Usage Example of Sec. 5.2.) Unlike many other programs, the option arguments are not preceded by hyphen or double-hyphen, for ease of input. The argument list `sys.argv` passed to the script is wrapped in the instance `sysargv` of the `CmdArgs` class, which keeps track of all arguments that have been parsed. This allows kdotpy to track if there are any unparsed arguments; if that is the case, a warning message is shown at the end.

The source file `cmdargs/cmdargs.py` provides several wrapper functions for extracting information from the command line arguments. For example, the functions `cmdargs.options()`, `cmdargs.plot_options()`, and `cmdargs.bandalign()` return `dict` instances with option values used by several other functions in `kdotpy`. Some features are enabled by simply putting the appropriate argument in the command line, e.g., `extrema` in the above example for doing the extrema analysis. This is simply tested by `"extrema" in sysargv`; the `CmdArgs` class automatically marks the argument as parsed.

We have provided a comprehensive list of all command line options in Appendix C, sorted thematically. For up-to-date information on commands (for versions other than the present one, v1.0.0), we recommend to consult the wiki [34] and/or the built-in help.

### 3.2.2 Configuration options

For user preferences that are unlikely to change between different calculations, `kdotpy` uses a system of configuration settings. A comprehensive reference for the configuration options is provided as Appendix B.7. The user may also consult the wiki [34] and/or built-in help for up-to-date information.

The configuration settings are stored in the file `~/.kdotpy/kdotpyrc`, which is read and parsed at the start of the program. This file contains `key=value` pairs for all known configuration options defined in `config.py`. If any configuration option is missing from `~/.kdotpy/kdotpyrc`, the `key=value` pair is added as a comment (i.e., preceded by `#`) with the default value. Using this system, the configuration file remains usable if new features are added to `kdotpy` and the new configuration options will become visible to the user.

It is also possible to use a custom configuration file by using `config filename` as a command line argument. Multiple files may be loaded in sequence, where the files are processed from left to right, and configuration values loaded later override those loaded earlier. For flexibility, it is also possible to adjust configuration values on the command line by providing `config` followed by semicolon-separated `key=value` pairs, like

```
kdotpy 2d ... config 'key1=value1;key2=value2' ...
```

where the single quotes are required by the shell. Both variants of the `config` argument may be combined in any order.

The configuration values are typically parsed *ad hoc*: they are evaluated at the moment they are needed. The configuration values are not checked for validity at the start of the program. Thus, error messages related to invalid configuration values may occur at any point during runtime.

An auxiliary subprogram `kdotpy config` is provided in order to read and manipulate settings easily. It also gives access to the help file which contains detailed information on every configuration option. The command-line syntax for the configuration tool `kdotpy config` is summarized in Sec 5.7.4.

### 3.2.3 Vector grids

The array of momenta $\mathbf{k}$ or magnetic fields $\mathbf{B}$ where the Hamiltonian is evaluated is called the 'grid'. In general, the command line parser yields a `ZippedKB` object, which contains the values of $\mathbf{k}$ and $\mathbf{B}$. This object may represent a dispersion ($\mathbf{k}$ is a `VectorGrid` instance, $\mathbf{B}$ is single valued), a magnetic-field dependence ($\mathbf{k}$ is single valued, $\mathbf{B}$ is a `VectorGrid` instance), or a single point (when both are single valued). A combination of $\mathbf{k}$- and $\mathbf{B}$-dependence is not permitted.

The `VectorGrid` class stores a multi-dimensional array of $\mathbf{k}$ or $\mathbf{B}$ values as a cartesian product of values. It may have any one of the following coordinate types:

- `x`, `y`, `z`: Values along one of the cartesian coordinate axes.

- `xy`: Cartesian coordinates $(x, y)$ in the $z = 0$ plane.

- `xyz`: Cartesian coordinates $(x, y, z)$.

- `pol`: Polar coordinates $(r, \phi)$, that convert to cartesian coordinates as $(x, y) = r(\cos\phi, \sin\phi)$ with $z = 0$.

- `cyl`: Cylindrical coordinates $(r, \phi, z)$, that convert to cartesian coordinates as $(x, y, z) = (r\cos\phi, r\sin\phi, z)$.

- sph: Spherical coordinates $(r, \theta, \phi)$, where $\theta$ is the polar angle in $[0, \pi] = [0, 180°]$ and $\phi$ the azimuthal angle. The relation to cartesian coordinates is given by $(x, y, z) = (r \sin \theta \cos \phi, r \sin \theta \sin \phi, r \cos \theta)$.

In `kdotpy`, the radial coordinate $r$ of the angular coordinate systems may be chosen negative, in contrast to the usual condition that $r \geq 0$.

The input of the grid on the command line is done by combination of the vector components on the command line, followed by a single value or a range, for example

```
kdotpy 2d ... k 0 0.2 / 20 kphi 45 bz 0.1 ...
```

yields the momentum values given by the polar coordinates $(k_i, 45°)$ with $k_i = \{0, 0.01, \ldots, 0.2\}\,\mathrm{nm}^{-1}$ in a magnetic field $\mathbf{B} = (0, 0, B_z)$ with $B_z = 0.1\,\mathrm{T}$. The valid component arguments for momenta, are k, kx, ky, kz, kphi, and ktheta; for magnetic fields, they are b, bx, by, bz, bphi, and btheta. The arguments k and b without explicit component refer to the radial direction if used in combination with angular coordinates. If they are used by themselves, k is equivalent to kx and b is equivalent to bz, i.e., the 'natural' directions for momentum and magnetic field, respectively. The values are in units of $\mathrm{nm}^{-1}$ and $T$ for momentum and magnetic field, respectively, and degrees for the angular components.

Values and ranges are input as follows:

- k a: Single value $a$.

- k a * b: Single value $a\,b$, where $a$ is an integer and $b$ is a floating-point value.

- k a b / c: Values $a$ and $b$ are the minimum and maximum of the range, respectively. If $c$ is a floating point number, it denotes the step size $\Delta = c$. If $c$ is an integer, the step size is $\Delta = (b - a)/c$. The resulting values are $\{a, a + \Delta, a + 2\Delta, \ldots, b\}$. Note that if $c$ is an integer, the number of values in this range is $c + 1$.

- k b / c: Equivalent to k 0 b / c.

- k a b c / d: The single momentum value $a + (b-a)c/d$, i.e., the $c$'th element from the range given by k a b / d. The values $c$ and $d$ must be integers.

Here, k can be replaced by any of the momentum and magnetic field components listed above. Also, a // (double-slash) operator can be used instead of /. In this case, quadratic stepping is used, which is useful in particular for magnetic fields: bz a b // c (where $c$ is an integer) yields the values $B_z = \{a, a + \Delta, a + 4\Delta, a + 9\Delta, \ldots, b\}\,\mathrm{T}$ where $\Delta = (b-a)/c^2$. The number of values is $c + 1$ like with the single slash /. The typical use case of // is with the perpendicular magnetic field in Landau-level mode, with $a = 0$.

The dimensionality of the grid is determined automatically by `kdotpy` based on the input of single values and ranges. For momenta, it is always smaller than or equal to the geometric dimension (i.e., the number of momentum components). For magnetic fields, multi-dimensional grids are not supported; the maximum dimension of the grid is 1.

### 3.2.4 Material parameters

Many of the coefficients of the Hamiltonian are material dependent. The values of these parameters are defined in separate input files. The program provides a default materials file (`materials/default`) with a set of parameters for HgTe, CdTe, (Hg,Cd)Te, (Hg,Mn)Te, and (Cd,Zn)Te. The user may override these materials and define new ones by providing custom material definitions in their own files in the directory `~/.kdotpy/materials`.

The materials files are formatted similar to the `.ini` format and are parsed by Python's `configparser` module. The 'sections' in the file serve as the material labels; for example the definitions for HgTe are preceded by the section head `[HgTe]`. The section contains several `param = value` pairs. The parser simply reads this data and puts them as keys and values of a `dict` instance. Comments (everything that follows #) are ignored. In addition, the user may also adjust material parameters on the command line with the argument `matparam` followed by several key-value pairs.

The 'left-hand size' `param` may be any of the recognized material parameters, as listed in Appendix B.2, or a user defined 'auxiliary' parameter which is used in the value of some other parameter. The 'right-hand size' `value` must represent a number or a valid Python expression where a restricted set of functions and operators may be used:

- The arithmetic operators +, -, *, /, **. Exponentiation is represented by **, not ^.

- The kdotpy defined polynomial functions linint, linearpoly, quadrpoly, cubicpoly, and poly. These represent the functions $x \mapsto a(1-x) + bx$, $x \mapsto c_0 + c_1 x$, $x \mapsto c_0 + c_1 x + c_2 x^2$, $x \mapsto c_0 + c_1 x + c_2 x^2 + c_3 x^3$, and $x \mapsto c_0 + c_1 x + \ldots + c_n x^n$, respectively.

- The comparison functions geq, leq, gtr, and less, which implement the binary comparison operators $\geq$, $\leq$, $>$, and $<$, respectively.

- Functions of the Python math module, like sqrt and log.

- The mathematical constants pi and e

- The physical constants defined in physconst.py, see Appendix B.1.

The value may also contain any other predefined or auxiliary material parameter. If the expression is not valid Python syntax or if any 'forbidden' keyword is used (e.g., import), kdotpy will raise an exception. The restricted function and variable labels also cannot be used as param on the left-hand side of =. (The following are also not permitted as param: All Python keywords except as, the numerical constants inf and nan, and the reserved variables x, y, z, and T.) The following examples are an excerpts from the default definitions for HgTe and $\mathrm{Hg}_{1-x}\mathrm{Cd}_x\mathrm{Te}$,

```
## HgTe, mercury telluride
[HgTe]
compound    = HgTe
composition = 1, 1
P           = sqrt(18800. * hbarm0)
Ev          = 0.0
Ec          = -303.0 + 0.495 * T ** 2 / (11.0 + T)  # Eg of HgCdTe for x = 0
gamma1      = 4.1
gamma2      = 0.5
gamma3      = 1.3

## HgCdTe (Hg_{1-x} Cd_x Te), mercury cadmium telluride
[HgCdTe]
compound    = HgCdTe
linearmix   = HgTe,CdTe,x
composition = 1 - x, x, 1  # is also set automatically by linearmix
P           = sqrt(18800. * hbarm0)
Eg          = -303 * (1 - x) + 1606 * x - 132. * x * (1 - x) + \
              (0.495 * (1 - x) - 0.325 * x - 0.393 * x * (1 - x)) * T ** 2 / \
              (11.0 * (1 - x) + 78.7 * x + T)  # identical to Ref. [HgCdTe2]
Eg0         = -303.0 + 0.495 * T ** 2 / (11.0 + T)
Evoff       = -570. * (Eg - Eg0) / (1606 - -303)
Ev          = Evoff
Ec          = Evoff + Eg
gamma1      = poly( 4.1, -2.8801,  0.3159, -0.0658, x)
gamma2      = poly( 0.5, -0.7175, -0.0790,  0.0165, x)
gamma3      = poly( 1.3, -1.3325,  0.0790, -0.0165, x)
```

The second example for $\mathrm{Hg}_{1-x}\mathrm{Cd}_x\mathrm{Te}$ also illustrates the 'special parameter' linearmix, which creates a third material as linear combination of two other materials. Here, linearmix = HgTe,CdTe,x indicates that all material parameters are interpolated as $p_{\mathrm{HgCdTe}} = (1-x) p_{\mathrm{HgTe}} + x p_{\mathrm{CdTe}}$. The subsequent definitions (like gamma1 in this example) override the linearly interpolated ones.

The expressions on the right-hand side are parsed by an abstract syntax tree (AST) parser based on Python's ast module, so that it accepts the 'whitelisted' functions and operators only. If an expression value evaluates to a finite value (not $\pm\infty$, not nan), the parser substitutes that value, otherwise it leaves the expression unevaluated. The latter is common when a material parameter depends on the composition (e.g., $x$ in $\mathrm{Hg}_{1-x}\mathrm{Cd}_x\mathrm{Te}$). This is acceptable until the layer stack is built; at that point, all material parameters must evaluate to numerical values.

Using the built-in and custom defined materials is done by using the material label (the section label in the materials file, e.g., [HgTe], without the brackets), possibly followed by up to three numerical values. For example, HgTe simply yields the material HgTe and HgCdTe 68% yields the material HgCdTe with the substitution $x \rightarrow 0.68$. As the material label can be any alphanumerical string starting with a letter (numbers, hyphens and underscores are also allowed at following positions), it is not needed to use molecular formulas as labels. For example, the label mercury_telluride would be acceptable too. This also gives the freedom to define multiple sets of parameters for one material, e.g., using labels CdTe_Novik and CdTe_Weiler for CdTe parameters based on the different literature sources (Refs. [3] and [16], respectively).

### 3.2.5 Building the layer stack

The structure of the sample as function of the growth direction $z$ is known as the *layer stack* in the language of kdotpy. The relevant properties of the materials in the device are stored in a LayerStack instance. This includes the material and thickness of each layer, the $z$ resolution, the coordinates of the interfaces, the width of the interface smoothening, and the 'layer names' (like 'well', 'barrier', etc.). The LayerStack object is itself an attribute of the PhysParams instance that is used to construct the Hamiltonian.

Importantly, the LayerStack class provides a function that creates a cache of all material parameters, which is stored within PhysParams. The $z$ dependence of each material parameter $Q$ is calculated in the following manner. We first construct a set of layer weights functions $w_l(z)$ for each layer $l$. Without interface smoothening, $w_l(z) = 1$ for $z$ inside the layer $l$ and $0$ elsewhere. For a finite interface smoothening width $\delta_{\mathrm{if}}$ (by default $\delta_{\mathrm{if}} = 0.075\,\mathrm{nm}$), the weight is

$$w_l(z) = \frac{1}{2}\left[\tanh\left(\frac{z - z_{\mathrm{min},l}}{\delta_{\mathrm{if}}}\right) - \tanh\left(\frac{z - z_{\mathrm{max},l}}{\delta_{\mathrm{if}}}\right)\right], \tag{77}$$

where the layer $l$ spans the interval $[z_{\mathrm{min},l}, z_{\mathrm{max},l}]$. The weights are normalized as $\tilde{w}_l(z) = w_l(z)/\sum_{l'} w_{l'}(z)$, so that $\sum_l \tilde{w}_l(z) = 1$ everywhere inside the layer stack. If we write the value of the material parameter $Q$ in layer $l$ as $Q_l$, the function $Q(z)$ is constructed as

$$Q(z) = \sum_l \tilde{w}_l(z) Q_l. \tag{78}$$

This function has the desired property that it interpolates smoothly between $Q_l$ and $Q_{l+1}$ if $\delta_{\mathrm{if}}$ is finite, while it approaches the values $Q_l$ away from the interfaces.

In view of the discrete representation of the $z$ coordinates, we need to evaluate $Q(z)$ only in a discrete array of $z$ values. If the $z$ resolution is $\Delta z$, we require evaluation at integer and half-integer multiples of $\Delta z$, because the action of the first and second degree derivatives in $z$, given by Eqs. (30) and (35), respectively, involves $Q(z_j \pm \frac{1}{2}\Delta z)$. The functions $Q(z)$ are thus evaluated on a discretized set of coordinates

$$\{z_{\mathrm{min}} - \tfrac{1}{2}\Delta z, z_{\mathrm{min}}, z_{\mathrm{min}} + \tfrac{1}{2}\Delta z, \ldots, z_{\mathrm{max}} - \tfrac{1}{2}\Delta z, z_{\mathrm{max}}, z_{\mathrm{max}} + \tfrac{1}{2}\Delta z\} \tag{79}$$

where $z_{\mathrm{min}}$ and $z_{\mathrm{max}}$ refer to the bottom and top of the complete layer stack. In physparams.py, this is achieved by setting

```
self.cache_z = -0.5 + 0.5 * np.arange(2 * self.nz + 1)
```

where coordinates are interpreted in units of $\Delta z$ with $0$ located at $z_{\mathrm{min}}$ at the bottom of the layer stack.

### 3.2.6 Handling of strain

In kdotpy, we use a simplified strain model for the diagonal components of the strain tensor. (The shear components are assumed to be zero.) By default, we strain each layer to match the in-plane lattice constants to that of the substrate. This defines

$$\epsilon_{\parallel} = \epsilon_{xx} = \epsilon_{yy} = \frac{a_{\mathrm{s}} - a_0}{a_0} \tag{80}$$

in terms of the target ('strained') lattice constant $a_{\mathrm{s}}$ and the equilibrium lattice constant $a_0$ of each layer material, cf. Sec. 2.7. The target lattice constant $a_{\mathrm{s}}$ is determined by one of three command-line arguments

- `msubst` followed by a material: The target lattice constant $a_s$ is taken as the equilibrium lattice constant of the substrate. The latter is defined as a material parameter a (see Appendix B.2).

- `alattice` followed by a value: The target lattice constant $a_s$ is taken to be the input value in **nm**.

- `strain` followed by a value: Equation (80) is bypassed and $\epsilon_\parallel$ is taken directly from the input value.

From minimization of the strain-energy tensor [30, 35], under the assumptions that the substrate is thick (so that its lattice constant stays at the equilibrium value) and that shear strain is absent, we find that the out-of-plane strain of the epitaxial layers equals

$$\epsilon_\perp = \epsilon_{zz} = -\frac{2C_{12}}{C_{11}}\epsilon_\parallel \tag{81}$$

For both HgTe and CdTe, $C_{12}/C_{11} \approx 0.69$ [2], which is presently hardcoded as a constant. (In a future kdotpy release, strain handling will be improved, by treating the elasticity modules $C_{11}$, $C_{12}$, and $C_{44}$ as material parameters.)

The strain command-line argument also allows for several other strain configurations. In short, the values following strain are interpreted as $\epsilon_{xx}$, $\epsilon_{yy}$, and $\epsilon_{zz}$, in order. If any of these values is not specified (omitted or input as –), it is determined from the other value(s), following these rules:

- If $\epsilon_{xx}$ is specified but $\epsilon_{yy}$ is not, then set $\epsilon_{yy} = \epsilon_{xx}$, and vice versa.

- If $\epsilon_{zz}$ is specified but $\epsilon_{xx}$ and $\epsilon_{yy}$ are not, then set

$$\epsilon_{xx} = \epsilon_{yy} = -\frac{C_{12}/C_{11}}{1 + C_{12}/C_{11}}\epsilon_{zz}. \tag{82}$$

  This relation minimizes the strain-energy tensor with $\epsilon_{xx}$ and $\epsilon_{yy}$ as free variables [30].

- If $\epsilon_{xx}$ and $\epsilon_{yy}$ are specified but $\epsilon_{zz}$ is not, then set

$$\epsilon_{zz} = -(C_{12}/C_{11})(\epsilon_{xx} + \epsilon_{yy}) \tag{83}$$

  which generalizes Eq. (81) to the case where $\epsilon_{xx} \neq \epsilon_{yy}$. (Presently, only $\epsilon_{xx} = \epsilon_{yy}$ is supported, but this will change in a future version.)

The relations (82) and (83) apply to growth direction (001). If strain is combined with a nontrivial orientation (command-line argument `orient`), a correct result is not guaranteed.

### 3.2.7 Electrostatic potentials

In addition to the intrinsic potentials defined by the band edges, kdotpy can also model electrostatic potentials $V(z)$. The potential is added to the Hamiltonian as a diagonal matrix, adding $V(z_j)$ to each point $z_j$ in the $z$-coordinate basis. The potential can be read from a file or calculated from a carrier density profile with certain boundary conditions. In kdotpy 2d and kdotpy ll, the potential $V(z)$ is initialized in the following order:

- Read the potential from a file, if the command line argument `potential filename` is provided. The input file must be a CSV file with two columns, labelled `z` and `potential` in the first row. The values that follow represent the coordinates $z_i$ and the values $V(z_i)$. The input coordinate values need not align with the $z$ coordinates of the lattice: Interpolation and extrapolation is used to calculate $V(z)$ on all $z$ coordinates of the lattice. (Thus, certain coordinate values may be omitted, e.g., when $V(z) = 0$ for all $z \in [z_1, z_2]$, one can specify the potential at $z_1$ and $z_2$ and omit all intermediate values.) A numerical multiplier can be added in order to scale the potential, for example `potential v.csv 10` adds $10V(z)$ to the Hamiltonian, where $V(z)$ is the potential defined in `v.csv`. Multiple arguments and numerical multipliers may be added in order to input linear combinations, for example with `potential v1.csv 10 potential v2.csv -5` the potential is $10V_1(z) - 5V_2(z)$. The parsing of potential files is done by the function `read_potential()` in `potential.py`.

- Calculate a Hartree potential self-consistently, if the command line argument `selfcon` is given. This method considers the electric charge density profile due to the occupied eigenstates, solves $V(z)$ from the Poisson equation, and then feeds this back into the Hamiltonian, after which diagonalization yields a new set of eigenstates. If this iterative process converges, it yields a self-consistent solution of the Schrödinger and the Poisson equation. We discuss this method in greater detail in Sec. 3.11.

- Define a potential from boundary conditions, specified on the command line. This is a 'static' calculation, not done self-consistently. The following commands are the most common ones to define a potential:

  - `vinner v0`, where v0 is a numerical value, representing an energy $V_0$ in meV. The potential $V(z)$ is fixed at $V(z_0) = 0$ in the centre of the 'well' layer and a potential difference $V_0$ is applied between the bottom and top interface of this layer. (This is equivalent to setting the potential to $-\frac{1}{2}V_0$ and $\frac{1}{2}V_0$ at the bottom and top interface of the well layer.) Extrapolation to the other layers is done by assuming a constant electric field.

  - `vouter v0`, where v0 is a numerical value, representing an energy $V_0$ in meV. This command is similar to `vinner`, with the difference that a potential difference $V_0$ is applied between the bottom and top of the full layer stack.

  - `vsurf v0 w`, where v0 and w represent an energy $V_0$ in meV and a width $w$ in nm. The resulting potential depends on the minimal distance $d_{\text{if}}(z)$ between $z$ and each interface $z_i$. The potential is

$$V(z) = V_0(1 - d_{\text{if}}(z)/w) \tag{84}$$

   if $d_{\text{if}}(z) \leq w$, otherwise $0$. If the command is followed by `q`, this is changed to the quadratic dependence $V(z) = V_0(1 - d_{\text{if}}(z)/w)^2$

The parsing of the potential options is done by `gate_potential_from_opts()`, whereas the solution of $V(z)$ given the boundary conditions is done by `solve_potential()`, both in `potential.py`. We provide more details on solving $V(z)$ in Appendix A.4.

These options may be combined, e.g., if one uses a potential input file together with the self-consistent Hartree method, then the input potential serves as the initial potential for the iterative solution process. For `kdotpy 1d`, the self-consistent Hartree method is not available. The extra option `potentialy filename` provides the capability of defining a potential $V(y)$. For `kdotpy bulk` and `kdotpy bulk-ll`, none of the potential options is available.

## 3.3 Construction of Hamiltonian

### 3.3.1 Bulk geometry

The $\mathbf{k} \cdot \mathbf{p}$ Hamiltonian is an $8 \times 8$ matrix $H_{pq}(k_x, k_y, k_z)$ with orbital indices $p, q$ and depending on the momentum coordinates $(k_x, k_y, k_z)$. For bulk materials, the band structure can simply be obtained by substituting values $(k_x, k_y, k_z)$ and *diagonalizing* the $8 \times 8$ matrix as to obtain $8$ eigenvalues and $8$ eigenvectors.

### 3.3.2 Two-dimensional geometry

As discussed in Sec. 2.2, layered structures break translational symmetry in the $z$ direction. The momentum $k_z$ is substituted by the operator $-i\partial_z$ and the coordinates are discretized to $z = \{z_j\}$. This set has to be finite for the sake of calculation. With $n_z$ coordinates in this set, the dimension of the Hilbert space is $8n_z$. The Hamiltonian is thus represented as an $8n_z \times 8n_z$ matrix $H_{i,p;j,q}(k_x, k_y)$ where $i, j$ represent the $z$ coordinates and $p, q$ the orbitals.

In kdotpy, the matrix $H_{i,p;j,q}(k_x, k_y)$ is constructed as a sparse matrix from $8 \times 8$ blocks. (The blocks are $6 \times 6$ is the six-orbital Kane model is used. For the sake of clarity, we assume the eight-orbital model in the following discussion.) The $8 \times 8$ blocks that constitute several terms of the Hamiltonian are defined in the submodule `hamiltonian.blocks`. They are summed up in a function from `hamiltonian.full`. For the 2D geometry, the appropriate function is `hz()`, defined with the argument signature

```
hz(z, dz, k, b, params, **kwds)
```

This function returns an $8 \times 8$ matrix for each combination of arguments. The indices $i, j$ for the $z$ coordinates encoded as z and z + dz, respectively. The arguments k and b are momentum and magnetic field. The argument params is the PhysParams instance needed to construct the matrices and **kwds denotes several additional options.

In line with the algorithm discussed in Sec. 2.2, hz returns a nonzero result only for dz being $\mathbf{0}$ or $\pm\mathbf{1}$. For dz = 0, the results are the $8 \times 8$ submatrices $H_{i,p;i,q}$ appearing on the block diagonal of the full Hamiltonian matrix. For dz = 1 and dz = –1, we find the off-diagonal blocks $H_{i,p;i\pm1,q}$ that contain terms involving $z$ derivatives, see, e.g., Eqs. (30) and (35).

The full Hamiltonian matrix, a sparse matrix of dimension $8n_z \times 8n_z$, is constructed as follows.

- Initialize lists allrows, allcols, allvals. These will be used later to construct a sparse matrix in COO (coordinate) format, where nonzero matrix elements are represented as triplets $(I, J, v)$ of row index, column index, and value.

- Iterate over the diagonal blocks: For $z = 0, \ldots, n_z - 1$, call hz(z, 0, k, b, params, **kwds). Flatten the resulting $8 \times 8$ matrix m and append the values to allvals,

  ```
  allvals.append(m.flatten())
  ```

  The corresponding row and column indices are $I = 8z + p$, $J = 8z + q$ where $p, q$ are the orbital indices. This is implemented as

  ```
  allrows.append(z * norb + rows0)
  allcols.append(z * norb + cols0)
  ```

  where rows0 and cols0 are (precalculated) arrays of length $\mathbf{64}$ with the row and column indices $(\mathbf{0}, \ldots, \mathbf{7})$ corresponding to the flattened matrix.

- Iterate over the off-diagonal blocks. For $z = 0, \ldots, n_z - 2$, calculate

  ```
  mm = 0.5 * (hz(z + 1, -1, k, b, params, **kwds) \
      + hz(z, 1, k, b, params, **kwds).conjugate().transpose())
  mp = mm.conjugate().transpose()
  ```

  In principle, we could have used hz(z + 1, -1, k, b, params, **kwds) and hz(z, 1, k, b, params, **kwds) as the two blocks above and below the diagonal. By construction they are each other's hermitian conjugate, but we symmetrize them in order to eliminate numerical errors and to make the Hamiltonian exactly hermitian. We proceed by filling the value, row, and column lists as

  ```
  allvals.append(mp.flatten())
  allrows.append(z * norb + rows0)
  allcols.append((z + 1) * norb + cols0)
  allvals.append(mm.flatten())
  allrows.append((z + 1) * norb + rows0)
  allcols.append(z * norb + cols0)
  ```

- Turn allvals, allrows, and allcols into one-dimensional arrays and construct the sparse matrix from all nonzero values,

  ```
  non0 = (allvals != 0)
  s = coo_matrix(
      (allvals[non0], (allrows[non0], allcols[non0])),
      shape = (norb * nz, norb * nz), dtype = complex
  )
  ```

  using coo_matrix from scipy.sparse.

The COO format is suitable for construction of a sparse matrix, but not so much for other operations. Diagonalization is done most efficiently in the CSC (compressed sparse column) format. For obtaining a sparse matrix in CSC format, constructing the matrix in COO format and converting it to CSC is more efficient than constructing it directly as a CSC matrix [2].

### 3.3.3  One-dimensional geometry

The construction of the Hamiltonian for 1D geometries follows the same line of reasoning. With the additional discretization of the $y$ coordinates, the Hilbert space has dimension $8n_y n_z$, where $n_y$ and $n_z$ are the number of coordinate values in $y$ and $z$ direction, respectively. The Hamiltonian is thus an $8n_y n_z \times 8n_y n_z$ matrix $H_{l,i,p;m,j,q}(k_x, k_y)$ where the indices $l, m$ encode the $y$ coordinates, $i, j$ the $z$ coordinates and $p, q$ the orbitals. The construction is by filling a sparse matrix with $8 \times 8$ blocks; the appropriate function for a 1D geometry (in absence of magnetic fields) is hzy(), defined with the argument signature

```
hzy(z, dz, y, dy, kx, params, **kwds)
```

where the indices $l, m$ are represented by y and y + dy, and $i, j$ by z and z + dz as before. The sparse matrix constructor iterates over $z$ and $y$. For example, for the diagonal terms (dz = 0 and dy = 0),

```
allvals.append(m.flatten())
allrows.append(y * norb * nz + z * norb + rows0)
allcols.append(y * norb * nz + z * norb + cols0)
```

where the row and column indices are obtained as $I = 8n_z y + 8z + p$ and $J = 8n_z y + 8z + q$, respectively. The constructor also does a similar iteration for three types of off-diagonal blocks, $(dz, dy) = (0, \pm 1)$, $(\pm 1, 0)$, and $(\pm 1, \pm 1)$. Like the 2D geometry, symmetrization is applied to the off-diagonal blocks to make the Hamiltonian matrix exactly hermitian. The constructor function also has an option periodicy by means of which periodic boundary conditions can be applied in the $y$ direction. This is done by adding the blocks corresponding to $(l, m) = (0, n_y - 1)$ and $(n_y - 1, 0)$. In the presence of magnetic fields, the blocks are obtained by using hzy_magn(), instead of hzy(), but the sparse construction itself is fully analogous.

### 3.3.4  Landau levels in axial approximation

In the Landau level formalism, the in-plane momentum operators $\hat{k}_x$ and $\hat{k}_y$ are substituted by the ladder operators $a$ and $a^\dagger$ acting on the Landau level states, see Sec. 2.4. In the axial approximation, the Landau level index $n$ is a conserved quantum number, so that the Hamiltonian may be split in several independent blocks $H^{\mathrm{ax},(n)}$. The diagonalization is then simply done for each $n = -2, -1, 0, \ldots, n_{\mathrm{max}}$, where $n_{\mathrm{max}}$ is the maximum index set by the command line argument nll.

For the axial model, we use the *symbolic* Landau level mode. (The label *symbolic* distinguishes it from the *legacy* mode, which was implemented earlier, but is no longer used.) In this mode, a symbolic form of the 2D Hamiltonian $H(k_x, k_y, z)$ is constructed. Subsequently, the operators $\hat{k}_x$ and $\hat{k}_y$ are substituted by $a$ and $a^\dagger$ following Eq. (51). The block $H^{\mathrm{ax},(n)}$ is obtained by applying the ladder operators which yields factors involving $n$. In detail, these steps are performed as follows:

- The Taylor expansion of the Hamiltonian in $k_x$ and $k_y$ up to quadratic order is

$$H(k_x, k_y, z) = H(0, 0, z) + k_x(\partial_{k_x} H)(0, 0, z) + k_y(\partial_{k_y} H)(0, 0, z)$$
$$+ \tfrac{1}{2}k_x^2(\partial_{k_x}^2 H)(0, 0, z) + \tfrac{1}{2}k_y^2(\partial_{k_y}^2 H)(0, 0, z) + k_x k_y(\partial_{k_x}\partial_{k_y} H)(0, 0, z). \quad (85)$$

We evaluate $H_0(z) \equiv H(0, 0, z)$ and its first and second degree derivatives at $(k_x, k_y) = (0, 0)$.

---

[2]See the documentation of scipy.sparse at https://docs.scipy.org/doc/scipy/reference/sparse.html

We calculate the derivatives as

$$
\begin{aligned}
H_x(z) &= (\partial_{k_x} H)(0,0,z) = \frac{H(\Delta k_x, 0, z) - H(-\Delta k_x, 0, z)}{2\Delta k_x} \\
H_y(z) &= (\partial_{k_y} H)(0,0,z) = \frac{H(0, \Delta k_y, z) - H(0, -\Delta k_y, z)}{2\Delta k_y} \\
H_{xx}(z) &= (\partial_{k_x}^2 H)(0,0,z) = \frac{H(\Delta k_x, 0, z) - 2H(0,0,z) + H(-\Delta k_x, 0, z)}{2(\Delta k_x)^2} \\
H_{yy}(z) &= (\partial_{k_y}^2 H)(0,0,z) = \frac{H(0, \Delta k_y, z) - 2H(0,0,z) + H(0, -\Delta k_y, z)}{2(\Delta k_y)^2} \\
H_{xy}(z) &= (\partial_{k_x}\partial_{k_y} H)(0,0,z) \\
&= \frac{H(\Delta k_x, \Delta k_y, z) - H(\Delta k_x, -\Delta k_y, z) - H(-\Delta k_x, \Delta k_y, z) + H(-\Delta k_x, -\Delta k_y, z)}{\Delta k_x \Delta k_y}.
\end{aligned}
\tag{86}
$$

Here, the values of $\Delta k_x$ and $\Delta k_y$ can be set arbitrarily, because the Hamiltonian is quadratic and does not contain higher-order terms.

- The symbolic Hamiltonian can be formed by simply substituting $k_x \to \hat{k}_x$ and $k_y \to \hat{k}_y$ in Eq. 85,

$$
H(\hat{k}_x, \hat{k}_y, z) = H_0(z) + \hat{k}_x H_x(z) + \hat{k}_y H_y(z) + \tfrac{1}{2}\hat{k}_x^2 H_{xx}(z) + \tfrac{1}{2}\hat{k}_y^2 H_{yy}(z) + \tfrac{1}{2}\{\hat{k}_x, \hat{k}_y\} H_{xy}(z). \tag{87}
$$

Note that the order of the operators is important. To simplify the substitution of ladder operators, we first transform to $\hat{k}_\pm = \hat{k}_x \pm i\hat{k}_y$. We thus obtain the symbolic Hamiltonian

$$
\begin{aligned}
H(\hat{k}_+, \hat{k}_-, z) = H_0(z) &+ \hat{k}_+ H_+(z) + \hat{k}_- H_-(z) \\
&+ \tfrac{1}{2}\hat{k}_+^2 H_{++}(z) + \tfrac{1}{2}\hat{k}_-^2 H_{--}(z) + \tfrac{1}{2}\hat{k}_+\hat{k}_- H_{+-}(z) + \tfrac{1}{2}\hat{k}_-\hat{k}_+ H_{-+}(z).
\end{aligned}
\tag{88}
$$

with $H_\pm = \tfrac{1}{2}(H_x \mp H_y)$, $H_{\pm\pm} = \tfrac{1}{2}(H_{xx} - H_{yy} \mp 2iH_{xy})$, and $H_{+-} = H_{-+} = \tfrac{1}{2}(H_{xx} + H_{yy})$.

- The symbolic Hamiltonian of Eq. (88) is implemented as a `SymbolicMatrix` object, which is essentially a wrapper around an operator sum represented by the `dict`

```
self.opsum = {
    "": h0, "+": hkp, "-": hkm,
    "++": hkpkp, "--": hkmkm, "+-": hkpkm, "-+": hkpkm
}
```

where `self` refers to the `SymbolicMatrix` instance. The keys of the operator sum `dict` are strings that represent combinations of operators $\hat{k}_\pm$; the values are the $8n_z \times 8n_z$ matrices $H_0$, $H_+$, $H_-$, etc.

- The `SymbolicMatrix` class has a member function `SymbolicMatrix.ll_evaluate()` that effectively evaluates matrix elements of the form $\langle \Phi^{(n')} | H(a, a^\dagger, z) | \Psi^{(n)} \rangle$, cf. Eq. (52). (For the axial model, $n' = n$.) It acts with the ladder operators on the Landau level states $|n + \delta n\rangle$ ($\delta n = -1, 0, 1, 2$) and replaces them by the appropriate values (that depend on $n$ and the magnetic field $B_z$). The matrix elements are calculated for each term in the operator sum separately. The result is the sum over these terms. It represents $H^{\mathrm{ax},(n)}$ as an $8n_z \times 8n_z$ matrix. (Except for $n = -2, -1, 0$, where the number of orbitals is $1, 4, 7$, respectively, instead of $8$.)

We note that the construction of the `SymbolicMatrix` object needs to be done only once. This object can then be evaluated repeatedly by calling `ll_evaluate()` for each $n = -2, -1, 0, \ldots, n_{\max}$ and for multiple values of $B_z$.

### 3.3.5 Landau levels in 'full' mode

In absence of axial symmetry, the Landau level index $n$ fails to be a conserved quantum number. As argued in Sec. 2.4, the nonaxial terms $R^{\text{nonax}}$ in the $\mathbf{k} \cdot \mathbf{p}$ Hamiltonian couple Landau levels $n'$ and $n$ with $n' - n = \pm 4$. (That is, $\langle \Phi^{(n')} | H | \Psi^{(n)} \rangle$ is nonzero for $n' - n = -4, 0, 4$.) For lower symmetry, there may be other nonzero matrix elements as well for other combinations of $n'$ and $n$ with $|n' - n| \leq 4$.

The 'full' Landau level Hamiltonian contains the Landau-level degrees of freedom in the Hilbert space. The matrix structure of the Hamiltonian can thus be expressed as $H_{n',i,p;n,j,q}(B)$, with $n', n$ being the Landau level indices, $i, j$ the $z$ coordinates, and $p, q$ the orbitals. This matrix is constructed as a sparse matrix as follows:

- Initialize lists `allrows`, `allcols`, `allvals`.

- Iterate over the Landau indices $n = -2, -1, 0, \ldots, n_{\max}$. For each index $n$, iterate over $n' = n, \ldots, n + 4$ with $n' \leq n_{\max}$. Evaluate the $(n', n)$ block of the symbolic Hamiltonian as

  ```
  ham = hsym.ll_evaluate((nprime, n), magn, ...)
  ```

  This yields a (sparse) matrix of dimension $n_{\text{orb}}(n')n_z \times n_{\text{orb}}(n)n_z$, where $n_{\text{orb}}(n)$ is the number of orbitals for Landau level index $n$, i.e., 1, 4, 7, or 8 for $n = -2, -1, 0$, and $n \geq 1$, respectively.

- If the block `ham` is nonzero (i.e., `ham.nnz > 0`), convert it to a sparse matrix `hamcoo` in COO format and extract its values, row indices, and column indices. The block is added the full matrix by copying its values and taking the indices shifted by the correct index offsets

  ```
  allvals.append(hamcoo.data)
  allrows.append(index_offsets[nprime + 2] + hamcoo.row)
  allcols.append(index_offsets[n + 2] + hamcoo.col)
  ```

  where `index_offsets` is a (precalculated) array of index offsets, defined as the cumulative number of degrees of freedom $n_{\text{orb}}(n')n_z$ for all Landau levels $n' < n$. The values of `index_offsets[n + 2]` are equal to $0, n_z, 5n_z, 12n_z$ for $n = -2, -1, 0, 1$ and $(4 + 8n)n_z$ for $n \geq 1$.

- Turn `allvals`, `allrows`, `allcols` into one-dimensional arrays and create the sparse matrix

  ```
  s = coo_matrix(
      (allvals, (allrows, allcols)), shape = (dim, dim), dtype = complex
  )
  ```

  where $\text{dim} = (12 + 8n_{\max})n_z$. Convert it to CSC format and return the result.

This algorithm is implemented as the function `hz_sparse_ll_full()` in the submodule `hamiltonian.hamiltonian`.

### 3.3.6 Transformable Hamiltonian

The default way of constructing the $8 \times 8$ blocks is by direct definition of the matrices by element. These built-in definitions are appropriate for crystals grown along the (001) direction, the most common direction used in crystal growth. Unfortunately, constructing the Hamiltonian for a generic growth direction from the Hamiltonian for (001) is not straightforward [2].

In kdotpy, we follow a different approach and construct the Hamiltonian bottom up. The idea is that the terms in the Hamiltonian are written as products of momenta $k_i$ and angular momentum matrices $\sigma_j, J_j, T_j, T_{lm}$ ($i, j, l, m = x, y, z$), similar to Table C.5 in the book by Winkler [15]. Each quantity has known transformation rules given by a representation of the group $\mathbf{SO(3)}$ of the proper rotations of three-dimensional space.

- Trivial representation: Some terms, like $k_x \sigma_x + k_y \sigma_y + k_z \sigma_z$ in the $H_{66}$ block, are invariant under all $\mathbf{SO(3)}$ rotations.

- Vector representation: The irreducible three-dimensional representation of **SO(3)** describes the transformations of vector-like quantities. Examples are momentum $(k_x, k_y, k_z)$, and angular momentum $(\sigma_x, \sigma_y, \sigma_z)$ and $(J_x, J_y, J_z)$.

- Five-dimensional representation: The five-dimensional irreducible representation of **SO(3)** is a term in the symmetric product of two vector representations. The quintuplet

$$(2T_{yz}, 2T_{zx}, 2T_{xy}, T_{xx} - T_{yy}, (2T_{zz} - T_{xx} - T_{yy})/\sqrt{3}) \tag{89}$$

(in the $H_{68}$ block) transforms in this representation.

Inversion does not exist in **SO(3)**, hence we do not distinguish axial vectors from regular vectors.

We use this information define the transformation between the *lattice coordinates*, related to the crystal lattice, and the *device coordinates*, attached to the geometry of the device. In this section, we denote the device (or sample) coordinates as $(x, y, z)$. The $z$ direction is by definition the growth direction. For a strip geometry, the longitudinal direction (in which momentum is defined) is $x$ and the transversal direction (confined) is $y$. For lattice coordinates, we use the primed symbols $(x', y', z')$. The 'primed' axes are aligned with the axes of the crystal lattice. The two coordinate systems are related by a pure rotation, $(x', y', z') = R(x, y, z)$ where $R$ is a matrix in **SO(3)**, i.e., an orthogonal $3 \times 3$ matrix with determinant $+1$. Note that for the growth direction (001), the two coordinate systems coincide; in other words, $R$ is the identity transformation.

The definition of the Hamiltonian is in terms of the lattice coordinates $(x', y', z')$, because the electrons are subject to the (local) crystal environment. In order to calculate the band structure for a non-trivial lattice orientation, the Hamiltonian defined in terms of momenta $k_i$ and angular momenta $J_j$ is transformed into device coordinates $(x, y, z)$ using the transformation matrix $R$. The following example illustrates this principle: Take the $\gamma_3$ term in the $\Gamma_8$ block [cf. Eq. (15)], which can be written as

$$-\gamma_3[\{J_{x'}, J_{y'}\}\{k_{x'}, k_{y'}\} + \{J_{y'}, J_{z'}\}\{k_{y'}, k_{z'}\} + \{J_{z'}, J_{x'}\}\{k_{z'}, k_{x'}\}] \tag{90}$$

where $A, B = AB + BA$ and we have explicitly indicated that it is defined in the primed (crystal) coordinate system. The unprimed version of this term is then found by setting

$$(J_{x'}, J_{y'}, J_{z'}) = R(J_x, J_y, J_z) \qquad \text{and} \qquad (k_{x'}, k_{y'}, k_{z'}) = R(k_x, k_y, k_z). \tag{91}$$

The Hamiltonian is stored in tensor form, where each term is stored as a separate tensor. The term in this example is encoded as tensor $T_{ij;kl}$; the relation Hamiltonian term is then found by setting $T_{ij;kl} k_i k_j J_k J_l$. (Einstein summation convention assumed.) The transformation to device coordinates is then given by

$$T_{ij;kl} \to T'_{ij;kl} = R_{ia} R_{jb} R_{kc} R_{ld} T_{abcd}, \tag{92}$$

This transformation is done for all terms in the Hamiltonian separately.

For the implementation in `kdotpy`, each term in the Hamiltonian is represented by instances of the `KJTensor` class. These are then transformed and evaluated by substituting the angular momentum matrices $J_j$. The result is a `KTermsDict` instance that encodes sum of terms $M_{ij} k_i k_j$ where $M$ is a matrix. In detail, the algorithm for constructing and transforming the Hamiltonian is as follows:

- For each term, a `KJTensor` is initialized through a `dict` which defines the tensor components. The example term of Eq. (90) is defined as

```
g3_tens = KJTensor(
    {'yzyz':1, 'xzxz':1, 'yxyx':1}, nk = 2
).symmetrize((0,1), fill = True).symmetrize((2,3), fill = True)
```

- The transformation is then done by calling `KJTensor.transform(R)` for each term in the Hamiltonian. This yields the transformed version of the `KJTensor` for each term. Terms that are known to be invariant are left alone:

```
for tens, tens_name in zip(all_tens, tens_names):
    invariant = tens.is_invariant_under_transform(params.lattice_trans)
    if not invariant:
        tens.transform(params.lattice_trans, in_place = True).chop()
```

- Subsequently, calling `KJTensor.apply_jmat()` on the transformed KJTensor instances turns them into matrix form by substituting the spin matrices $J_j$ for their matrix representations. This returns a KTerms instance, which encodes the sums of the form $M_{ij}k_ik_j$ where $M$ is a matrix. Multiple KTerms instances are collected in the KTermsDict instance `kterms`. For the example term [Eq. (90)]

```
kterms = KTermsdict()
...
kterms['g3_88'] = \
    g3_tens.apply_jmat(spin.j3basis, symmetrize_k = True).chop()
```

Here, the term labelled {'g3_88' in `kterms` is a KTerms instance defined by substituting the '$J$ basis' (simply the set of matrices $J_x, J_y, J_z$) on the already transformed KJTensor labelled `g3_tens`. We enforce symmetrization in the momentum components and use `.chop()` to chop off almost-zero values.

A non-trivial lattice orientation can be requested from the command-line by using the argument `orientation` with one to three further parameters that define the transformation matrix $R$ (see Appendix B.3 for details). If this is done, the KTermsDict object is constructed as described above and passed to the Hamiltonian block construction function as argument `kterms`. These functions are then responsible for composing the Hamiltonian from all different terms, and for substituting the momentum components $k_i$ as usual: By discretized derivatives in the confined directions and by momentum values in the unconfined directions.

The orientation may be set as `orientation – 001` for the default growth direction. This applies the formalism described here with the rotation matrix $R$ being the identity. In absence of an `orientation` (or an equivalent) argument, the regular construction method of the Hamiltonian is used. Comparison between the two is useful for verification that the two representations of the Hamiltonian yield the same result. We do not use the transformable representation of the Hamiltonian by default, because the regular construction method is significantly faster.

## 3.4 Diagonalization

### 3.4.1 Default eigensolver: Arnoldi with shift-and-invert

Diagonalization is the process of finding the eigenvalues and eigenvectors of a square matrix $M$. This is an ubiquitous numerical problem, for which several algorithms have been developed over the years. Perhaps the most well-known algorithm adapted to large sparse matrices is the Lanczos algorithm, that finds a subset of the eigenvalues (typically the ones largest in magnitude) and the corresponding eigenvectors for a hermitian matrix $M$ [36]. The matrix $M$ is applied iteratively to a set of independent vectors $v_j$ until sufficient convergence is reached. The idea is that matrix-vector multiplication is a relatively cheap operation if the matrix $M$ is sparse. The naive implementation of this method is numerically unstable, but it can be adapted to yield reliable results [37]. The *Arnoldi iteration* method extends the Lanczos method to generic complex matrices. The *implicitly restarted Arnoldi method* is a commonly applied method thanks to its implementation in the popular ARPACK package [38], that has been linked to by many software packages, including the SciPy library for Python.

In order for these methods to be usable for `kdotpy`, we need to make an additional intermediate step. The Lanczos and Arnoldi methods find the eigenvalues with the largest magnitude, yet for band structures, we are typically most interested in the eigenvalues close to the charge neutrality point. Instead of finding the eigenvalues of the Hamiltonian $H$ itself, we apply the *shift-and-invert* algorithm and apply Arnoldi/Lanczos to the matrix

$$M = (H - \sigma I)^{-1} \tag{93}$$

where $I$ is the identity matrix and $\sigma$ is the *target energy*. Indeed, finding the $n_{\mathbf{eig}}$ largest eigenvalues of $M$ is equivalent to finding the $n_{\mathbf{eig}}$ eigenvalues of $H$ with smallest distance to $\sigma$. Implementations of the shift-and-invert algorithm generally do not store the matrix $(H - \sigma I)^{-1}$ in memory, as it is usually not sparse even if $H$ is. Instead, as it only needs to calculate matrix-vector products of the form $A^{-1}\mathbf{b} = \mathbf{x}$, it solves the equivalent system of equations $A\mathbf{x} = \mathbf{b}$. This can be solved efficiently with a sparse version of LU factorization.

In kdotpy, the default *eigensolver* is the function eigsh() from the SciPy module scipy.sparse.linalg. In essence, this SciPy function is just an interface to the ARPACK implementation. For the shift-and-invert step, it uses the SuperLU library. The function eigsh() is applied to the Hamiltonian matrix in CSC sparse format. The number of eigenvalues $n_{eig}$ is set by the command line using argument neig. The eigsh() argument sigma is set to the target energy, given by targetenergy on the command line.

The diagonalization step is usually the computationally heaviest part of kdotpy calculations, and as such the overall performance of the program strongly depends on this step. In order to maximize performance, kdotpy applies parallelization to reduce calculation time by making use of all available CPU resources. In addition, custom eigensolvers can increase performance by providing optimized methods for the available hardware, for example using the Intel MKL framework for optimization on supported CPUs or using CUDA for GPU acceleration; see Appendix B.4 for a detailed discussion.

### 3.4.2 Parallelization of the main loop

In kdotpy, calculation of the eigenvalues at each value of momentum $k$ and/or magnetic field $B$ constitutes a separate eigenvalue problem, and can be treated as an independent task. For each type of Hamiltonian (1D, 2D, LL, etc.), the diagonalization module provides a diagonalization function that performs the following steps for a single value of $k$ and/or $B$:

- Construct the Hamiltonian;

- Diagonalize the Hamiltonian (apply eigensolver);

- Process the eigenstates (e.g., calculate observables and Berry curvature).

This function returns a single DiagDataPoint instance. For a dispersion, the complete process of finding a band structure can be summarized as

```
datapoints = [diag_function(k, *args, **kwds) for k in kvalues]
data = DiagData(datapoints, grid=kvalues)
```

where kvalues is the VectorGrid instance containing the $k$ values. The iterative application of diag_function is very suitable for parallelization, because each call to diag_function requires a similar amount of resources.

Unfortunately, parallelization in Python is challenging as a result of the global interpreter lock (GIL). However, Python's multiprocessing module avoids this obstacle by running Python code in parallel subprocesses, each of which is affected only by its own GIL.

The basic form of parallelization in kdotpy is facilitated by the parallel_apply() function in the parallel submodule, that is based on the Pool.apply_async() method of Python's multiprocessing module. The iteration in the example above is replaced by

```
datapoints = parallel_apply(
    diag_function, kvalues, args, kwds, num_processes=num_cpus
)
data = DiagData(datapoints, grid=grid)
```

where num_cpus is the number of CPU cores in the system, or a custom value set by the command line argument cpu. The function parallel_apply() also implements a custom *signal handler* for proper handling of keyboard interrupts (Ctrl-C by the user) and terminate and abort signals. For details, refer to Appendix A.1.

The parallel_apply() method is adequate for the default eigsh solver from SciPy, where the three steps of each diagonalization task run on the same CPU. This approach is no longer adequate when a GPU-accelerated eigensolver is used: While the application of the eigensolver is done by the GPU, the other steps are CPU bound. The need for additional flexibility has motivated us to implement a parallelization framework with a higher level of abstraction. This *Task-Model framework* is built around the ModelX, Task, and TaskManager classes. Each ModelX class (where X can be 1D, 2D, LL, etc.) implements the recipe analogous to the diagonalization function. It defines the steps as separate functions so that they can be run as separate tasks. The Task class is a wrapper around each such task. The TaskManager initializes, manages, and closes the process and/or thread pools, and thus plays a similar

role as `parallel_apply()`. Like the parallelization framework with `parallel_apply()`, the Task-Model framework is built around the `Pool.apply_async()` method of Python's `multiprocessing` module. The benefit of using the Task-Model framework is the higher degree of flexibility and configurability. We discuss the implementation details in Appendix A.1.

## 3.5 Post-diagonalization

The eigenvectors obtained from the diagonalization have as many components as the size of the Hamiltonian matrix, and thus they require a substantial space to store them into memory or on disk. The default approach of `kdotpy` is to extract information from the eigenvectors immediately after each point is diagonalized, and to discard the eigenvectors themselves upon creating a `DiagDataPoint` instance, i.e., usually the attribute `DiagDataPoint.eivec` remains `None`.

### 3.5.1 Observables

The `DiagDataPoint` class has a separate attribute `DiagDataPoint.obsvals` which stores a two-dimensional array with the observable values for each eigenstate in the `DiagDataPoint` instance. The attribute `DiagDataPoint._obsids` contains the labels for the observables stored in `DiagDataPoint.obsvals`. The `DiagDataPoint` class also defines several functions for getting the values for one or more observables for all eigenstates in the instance.

The observables are defined in `observables.py`. Each one is an instance of the `Observable` class. The key attributes in this object are

- `obsid`: The observable label.

- `obsfun`: A function that calculates the values from the eigenvectors.

- `obsfun_type`: Determines the argument pattern passed to this function.

Further properties determine the textual representation of the observable, its unit, as well as the colour map used in plots.

Each observable has two variants, namely a dimensionless and a dimensionful one. For example, the expectation value along the growth axis $\langle z \rangle$ is a dimensionful value, while its dimensionless partner is $\langle z \rangle / d$ where $d$ is the total thickness of the layer stack. The `obsfun` function typically calculated the dimensionless variant. The `Observable` instance also contains attributes for the conversion between the two, and both the dimensionless and dimensionful variants have their own textual representations for quantity and unit.

The source file `observables.py` also defines an `ObservablesList` class with `all_observables` as its single instance. This instance contains all the `Observable` objects with the observable definitions. The output functions import `all_observables` in order to access the relevant properties. An overview of all observables defined in `observables.py` can be found in Appendix B.5.

All observable values saved in `DiagDataPoint.obsvals` are written to the output files. If the command line argument `obs obsid` is provided, then that observable is used to generate extra output files, e.g., csv files with just the values of that observable as function of momentum, and to colourize the curves in a dispersion or magnetic-field dependence plot. The appropriate colour scale is determined by the definitions in the appropriate `Observable`. Depending on the type of observable, a discrete or continuous set of colours is used. It is also possible to combine observables into a dual colour scale or an red-green-blue (RGB) colour map. In particular, `obs orbitalrgb` visualizes the orbital content of the states by using colours where the red, green, and blue channels ($(r, g, b)$ with normalized values on the interval $[0, 1]$) represent the `gamma6`, `gamma8l`, and `gamma8h` observables, respectively; the `gamma7` observable is implicitly represented as the 'blackness' $1 - r - g - b$ of the colour. Figure 2 illustrates the legend of figures with `obs orbitalrgb` with a few example colours.

### 3.5.2 Overlaps

The overlap between eigenvectors at different momenta or magnetic-field values is often useful to identify states in complicated dispersions or Landau fans.

For dispersions in the two-dimensional geometry (`kdotpy 2d`), the eigenvectors $|\psi_i(0)\rangle$ of the subbands at $\mathbf{k} = 0$ are stored when the extra command line argument `overlaps` is provided. Then,

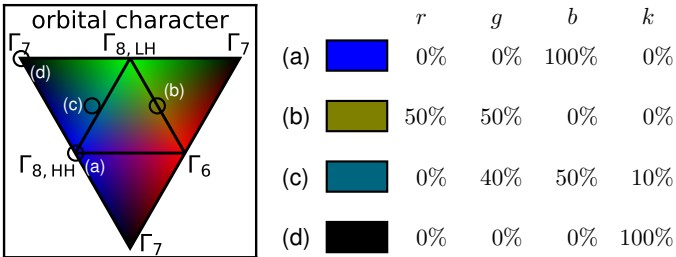

Figure 2: The legend of figures with `obs orbitalrgb` as command-line input. The red, green, and blue colour channels $(r, g, b)$ indicate the expectation values of the `gamma6`, `gamma8l`, and `gamma8h` observables, respectively. For example, blue (a) represents a pure heavy-hole state and dark yellow (b) an equal mixture of $|\Gamma_6, \pm\frac{1}{2}\rangle$ and $|\Gamma_8, \pm\frac{1}{2}\rangle$ states. The blackness value $k = 1 - r - g - b$ is implied, and represents the `gamma7` observable. We show examples with $k = 10\%$ (c) and pure black $k = 100\%$ (d), where the latter represents a pure $|\Gamma_7, \pm\frac{1}{2}\rangle$ state.

for each momentum $\mathbf{k}$, `kdotpy` calculates the subband overlaps $|\langle\psi_i(0)|\psi_j(\mathbf{k})\rangle|^2$ for all eigenstates $|\psi_j(\mathbf{k})\rangle$. This quantity can be interpreted as the expectation values of the projection operator

$$O^{(i)} = |\psi_i(0)\rangle\langle\psi_i(0)| \tag{94}$$

on the eigenstates $|\psi_j(\mathbf{k})\rangle$. The same also works for Landau level calculations (`kdotpy ll`), where the overlaps are calculated with the subbands $i$ at zero magnetic field.

The values of the subband overlaps are stored in `DiagDataPoint.obsvals` like any other observable and thus also written to the output files. If one provides `obs subbandrgb` on the command line, the plot colour of the dispersions is defined as the RGB triplet

$$(r, g, b) = (\langle O^{E1+}\rangle + \langle O^{E1-}\rangle, \langle O^{H1+}\rangle + \langle O^{H1-}\rangle, \langle O^{H2+}\rangle + \langle O^{H2-}\rangle), \tag{95}$$

where the red, green, and blue channels are normalized values on the interval $[0, 1]$. Likewise, `obs subbande1h1l1` does the same with the H2± subbands replaced by L1±. A larger number of subband labels may also been given, e.g., `obs subbande1h1h2l1`, where each label refers to a pair of subbands with opposite spin states, like $\langle O^{E1+}\rangle + \langle O^{E1-}\rangle$. For four labels, the colours are mixed from red, yellow, green, and blue. For $n_l > 4$ labels, the colours are mixed from $n_l$ equidistant hues (maximally saturated colours).

In Landau level mode, the command line argument `lloverlaps` can be used in order to calculate the probability density $\langle n|\psi\rangle$ in the Landau levels $n = -2, -1, 0, \ldots, n_{\max}$. These quantities can also be viewed as expectation values of the projection operators $|n\rangle\langle n|$. This option is available in the full LL mode only; in the symbolic LL mode, the index $n$ is a conserved quantum number, so that the expectation value is always $1$ for one index and $0$ for all others.

### 3.5.3 Berry curvature, Chern numbers, Hall conductance

The calculation of the Berry curvature in dispersion mode is based on Eq. (75) (see Sec. 2.8.1), with the slight modification that the summation over $|\psi\rangle$ includes all eigenstates within the energy window that has been calculated, which may be a subset of the full spectrum. Due to the denominator in Eq. (75), states $|\psi'\rangle$ that are far in energy from $|\psi\rangle$ only contribute weakly to the Berry curvature. Nevertheless, one should be aware that the omission of remote states can lead to deviations of the calculated Chern numbers from integer values.

The implementation in `berry.py` involves numerical evaluation of the derivatives $\partial_{k_x} H$ and $\partial_{k_y} H$ of the Hamiltonian. These are used to evaluate the matrices $V_i$ ($i = x, y, z$) with

$$(V_i)_{pq} = \frac{\langle\psi_p|\partial_{k_i} H|\psi_q\rangle}{E_p - E_q} \tag{96}$$

if $p \neq q$ and $V_{pp} = 0$. Here, $p$ and $q$ label the eigenstates and they run over all eigenstates obtained from the diagonalization. In terms of $V_i$, Eq. (75) may be evaluated as

$$\Omega_i^q(\mathbf{k}) = -\mathrm{Im} \sum_p \sum_{l,m} \epsilon_{ilm} (V_l)_{qp}^* (V_m)_{pq} = -\mathrm{Im} \sum_{l,m} \epsilon_{ilm} (V_l^\dagger V_m)_{qq} \tag{97}$$

The summation over $p$ is a matrix multiplication that can be evaluated efficiently with NumPy functions. Because we only need the diagonal elements of $V_l^\dagger V_m$, we extract the relevant columns and rows from $V_l^\dagger$ and $V_m$ and evaluate vector-vector dot products as

```
bcurv = [
    -np.imag(np.dot(vxd[q, :], vy[:, q]) - np.dot(vyd[q, :], vx[:, q]))
    for q in range(0, neig1)
]
```

for evaluating $\Omega_z^q(\mathbf{k})$ for all eigenstates $q$. In view of efficiency, we can choose to restrict the set of eigenstates $q$ for which we evaluate the Berry curvature. It is important to note that the set of eigenstates $p$ in the summation (or vector-vector product) must be a large as possible as to not lose significant contributions from remote eigenstates.

In two-dimensional dispersion mode (kdotpy 2d), only $\Omega_z^q(\mathbf{k})$ is evaluated, as the Berry curvature is essentially scalar in two dimensions. The Berry curvature observable may be accessed with the argument obs berry or obs berryz (equivalent). For three dimensions (kdotpy bulk), the Berry curvature is a vector quantity $\Omega_i^q(\mathbf{k})$ with $i = x, y, z$. The relevant observable labels are berryx, berryy, and berryz.

For Landau levels, Berry curvature $\Omega^q$ is calculated in the same manner as for dispersions, with the exception of how the Hamiltonian is treated: In the symbolic and full LL modes, the derivatives $\partial_{k_i} H$ of the Hamiltonian are calculated symbolically. The remaining factors $k_x$ and $k_y$ are substituted by the appropriate ladder operators $a$ and $a^\dagger$ before $\partial_{k_i} H$ is inserted into into Eq. (96).

The Chern number $C^q$ of each eigenstate $q$ is obtained by multiplying the Berry curvature by the Landau level degeneracy,

$$C^q = \frac{e B_z}{\hbar} \Omega^q \tag{98}$$

in terms of the perpendicular magnetic field $B_z$, cf. Sec. 2.8.2. The degeneracy factor is related to the magnetic length $l_B$ as $e B_z/\hbar = 1/l_B^2$. The Chern number from Eq. (98) is stored as observable labelled chern. These values can be used to find the Hall conductance $\sigma_H$ by summing over all occupied states, see Eq. (76).

Due to the spectrum being incomplete, the Chern numbers calculated with Eq. (98) may deviate from integer values. In addition to observable chern, kdotpy also provides the observable chernsim which 'simulates' exactly integer Chern numbers. The value is simply $1$ for all states if $B_z \neq 0$ and $0$ if $B_z = 0$. We note that this simple assumption violates the property that the sum of all Chern numbers must vanish, hence the resulting Hall conductance [from Eq. (76)] is valid only in the part of the spectrum close to the charge neutrality point.

## 3.6 Band alignment

### 3.6.1 Motivation

For all calculation modes except for the bulk mode, kdotpy uses sparse diagonalization which yields only a subset of all eigenvalues and eigenvectors. Thus, it is not possible to label the eigenvectors from $1$ to $N$ (the matrix size) from smallest to largest eigenvalue. Given two sets of eigenvalues at two points in momentum space, it is not known a priori how to connect the eigenvalues as to form a (sub)band.

The band alignment algorithm *connects the dots*: Considering two adjacent points in momentum space (or in magnetic-field value), assign a 'band index' to each eigenvalue. States with identical band indices across momentum point are considered to be one connected band. This band index is used by many other functions of kdotpy, for example, plot and CSV output, calculation of density of states, dispersion derivatives, etc.

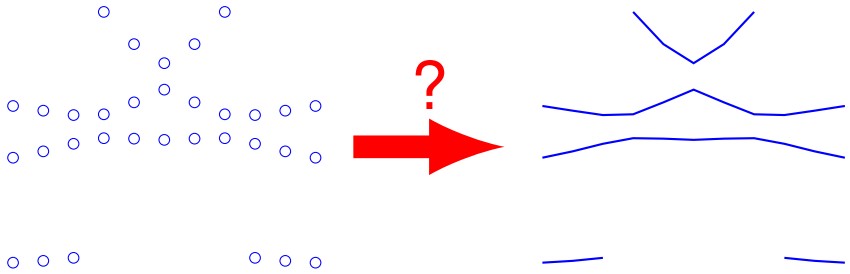

Figure 3: Illustration of the problem that is solved by the band alignment algorithm.

### 3.6.2 Band indices definition

Before we look at the actual recipe, let us discuss in detail the properties and the role of the band indices. They satisfy the following basic properties:

- The band indices are nonzero integers.

- The band indices are always monotonic in the energy (eigenvalues). As a consequence, dispersions never cross (e.g., in the output plots). Whereas this is not always the most natural choice from the perspective of the eigenstates, the assumption of monotonicity greatly simplifies the band alignment algorithm. The advantage of this method is that it does not need the eigenvectors, so that the RAM usage can remain limited. Earlier attempts with use of the eigenvectors have proved to be much less reliable than the present implementation.

- Additionally, `kdotpy` tries to determine the charge neutrality point from the band characters at zero momentum and zero magnetic-field. At this point, states above this energy get positive band indices ($1, 2, 3, \ldots$) and those below get negative indices ($-1, -2, -3, \ldots$). The index $0$ is never assigned. Thus, the sign of the band index is used to determine 'electron-like' or 'hole-like' states. (Here, the notions 'electron' and 'hole' are defined with respect to the band energy with respect to charge neutrality.)

- For Landau-level calculations in the axial approximation, the band indices are calculated for each Landau level index separately. In this mode, states are labelled by a pair of indices (LL index, band index). This is not true for full LL mode, where `kdotpy` uses a single band index similar to dispersion mode.

- The band indices are quite essential for many functions to do their job. For example, in order to calculate the derivative of the dispersion, one needs two points of the same band at different points in momentum space, as to be able to calculate the differential quotient. For the integrated density of states (IDOS), the assignment of positive and negative band indices determines at which energy the **IDOS = 0**. It also increases the accuracy of the (integrated) DOS, because it is based on interpolation of the dispersion between the available data points.

### 3.6.3 Band alignment algorithm

The band alignment algorithm is given by the following 'recipe'.

1. Calculate band characters at zero ($\mathbf{k} = 0$ and $\mathbf{B} = 0$). Determine the charge neutrality point and assign positive band indices above, and negative band indices below this value.

2. Take a data point adjacent to zero. Align the two sets of eigenvalues (see below). States aligned with each other get the same band index, i.e., the band indices for the 'new' data point get them from the data point at zero.

3. For all band indices that the two points have in common, extrapolate the energies to the next momentum or magnetic field value. That is, given the energies $\{E_i^{(0)}\}$ and $\{E_{i'}^{(1)}\}$, calculate

$$\tilde{E}_j^{(2)} = E_j^{(0)} + \frac{k_2 - k_0}{k_1 - k_0}(E_j^{(1)} - E_j^{(0)}) \tag{99}$$

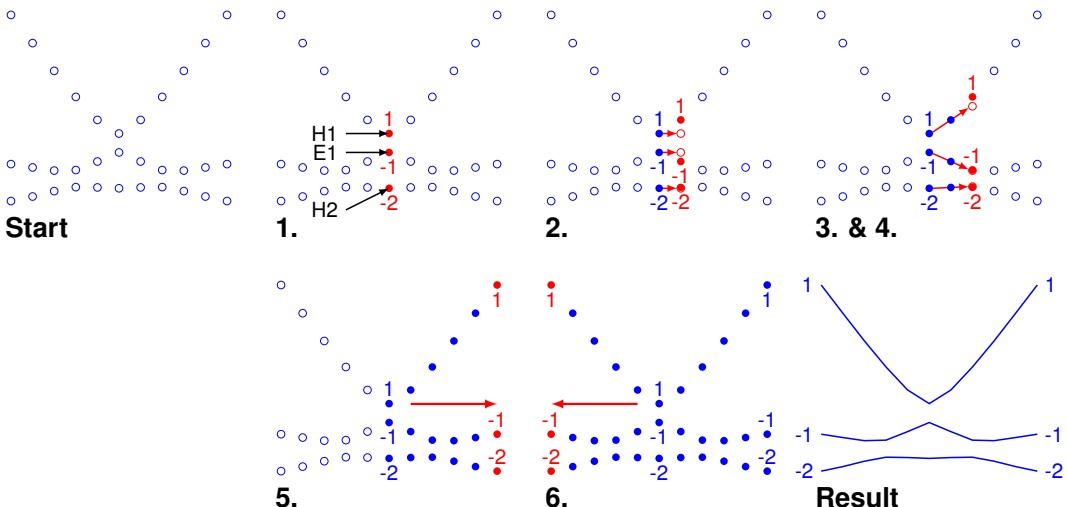

Figure 4: Band alignment recipe. 1. Determine band characters and assign band indices at zero momentum. 2. Align energies at adjacent point. 3–4. Extrapolate and align at third point. 5. Repeat previous steps forwards till the end of the domain. 6. Repeat previous steps backwards till the other end of the domain.

for all $j$ for which $E_j^{(0)}$ and $E_j^{(1)}$ are both defined.

4. Align the actual eigenvalues at the new data point to the extrapolated values $\tilde{E}_j^{(2)}$. Thus, set the band indices for the new data point from the extrapolated energies.

5. Repeat steps 3. and 4. until one reaches the end of the calculated domain.

6. Repeat steps 2. for the data point on the other side of the zero point. Perform steps 3. through 5. towards the other end of the domain.

This algorithm is essentially one-dimensional. For two dimensional grids, first perform the algorithm along the $k_x$ axis, then in the perpendicular direction starting at $(k_x, 0)$, in a 'fish bone' pattern. Analogously, in polar coordinates, the algorithm is performed along the radial direction first, then in the angular direction.

The alignment of two sets of energies as described above, is essentially a minimization of the energy differences. Assume two ordered sets of energies $\{E_i^{(1)}\}$ and $\{E_j^{(2)}\}$ (i.e., the energies are monotonic). Then vary the two indices relative to each other and find the *minimum average energy difference*. We also add a 'penalty' for non-matching energies, i.e., values in one set that do not have a partner in the other set.

Formalizing this idea, we find the value $d$ for which

$$\Delta(d) = \left( \sum_i |E_i^{(1)} - E_{i+d}^{(2)}|^e \right) / n(d) + W/n(d). \tag{100}$$

is minimal. The sum runs over all $i$ for which both values $E_i^{(1)}$ and $E_{i+d}^{(2)}$ are defined. The value $n_i(d)$ is the number of terms in the sum. The coefficient $e$ is an exponent and $W$ is the 'bonus weight' for each term in the sum. These values are set by the configuration settings `band_align_exp` (default is 4) and `band_align_ndelta_weight` (default is 20). (Setting `band_align_exp=max` replaces the sum by $\max_i |E_i^{(1)} - E_{i+d}^{(2)}|$.)

Given the value $d$ for which $\Delta(d)$ is minimal, we say that the values $E_i^{(1)}$ and $E_{i+d}^{(2)}$ are aligned. The result is that they will receive the same band indices.

Note that the algorithm is based on the energies *collectively*, not *individually*. This property makes the algorithm quite reliable.

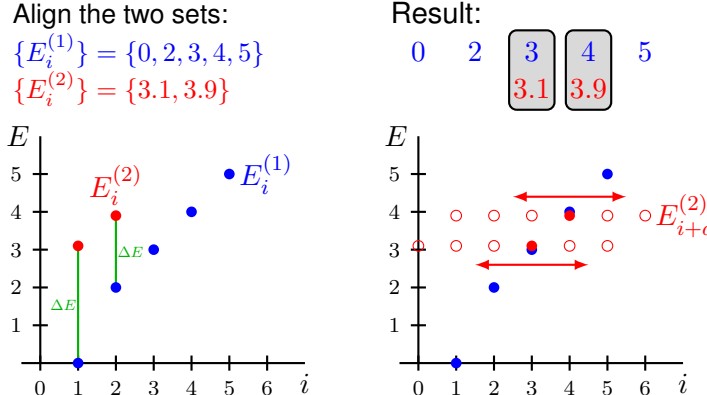

Figure 5: The alignment algorithm. (Left) Given two lists of energy values, we calculate the energy differences $\Delta E = |E_i^{(1)} - E_i^{(2)}|$ for each pair of values. The second set is shifted uniformly by index relative to the first one, $E_i^{(2)} \to E_{i+d}^{(2)}$. The sum of energy differences $\Delta(d)$ [see Eq. (100)] is calculated for each shift $d$ and minimized over $d$.

### 3.6.4 Manually assisted band alignment

Should the result of the band alignment not be satisfactory (e.g., incorrect), `kdotpy merge` includes two options to manually 'guide' the band alignment to a correct result.

- Using the `bandalign # [#]` command line argument. The first value is an energy and the second one a gap index (omitted value means **0**). This pins the gap with that index to the given energy at zero momentum or magnetic field. For example, if one uses `bandalign -10 2`, then the first eigenstate below $-10\,\text{meV}$ gets the band index **2**, the first one above band index **3**. The given energy need not be very precise; it must just lie in between two bands. For convenience, one could choose a large gap.

  NOTE: The gap with index **0** lies between the bands with indices $-1$ and **1**. The gap with index $g > 0$ between bands $g$ and $g + 1$, the gap with index $g < 0$ between bands $g - 1$ and $g$. Zero gaps (i.e., between degenerate states) also count.

  This option replaces step 1. in the above recipe only. The algorithm then uses the same alignment strategy as described in steps 2. through 6.

- Using an input file in CSV format with band indices and energies as function of momentum or magnetic field. The file format is the same as the *output* files `dispersion.byband.csv`. This output file is essentially the result of the band alignment, i.e., the assignment of band indices to energies. This file may be edited in a spreadsheet editor by moving around values using cut-and-paste, for example. Then it can be saved as a CSV file and loaded into `kdotpy` by using the command-line option `reconnect filename.csv`.

  The input file may be incomplete: Missing energy values at the left- or right-hand side of each line (not in the middle) will be filled in automatically, by extrapolation of band indices. Momentum (or magnetic field) values for which there is no data are filled in by interpolation or extrapolation from adjacent points, i.e., steps 3. and 4. of the above recipe. The energies only need to be approximately exact. This tolerant behaviour of the input is made possible by the implementation: The file input is processed by aligning the given energies to the actual eigenvalues. The band alignment algorithm does not need the energies to be exact or complete in order to be reliable.

  In this manner, it is also possible to get some hybrid result between manual input and the automatic algorithm.

| | A | B | C | D | E | F | G | H | I |
|---|---|---|---|---|---|---|---|---|---|
| 1 | | | -4 | -3 | -2 | -1 | 1 | 2 | |
| 2 | | | H2- | H2+ | E1- | E1+ | H1- | H1+ | |
| 3 | k | kphi | E | E | E | E | E | E | |
| 4 | -0.25 | 45 | -82.2 | -82.1 | -48.1 | -48 | | | |
| 5 | -0.2 | 45 | -78.1 | -78 | -51.2 | -51.1 | | | |
| 6 | -0.15 | 45 | -73 | -73 | -54 | -53.9 | | | |
| 7 | -0.1 | 45 | -69.3 | -69.2 | -53.5 | -53.4 | 14.2 | 14.2 | |
| 8 | -0.05 | 45 | | | -45.6 | -45.5 | -7.1 | -7 | |
| 9 | 0 | 45 | | | -37.2 | -37.1 | -19.8 | -19.7 | |
| 10 | 0.05 | 45 | | | -45.6 | -45.5 | -7.1 | -7 | |
| 11 | 0.1 | 45 | -69.3 | -69.2 | -53.5 | -53.4 | 14.2 | 14.2 | |
| 12 | 0.15 | 45 | -73 | -73 | -54 | -53.9 | | | |
| 13 | 0.2 | 45 | -78.1 | -78 | -51.2 | -51.1 | | | |
| 14 | 0.25 | 45 | -82.2 | -82.1 | -48.1 | -48 | | | |
| 15 | | | | | | | | | |
| 16 | | | | | | | | | |
| 17 | | | | | | | | | |

Figure 6: Manually assisted band alignment. Example of the manual input of band indices in a CSV file, as displayed in a spreadsheet editor.

### 3.6.5 Challenges

The band alignment algorithm is quite reliable, but there are scenarios that commonly lead to incorrect results, for example:

- The alignment algorithm may misalign states if the two sets consist of almost equidistant sub-bands. Such patterns typically occur in both the conduction and the valence band individually. In order to avoid this condition, choose the diagonalization parameters such that at least the highest valence subband *and* the lowest conduction subband are included at every point in momentum space. The alignment algorithm is then stabilized by the large gap.

- The sets of eigenvalues must be connected and complete, i.e., between the lowest and highest eigenvalues, there must be no missing ones. For a single calculation, this is always true, but if one uses `kdotpy merge` for merging multiple data sets at the same momentum values, make sure that no eigenstates are missing. If the two sets overlap (share at least one eigenvalue, preferably more), then one is on the safe side.

- The band alignment algorithm does not admit real crossings. Even when real crossings are expected in the spectrum, e.g., between states that have different values of a conserved quantum number, they are rendered as anticrossings. This also affects derived quantities: For example, density of states may be inaccurate around the absent crossing points. This can partially be mitigated by increasing the momentum or magnetic-field resolution.

## 3.7 Density of states

### 3.7.1 Motivation

The density of states plays an essential role in many physical quantities observed in experiment. The position of the Fermi level determines whether the material behaves as an insulator or conductor in transport physics. In spectroscopy, the occupation of each state determines which optical transitions are visible. The total carrier density is often known from experiments, is used to determine the Fermi energy. The tuning of the Fermi level with one or more gate electrodes is understood most straightforwardly in terms of carrier density, as the latter typically depends linearly on the gate voltage.

The density functions of `kdotpy` calculate the relation between carrier density $n$ and energy $E$, and places the Fermi level at the correct position if the carrier density is provided with the command line argument `cardens`. The relation also allows `kdotpy` to convert **k**- or **B**-dependence plots by putting density $n$ on the vertical axis instead of energy $E$. This is particularly useful in the Landau-level mode, where Landau fans with $n$ on the vertical axis may be compared directly with Landau fan plots from magnetotransport experiments.

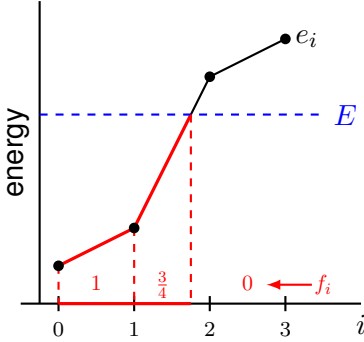

Figure 7: Calculation of the integrated density of states (IDOS) for a single band. The band energies at momenta $k_i$ are extracted from DiagData as the values $e_i$. Interpolate linearly to find a continuous function $e(k)$. For each interval $(i, i+1)$, find the fraction $f_i(E)$ of the interval for which $e(k) < E$. Iterate this procedure over many values of $E$. For holes, subtract $1$ from each $f_i$; in this example, this would yield $f_i - 1 = (0, -\frac{1}{4}, -1)$.

### 3.7.2 Integrated density of states (IDOS)

The key quantity in the density calculations in kdotpy is the integrated density of states (IDOS; equivalent to carrier density), defined as the number of occupied states lying between the charge neutrality point and the Fermi level. The counting of occupied states is based on the principle that each 'mode' (given by its momentum) contributes equally. An isolated, fully occupied band would correspond to one state in a volume of $a^d$, assuming a (hyper)cubic lattice in $d$ dimensions with lattice constant $a$. In momentum space, this state occupies the full Brillouin zone with a volume of $(2\pi/a)^d$. A partially occupied band, which occupies a volume of $V_{\text{occ}}$ in momentum space, thus corresponds to a density of $V_{\text{occ}}/(2\pi)^d$.

In order to calculate the IDOS, we thus need to calculate the occupied volume in momentum space for all bands and sum over them. In order to get density $n(E)$ as function of energy, we choose a discrete array of energy values, which we call the 'energy range'. We sketch the algorithm for the IDOS calculation of a dispersion in $d = 1$ dimension. For each band $b$:

- Extract $e_i = E^{(b)}(k_i)$, the energy dispersion of the band with band index $b$ from DiagData. The index $i = 1, \ldots, n_k$ runs over $n_k$ data points.

- Let $E$ be the Fermi energy. Do a piecewise linear interpolation $e(k)$ of the $e_i$, i.e., drawing straight lines between $(k_i, e_i)$ and $(k_{i+1}, e_{i+1})$. For all intervals $(k_i, k_{i+1})$, determine the fraction of the interval where $e(k) \leq E$. This is given by

$$f_i(E) = \begin{cases} 1 & \text{if } e_i < e_{i+1} \leq E \\ \frac{E - e_i}{e_{i+1} - e_i} & \text{if } e_i \leq E < e_{i+1} \\ 0 & \text{if } E < e_i < e_{i+1} \end{cases} \tag{101}$$

where we have assumed (without loss of generality) that $e_i < e_{i+1}$. See Fig. 7 for an illustration. This step is done by linear_idos_element(), which does this for all energies $E$ in the energy range simultaneously, by clever use of NumPy arrays.

- If the band is hole-like (band index $b < 0$), subtract $1$ from all $f_i$. Thus, the volume below the Fermi energy is counted as $0$ and the volume above it as $-1$. In this way, hole-like densities are automatically counted as being negative.

- Integrate over momentum. For all intervals $(k_i, k_{i+1})$, calculate the integration volume element $dk_i$. In one dimension, the volume element would be $dk_i = k_{i+1} - k_i$ in cartesian coordinates and $dk_i = \pi(k_{i+1}^2 - k_i^2)$ if $k$ is a radial coordinate in polar coordinates. The IDOS contribution for band $b$ is thus

$$n^{(b)}(E) = \frac{1}{(2\pi)^d} \sum_{i=1}^{n_k - 1} f_i(E) dk_i, \tag{102}$$

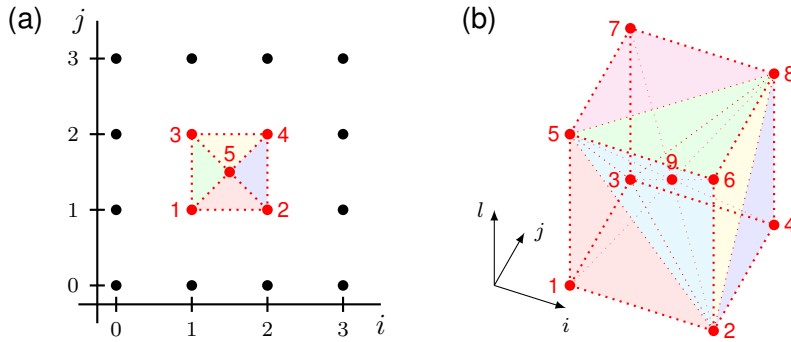

Figure 8: Division of momentum space for two and three dimensions. (a) In two dimensions, the momentum space is triangulated into triangular simplices. To this end, a fifth point is inserted in each elementary square (red) by setting $e_{5,i,j}$ as the average over the four corner points. The four simplices for each square are **125**, **135**, **245**, and **345**. (b) In three dimensions, the space is divided into tetrahedral simplices. A ninth point is inserted in the centre of each elementary cube, by setting $e_{9,i,j,l}$ as the average over the eight corner points. The illustrated division is the set of twelve tetrahedra **1239**, **2349**, **1259**, **2569**, **2489**, **2689**, **1359**, **3579**, **5689**, **5789**, **3489**, and **3789**. Another choice can be obtained by spatial inversion.

where we note that the $f_i(E)$ are still functions of the Fermi energy $E$, implemented as one-dimensional NumPy array. The result is thus also a function of $E$.

To find the total density as function of $E$, we sum over all bands. If desired, the sum may also be restricted to either electron-like or hole-like bands.

The function `loc_int_dos_by_band()` in `density/base.py` takes care of obtaining the $f_i$, where Eq. (101) is implemented in `density/elements.py`. The integration over momentum is done by `int_dos_by_band()` in `density/base.py`. The result is a one-dimensional array, where the elements encode the IDOS values $n(E_i)$, where $E_i$ are the values in the energy range. The summation over bands may also be skipped in order to find the contributions by band; this result is represented by a `dict` instance where the keys are the band labels $b$ and the values are the one-dimensional arrays $n^{(b)}(E_i)$.

For higher dimensions, an additional intermediate step is required. In order to calculate $f_i$ in two or three dimensions, we need to divide the space into triangular or tetrahedral simplices. In two dimensions, for an elementary square $[k_{x,i}, k_{x,i+1}] \times [k_{y,j}, k_{y,j+1}]$, take the centre point $k_{5,i,j} = (\frac{1}{2}(k_{x,i} + k_{x,i+1}), \frac{1}{2}(k_{y,j} + k_{y,j+1}))$ and divide the plaquette into four triangles with $k_{5,i,j}$ as one of its vertices, see Fig. 8(a). The band energy at $k_{5,i,j}$ is determined by interpolation,

$$e_{5,i,j} = \frac{1}{4}\left(E^{(b)}(k_{x,i}, k_{y,j}) + E^{(b)}(k_{x,i+1}, k_{y,j}) + E^{(b)}(k_{x,i}, k_{y,j+1}) + E^{(b)}(k_{x,i+1}, k_{y,j+1})\right). \qquad (103)$$

Labelling each simplex shape as $t = 1, 2, 3, 4$ (corresponding to the four colours in Fig. 8(a)), we find all $f_{t,i,j}(E)$ with a function analogous to Eq. (101), given as Eq. (A.2) in Appendix A.2 and implemented in `triangular_idos_element()`. The integral over momentum involves volume elements $dk_{t,i,j}$ for all simplices,

$$n^{(b)}(E) = \frac{1}{(2\pi)^d} \sum_{t=1}^{4} \sum_{i=1}^{n_{k_x}-1} \sum_{j=1}^{n_{k_y}-1} f_{t,i,j}(E) dk_{t,i,j}. \qquad (104)$$

For cartesian coordinates, $dk_{t,i,j} = \frac{1}{4}\Delta k_x \Delta k_y$. For polar coordinates, we take the approximate expression $dk_{t,i,j} = \frac{1}{4}\overline{k_r}\Delta k_r \Delta k_\phi$, where $\overline{k_r}$ is the average of the radial coordinates of the three vertices of the triangle defined by $t, i, j$ and $\Delta k_\phi$ is measured in radians.

For three dimensions, an analogous method is used, with each elementary cube $[k_{x,i}, k_{x,i+1}] \times [k_{y,j}, k_{y,j+1}] \times [k_{z,l}, k_{z,l+1}]$ being divided into **12** tetrahedra with the body-centred point $k_{9,i,j,l} = (\frac{1}{2}(k_{x,i} + k_{x,i+1}), \frac{1}{2}(k_{y,j} + k_{y,j+1}), \frac{1}{2}(k_{z,l} + k_{z,l+1}))$ as one of its vertices, see Fig. 8(b). The energy value $e_{9,i,j,l}$ is calculated as the average over all eight vertices of the elementary cube. The method

for finding value $f_{t,i,j,l}(E)$ with a function analogous to Eq. (101) is given as Eq. (A.3) in Appendix A.2 and is implemented in `tetrahedral_idos_element()`. The volume elements are $dk_{t,i,j,l} = \frac{1}{12}\Delta k_x \Delta k_y \Delta k_z$ for cartesian coordinates, $dk_{t,i,j,l} = \frac{1}{12}\overline{k_r}\Delta k_r \Delta k_\phi \Delta k_z$ for cylindrical coordinates, and $dk_{t,i,j,l} = \frac{1}{12}\overline{k_r^2}\,\overline{\sin k_\theta}\,\Delta k_r \Delta k_\theta \Delta k_\phi$ for spherical coordinates, where the overlines denote averages over all four vertices of each tetrahedron.

### 3.7.3 Density in Landau level mode

In Landau level mode, the calculation of IDOS is much simpler due to the absence of the momentum coordinates. For each magnetic field value $B = B_z$, apply the following recipe.

- For each band labelled $b$ (full LL mode) or $(n, b)$ (symbolic LL mode), extract the energy values $E^{(b)}(B)$ from `DiagData`.

- Determine the charge neutrality point $E_{\text{CNP}}(B)$. For full LL mode, this is simply the centre of the gap between bands $-1$ and $1$. For symbolic LL mode, the band labels $(n, b)$ are first converted to a single *universal band index* $u$; $E_{\text{CNP}}(B)$ then lies between the states with $u = -1$ and $u = 1$. (Details on the universal band index in Appendix A.3.)

- Let the Fermi energy be $E$. Count the number of states with energies between $E_{\text{CNP}}(B)$ and $E$.

- Multiply by the LL degeneracy factor $eB/2\pi\hbar$.

This algorithm is applied for each magnetic field value independently, and yields a function of energy $E$ (represented by an array) for each $B$. The result can be interpreted as 'local' integrated density of states. The function that takes care of this calculation is `loc_int_dos()`, with the determination of the charge neutrality points being handled by the member function `DiagData.get_e_neutral()` of the `DiagData` class.

### 3.7.4 Data structure

The IDOS data is stored in an instance of the class `DensityData`. This class has three important array-like attributes:

- `densdata`: An array of at least one dimension, holding the IDOS values.

- `ee`: A one-dimensional array of the energy values.

- `xval`: Optionally, an array or `VectorGrid` object that holds the momentum or magnetic field values for local IDOS data.

The shape of `densdata` must match those of `xval` and `ee`, i.e., the condition

```
densdata.shape == (*xval.shape, *ee.shape)
```

must evaluate to `True`, with `xval.shape` being replaced by `()` if `xval` is `None`. The class also remembers the dimensionality (as `kdim`) and whether or not Landau levels are considered (as the boolean value `ll`).

The class has several member functions for common operations on the integrated density of states. The most important ones are:

- `integrate_x()`: Integrate over the momentum coordinates. NOTE: Technically, magnetic field values can also be integrated over, but there is no physically relevant reason to do so.

- `get_idos()`: Return the IDOS data, scaled to the desired units.

- `get_dos()`: Return the density of states, defined as the energy derivative of the IDOS. Scale the values to the desired units, like with `get_idos()`.

- `get_validity_range()`: Estimate the energy range where the IDOS is valid, i.e., in which energy range we can be reasonably sure that no states have been overlooked due to the finite momentum range. This is done by considering the dispersion energies and their derivatives at the edge of the momentum range. For example, if the momentum range is $[k_{min}, k_{max}]$ and the dispersion $E(k)$ approaches $E(k_{max})$ from below (with $dE/dk > 0$), then we can infer that there are states at $E > E(k_{max})$ for $k > k_{max}$, which are not considered for the DOS calculation. Thus, the upper bound of the validity range must be $\leq E(k_{max})$. The upper and lower limits of the validity range are determined by iterating this algorithm over all bands. The validity range may be empty or 'negative' (when the upper bound lies below the lower bound) in some cases. The result is printed on the terminal and is visualized in the DOS and IDOS plots by shading of the regions outside the validity range.

- `idos_at_energy()`: Get the IDOS value at a given energy. This is done by interpolation along the energy axis.

- `energy_at_idos()`: Get the energy value at a given IDOS $n_{target}$ (carrier density). This is done by solving $E$ from the equation $n(E) = n_{target}$, where $n(E)$ is a linearly interpolated function from the IDOS data (`densdata` versus `ee`) stored in the `DensityData` instance.

There are several additional attributes and member functions related to *special energies*, for example the Fermi level at zero density and at the desired density, and the charge neutrality point.

### 3.7.5 Units and scaling

The internal density units for `DensityData.densdata` are always $\mathbf{nm}^{-d}$, where $d$ is the dimensionality. In order to be able to extract quantities proportional to DOS and IDOS values expressed in a user-preferred set of units, the `DensityData.scale` may be set. If this attribute is set as an instance of the `DensityScale` class, functions like `DensityData.get_idos()` and `DensityData.get_dos()` automatically return the values in the appropriate units.

The role of the `DensityScale` class is to store the desired quantity and unit, and to scale the exponent $p$ automatically, such that the values can be written as $x \times 10^p$ with $x$ having a 'convenient' magnitude. The default quantity is simply the IDOS as defined before. The area or volume in momentum space is related by a factor of $(2\pi)^d$. Charge density has the same value as IDOS, but with units of $e\,\mathbf{nm}^{-d}$ instead of $\mathbf{nm}^{-d}$. For the units, one may choose $\mathbf{nm}$, $\mathbf{cm}$, or $\mathbf{m}$ as the units for length. For the extracted IDOS, the values are expressed in units of $10^p\,\mathbf{nm}^{-d}$, $10^p\,\mathbf{cm}^{-d}$, or $10^p\,\mathbf{m}^{-d}$, with $p$ set to a reasonable value for the chosen units. For DOS, the appropriate unit is obtained by multiplying with $\mathbf{meV}^{-1}$. The `DensityScale` class provides the functions `DensityScale.qstr()` and `DensityScale.unitstr()` for formatting the quantity and unit as a string for use in output files (text and graphics). For convenience, `DensityData` provides these member functions as well.

### 3.7.6 Integrated observables

A special application of the density functions is the notion of integrated observable. In the summation over the bands [similar to Eq. (102)], we include the value of an observable $O$ as an additional factor. The integrated observable is defined by the sum $O(E) = \sum_b O^{(b)}(E)$ over the contributions from all bands,

$$O^{(b)}(E) = \frac{1}{(2\pi)^d} \sum_{i=1}^{n_k} f_i(E) O^{(b)}(k_i) dk_i, \tag{105}$$

where $k_i$ are the momentum values of the grid and $O^{(b)}(k_i)$ is the observable value of band $b$ at $k_i$. Note that unlike the IDOS calculation, we sum over the momentum values themselves, not the intervals between them. The implementation (function `integrated_observable()`) returns an `IntegratedObservable` class derived from the `DensityData` class.

The integrated observable $O(E)$ as function of energy can be transformed into the corresponding function $\tilde{O}(n)$ as function of density by means of a *pushforward* transformation over the function defining density $N(E)$ as function of energy. The result is $\tilde{O} = O \circ N^{-1}$, where $\circ$ denotes composition of functions and $N^{-1}$ is the inverse function of $N$. The function $N^{-1}$ takes a density value $n$ and calculates the corresponding energy $E$. Then, $O$ acting on $E$ yields $O(E)$. Thus, one may think of $\tilde{O}$ as the function $n \mapsto O(E(n))$, with $E(n) = N^{-1}(n)$.

This transformation is implemented as `IntegratedObservable.pushforward()`, which determines $\tilde{O}$ by inverse interpolation using $O(n)$ defined from the class instance itself and the function $N(E)$ (a `DensityData` instance) as the first argument. The second argument is the array of density values $n$ at which $\tilde{O}(n) = O(N^{-1}(n))$ is evaluated. In the most simplified form, the essence of the implementation is[3]

```
def pushforward(self, other, values):
    return np.interp(values, other.densdata, self.densdata)
```

where `self` represents $O(E)$ and `other` represents $N(E)$. In this example, we assume that `self` and `other` are defined on the same array of energies and that the attribute `xval` is `None` for both.

### 3.7.7 Broadening

In essence, the density of states (DOS) is defined as the size (length, area, volume) of the level sets of the dispersion $E(\mathbf{k})$. From the principles of minimal energy and Pauli exclusion, electronic states fill up to the Fermi energy, such that the number of carriers equals the integral over the DOS up to the Fermi energy. This picture assumes zero temperature, so that the occupation function $F(E)$ has a hard cutoff at the Fermi energy.

For finite temperature $T$, the occupation function smoothly goes from $1$ to $0$ in an energy window of width proportional to $k_B T$. The effective density of carriers in the product of the DOS $g(E)$ and the smooth occupation function $F(E)$. Due to the smoothness of this function, an electron in quantum state $E$ can also be found at energies nearby. The probability distribution for finding the particle at energy $E$ is $f(E) = dF(E)/dE$, which is centred around the eigenenergy of the state. The fact that this distribution is somewhat spread out explains the term *broadening*. Other probabilistic phenomena such as disorder can broaden also the density of states. These can be treated on the same footing as the broadening due to finite temperature.

Broadening is applied essentially as the convolution of the density of states $g(E)$ with the broadening function $f(E)$.

$$d(E) = (f * g)(E) = \int_{E_0}^{E} f(E - E')g(E')dE' \tag{106}$$

In `kdotpy`, the central quantity is the integrated density of states $G(E) = n(E)$. The broadened version $D(E)$ is similarly obtained as the convolution

$$D(E) = (f * G)(E) = \int_{E_0}^{E} f(E - E')G(E')dE' = (F * g)(E), \tag{107}$$

where the latter equality is a property of convolution in general.

Since the energy values are defined on a grid, we need to take special care of numerical inaccuracies. In particular, we need to make sure that the integral over the broadening function equals 1 exactly. Note that if we simply take some values $f(E_j)$ ($E_j = j\delta E$ with $j$ being integers) the (Riemann) sum $\sum_j f(E_j)\delta E$ that approximates the integral $\int f(E)dE$ may deviate from 1, especially if the broadening parameter is comparable in size to $\delta E$. In order to prevent this from happening we rather use the discrete derivative of $F(E) = 1 - \int^{E} f(E')dE$. The reason for '1 minus' is to make the interpretation of $F(E)$ to be the *occupation functions* associated with the *broadening kernel* $f(E)$.

The broadening functions and their occupation functions implemented in `kdotpy` are listed in Appendix B.6. In general, the command line argument is of the form `broadening w0 type dep`, where $w_0$ is the width parameter and `type` determines the type (shape of the broadening function), i.e., `fermi`, `thermal`, `gauss`, or `lorentz`). The dependency argument `dep` determines the dependence of the width parameter on $k$ or $B$; for example, when `sqrt` is used with a magnetic-field dependence, the broadening width applied to the IDOS is $w(B) = w_0\sqrt{B}$ with $B$ in T.

Generally, some of the arguments after `broadening` may be omitted, in which case the following defaults are used:

- If `broadening` is not given at all, do not apply broadening. NOTE: `dostemp T` (where $T$ is a temperature in K) qualifies as broadening argument and is equivalent to `broadening T thermal`.

---

[3]This function is currently a member of the parent class `DensityData`.

- If `broadening` is given without type, then use the default type: `thermal` in dispersion mode, `gauss` in LL mode.

- If `broadening thermal` is given without width parameter (temperature), use the temperature given by `temp` T. This defaults to $T = 0$ if `temp` is absent.

- If the dependence argument is omitted, broadening type `gauss` assumes `sqrt` dependence, all others use `const`.

Finally, compound broadening might be achieved with multiple broadening arguments. The broadening arguments are iteratively applied to the density of states. For example, if one gives `broadening gauss 1.0 broadening thermal 0.5`, first the Gaussian broadening is applied to the DOS, then the thermal broadening. If the broadening kernels are denoted $f_1$ and $f_2$, then the result is

$$d(E) = (f_2 * (f_1 * g))(E) = ((f_2 * f_1) * g)(E) \tag{108}$$

By virtue of associativity of convolution, the combined broadening kernel is $f_2 * f_1$, which generalizes to $f_n * \cdots * f_1$ if one combines $n$ broadening kernels.

Since convolution is commutative, $f_2 * f_1 = f_1 * f_2$, the order of applying the broadening kernels is irrelevant *in principle*. Due to limitations of the numerics—convolution is calculated by numerical integration on a finite interval with finite resolution—changing the order might lead to small numerical differences. Some combinations of broadening may be simplified analytically, for example the convolution of two Gaussian distribution functions with standard deviations $\sigma_1$ and $\sigma_2$ is a Gaussian distribution function with standard deviation $\sigma = \sqrt{\sigma_1^2 + \sigma_2^2}$. (For Lorentzians, the widths add up, $\gamma = \gamma_1 + \gamma_2$, and a combination of Fermi distributions cannot be simplified analytically.) These simplifications may be useful to perform manually prior to input, as to avoid unnecessary numerical errors.

### 3.7.8  Density as function of $z$

Knowledge of the spatial distribution of charge $\rho(z)$ given a certain carrier density can aid significantly in understanding the observable behaviour of a device. It is also essential for the self-consistent Hartree algorithm (see Sec. 3.11) in order to find the electric potential induced by the charges in the material.

The idea of calculating $\rho(z)$ is analogous to the methods for the IDOS and the integrated observable. We insert the probability density appropriate for the interval $[k_i, k_{i+1}]$ into Eq. (102): for each band, we calculate

$$\rho^{(b)}(z, E) = \frac{1}{(2\pi)^d} \sum_{i=1}^{n_k-1} \frac{1}{2} \left( |\psi_{k_i}^{(b)}(z)|^2 + |\psi_{k_{i+1}}^{(b)}(z)|^2 \right) f_i(E) dk_i. \tag{109}$$

and we sum over all bands, $\rho(z, E) = \sum_b \rho^{(b)}(z, E)$ (or if desired, over either electrons or holes only). In the implementation of `density_energy()`, the probability densities $|\psi_{k_i}^{(b)}(z)|^2$ are extracted from the eigenvectors, summed over the orbital degree of freedom, and supplied as an argument to `int_dos_by_band()`. The result is a two-dimensional array that encodes $\rho(z, E)$ over an array of $z$ values and an array of energy values $E$. The function `densityz()` evaluates $\rho(z, E)$ for a specific energy $E$ and returns a one-dimensional array. The latter function is used by the self-consistent Hartree method, discussed in Sec. 3.11.

In two dimensions, the algorithm is analogous, but the interpolated probability density in (109) is replaced by a weighted sum over the four corner points of each momentum space plaquette,

$$\sum_{\nu=1}^{4} c_\nu |\psi_{k_{i_\nu}}^{(b)}(z)|^2, \tag{110}$$

and the sum over momenta is replaced by the simplices over triangles in the triangulated lattice [cf. Eq. (104)] One can choose between equal weights $c_\nu = (\frac{1}{4}, \frac{1}{4}, \frac{1}{4}, \frac{1}{4})$ for all triangles or adjusted weights taking into account the shape of the triangle, i.e., $c_\nu = (\frac{5}{12}, \frac{5}{12}, \frac{1}{12}, \frac{1}{12})$ for the triangle with vertices **125**, etc.

In Landau level mode, the summation over momentum is absent, and we sum all contributions $|\psi_B^{(b)}(z)|^2$ for all states with eigenvalues $E^{(b)}$ between the charge neutrality point and the Fermi level. Like the IDOS calculation, the Landau level degeneracy factor $eB/2\pi\hbar$ is taken into account as well.

The $z$-dependent density $\rho(z)$ is always calculated such, that $\int \rho(z)dz$ is equal to the IDOS $n$ at the same energy. We note that until here, we have assumed that $\rho(z)$ is uniformly zero at the charge neutrality point, but this is not necessarily the case: In fact, at zero total density ($n = 0$), the system need only be neutral on average ($\int \rho(z)dz = 0$), but the local charge density $\rho(z)$ may vary. In kdotpy, one can define a density offset $\rho_{\mathrm{offset}}(z)$ at zero density, or use a different reference point. An in-depth analysis of physically sensible choices of the reference density is outside the scope of this work and will be presented elsewhere.

## 3.8 Optical transitions

### 3.8.1 Background

Optical spectroscopy is a powerful experimental technique for characterization and investigation of band structures of semiconductors. By analysing the optical transitions between two states, conclusions about e.g. the band gap, the band order, etc. can be drawn. Simulating these transitions and their dependence on e.g. magnetic field in kdotpy and, thus, being able to easily compare them to experimental data, provides a great tool to improve the modelling capabilities of kdotpy and to validate which approximations/symmetries suffice for specific system configurations. Ultimately, this refines the predictions that can be made with kdotpy with regards to band structure engineering, etc.

We achieve this by including the contributions of the electromagnetic (EM) field, that drives optical transitions, as a separate vector potential $\mathbf{A}_{\mathrm{EM}}$ in the Hamiltonian

$$H_{\mathrm{pert}} = \frac{(\mathbf{p} + e\mathbf{A}_0 + e\mathbf{A}_{\mathrm{EM}})^2}{2m} + V \equiv H + H_{\mathrm{EM}} \tag{111}$$

with

$$H = \frac{(\mathbf{p} + e\mathbf{A}_0)^2}{2m} + V \quad \text{and} \quad H_{\mathrm{EM}} = \frac{e\mathbf{A}_{\mathrm{EM}} \cdot \mathbf{p}}{m}. \tag{112}$$

The identity $\mathbf{p} \cdot \mathbf{A} = \mathbf{A} \cdot \mathbf{p}$, which is always valid for divergence free fields of EM waves, was used and the energy offset terms $\mathbf{A}_{\mathrm{EM}}^2 + 2\mathbf{A}_0 \cdot \mathbf{A}_{\mathrm{EM}}$ were neglected to get to the second line. Using the dipole approximation $\mathbf{q} \to \mathbf{0}$, neglecting any spatial contribution to the phase factor, the vector potential of EM waves can be expressed as

$$\mathbf{A}_{\mathrm{EM}} = \frac{\mathbf{E}}{2\omega}\left(e^{i\omega t} + \mathrm{c.c.}\right) \tag{113}$$

For the sake of simplicity, we drop the time varying phase factors (these will become relevant in Fermi's Golden Rule) and we assume an EM wave travelling in $z$ direction. The operator product can then be written as

$$\mathbf{A}_{\mathrm{EM}} \cdot \mathbf{p} \propto \mathbf{E} \cdot \mathbf{p} = E_x p_x + E_y p_y = \frac{1}{\sqrt{2}}\left(E_+ p_+ + E_- p_-\right) \tag{114}$$

where $E_\pm = \frac{1}{\sqrt{2}}\left(E_x \mp iE_y\right)$ and $p_\pm = p_x \pm ip_y$ as circular polarization basis.

The optical transition rate $\Gamma_{i \to f}$ from initial state $|\psi_i\rangle$ to final state $|\psi_f\rangle$ can be calculated using Fermi's Golden Rule

$$\Gamma_{i \to f} = \frac{2\pi}{\hbar}\left|M^{(fi)}\right|^2 \delta\left(\hbar\omega_{fi} - \hbar\omega\right) \tag{115}$$

with transition matrix element

$$M^{(fi)} = \langle\psi_f|H_{\mathrm{EM}}|\psi_i\rangle = \frac{e\mathbf{E}}{2\omega}\langle\psi_f|\mathbf{v}|\psi_i\rangle \tag{116}$$

The omitted time dependence of the electric fields is responsible for the $\delta$ distribution, assuring energy conservation for absorption and emission of photons.

### 3.8.2 Evaluation of transition matrix elements

The transition matrix elements given by (116) are calculated by get_transitions() (symbolic mode) and get_transitions_full() (full mode) in transitions.py. This calculation requires the eigenvectors, and therefore this is done immediately after diagonalization of the Hamiltonian, before the eigenvectors are discarded from memory.

The velocity operator is evaluated by using the Ehrenfest theorem

$$v_x = -\frac{\mathrm{i}}{\hbar}\left[x, H_{\mathrm{pert}}\right] \approx -\frac{\mathrm{i}}{\hbar}\left[x, H\right] = \frac{\mathrm{d}H}{\mathrm{d}p_x} = \frac{1}{\hbar}\frac{\mathrm{d}H}{\mathrm{d}k_x} \tag{117}$$

and similar for $y$. Let us relabel this operator $O_{x/y} = v_{x/y}$ and define the corresponding operators in circular basis

$$O_\pm = O_x \pm \mathrm{i}O_y = \frac{2}{\hbar}\frac{\mathrm{d}H}{\mathrm{d}k_\mp} \tag{118}$$

using $k_\pm = k_x + \mathrm{i}k_y$ and $\frac{\mathrm{d}}{\mathrm{d}k_\pm} = \frac{1}{2}\left(\frac{\mathrm{d}}{\mathrm{d}k_x} \mp \mathrm{i}\frac{\mathrm{d}}{\mathrm{d}k_y}\right)$. Independent of the used LL mode, the derivation of the basic LL Hamiltonian $H$ in $k_\mp$ is performed in its symbolic representation. After derivation, the numerical values of the operator are evaluated for LLs $n$ and $n+1$. Then, the matrix product

$$|O_\pm^{fi}|^2 = \left|\langle\psi_f|O_\pm|\psi_i\rangle\right|^2 \tag{119}$$

is evaluated for all combinations of $\langle\psi_f|$ and $|\psi_i\rangle$. This evaluation is done efficiently as a sequence of two matrix multiplications on NumPy matrices

```
opeivec1T = op @ eivec1T
ov = eivec2T_H @ opeivec1T
ov2 = np.real(np.abs(ov)**2)
```

where op represents $O_\pm$, `eivec1T` holds the eigenvectors $|\psi_i\rangle$ as columns, and `eivec2T_H` holds the conjugate eigenvectors $\langle\psi_f|$ as rows. The result is filtered, by discarding all transitions with an amplitude below a threshold `ampmin`, schematically

```
sel = (ov2 >= ampmin)
ov2_filtered = ov2[sel]
```

which yields the filtered transitions as the one-dimensional array `ov2_filtered`. In the same manner, one-dimensional arrays containing the energies and Landau level index of initial and final state are constructed. These arrays are stored together in a `TransitionsData` instance (class definition in `transitions.py`) inside `DiagDataPoint.transitions`, see `ModelLL._post_solve()` in `models.py`, for further analysis later on.

Note that we only calculate $|O_+^{fi}|^2$ in `kdotpy`, to reduce calculation time. Because of the relation $\langle\psi_f|O_+|\psi_i\rangle = \langle\psi_i|O_-|\psi_f\rangle^\dagger$ and the fact that we only keep the absolute square of these matrix elements, we can reinterpret the emission matrix elements of the $|O_+^{fi}|^2$ matrix (negative energy difference from final to initial state) as absorption matrix elements of $|O_-^{fi}|^2$.

### 3.8.3   Postprocessing and filtering

In `postprocess.py` the function `transitions()` is responsible for any further analysis of optical transitions and corresponding spectra. Depending on input parameters (e.g. using `cardens`) the calculated transitions are filtered by the method `filter_transitions()` of a `DiagData` instance. If no filtering is performed, all transitions are plotted and written into a table. For filtered data, plotting can be suppressed to speed up code execution, but the data will always be written into a table. Further, also only for filtered transitions, other quantities related to optical transitions can be optionally calculated, e.g., rotation and ellipticity. This heavily impacts calculation time.

To filter transitions the `TransitionsData` instance method `at_energy()` is called, which only returns transitions that cross the given energy level, i.e., the initial state is (partially) occupied while the final state is (partially) unoccupied, as a new `TransitionsData` instance. If no broadening is used (delta-peak shaped energy states), every transition is checked if and only if one of the two involved states is above while the other is below the given energy. Otherwise, the implemented occupation function for the type of broadening is used to filter transitions (e.g., Fermi distribution for thermal broadening). The occupation factor is

$$P_{i\to f} = \left|f(E_i) - f(E_f)\right| \tag{120}$$

where $f(E_{i/f})$ is the occupation function for the given broadening type evaluated at the energy of the initial/final state, respectively. Only transitions for which $\left|\langle\psi_f|O_\pm|\psi_i\rangle\right|^2 \cdot P_{i\to f} \geq A_{\min}$ will be kept as

filtered transition, where $A_\text{min}$ can be adjusted by the command line argument `transitions` followed by a floating point value or by the configuration option `transitions_min_amplitude` (default value is **0.01**).

### 3.8.4 Output (tables and plots)

Plotting and saving of transition data is performed with the same routines for both unfiltered and filtered transition data, using different file names (`transitions-all...` vs. `transitions-filtered...`). These functions can be found in `ploto/auxil.py` and `tableo/auxil.py` and are both called `transitions()`.

Only a single quantity is plotted as transition plot. By using the configuration option `plot_transitions_quantity` (valid choices see below) the user can choose which quantity is plotted, the default is `rate`. Independent of choice, the amplitude of the chosen quantity is colour-coded and plotted onto a $B$ vs. $\Delta E$ grid.

Contrary to the plot output, as many quantities as possible are saved into the csv file. Depending on input and material parameters, some may not be possible to calculate. Additional quantities that will always be saved to the csv, but are not available for plotting, are magnetic field values, initial/final LL indices and initial/final band indices. The remaining quantities mostly overlap with the plot quantities and are given in the following list:

- `deltae`: The energy difference $\Delta E$ between initial and final state in meV.

- `freq`: The corresponding frequency for the energy difference $\Delta E$ in THz.

- `lambda`: The corresponding wavelength for the energy difference $\Delta E$ in μ**m**.

- `amplitude`: The absolute-squared transition matrix element $|O_\pm^{fi}|^2 \cdot \delta(\hbar\omega_{fi} - \hbar\omega)$ [see Eq. (119); including delta distribution from Fermi's Golden Rule] in $\text{nm}^2\,\text{ns}^{-2}\,\text{meV}^{-1}$.

- `occupancy`: The occupation factor $P_{i\to f}$ [see Eq. (120)], dimensionless. (Only included if occupancies can be calculated. In this case, also the LL degeneracy factor $G = eB/2\pi\hbar$ is included.)

- `rate`: Calculates a transition rate density per electric field intensity,

$$R = \frac{\pi}{4\hbar\omega_{fi}^2} \cdot |O_\pm^{fi}|^2 \cdot G \cdot P_{i\to f} \tag{121}$$

in $\text{ns}^{-1}\,\text{mV}^{-2}$, taking LL degeneracy $G$ and occupancy into account.

- `absorption`: Local 2D absorption coefficient $\alpha$ as in $I(d) = e^{-\alpha} \cdot I(0)$

$$\alpha = \frac{1}{\epsilon_0}\frac{2}{cn}\hbar\omega_{fi}R \tag{122}$$

in ‰, where $R$ is the transition rate per electric field intensity, with velocity of light $c$ and refractive index $n$. (Only included if the refractive index of the QW layer is known. In this case, also a signed absorption coefficient is included, where the sign indicates which circular polarization basis is absorbed.)

### 3.8.5 Outlook on polarimetry spectra

Expanding on optical absorption spectroscopy, polarimetry experiments investigate the polarization state of the EM wave after interaction with a sample. The generally elliptic polarization can be described in terms of a complex angle. For the above mentioned applications, the imaginary part of this complex angle is of great interest, which can be translated into an ellipticity $\epsilon_F$. The ellipticity describes the ratio of minor $b$ to major axis $a$ by $\tan\epsilon_F = b/a$ and is a measure for the shape of the ellipse, where the sign indicates the pseudo-spin of the EM wave. Therefore, it not only carries information on the absorption amplitude but also on the relative absorption difference between both circular polarization modes, which allows observing a change in orbital composition of states and thus, achieving further insight into the band structure.

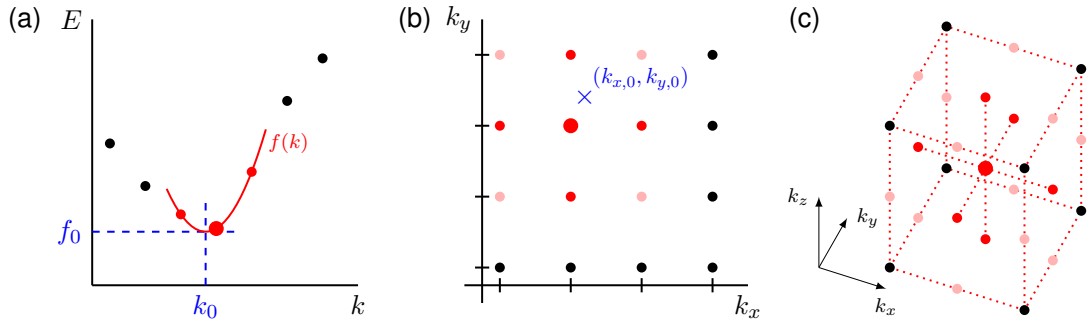

Figure 9: Extrema solvers in 1, 2, and 3 dimensions. (a) The three-point extrema solver for one dimension fits the function $f(k) = f_0 + c(k - k_0)^2$ to three consecutive data points. The location of the minimum in momentum space is $k_0$, which may be between the grid points. The local minimum has energy $f_0$. (b) In two dimensions, a nine-point extrema solver is used. The fit function [Eq. (125)] considers the energy values at the five red points [central point $(k_{x,i}, k_{y,j})$ highlighted with larger size]. The four pink points are considered partially, i.e., only some linear combinations of the dispersion values, see Appendix A.5. The location $(k_{x,0}, k_{y,0})$ of the minimum is generally between the grid points. (c) In three dimensions a nineteen-point extrema solver is used. Seven points (red) are considered fully, twelve more points (pink) are treated partially. The corner points (black) are not considered at all.

Knowledge of the relative absorption $A_\pm$ allows direct access to the ellipticity $\epsilon_F$, by $\tan(\epsilon_F) = \frac{E_+ - E_-}{E_+ + E_-}$, where $E_\pm \propto \sqrt{I_\pm(d)}$. Assuming $I_+(0) = I_-(0)$ and using $A_\pm = \frac{I_\pm(d)}{I_\pm(0)} = \exp(-\alpha_\pm)$ we get

$$\epsilon_F = \arctan\left(\frac{\sqrt{A_+} - \sqrt{A_-}}{\sqrt{A_+} + \sqrt{A_-}}\right) = \arctan\left(\frac{\exp(-\alpha_+/2) - \exp(-\alpha_-/2)}{\exp(-\alpha_+/2) + \exp(-\alpha_-/2)}\right) \tag{123}$$

At the time of publication of this article, only the absorption coefficients are calculated by `kdotpy`, ellipticity may be added at a later stage in development. Until then, users may use the csv output to calculated the ellipticity on their own. Note that these calculated ellipticity spectra are delta peak transitions. In experimental data these transitions always have finite linewidth. This can be accounted for by broadening all transitions by Cauchy-Lorentz distributions, after calculating them with `kdotpy`.

## 3.9 Other postprocessing functions

### 3.9.1 Extrema

Local extrema in the band dispersions are important features for characterizing the band structure. In `kdotpy`, extrema analysis is performed if the command-line argument `extrema` is given. The algorithm relies on the notion of bands, hence the band indices are essential. The extrema analysis does not just find the local extrema in the calculated eigenvalues, but also does a quadratic interpolation to localize each extremum more precisely, and to determine the effective mass as the inverse of the second derivative. Each band extremum is stored in a light-weight class `BandExtremum` that contains the momentum value, energy value, whether minimum or maximum, and the effective mass. The results are displayed as standard output and written to a csv file. The extrema are also shown in two-dimensional dispersion plots.

In one dimension, the following algorithm is applied to each band $b$:

- Extract the dispersion $e_i = E^{(b)}(k_i)$ as function of the momenta $k_i$ on the grid.

- Scan over all $i = 2, \ldots, n_k - 1$. If a triplet $(e_{i-1}, e_i, e_{i+1})$ satisfies $e_{i-1} > e_i$ and $e_{i+1} > e_i$, label it as a minimum; if $e_{i-1} < e_i$ and $e_{i+1} < e_i$, label it as a maximum.

- For each minimum and maximum, apply `three_point_extremum_solver()`. This function takes the points $(k_{i-1}, e_{i-1})$, $(k_i, e_i)$, and $(k_{i+1}, e_{i+1})$, and calculates the coefficients $f_0$, $k_0$, and

$c$ of the quadratic function $f(k) = f_0 + c(k - k_0)^2$ that passes through these points. Assuming that the momentum values are spaced evenly, $k_{i+1} - k_i = k_i - k_{i-1}$, we have

$$
\begin{aligned}
c &= \frac{e_{i+1} - 2e_i + e_{i-1}}{2(k_i - k_{i-1})^2}, \\
k_0 &= k_i - \frac{e_{i+1} - e_{i-1}}{2(k_{i+1} - k_{i-1})c}, \\
f_0 &= e_i - c(k_i - k_0)^2.
\end{aligned}
\tag{124}
$$

The coefficient $k_0$ is the momentum position of the extremum and $f_0$ is the energy value, see Fig. 9(a) for an illustration. The second derivative is equal to $2c$, from which we determine the effective mass $m_{\mathrm{eff}} = -\hbar/m_0 c$, where $m_0$ is the bare electron mass.

If the momentum range starts or ends at $k = 0$, it is automatically extended by reflection, so that extrema at zero are found as well.

In two dimensions, we use a similar method to detect extrema at the grid points $(k_{x,i}, k_{y,j})$. The location, energy, and band mass of the extremum are determined with the `nine_point_extremum_solver()`, that tries to fit the quadratic equation

$$
f(k_x, k_y) = f_0 + a(k_x - k_{x,0})^2 + b(k_y - k_{y,0})^2 + c(k_x - k_{x,0})(k_y - k_{y,0})
\tag{125}
$$

to the energies at the nine points defined by indices $\{i-1, i, i+1\}$ and $\{j-1, j, j+1\}$ in the $k_x$ and $k_y$ direction, respectively. There are only six unknowns, namely $f_0$, $(k_{x,0}, k_{y,0})$, and $(a, b, c)$, for nine input variables. Of the four corner point values $e_{i\pm1,j\pm1}$, only the linear combination $e_{i+1,j+1} - e_{i-1,j+1} - e_{i+1,j-1} + e_{i-1,j-1}$ is considered see Fig. 9(b) and Appendix A.5. The extremum energy is $f_0$, its location $(k_{x,0}, k_{y,0})$, and the band mass follows from the eigenvalues $\lambda_{1,2}$ of the Hessian matrix

$$
\begin{pmatrix} 2a & c \\ c & 2b \end{pmatrix}
\tag{126}
$$

as $m_{\mathrm{eff},a} = -\hbar/m_0 \lambda_a$. Momentum space extension is also used if the range of $k_x$ and/or $k_y$ values ends at zero. For polar coordinates, we use the same algorithm as for cartesian coordinates, except for $k = 0$, where we use a special version of the one-dimensional solver. The band masses are coordinate-system independent in principle: prior to obtaining the eigenvalues, the Hessian matrix is transformed to cartesian coordinates (see Appendix A.5). If one compares the results from dispersion calculations in cartesian and polar coordinates, they are equal up to some numerical differences from the coordinate conversion and from the grid point interpolation.

For three dimensions, we use a `nineteen_point_extremum_solver()`. The inputs are arranged on a $3 \times 3 \times 3$ grid, but the eight corner points are ignored. The fitted function is

$$
f(k_x, k_y, k_z) = f_0 + \begin{pmatrix} k_x' & k_y' & k_z' \end{pmatrix} \begin{pmatrix} 2a & d & e \\ d & 2b & f \\ e & f & 2c \end{pmatrix} \begin{pmatrix} k_x' \\ k_y' \\ k_z' \end{pmatrix},
\tag{127}
$$

where $(k_x', k_y', k_z') = (k_x, k_y, k_z) - (k_{x,0}, k_{y,0}, k_{z,0})$. The number of unknowns is ten, i.e., the energy $f_0$, the location $(k_{x,0}, k_{y,0}, k_{z,0})$ and the six independent entries $(a, b, c, d, e, f)$ of the Hessian matrix. Like with the nine-point extrema solver, some points are considered partially, see Fig. 9(c) and Appendix A.5. The band masses are derived from the eigenvalues of the Hessian matrix, with coordinate transformations applied for cylindrical and spherical coordinate systems.

### 3.9.2   Wave functions

The band structure scripts (`kdotpy 1d`, `kdotpy 2d`, etc.) have the option to plot the wave functions. This is done with the command line argument `plotwf` and further arguments that determine the plot style and the locations (momenta or magnetic fields) at which the plots are made.

In most cases, `plotwf` will output data files in csv format together with the plots. The csv files typically contain the same data as the plot in the plot PDF. For styles where multiple plots are collected in a single PDF, there will be a separate csv file for each page in the PDF (i.e., for each state), where the file name labels the state by band index, LL index, character, and/or energy. If the configuration

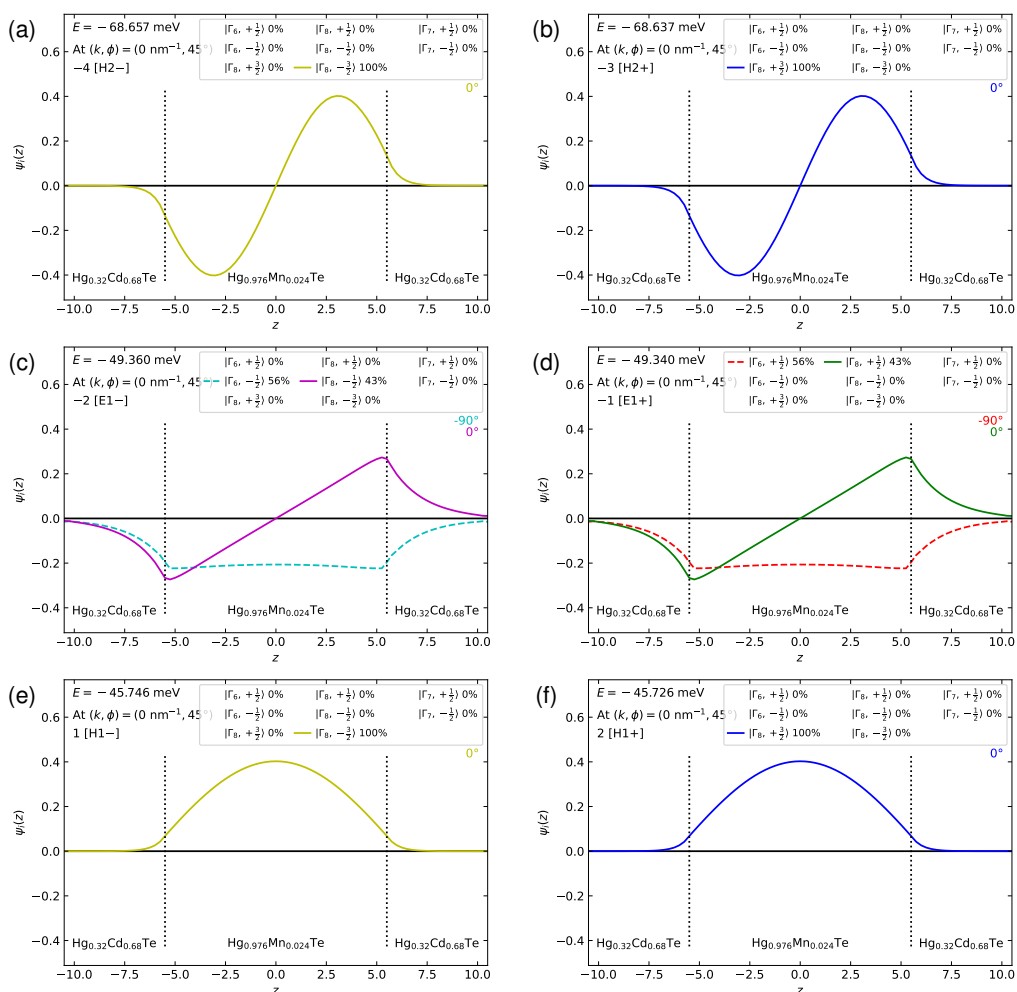

Figure 10: Examples of wave function plots from `kdotpy 2d` with `plotwf separate`. The output is a multipage PDF; here, six pages are shown as panels (a)–(f), each representing a different eigenstate.

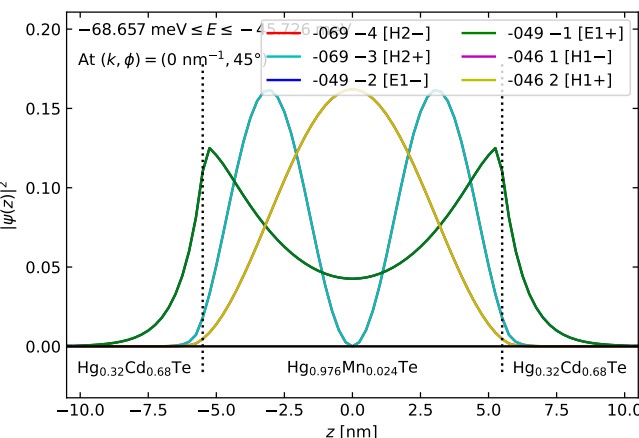

Figure 11: Example of a wave function plot from `kdotpy 1d` with `plotwf together`. The output is a single page PDF.

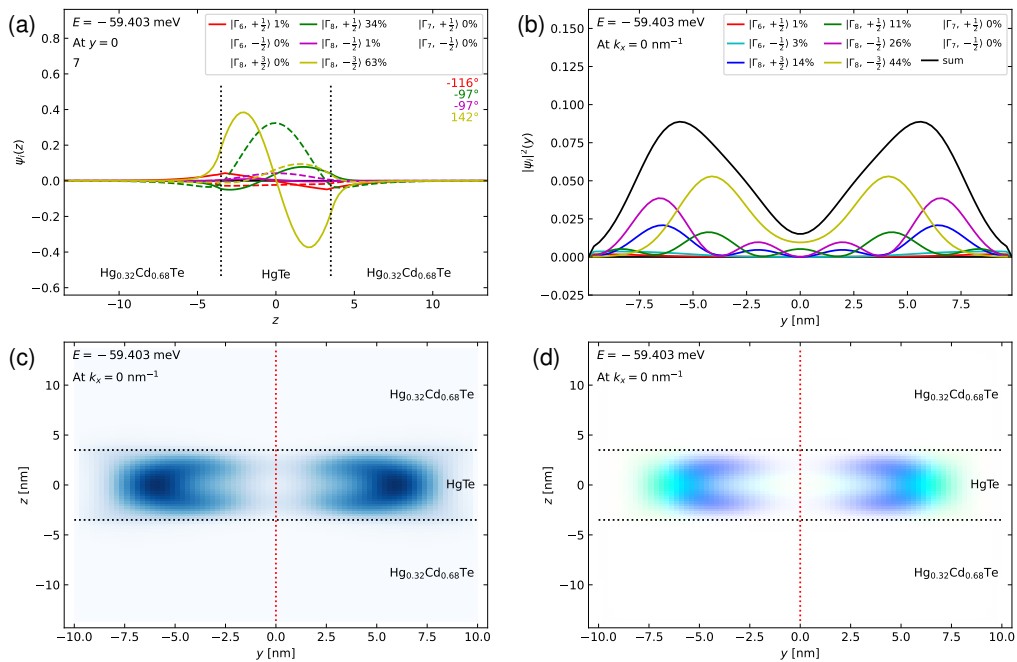

Figure 12: Examples of wave function plots from `kdotpy 1d` with (a) `plotwf z`, (b) `plotwf y`, (c) `plotwf zy`, and (d) `plotwf color`. These examples are based on the same eigenstate. The actual output contains the wave functions for multiple eigenstates.

option `table_wf_files` is set to `tar`, `targz`, `zip`, or `zipnozip` the files are collected into a single archive file.

For the two-dimensional geometry, there are two main plot styles:

- `separate`: As function of z, with different curves for each orbital. For each state, the wave function is expanded into its orbital components $\psi_i(z)$. Each orbital is represented by a separate colour and solid and dashed lines indicate the real and imaginary parts, respectively. Orbitals for which the amplitude is small, are not drawn. The plot is saved as a multi-page PDF, with each eigenstate on a separate page, see Fig. 10.

- `together`: Absolute-squared, as function of z, together in one plot, see Fig. 11.

For plot style `separate`, the phase for each orbital wave function is normalized to the orbital with largest amplitude (over z, not integrated): The maximum value is set to a positive real number, with relative phases being preserved. If the orbital have a definite complex phase, i.e., $\psi_i(z_) = f_i(z)e^{i\phi_i}$ for some real function $f_i(z)$ and phase value $\phi_i$, the phases $\phi_i$ are listed as angles in degrees as additional inset on the right side of the plot. The behaviour can be adjusted with the configuration option `plot_wf_orbitals_realshift`.

In LL mode `kdotpy ll`, one can use the same plot styles `separate` and `together`. In symbolic LL mode, the wave function $\psi(z)$ is plotted for the corresponding LL index $n$. In full LL mode, plot only the largest contribution, i.e., $\psi_n(z)$ for which $\int |\psi_n(z)|^2 dz$ is maximal over $n$; in this case, the probability density $\int |\psi_n(z)|^2 dz$ may be smaller than 1. For plot style `separate`, this LL index $n$ and the probability density are displayed as $LL = n$ and $|\psi_{LL}|^2 = $ **value**, respectively, for each state.

For the one-dimensional geometry, the wave functions $\psi_i(z, y)$ depend on two spatial coordinates. The following plot styles can be used:

- z: As function of $z$, at $y = 0$ (at the middle of the sample), i.e., $\psi(z, 0)$. Like `separate` for 2D, the wave function is decomposed into orbitals.

- y: Absolute-squared, as function of $y$, integrated over the $z$ coordinate, i.e., $|\psi|^2(y) = \int |\psi(z, y)|^2 dz$. The wave function is decomposed into orbitals or subbands.

- zy: Absolute-squared, as function of $(z, y)$, i.e., $|\psi(z, y)|^2$, summed over all orbitals. The values $|\psi(z, y)|^2$ are represented as colours from a colour map.

- color: Absolute-squared, with different colours for the orbitals, as function of $(z, y)$.

Examples are shown in Fig. 12. With kdotpy 1d, wave function output is possible only if the grid contains a single point. This restriction has been imposed to avoid problems with the large memory footprint needed given two spatial coordinates.

### 3.9.3 Symmetry analysis and symmetrization

The point group symmetry of the crystal structure imposes symmetry constraints on the Hamiltonian. The band structure exhibits the symmetries of the Hamiltonian, and kdotpy provides a way to verify the symmetry properties, known as *symmetry analysis*.

In kdotpy, symmetry analysis is implemented as the member function DiagData.symmetry_test(). This function applies the following algorithm:

- Define a symmetry transformation $T$, being a reflection, rotation, or roto-reflection of the point group. This object is implemented as instance of the VectorTransformation class.

- For each $\mathbf{k} \neq \mathbf{0}$ in the grid, verify if its image $T\mathbf{k}$ is also in the grid. If $T\mathbf{k}$ is not in the grid or if $T\mathbf{k} = \mathbf{k}$, skip this point.

- Compare the set of eigenvalues at $\mathbf{k}$ and the set of eigenvalues at $T\mathbf{k}$ and verify whether they are identical up to small numerical errors. If not, there is no symmetry between $\mathbf{k}$ and $T\mathbf{k}$.

- If the eigenvalues are the same, compare the observables at $\mathbf{k}$ and $T\mathbf{k}$. For scalar observables $O$, verify if $O_\mathbf{k} = \pm O_{T\mathbf{k}}$ for all eigenstates. (Special consideration is given to degenerate states.) For vector observables $\mathbf{O} = (O_x, O_y, O_z)$, for example (jx, jy, jz), verify if it transforms as one of the vector representations from the point group $O_h$.

- Collect the results by iterating over the grid points $\mathbf{k}$. For each transformation $T$ and for each observable $O$, print the $O_h$ representations compatible with the transformation properties of the observable.

- As a final step, analyze the symmetries over all transformations $T$. Try to deduce the relevant point group (subgroup of $O_h$) and the possible representations for each observable.

In version 1.0 of kdotpy, the implementation is experimental, and may be improved in a future version.

The *symmetrization* feature of kdotpy works in the opposite direction: Knowing that the dispersion satisfies a set of symmetries, the dispersion can be calculated on some range of momentum space and be extended to a larger range of momenta, thus saving time compared to doing a calculation on the full range. For the momenta $T\mathbf{k}$ obtained by symmetrization from the original grid points $\mathbf{k}$, the eigenvalues are copied from $\mathbf{k}$, and the observable values are calculated from those at $\mathbf{k}$ with the appropriate transformation applied, given by a representation of the point group.

A common use case is the reflection around zero for a one-dimensional momentum scan in a two-dimensional geometry (kdotpy 2d),

```
kdotpy 2d 8o noax msubst CdZnTe 4% mlayer HgCdTe 68% HgTe HgCdTe 68%
llayer 10 7 10 zres 0.25 k 0 0.6 / 60 kphi 45 erange -80 0 split 0.01
splittype isopz obs jz legend char out -jz outdir data-qw localminmax
symmetrize
```

where the eigenvalues and observable values for negative $\mathbf{k}$ are obtained from those at $-\mathbf{k}$, see Fig. 13(a). (The observable $O = \langle J_z \rangle$ is symmetric under this transformation.) A similar construct works for two-dimensional dispersions: The dispersion is calculated in the first quadrant ($k_x > 0$, $k_y > 0$ in cartesian coordinates or $k_\phi \in [0°, 90°]$ for polar coordinates) and extended to all four quadrants, see Fig. 13(b).

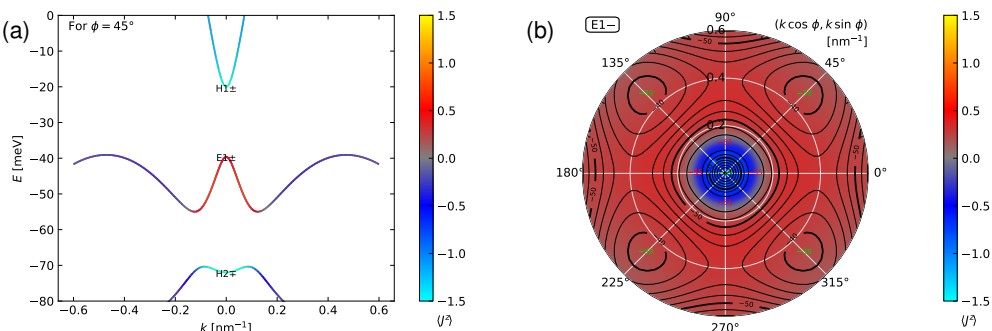

Figure 13: (a) Symmetrization of a one-dimensional momentum scan. The data for $k < 0$ are obtained by symmetrization from those at $k > 0$. (b) Symmetrization of a two-dimensional dispersion in polar coordinates. The data has been calculated explicitly for angles $k_\phi \in [0°, 90°]$. Symmetrization extends it to the full circle.

### 3.9.4 Löwdin perturbation theory (BHZ-like models)

A substantial number of works in the field relies on the simplified model made famous by Bernevig, Hughes, and Zhang (BHZ) [39]. This effective model takes the four subband degrees of freedom $|E1\pm\rangle, |H1\pm\rangle$ as its basis. This model is adequate for describing the topological phase transition in quantum wells as function of the well thickness $d$: At the critical thickness $d_c$, the energetic positions of the electron-like subbands $|E1\pm\rangle$ and the heavy-hole-like $|H1\pm\rangle$ change. For $d < d_c$, the device is trivially insulating and for $d > d_c$, the device is a two-dimensional topological insulator that hosts the quantum spin Hall effect [39] that can be measured in a Hall bar geometry [12].

Outside of the context of HgTe quantum wells with $d \approx d_c$, the four-band BHZ model is often inadequate to describe the essential physics. However, similar models with a modified or extended basis may be useful in order to gain useful insight, in some cases. Although we do not advocate the use of these simplified models in subband basis, we have included the functionality to derive them so that they can be compared against the more accurate $\mathbf{k} \cdot \mathbf{p}$ models.

The method of projecting the $\mathbf{k} \cdot \mathbf{p}$ Hamiltonian onto a basis of subband states is known as Löwdin perturbation theory or quasi-degenerate perturbation theory [15, 40]. The basis is a set of subband states at $\mathbf{k} = \mathbf{0}$. The contributions from the other subband states (outside of the basis) and from $\mathbf{k} \neq \mathbf{0}$ are treated by perturbation theory. The result is an effective Hamiltonian in the chosen subband basis that is valid near $\mathbf{k} = \mathbf{0}$.

In kdotpy, Löwdin perturbation theory is implemented up to second order in momentum. We use the framework of symbolic Hamiltonians (class SymbolicHamiltonian) that we also use for deriving Landau level Hamiltonians. The recipe is as follows:

- Take the symbolic Hamiltonian $H$ as well the set of eigenvalues and eigenstates at $\mathbf{k} = \mathbf{0}$ as input.

- Select the basis for the effective model based on the command-line argument bhz. Usually, this is an even number $n_A$ of subband states near the charge neutrality point. We label the subband states in the basis the 'A states'.

- Select the subband states that are treated perturbatively. The amount can optionally be specified from the command line. If it is not specified, all states confined in the quantum well are taken by default. (Deconfined states are not considered, because including them leads to unpredictable and unphysical results.) We label these states the 'B states'. For the selection procedure, they are separated into 'lower' and 'upper' B states, i.e., with energies below and above those of the A states.

- For the perturbative expansion, we use the expressions listed in the textbook by Winkler [15]. The zeroth term is simply the diagonal matrix of the eigenvalues $E_m$ of the A states,

$$H^{(0)} = \text{diag}(E_m). \tag{128}$$

The matrix size is $n_A \times n_A$.

- The construction of the first order perturbation term $H^{(1)}$ is implemented in `SymbolicHamiltonian.hper1()`. It takes the matrix of the eigenvectors $|\psi_m\rangle$ (where $m$ label the A states) as its only argument. It is defined as [15]

$$H^{(1)}_{mm'} = \langle \psi_m | H | \psi_{m'} \rangle, \tag{129}$$

where $H$ is the symbolic Hamiltonian. The resulting matrix $H^{(1)}$ is an $n_A \times n_A$ matrix containing the operators $\hat{k}_\pm$ up to second order. The function `SymbolicHamiltonian.hper1()` returns a list of lists of `SymbolicObject` instances, representing the matrix elements $H^{(1)}_{mm'}$.

- The construction of the second order perturbation term $H^{(2)}$ is implemented in `SymbolicHamiltonian.hper2()`. As arguments, it takes the eigenvalues $\{E_m\}$ and $\{E_l\}$ of the A and B states, respectively, as well as the associated eigenvectors $|\psi_m\rangle$ and $|\phi_l\rangle$. The second-order term is [15]

$$H^{(2)}_{mm'} = \frac{1}{2} \sum_l \langle \psi_m | H | \phi_l \rangle \langle \phi_l | H | \psi_{m'} \rangle \left( \frac{1}{E_m - E_l} + \frac{1}{E_{m'} - E_l} \right), \tag{130}$$

where the index $l$ of the sum runs over all B states. The result $H^{(2)}$ is once more an $n_A \times n_A$ matrix involving the operators $\hat{k}_\pm$. The function `SymbolicHamiltonian.hper2()` returns the matrix elements $H^{(2)}_{mm'}$ as a list of lists of `SymbolicObject` instances.

- The results are summed together, $H^L = H^{(0)} + H^{(1)} + H^{(2)}$. Negligible coefficients (absolute value $< 10^{-7}$) and terms of order $> 2$ in $\hat{k}_\pm$ are discarded.

- The basis vectors may be multiplied by complex phases in an attempt to make the matrix elements $H^L_{mm'}$ purely real or purely imaginary.

- If the basis can be separated into two uncoupled sectors, it is reordered, such that the matrix $H^L$ separates in two blocks on the diagonal.

The result of this method is a `SymbolicHamiltonian` object, that encodes the operator sum

$$H^L = H^L_0 + H^L_+ \hat{k}_+ + H^L_- \hat{k}_- + H^L_{++} \hat{k}_+^2 + H^L_{--} \hat{k}_-^2 + H^L_{+-} \hat{k}_+ \hat{k}_- + H^L_{-+} \hat{k}_- \hat{k}_+., \tag{131}$$

where the factors $H^L_0$, $H^L_0\pm$, etc. are $n_A \times n_A$ matrices.

The submodule `bhzprint` outputs the result as a matrix in LaTeX notation and compiles it if PDFLaTeX is available. An example result is shown in Fig. 14. The result is expressed in a notation that generalizes the notation of Ref. [39]. For a Löwdin perturbation similar to the BHZ model, i.e., two blocks of two subbands each, the configuration setting `bhz_abcdm=true` may be used in order to express the result in the canonical BHZ notation with coefficients $A, B, C, D$ and $M$. In `verbose` mode, `kdotpy` also writes the matrix elements $H^L_{mm'}$ and several intermediate results to standard output.

## 3.10 Data output

### 3.10.1 Types of data

Output is produced by `kdotpy` in various formats that serve various (in part complementary) purposes.

- Long-time storage: Scientific data must be maximally reproducible and is expected to be kept for a longer period of time. For this purpose, we use an XML-based data format, which contains both data and metadata.

- Immediate further processing: Data must be in a format that many other applications can read or import with as little effort as possible. We rely on CSV files (comma-separated values) for this purpose.

- Human interaction: Immediate feedback aids users in judging the calculation results even without further processing. Examples are graphical output (PDF and PNG) and console output.

Plots and CSV files are typically produced in pairs, with the CSV file containing exactly the data that is shown in the figure.

# 1 BHZ model

Basis: $|E1, +\rangle, |H1, +\rangle, |E1, -\rangle, |H1, -\rangle$

Hamiltonian:

$$\begin{pmatrix} E_1 + B_1 k^2 + G_1 \mathcal{B} & -iA_{1,2} k_+ & 0 & 0 \\ iA_{1,2} k_- & E_2 + B_2 k^2 + G_2 \mathcal{B} & 0 & 0 \\ 0 & 0 & E_3 + B_3 k^2 + G_3 \mathcal{B} & iA_{3,4} k_- \\ 0 & 0 & -iA_{3,4} k_+ & E_4 + B_4 k^2 + G_4 \mathcal{B} \end{pmatrix}, \quad (1)$$

with:

$A_{1,2} = 373\,\text{meV nm}, \quad B_3 = 1795\,\text{meV nm}^2, \quad E_3 = -49.36\,\text{meV}, \quad G_3 = -1.553\,\text{meV/T},$
$A_{3,4} = 373\,\text{meV nm}, \quad B_4 = 502\,\text{meV nm}^2, \quad E_4 = -45.75\,\text{meV}, \quad \text{and}$
$B_1 = 1794\,\text{meV nm}^2, \quad E_1 = -49.34\,\text{meV}, \quad G_1 = 1.551\,\text{meV/T}, \quad G_4 = -1.028\,\text{meV/T}.$
$B_2 = 502\,\text{meV nm}^2, \quad E_2 = -45.73\,\text{meV}, \quad G_2 = 1.028\,\text{meV/T},$

NOTE: $G$ factors are orbital contributions only.

# 2 Dispersion

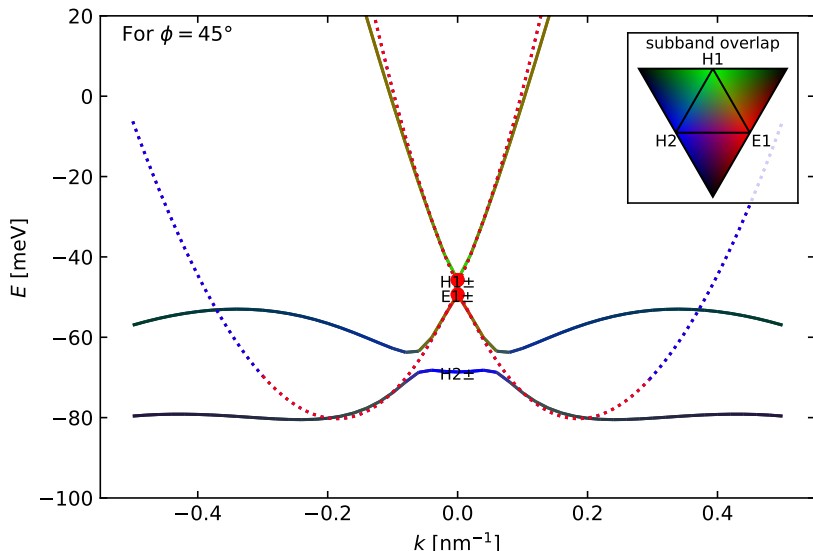

Figure 14: Example output of `kdotpy` with the `bhz` command line argument. This calculation is done for a quantum well of **11 nm** thick $\text{Hg}_{1-x}\text{Mn}_x\text{Te}$ (**$x = 0.024$**). This Löwdin perturbation uses the same basis as the canonical BHZ model of Ref. [39]. In the plot, the solid curves indicate the $\mathbf{k} \cdot \mathbf{p}$ dispersion. The dotted curves are the bands calculated from the Löwdin Hamiltonian $H^{\mathbf{L}}$ (here, equivalent to the BHZ Hamiltonian). The red and blue colours of the dotted curves separate the two blocks; here, they are degenerate. (The layout has been adjusted for reasons of illustration. The actual `kdotpy` output appears on two landscape A4 pages.)

### 3.10.2 XML files

The XML-format produced by kdotpy is a hierarchical data format originally designed for communication between different sessions of kdotpy, in particular for combining data sets with kdotpy merge. An important design principle to make this possible is maximum compatibility between different versions of kdotpy. The hierarchical nature of the XML format is well suited to achieve this goal. Reading the XML data is done by means of extracting the relevant tags. When a new type of data is added to the XML file (e.g., by a newer version of kdotpy), the new tag is simply ignored by older versions, and the XML file remains compatible.

The format includes a large number of metadata attributes, which is an important asset for reproducibility. The data file contains the following tags with metadata:

- `<info>`: Information on the program. This tag contains the command line, information on the host (computer) and its operating system, and the version numbers of kdotpy (including the Git hash if available), Python, and the installed modules.

- `<configuration>`: Configuration values.

- `<parameters>`: All information in the PhysParams object, including the evaluated material parameters.

- `<options>`: Options set on the command line. These are the values stored in the dict instance opts, which affects the calculations.

Especially the command line arguments and version numbers are essential for repeating a calculation. The metadata can also be used for filtering a large collection of data files by a certain attribute. The metadata are placed at the top of the file to allow for human inspection.

The actual data follows the metadata. The data in the DiagData object is serialized into the tag `<dispersion>` or `<dependence variable="b">`, in case of a dispersion and a magnetic-field dependence, respectively. In both cases, it contains the `<vectorgrid>` tag for the VectorGrid object. All data points (DiagDataPoint instances) are serialized as `<momentum>` (for dispersion) or `variabledata` (for magnetic-field dependence); they contain lists of the eigenenergies, band indices, Landau level indices, and observables.

Finally, the `<extrema>` tag is included for serialization of the BandExtremum. The serialization of further data container objects (e.g., DensityData) is scheduled for future versions of kdotpy.

### 3.10.3 CSV files

Data for further processing is saved in CSV files. The CSV (comma separated value) format is a universally understood lightweight format for storing one- or two-dimensional arrays of data. The workflow in kdotpy separates the composition (or preparation) of the data and writing it to a file. In the composition stage, kdotpy prepares a list or a dict instance, which contains a set of one-dimensional arrays of values, representing the columns. This data is passed to the writer function, together with formatting information and labels for each column. In some cases, a post-write function adds extra data to the file, e.g., extra rows with band labels.

We use a small number of common layouts of the data. The uniformity is a boon to both users and developers. Fewer layouts means less effort for external processing of the data. On the development side, there is less code to maintain. The following layouts are the most common ones:

- Column-wise data: Each column in the table represents a property or quantity and each row represents another item. For example, in extrema.csv, there are columns for band index, band character, minimum or maximum, momentum, energy, and band mass; see Fig. 15(a) for an illustration. The column headers on the first row indicate the quantity, and units are included (optionally) on the second row.

- Functions of one dimension: A set of functions $f_i(x)$ is represented column-wise. The $x$ values appear in the first column, and the subsequent columns represent $f_i(x)$. Examples: dos.csv [Fig. 15(b)], with DOS and IDOS as function of energy $E$ (first column); dispersion-byband.csv [Fig. 15(c)], with the dispersion $E_i(\mathbf{k})$ of many bands $i$ as function of momentum $\mathbf{k}$. The latter uses two columns, here with $k$ and $\phi$ in the polar coordinate system. The band indices and characters appear in two extra rows at the top; the position is configurable.

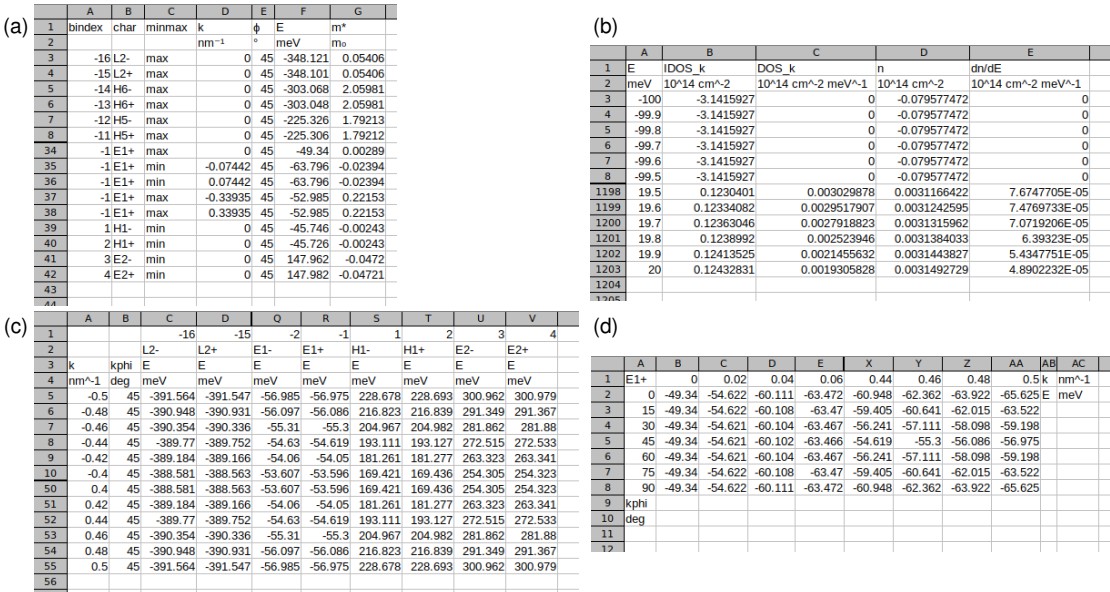

Figure 15: Examples of CSV output, as imported by a spreadsheet program. Some rows and columns have been hidden for clarity. (a) `extrema.csv`: Column-wise data, where each row lists the properties of a band extremum. The first and second row label the quantities and units. (b) `dos.csv`: The DOS and the IDOS as function of energy $E$. (c) `dispersion-byband.csv`: Dispersions $E_i(\mathbf{k})$, i.e., band energies as function of momentum. Each column represents a band. The bands are labelled by band index and character in the first and second row. (d) `dispersion.bi.csv`: Two-dimensional dispersion $E(k, \phi)$ of a single subband.

- Two-dimensional array: The data represents a function $f(x, y)$. The first row contains the values $x$ and the first column contains the values $y$. The data $f(x, y)$ starts at the second row and second column. The quantities and units for $x$ are printed at the end of the first row and those for $y$ at the end of the first column. The quantity and unit for $f$ are at the end of the second row, i.e., the first data row. The first row and first column provides an extra label for the data. Examples: `dispersion.bi.csv` [Fig. 15(d)], which provides the dispersion $E_i(k, \phi)$ for a single subband. The file is labelled with the band index $i$. This layout is also used frequently in LL mode for functions $f(B, E)$ or $f(B, n)$, i.e., quantities as function of magnetic field $B$ and energy or carrier density.

- Three-dimensional array: Used for bulk dispersions $E(k_x, k_y, k_z)$ (as well as for cylindrical and spherical coordinates). The third coordinate is inserted as the first column. The result is arranged as a stack of two-dimensional arrays.

At the time of composition, the data is arranged in columns. The composition functions also determines the formatting for the data, i.e., how numerical values are converted to strings. For several quantities X, the number of digits for numerical values can be set by configuration values `table_X_precision`. The composition function also takes care of the formatting of quantities and units in the headers. The result depends on the configuration settings `table_X_style`, with the following choices:

- `raw`: 'Without' formatting; use the raw labels (internal representation) for quantities and units. NOTE: The output is not guaranteed to be ASCII only. For example, the 'micro' prefix $\mu$ may be used with some units.

- `plain`: Plain-text formatting using common symbols, e.g., square is ^2 and Greek letters are spelled out.

- `unicode`: Formatting using 'fancy' Unicode symbols, e.g., square is the superscript-2 symbol and Greek letters use their corresponding Unicode symbol.

- `tex`: LaTeX formatting.

The composition functions call the general writer function `tableo.write.write()`. This function delegates the actual work to one of the following functions, depending on the value of the configuration value `csv_style`:

- `csvwrite()`: Uses the `csv` module provided with Python. The default *dialect* is to use `,` as delimiter (separator between values), and `"` as quoting character. This writer is used if `csv_style=csvinternal` or if `csv_style=csv` and the Pandas package is not installed.

- `alignwrite()`: Aligns the formatted strings in columns, using spaces as column separators and fill characters. The result is well suited for direct viewing in a text editor (with a monospace font). Importing it into a spreadsheet program is more tedious than with the other writers. This writer is slightly slower than `csvwrite()` due to the extra step needed for determination of the column widths. To use this writer, set `csv_style=align`.

- `pdwrite()`: Uses the Pandas package to produce a CSV file. The data is first converted into a Pandas `DataFrame` object. The file is then produced by `DataFrame.to_csv()`. The result is very similar to `csvwrite()`, but slightly less flexible in terms of setting the numerical precision of floating-point values. This writer is used if `csv_style=csvpandas` or `csv_style=csv`, when Pandas is available.

More writers may be added in the future, depending on user request. After the work done by one of these three writers, `tableo.write.write()` may call the post-write functions `write_axislabels()`, for the quantities and units in the two- and three-dimensional array layout, and/or `write_extraheader()`, for the band labels (see Fig. 15(c), for example).

### 3.10.4 Graphics

For graphics, `kdotpy` uses the Matplotlib package with the `pdf` backend. This backend produces vector graphics, which delivers the highest quality with reasonable file sizes. (Rasterization is used with some

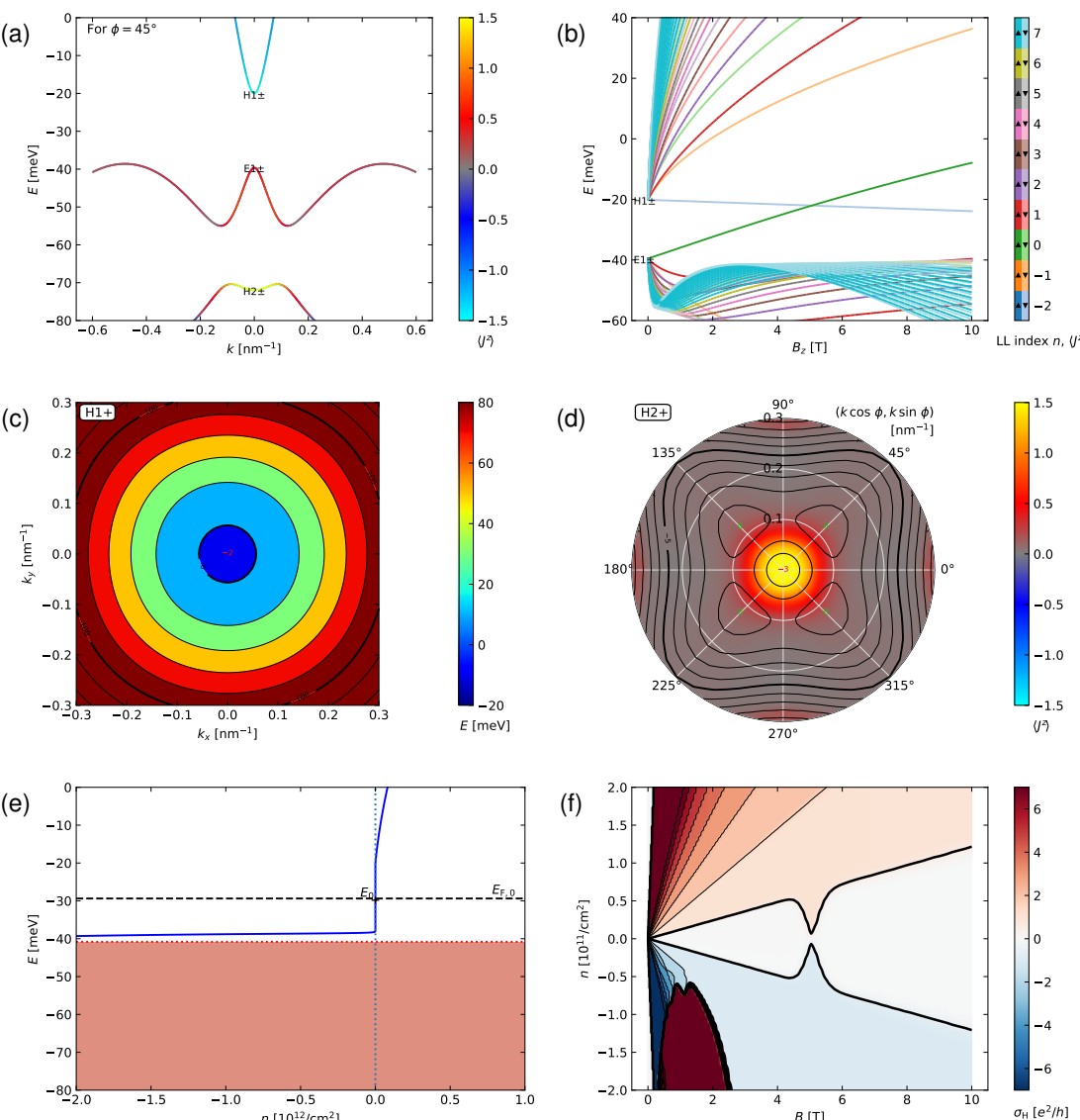

Figure 16: Examples of graphical output of `kdotpy`. (a) A one-dimensional dispersion plot for a quantum well device with **7 nm** HgTe as the active layer. The plot visualizes the dispersions of many bands $E^{(i)}(k)$ as function of momentum $k$. The colours encode the values of the observable $\langle J_z \rangle$. (b) Magnetic-field dependence of Landau levels, for the same system as in (a). (c) A two-dimensional dispersion plot $E^{(i)}(k_x, k_y)$ for one of the surface states of a 3D topological insulator device (**70 nm** HgTe). The colour scale refers to energy $E$ and the red value $-2$ at $\mathbf{k} = 0$ indicates a local minimum at that energy. (d) A dispersion plot in polar coordinates for the highest valence band state of the same system as (c). The colour scale refers to the observable $\langle J_z \rangle$. (e) Integrated density of states (also known as carrier density $n$) corresponding to (a). The vertical axis is energy and aligns with the vertical axis of (a). (f) Landau fan with carrier density on the vertical axis, corresponding to the magnetic field dependence in (b). The colour scale is given by the Hall conductance $\sigma_H$ in units of $e^2/h$.

two-dimensional colour maps, which would lead to large files and long rendering times as pure vector graphics.)

The results are highly configurable. First of all, command-line arguments can be used to enable or disable plot elements, for example, legends, plot titles, and band character labels. Secondly, configuration options determine the plot geometry (sizes and margins), the colour scale being used, and the style of the axis labels, for example. Finally, Matplotlib provides a variety of customization options, named `rcParams`, that can be manipulated with style sheets. Matplotlib style sheets are supported by kdotpy, and can be loaded with the configuration value `fig_matplotlib_style`. The default style file `kdotpy.mplstyle` is provided with kdotpy and is copied to the `~/.kdotpy` directory. The user may edit the default style file or provide additional style files.

We provide several representative examples of plots generated by kdotpy. Like with the CSV output, we use a small number of generic layouts for uniformity of the output and for maintainability. Commonly used plots are:

- One-dimensional dispersion or magnetic-field-dependence plot: The function `ploto.bands_1d()` produces the typical visualization of a band structure or Landau level fan, as illustrated by Figs. 16(a) and (b) in the Introduction. The horizontal axis is a momentum or magnetic field component, and the vertical axis is energy $E$. When an observable is specified with the command line argument `obs`, it is used to colour the bands; depending on whether the observable is constant or variable, the band is plotted as a single line object with a flat colour or as a collection of line segments with variable colours. Optionally, the energy axis may be transformed to density $n$.

- Two-dimensional dispersion: The function `ploto.bands_2d()` implements the visualization of dispersions $E(k_x, k_y)$ or $E(k, \phi)$ in cartesian or polar coordinates, respectively; see Figs. 16(c) and (d). The dispersions are represented as contour plots in a cartesian or polar coordinate system. The result is a multi-page PDF file, with one page for each band within the energy range. The colouring is done based on the observable given by the command-line argument `obs`; if this argument is omitted, the colour is based on energy.

- Total (integrated) density of states: The functions `ploto.dos()` and `ploto.integrated_dos()` plot the DOS and IDOS, respectively, as stored in a `DensityData` instance. By default, the energy axis is vertical, and aligns with the vertical axis of dispersion plots. See Fig. 16(e) for an example.

- Generic two-dimensional colour maps: The function `ploto.density2d()` provides the framework for generic two-dimensional density plots, where a function $f(x, y)$ is represented by a colour map. The horizontal axis is typically momentum or magnetic field and the vertical axis is energy or density. The function `ploto.density2d()` is used for many quantities in the `postprocess` module, e.g., optical transitions and Hall plots, see Fig. 16(f).

### 3.11 Self-consistent Hartree method

The *self-consistent Hartree method* is an iterative solver of the Schrödinger equation and the one-dimensional Poisson equation[4]. A simultaneous solution of the Schrödinger and Poisson equation cannot be found analytically. The iterative method aims at finding a self-consistent solution, solving the Schrödinger equation and the Poisson equation alternatingly. The physical picture is that the occupied eigenstates of the Hamiltonian induce a carrier density, leading to an electrostatic potential that in turn affects the Hamiltonian. We emphasize that this method applies to the 2D geometry only. The self-consistent Hartree method can be invoked by using `selfcon` on the command line for kdotpy 2d and kdotpy ll.

#### 3.11.1 Program Flow

The program flow of the iterative `SelfConSolver` is sketched in Fig. 17. After initializing the starting conditions of the solver, a starting potential is initialized. If no initial potential is provided and the user did not request to initialize the potential based on a uniform charge density or fixed background density, the first iteration starts with a flat potential. Next, the Hamiltonian including the Hartree potential is

---

[4]This version of the Poisson equation describes one-dimensional electrostatics in the $z$ direction only.

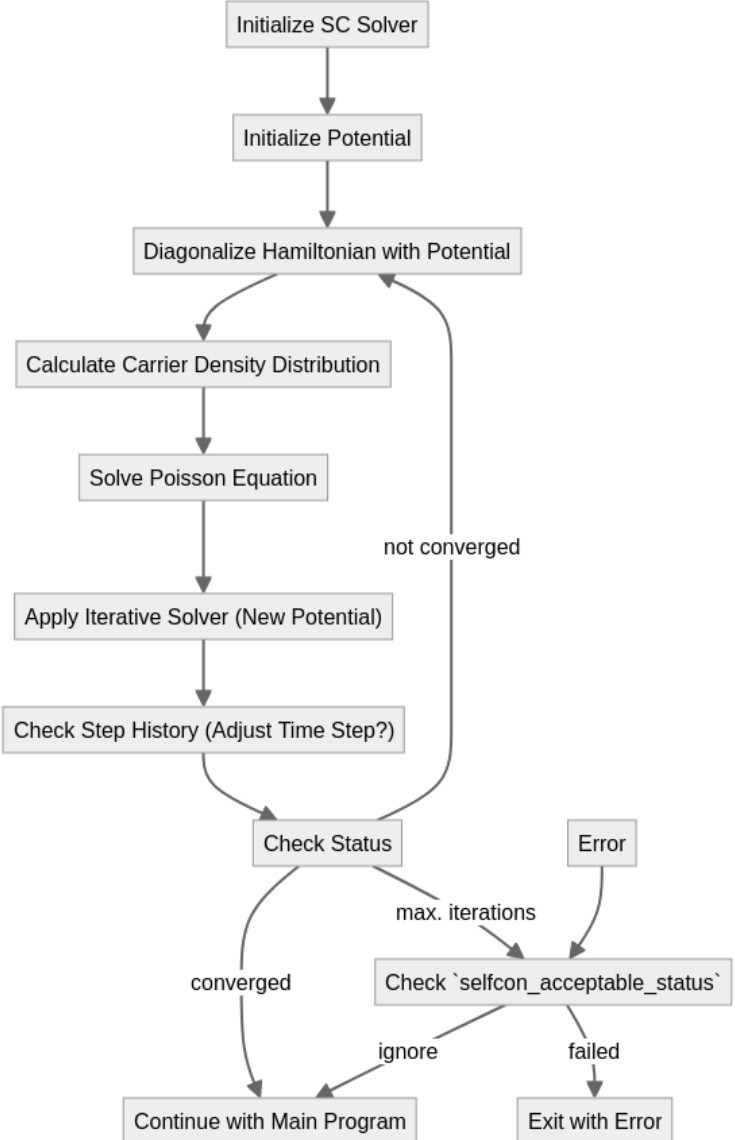

Figure 17: Flowchart for the self-consistent Hartree part of kdotpy.

diagonalized and the resulting charge carrier density $\rho(z)$ along the growth direction $z$ is evaluated (see Section 3.7.8). The Hartree potential $V_H(z)$ is then solved from the Poisson equation,

$$\partial_z \left[ \varepsilon(z) \, \partial_z V_H(z) \right] = \frac{e}{\varepsilon_0} \rho(z), \tag{132}$$

where $\varepsilon(z)$ is the relative dielectric constant of the layers and $\varepsilon_0$ is the permittivity of the vacuum. The electric-field or potential boundary conditions can be specified for solving Eq. 132, see Appendix A.4 for details. It is also possible to specify a fixed background density which is added to $\rho(z)$ to model for example ionic cores from modulating doping.

In order to ensure better convergence, the solution of Eq. (132) is not directly used as a new potential in the next iteration. Instead, the potential $V_H^{(i+1)}(z)$ of iteration $i + 1$ is calculated from the old potential $V_H^{(i)}(z)$ as

$$V_H^{(i+1)}(z) = V_H^{(i)}(z) + \tau \left( \mathcal{S}[V_H^{(i)}](z) - V_H^{(i)}(z) \right), \tag{133}$$

where $\mathcal{S}[V_H^{(i)}]$ is the potential obtained from solving Eq. (132) with the eigenstates of $H + V_H^{(i)}(z)$, and $\tau$ is a number between 0 and 1. Viewing Eq. (133) as a discrete step in solving a differential equation, we can interpret the value $\tau$ as a 'virtual time step'. The initial time step is 0.9 by default, and can be adjusted by an additional numerical argument after `selfcon`.

Next, the history of potential differences $V_H^{(i+1)}(z) - V_H^{(i)}(z)$ between iterations is checked for periodic orbits or chaotic oscillations. If either are detected, `kdotpy` can automatically reduce the time step in an attempt to 'escape' the periodic orbit or chaos and to reach convergence. An large initial time step leads to faster convergences but makes the solver more vulnerable to developing periodic orbits or chaotic oscillations.

Finally, the convergence criterion is checked by comparing the last potential difference to a predefined convergence threshold. If a satisfactory convergence has not yet been achieved, the above process loops again. Once convergence is accomplished, `kdotpy` continues with the rest of the program as if the `selfcon` options was not given but using the self-consistently calculated Hartree potential In case the maximum number of specified iterations is reached or an error occurs in the above process, `kdotpy` either exits with an error (by default) or continues with the rest of the program ignoring the error (if configured to do so).

The source file `selfcon.py` defines two separate classes `SelfConSolver` and `selfConSolverLL` for momentum and Landau level mode, respectively. The latter is derived from the former, where the differences lie in how the charge carrier densities are calculated (cf. Section 3.7) and the fact that in the Landau level mode, a separate Hartree potential is calculated for each magnetic field value. Furthermore, as it is not possible to calculate the density up to arbitrarily low magnetic fields in a Landau level picture with a finite number of Landau levels, the potentials at the smallest fields are set equal to the first potential that can be calculated.

### 3.11.2   Two alternatives for calculating the density as function of $z$

`kdotpy` implements two different algorithms for calculating the charge carrier distribution $\rho(z)$ along the growth axis $z$. The first method is the one sketched in Sec. 3.7.8, with the 'naive' assumption that $\rho(z)$ is identically zero at the charge neutrality point. All states in a given band are either counted as pure holes or as pure electrons based on the position of the band relative to the charge neutrality point, i.e., whether the band index is negative or positive. This method was applied successfully to model modulation doping in thin HgTe quantum wells [3]. For larger thicknesses or strong Hartree potentials this method can fail, as it becomes difficult to clearly separate electron-like and hole-like states.

The second method avoids this problem: One applies full diagonalization, where one calculates all eigenstates of the Hamiltonian, and takes the top or bottom end of the energy spectrum as a reference for $\rho(z)$. In other words, all states are treated as a single carrier type and a suitable background density is subtracted (full-band envelope approach) [41]. It suffices to consider only the top of the spectrum and to calculate all conduction band eigenstates down to the charge neutrality point, plus a few valence band states in addition. (Taking the conduction band is computationally more efficient than the valence band, in view of the smaller number of states.) The subtracted density offset is

$$n_{\text{offset}} = 2n_z \frac{\Omega_{\text{grid}}}{(2\pi)^2 \Delta z}, \tag{134}$$

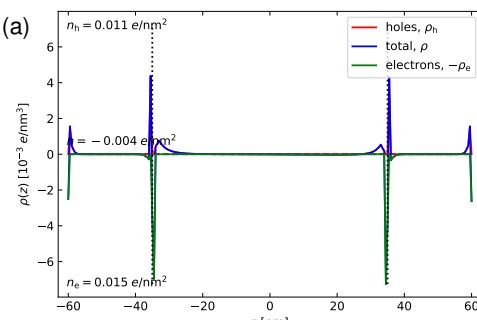 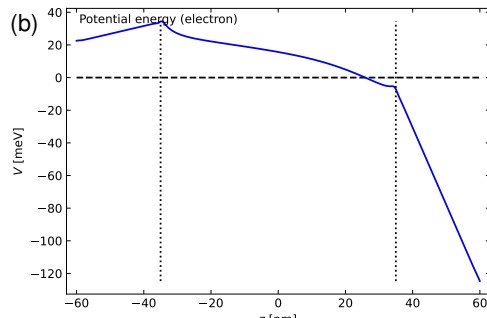

Figure 18: (a) Self-consistently calculated Hartree potential for a structure with **70 nm** thick HgTe layer and (Hg,Cd)Te barriers. The total carrier density has been set as **0.004 nm⁻²**. The boundary conditions are determined by imposing an electric field strength of **0.48 mV nm⁻¹** inside the bottom barrier. This calculation has been performed with the default setting `selfcon_full_diag=true`.

where $n_z$ is the total number of $z$ points and $\Omega_{\mathrm{grid}}$ is the volume of the grid in $k$-space over which the calculation is performed. For the Landau-level mode, the volumetric offset density is instead given by

$$n_{\mathrm{offset}} = n_z[(n_{\mathrm{max}} + 1) + (n_{\mathrm{max}} + 2)]\frac{eB}{h}\frac{1}{n_z \Delta z}, \tag{135}$$

where $n_{\mathrm{max}}$ is the maximum Landau level index (set by the command line argument `llmax`) and $B$ is the magnetic field.

The full-band envelope approach not only considers the active carriers in the quantum well but also results in interface dipoles and side bumps at the model edges (see Fig. 18). Note that the algorithm described in 3.11.1 keeps the total carrier density in the system constant, which includes these features. Hence the carrier density inside the quantum well will be different from the one specified by the argument `cardens`.

The implementation of the full-band approach is done by the `SelfConSolver` derived classes `SelfConSolverFullDiag` and `SelfConSolverLLFullDiag` for dispersion and Landau-level mode, respectively. By default, the full-band approached is used for the self-consistent Hartree method. To use the naive method, for gaining speed at the expense of becoming less physically sound, one may set the configuration value `selfcon_full_diag=false`.

### 3.11.3   Interpretation of the results

Due to the combination of diagonalization and integration (in the steps solving the Schrödinger and the Poisson equation, respectively), the iterative dynamics of $V_{\mathrm{H}}^{(i)}$ as function of iteration step $i$ is difficult to characterize and to predict. The convergence behaviour can depend sensitively on input parameters such as boundary conditions and the initial time step. If the solver converges, the result only represents a single solution, out of possibly many solutions. Whether the result is the most natural solution can only be determined by critical scrutiny.

When one relies on the self-consistent method, it is advisable to run the calculation multiple times with slightly different settings, in order to see if the algorithm is stable against such variations. The terminal output from `kdotpy` can also be a useful information source, especially for tracking convergence behaviour. To allow for systematic analysis of the iterative dynamics, the self-consistent solvers can provide debug output, with the internal state (carrier density $\rho(z)$ and Hartree potential $V_{\mathrm{H}}(z)$) being written to a CSV file at each iteration. A thorough systematic analysis of the self-consistent Hartree method is beyond the scope of this work and will be addressed elsewhere.

# 4   Installation

## 4.1   Prerequisites

kdotpy is packaged as a Python package that is intended to be installed using PIP. For installing kdotpy, you must have a working Python installation (version 3.9.0 or above) and an up-to-date version of PIP. When PIP installs kdotpy, it automatically checks if the packages numpy, scipy, and matplotlib are installed and have compatible versions. Packages that kdotpy depends on optionally may be installed manually with PIP.

If your Python installation already contains a variety of packages, the dependencies of the installed packages could potentially conflict with those of kdotpy. This problem may be avoided by installing kdotpy in a virtual environment, which is like a 'sandbox' where kdotpy and its dependencies do not interfere with other packages. The virtual environment can be created with python3 -m venv directoryname. After activating the virtual environment, you may now install kdotpy following any of the installation methods below. Consult the documentation of the Python venv module for more information[5].

Installation methods 2 and 3 below rely on git, which you must have installed in order for these methods to work. If you do not wish to use git, you may wish to use one of the other methods.

## 4.2   Download and installation

We list four methods for installing kdotpy. You can choose any one of them, but combining them is not recommended.

1. *Using PIP, from the Python Package Index (PyPI):*

   ```
   python3 -m pip install kdotpy
   ```

2. *Using PIP, from our Gitlab repository:*

   ```
   python3 -m pip install git+ssh://git@git.physik.uni-wuerzburg.de/kdotpy/kdotpy.git
   ```

3. *Using* git clone *and installing it from your local copy:*

   ```
   git clone https://git.physik.uni-wuerzburg.de/kdotpy/kdotpy.git
   python3 -m pip install ./kdotpy
   ```

   Note that this clones the files into the subdirectory kdotpy of your current working directory.

4. *Using a manual download and installing it from your local copy:*
   If you prefer to avoid the PyPI and git, then you can download a .zip or .tar.gz file from our Gitlab repository at https://git.physik.uni-wuerzburg.de/kdotpy/kdotpy [42]. Download the preferred file format from the dropdown menu under the 'Code' button. Unpack the files into an empty directory. Then use

   ```
   python3 -m pip install ./directoryname
   ```

   to install kdotpy, replacing ./directoryname by the appropriate directory name where you unpacked the files.

You may obtain version 1.0.0, the exact version discussed in this article, also from the Gitlab repository of this journal.

---

[5]See https://docs.python.org/3/library/venv.html.

## 4.3 Installation for active developers

If you wish to make modifications to the `kdotpy` code, then it is recommended to use PIP's `-e` option (short for `-editable`). This option dynamically links the package installation directory to the source files. If you change the source, you thus do not have to re-install the package. To use an editable install, use

```
git clone https://git.physik.uni-wuerzburg.de/kdotpy/kdotpy.git
python3 -m pip install -e ./kdotpy
```

If you intend to actively contribute to the project, you must change to a non-protected branch in order to be able to push your changes to the repository. (Where you are allowed to push depends on your membership role in our Gitlab project. The `master` branch is protected for all except the project owner. The journal repository shall not be used for development purposes.)

## 4.4 Testing the installation

If the install was successful, then entering

```
kdotpy version
```

on the command line should return the version number. If not, then something went wrong during the installation and you may wish to try one of the other methods. (This assumes that if you use a virtual environment, it has been set up and activated correctly.)

For testing the functionality of `kdotpy`, you can use

```
kdotpy test
```

for running the standardized tests. Note that the tests may take a few minutes to complete. They will generate output in the subdirectory `test` relative to where you start `kdotpy test`.

# 5 Usage

## 5.1 General remarks

kdotpy is designed as a standalone application. If you have followed the installation instructions above, you can simply run kdotpy from the command line, followed by the 'subprogram' label and further arguments. The first argument is always the subprogram, but the order of the following arguments is usually unimportant. (For an overview of the subprograms, see Sec. 3.1.2.) You can run kdotpy from any folder.

Alternatively, you can also use python3 -m kdotpy followed by the subprogram and further arguments. You can also import specific functions or submodules of kdotpy from other Python scripts or the interactive interpreter using from kdotpy import function. As we have designed kdotpy primarily as a command-line tool, such imports are not recommended for normal use.

Below, we provide detailed tutorials that describe how to set up a calculation in kdotpy step by step, and how to interpret the output. The curious user may also find the tests defined by kdotpy test useful; the command lines for these standardized tests can be extracted using kdotpy test showcmd.

## 5.2 Example calculation: Basic dispersion

### 5.2.1 Introduction

This basic tutorial describes a systematic manner to compose the command line from scratch, adding the necessary arguments step by step. As an example, we will calculate the dispersion of a **7 nm** HgTe quantum well: We assume the substrate is **$Cd_{0.96}Zn_{0.04}Te$** and the barriers are **$Hg_{0.32}Cd_{0.68}Te$**. We will calculate the dispersion along the diagonal axis (110) and try to answer the following questions about the maxima of the dispersion at finite momentum in the valence band (also known as *camel back*):

- Where is the finite-momentum maximum in momentum and energy?

- Do we have a direct or an indirect gap?

- What is the orbital character at the finite-momentum maximum?

### 5.2.2 Setting it up step by step

1. We first have to determine the geometry and/or mode, defined by the number of translationally invariant dimensions. In this example, we have a '2D' geometry, with momentum coordinates $(k_x, k_y)$. Thus, the appropriate subprogram to use here is kdotpy 2d.

   ```
   kdotpy 2d
   ```

2. Next, we determine the number of orbitals in the model. This is usually eight orbitals, for which we use the option 8o. Alternatively 6o for six orbitals may be used. Let us also specify that we do not want to use the axial approximation by entering noax.

   ```
   kdotpy 2d 8o noax
   ```

3. Let us now enter the substrate material and 'layer stack'. We use msubst for the substrate material, mlayer for the layer materials and llayer for the layer thicknesses. We also enter the resolution of the discretization in the **z** direction with zres; **0.25 nm** is a good value to start with.

   ```
   kdotpy 2d 8o noax msubst CdZnTe 4% mlayer HgCdTe 68% HgTe HgCdTe 68%
   llayer 10 7 10 zres 0.25
   ```

   Note that we have taken the barriers to be 10 nm thick. Usually, even if the barriers are much larger in reality, setting the thickness to **10 nm** gives an accurate representation of the dispersion while reducing calculation time. Note also how the materials are entered.

4. We define the parametrization of the momentum. Here, we take 60 points along the chosen momentum axis, symmetric around zero, with k -0.6 0.6 / 120. The (110) axis has a **45** degree angle with the $k_x$ axis, hence we enter kphi 45.

   ```
   kdotpy 2d 8o noax msubst CdZnTe 4% mlayer HgCdTe 68% HgTe HgCdTe 68%
   llayer 10 7 10 zres 0.25 k -0.6 0.6 / 120 kphi 45
   ```

   If we would run this command, we already get a result, dispersion.pdf. It looks promising, but kdotpy indicates that there are a few problems:

   ```
   Calculating bands (k=0)...
   Warning (band_type): Unable to determine band character and/or number of
   nodes for 50 eigenstates.
   Possible causes: spin degeneracy not broken, nonzero potential,
   one-dimensional geometry, etc.
   1 / 1
   ERROR (estimate_charge_neutrality_point): Failed, because E+ and/or E-
   bands are missing
   ```

5. The assignment of band characters has failed because the spin degeneracy has not being broken. The band characters are used to determine the position of the charge neutrality point and the band indices, and if this fails this can lead to subsequent problems, especially with post-processing functions that rely on the band indices, like density of states. We avoid these problems by splitting the degeneracy with split 0.01. The value **0.01** (in meV) hardly ever needs to be changed.

   ```
   kdotpy 2d 8o noax msubst CdZnTe 4% mlayer HgCdTe 68% HgTe HgCdTe 68%
   llayer 10 7 10 zres 0.25 k -0.6 0.6 / 120 kphi 45 split 0.01
   ```

6. With erange -80 0 we zoom in on the energy range of our interest. Moreover, let us choose a colour scale representing the orbital degree of freedom of the eigenstates. We set this by obs orbitalrgb. We also include the figure legend with legend. We add subband character labels with char.

   ```
   kdotpy 2d 8o noax msubst CdZnTe 4% mlayer HgCdTe 68% HgTe HgCdTe 68%
   llayer 10 7 10 zres 0.25 k -0.6 0.6 / 120 kphi 45 split 0.01 erange -80 0
   obs orbitalrgb legend char
   ```

7. We choose a label for the filenames with out -7nm and the target folder with outdir data-qw.

   ```
   kdotpy 2d 8o noax msubst CdZnTe 4% mlayer HgCdTe 68% HgTe HgCdTe 68%
   llayer 10 7 10 zres 0.25 k -0.6 0.6 / 120 kphi 45 split 0.01 erange -80 0
   obs orbitalrgb legend char out -7nm outdir data-qw
   ```

   Note that if outdir is omitted, the files will end up in the subfolder data if it exists, and in the current folder otherwise.

8. Finally, we add the post-processing option extrema, which will gives us useful information about the extrema.

   ```
   kdotpy 2d 8o noax msubst CdZnTe 4% mlayer HgCdTe 68% HgTe HgCdTe 68%
   llayer 10 7 10 zres 0.25 k -0.6 0.6 / 120 kphi 45 split 0.01 erange -80 0
   obs orbitalrgb legend char out -7nm outdir data-qw extrema
   ```

Other than the subprogram kdotpy 2d, which must be at the beginning of the command line, the other command-line arguments may be added in any order.

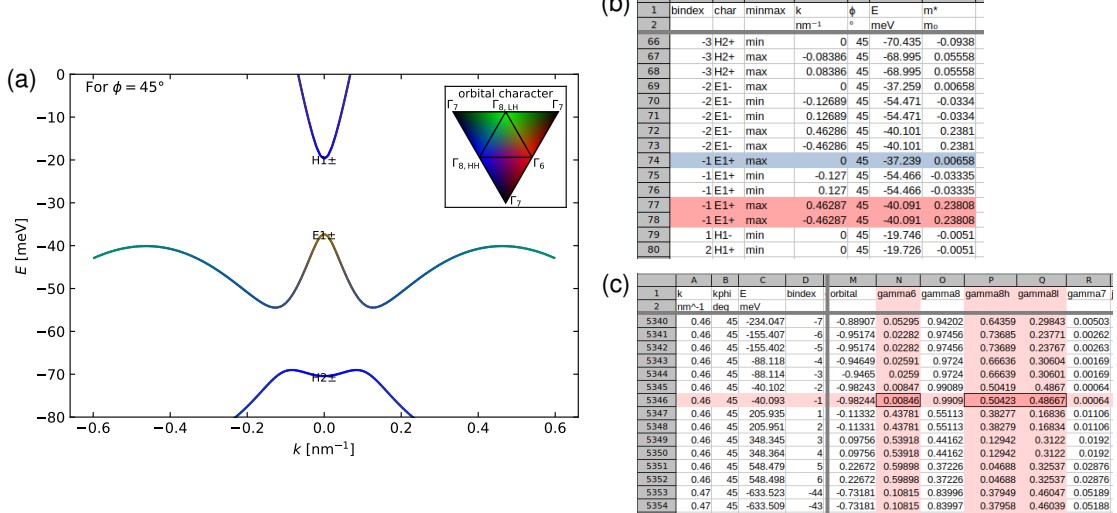

Figure 19: (a) Dispersion of a **7 nm** HgTe quantum well as shown by `dispersion-7nm.pdf`. The colours indicate orbital character, see the legend. At $k = 0$, the band character labels are shown, where '±' indicates that the states are approximately two-fold degenerate. (b) The properties of the extrema in `extrema-7nm.csv`. The maxima of the band with `bindex = -1` are highlighted (highlighting added manually, for illustration). (c) In `dispersion-7nm.csv`, we find the observable values for all eigenstates that `kdotpy` has calculated. The state closest to the maximum at finite momentum has been highlighted. We read off the expectation values for the observables `gamma6`, `gamma8h`, and `gamma8l` to gain information on the orbital character at this point.

### 5.2.3 Results and interpretation

After running this command, we will find the following files in the target output folder `data-qw`:

- `dispersion-7nm.pdf`: The dispersions $E(k)$ for all calculated eigenstates, see Fig. 19(a). The colours visualize the values of the observable(s) chosen with the option `obs`.

- `dispersion-7nm.csv`: A csv file where all data points (momentum, energy) are given with the values of the observables. The points are ordered by momentum.

- `dispersion-7nm.byband.csv`: A csv file with the band energies for each curve of `dispersion-7nm.pdf`, i.e., ordered by band.

- `extrema-7nm.csv`: A csv file listing all the extremal values.

- `output-7nm.xml`: The XML data file (see Sec. 3.10.2) that can be read again by the subprograms `kdotpy merge` and `kdotpy compare` for the purpose of replotting and comparing data sets and that contains a large amount of metadata that can be used for diagnostics and reproducing the results later. Discarding this file is strongly discouraged.

With these results, we can answer the questions above:

- *Where is the finite-momentum maximum in momentum and energy?*— In the plot, we see that these maxima appear in the band with character **E1±** at $k = 0$. We find the following data in `extrema.csv`, see Fig. 19(b). We find three maxima of the **E1+** band, one at $k = 0$ (highlighted in blue) and two at finite $k$ (highlighted in red), namely, at $k = \pm 0.463 \, \text{nm}^{-1}$ and $E = -40.1 \, \text{meV}$. Since we have calculated along the (110) axis, with `kphi` 45, these points are located at $k = \pm 0.463 \, \text{nm}^{-1} \times (\cos 45°, \sin 45°) = \pm(0.327, 0.327) \, \text{nm}^{-1}$.

- *Do we have a direct or an indirect gap?*— The maximum of the **E1+** band at $k = 0$ lies at $E = -37.2 \, \text{meV}$. The maxima at finite $k$ are lower, so the global maximum is at $k = 0$. The value `bindex = -1` confirms that this is the highest valence band state. The **H1—** state with `bindex`

= 1 is the lowest conduction band state. It only has one minimum, at $k = 0$ and $E = -19.7\,\text{meV}$. We find that the gap is direct (at $k = 0$) and its size is $|\Delta| = 17.5\,\text{meV}$.

- *What is the orbital character at the finite-momentum maximum?*— We extract this information from `dispersion-7nm.csv` that contains the values of the observables. The maximum is located at $k = \pm 0.463\,\text{nm}^{-1}$, so we check the states at the nearest value $k = 0.46\,\text{nm}^{-1}$ (as a first approximation). The band with `bindex = -1` is the correct one. We could also identify it by energy. The orbital character is determined by the probability densities in the $|\Gamma_6, \pm\frac{1}{2}\rangle$, $|\Gamma_8, \pm\frac{3}{2}\rangle$, and $|\Gamma_8, \pm\frac{1}{2}\rangle$ states, which is encoded by the observables `gamma6`, `gamma8h` and `gamma8l`, respectively. We thus find **0.8% $\Gamma_6$** ('electron or s orbital'), **50.4% $\Gamma_{8H}$** ('heavy hole'), and **48.7% $\Gamma_{8L}$** ('light hole'). We confirm in the dispersion plot that the colour at this position is approximately cyan, a mixture of **50%** green and **50%** blue.

## 5.3 Example calculation: Landau levels

### 5.3.1 Introduction

We set up a Landau-level calculation for the same structure as in Sec. 5.2. We aim to calculate the Landau levels up to $B = 10\,\text{T}$ and to answer the following questions, centred around the properties of the 'lowest Landau levels', i.e., those that border the band gap:

- Which Landau levels are the 'lowest' ones?
- What is the critical field $B_c$ where the inversion of the lowest LLs is undone?
- What orbital character do the states have at this crossing?

We set up the calculation step by step, reusing some of the steps of Sec. 5.2.

### 5.3.2 Setting it up step by step

1. First, we determine the subprogram. The geometry is 2D, and we use the Landau level formalism. We thus use

   ```
   kdotpy ll
   ```

2. As before, we use the eight-orbital model. In order to speed up the calculation, let us use the axial approximation. For `kdotpy ll`, this is achieved by omitting `noax`.

   ```
   kdotpy ll 8o
   ```

3. The layer stack parameters are the same as in Sec. 5.2.

   ```
   kdotpy ll 8o msubst CdZnTe 4% mlayer HgCdTe 68% HgTe HgCdTe 68%\
   llayer 10 7 10 zres 0.25
   ```

4. Next, we specify the grid for the magnetic field values. Let is choose 100 points with the upper limit **10 T**. We could choose evenly spaced points (every **0.1 T** with `b 0 10 / 100`, but for LL calculations, quadratic stepping is recommended. This is achieved by the 'double slash' notation `b 0 10 // 100`

   ```
   kdotpy ll 8o msubst CdZnTe 4% mlayer HgCdTe 68% HgTe HgCdTe 68%
   llayer 10 7 10 zres 0.25 b 0 10 // 100
   ```

5. Like in Sec. 5.2, we zoom in at the energy range `erange -80 0` and we use `split 0.01`. This time, we also enter the diagonalization parameters explicitly. We choose the maximum LL index to be 20 using `nll 20`. For each LL index, we aim to get approximately 12 (subband) states. For this we need about 240 eigenvalues: `neig 240`. We also put the target energy for the diagonalization somewhere near the gap, which we know is at approximately **−30 meV**. Note, however, that if we put the target energy at that value, the **E2±** subbands might be out of range. These subbands are required for the determination of the charge neutrality point to work properly, so we put the target energy higher up, i.e., we put `targetenergy 0`. (For the same reason, we aim for 12 subband states instead of the minimally required number of 8.)

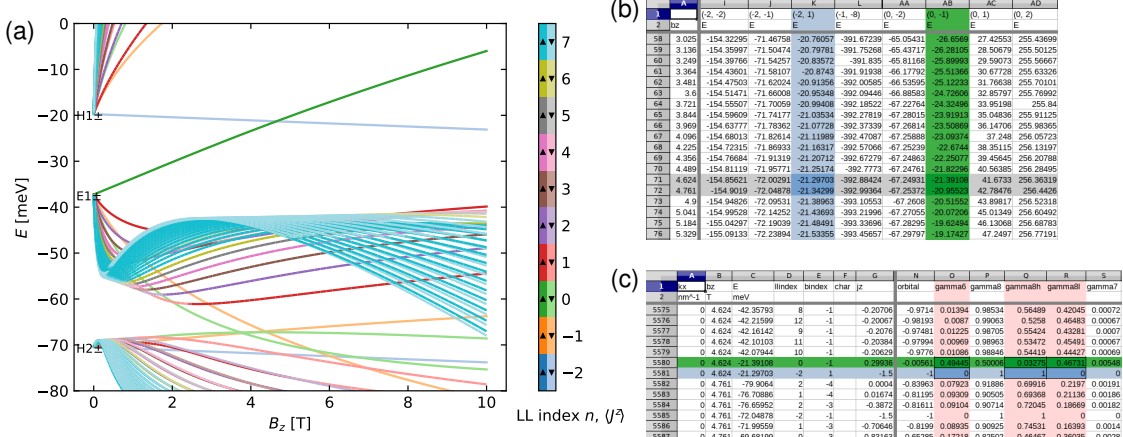

Figure 20: (a) The Landau level spectrum, `bdependence-7nm-landau.pdf`. The colours indicate the Landau level index $n$ and the shade (brighter/darker) indicates the sign of $\langle J_z \rangle$, see the colour legend. (b) Screenshot of `bdependence-7nm-landau.byband.csv` which provides the energies of the 'lowest' Landau levels $|n, b\rangle$ with $(n, b) = (-2, 1)$ and $(0, -1)$ as a function of the magnetic field $B$ (column A, labelled bz). (c) Screenshot of `bdependence-7nm-landau.csv` which contains the expectation values for all relevant observables for all states. In (b) and (c), some columns were hidden and the coloured highlights were added manually to lay emphasis on the lowest Landau level states near the crossing.

```
kdotpy ll 8o msubst CdZnTe 4% mlayer HgCdTe 68% HgTe HgCdTe 68%
llayer 10 7 10 zres 0.25 b 0 10 // 100 split 0.01 erange -80 0 nll 20
neig 240 targetenergy 0
```

6. For colouring the states, let us choose a combination of Landau level index and total angular momentum. We set this by `obs llindex.jz`. Again, we include the figure legend with `legend` and the band characters with `char`.

```
kdotpy ll 8o msubst CdZnTe 4% mlayer HgCdTe 68% HgTe HgCdTe 68%
llayer 10 7 10 zres 0.25 b 0 10 // 100 split 0.01 erange -80 0 nll 20
neig 240 targetenergy 0 obs llindex.jz legend char
```

7. We choose a label for the filenames `out -7nm-landau` and the folder where the files should go `outdir data-landau`.

```
kdotpy ll 8o msubst CdZnTe 4% mlayer HgCdTe 68% HgTe HgCdTe 68%
llayer 10 7 10 zres 0.25 b 0 10 // 100 split 0.01 erange -80 0 nll 20
neig 240 targetenergy 0 obs llindex.jz legend char out -7nm-landau
outdir data-landau
```

### 5.3.3  Results and interpretation

After running this command, we get the following files in the folder `data-landau`:

- `bdependence-7nm-landau.pdf`: A plot of the Landau level spectrum, i.e., the eigenenergies of the states $|n, b\rangle$ as function of the magnetic field, see Fig. 20(a).

- `bdependence-7nm-landau.csv`: A csv file where all data points (magnetic field, energy) are given with the values of the observables. The points are ordered by magnetic field value.

- `bdependence-7nm-landau.byband.csv`: A csv file with the band energies for each curve, i.e., ordered by LL band.

- `output-7nm-landau.xml`: The XML file with all parameters and data.

We can extract sufficient information to answer the questions above:

- *Which Landau levels are the 'lowest' ones?*— Let us examine the plot of Fig. 20(a). The lowest LL index formally is **−2**, but the actual lowest index differs by subband character. Let us focus on the **H1±** and **E1±** subbands, as these are the most relevant for the physics. The '±' indicates the expectation value $\langle J_z \rangle$ at zero momentum and zero magnetic field. This degree of freedom is encoded in the colour shading, see the legend: dark colours represent the + states ($\langle J_z \rangle > 0$) and bright colours the − states ($\langle J_z \rangle < 0$).

  - For **H1+**, dark colours emanating from '**H1±**', we find that the lowest index is **1** (red colour); **0**, **−1**, and **−2** are missing.
  - For **H1−**, bright colours emanating from '**H1±**', the baby-blue LL can be identified as index **−2**, so all indices are present for **H1−**.
  - For **E1+**, dark colours emanating from '**E1±**', the lowest index is **0** (dark green).
  - For **E1−**, bright colours emanating from '**E1±**', the lowest index is **−1** (bright orange green).

  These observations can be confirmed from the data in the csv files. The lowest LLs near the gap are thus (**E1+**, $n = 0$) and (**H1−**, $n = -2$).

- *What is the critical field $B_c$ where the inversion of the lowest LLs is undone?*— In principle we could estimate this from Fig. 20(a), but we use one of the csv files in order to answer this question more precisely: In bdependence-7nm-landau.byband.csv, see Fig. 20(b), the bands are ordered in columns labelled by $(n, b)$ in the first row, where $n$ is the LL index and $b$ is the band index. The band indices are counted for each LL separately, which makes intuition a bit tricky for the lowest LLs ($n = -2, -1, 0$), but it is guaranteed that the LLs at the crossing have $(n, b) = (0, -1)$ and $(-2, 1)$ respectively. The respective columns are highlighted. We find the crossing between **4.624 T** and **4.761 T**. Linear interpolation may be used as to obtain a more accurate value.

- *What orbital character do the states have at this crossing?*— From bdependence-7nm-landau.csv, depicted in Fig. 20(c), we extract the orbital character from the respective states at $B = 4.624$ T (which lies closer to the actual crossing than $B = 4.761$ T. The desired information is encoded in the observables gamma6, gamma8h, and gamma8l for the two respective states there. The lowest LL coming from the **E1+** band, $(n, b) = (0, -1)$, is a mixture of **49.4%** $\Gamma_6$, **46.7%** $\Gamma_{8L}$, and **3.3%** $\Gamma_{8H}$. We thus find that it is predominantly a $J_z = \pm\frac{1}{2}$ state. The one from the **H1−** band is purely $\Gamma_{8H}$, and has $\langle J_z \rangle = -\frac{3}{2}$ exactly.

## 5.4 Example calculation: Magneto-transport

### 5.4.1 Introduction

In the previous example calculation, we have set up a basic LL calculation. With some postprocessing in kdotpy, we are able to extract much more useful information and simulate magneto-transport experiments. The key ingredients here are density of states (DOS) and Chern numbers (or Berry curvature). We will set up an example simulation, where we will address the following questions:

- Which Landau levels are filled at the constant carrier density $2 \times 10^{11}$ cm$^{-2}$?
- Do we see plateaus in the Hall resistance $R_{xy}$?
- Are there Shubnikov-de Haas (SdH) oscillations at low magnetic fields?
- Can we simulate a Landau fan such that it resembles experimental data?

### 5.4.2 Setting up the simulation step by step

We take the the command line from Sec. 5.3 as a starting point:

```
kdotpy ll 8o msubst CdZnTe 4% mlayer HgCdTe 68% HgTe HgCdTe 68%
llayer 10 7 10 zres 0.25 b 0 10 // 100 split 0.01 erange -80 0 nll 20
neig 240 targetenergy 0 obs llindex.jz legend char out -7nm-landau
outdir data-landau
```

We modify and add command line arguments as follows:

1. We change some parameters slightly for this calculation. We choose `erange -60 40` instead of `erange -80 0`, in order to give a better view of the data. We increase the number of eigenvalues from `neig 240` to `neig 300`, so that we obtain a sufficient number of states to view this energy range. We also change the output filenames and output folder.

   ```
   kdotpy ll 8o msubst CdZnTe 4% mlayer HgCdTe 68% HgTe HgCdTe 68%
   llayer 10 7 10 zres 0.25 b 0 10 // 100 split 0.01 erange -60 40 nll 20
   neig 300 targetenergy 0 obs llindex.jz legend char out -7nm-hall
   outdir data-hall
   ```

2. In order to calculate the density of states, we add the options `dos` and `localdos` for calculating the total and local density of states, respectively. For the DOS, we use Gaussian broadening as in Ref. [3], with a broadening width (standard deviation) $\sigma(B) = \sigma_1 \sqrt{B\,[\mathrm{T}]}$ with $\sigma_1 = 0.5\,\mathrm{meV}$ (see also Appendix B.6). The appropriate command is `broadening gauss 0.5 sqrt`, but this may be shortened to `broadening 0.5`; for `kdotpy ll`, `gauss` and `sqrt` are default settings for the broadening.

   ```
   kdotpy ll 8o msubst CdZnTe 4% mlayer HgCdTe 68% HgTe HgCdTe 68%
   llayer 10 7 10 zres 0.25 b 0 10 // 100 split 0.01 erange -60 40 nll 20
   neig 300 targetenergy 0 obs llindex.jz legend char out -7nm-hall
   outdir data-hall dos localdos broadening 0.5
   ```

3. For calculating the Hall conductance, we need to calculate the Chern numbers and set its broadening correctly. We add `chern` to calculate the Chern numbers.

   ```
   kdotpy ll 8o msubst CdZnTe 4% mlayer HgCdTe 68% HgTe HgCdTe 68%
   llayer 10 7 10 zres 0.25 b 0 10 // 100 split 0.01 erange -60 40 nll 20
   neig 300 targetenergy 0 obs llindex.jz legend char out -7nm-hall
   outdir data-hall chern dos localdos broadening 0.5
   ```

4. We replace `broadening 0.5` by `broadening 0.5 10%` to set the broadening of the Hall conductance as function of energy to **10%** of that of the carrier density (integrated density of states).

   ```
   kdotpy ll 8o msubst CdZnTe 4% mlayer HgCdTe 68% HgTe HgCdTe 68%
   llayer 10 7 10 zres 0.25 b 0 10 // 100 split 0.01 erange -60 40 nll 20
   neig 300 targetenergy 0 obs llindex.jz legend char out -7nm-hall
   outdir data-hall chern dos localdos broadening 0.5 10%
   ```

5. For convenience, one may also use the command shortcut `hall`, which is equivalent to `chern dos localdos broadening 0.5 10%`.

   ```
   kdotpy ll 8o msubst CdZnTe 4% mlayer HgCdTe 68% HgTe HgCdTe 68%
   llayer 10 7 10 zres 0.25 b 0 10 // 100 split 0.01 erange -60 40 nll 20
   neig 300 targetenergy 0 obs llindex.jz legend char out -7nm-hall
   outdir data-hall hall
   ```

   This command line is equivalent to the one in step 4.

For this calculation, the following configuration options are relevant:

- `dos_unit`: The plots in Fig. 21 have been produced with `dos_unit=cm`, so that densities are expressed in units of **cm$^{-2}$** with an appropriate power of ten.

- `dos_energy_points`: The default energy resolution used for DOS calculations (etc.) is `dos_energy_points=1000`, which is usually is a good compromise between accuracy and computation time. If the value is insufficiently large, one may see visible artifacts, like jagged equal-density lines where one would expect smooth curves. If that is the case, one should increase this value.

- `berry_ll_simulate`: By default (`berry_ll_simulate=false`), the plots related to Hall conductance are extracted from the calculated Chern numbers (Berry curvature), stored as the observable `chern`. When the number of states, set by command-line argument `neig`, is not sufficient, some values are inaccurate or incorrect; see Sec. 3.5.3 for a detailed discussion. In bad cases, this renders the data to be partially or completely unusable. One can mitigate this problem by setting `berry_ll_simulate=true`. In this case, the output functions use a simulated Chern number, which is exactly **1** for all Landau-level eigenstates and is stored as the observable `chernsim`. This value coincides with the calculated Chern number (up to numerical error) for almost all states, except for states the edges if the spectrum, which should usually be mistrusted anyway. If `berry_ll_simulate=true`, the output file names will be changed by insertion of `sim` or `simul` in order to be able to easily spot whether the calculated or simulated Chern numbers have been used. It is recommended to run `kdotpy` with both settings and to compare the results at least once in order to get an intuition for the implications of this setting and for the physics behind it.

The configuration values may be set permanently with

```
kdotpy config 'dos_unit=cm;dos_energy_points=1000;berry_ll_simulate=true'
```

or used once by appending

```
config 'dos_unit=cm;dos_energy_points=1000;berry_ll_simulate=true'
```

to the command line that starts `kdotpy ll`.

### 5.4.3 Results and interpretation

The above commands produce many pdf and csv files, and here we shall only discuss the ones relevant for answering the questions above. In most cases, the pdf is accompanied by a csv file which contains exactly the data shown in the figure.

- *Which Landau levels are filled at the constant carrier density* $2 \times 10^{11} \, \text{cm}^{-2}$*?—* The answer to this question of course depends on the magnetic field. It is encoded in the file `bdependence-density.pdf`, see Fig. 21(a), which is generated when `kdotpy ll` is used with the `dos` option. Equal-density contours are overlaid onto the magnetic-field dependence plot, (LL energies as function of magnetic field $B$). In Fig. 21(a), we find the equal-density curve for $n = 2 \times 10^{11} \, \text{cm}^{-2}$ to be at approximately **20 meV** for low magnetic fields. A different visualization may be obtained by adding `cardens 0.002` to the command line, in which case only the contour line for the given carrier density ($0.002 \, \text{nm}^{-2} = 2 \times 10^{11} \, \text{cm}^{-2}$) is shown.

- *Do we see plateaus in the Hall resistance* $R_{xy}$*?—* The Hall resistance is shown in the plot `rxy-constdens-7nm-hall.pdf`, see Fig. 21(b). We find plateaus corresponding to the filling factors $\nu = 1, 2, \ldots$. At higher filling fractions (lower magnetic fields), the plateau transitions are more washed out due to the broadening. Here, the curve tends more towards the classical value $R_{xy} = B/ne$, indicated by the dashed line. This type of visualization can be compared directly to experimental results. For completeness, `kdotpy` also generates the plot `sigmah-constdens-7nm-hall.pdf`, which shows the Hall conductance $\sigma_H$ as function of magnetic field for constant carrier density. In both cases, these output files correspond to the density specified by `cardens`, if this argument is provided on the command-line. Otherwise, the output will be multi-page PDFs with multiple densities, namely integer multiples of $1 \times 10^{11} \, \text{cm}^{-2}$.

- *Are there Shubnikov-de Haas (SdH) oscillations at low magnetic fields?—* The constant-carrier-density curves in Fig. 21(c) clearly show oscillations with magnetic field. The plot shows density of states, which is thought to correlate with the longitudinal resistance $R_{xx}$. The red markers at the bottom indicate the values of $1/B$ equal to integer multiples of $e/hn$, where $n$ is the carrier density. The minima of the density of states thus align well with the multiples of $e/hn$ for $B \gtrsim 1 \, \text{T}$. At lower fields, the resolution of $B$ values is insufficient to make any conclusive statement. This may easily be mitigated by an additional calculation at lower fields, for example with `b 0 1 // 100`.

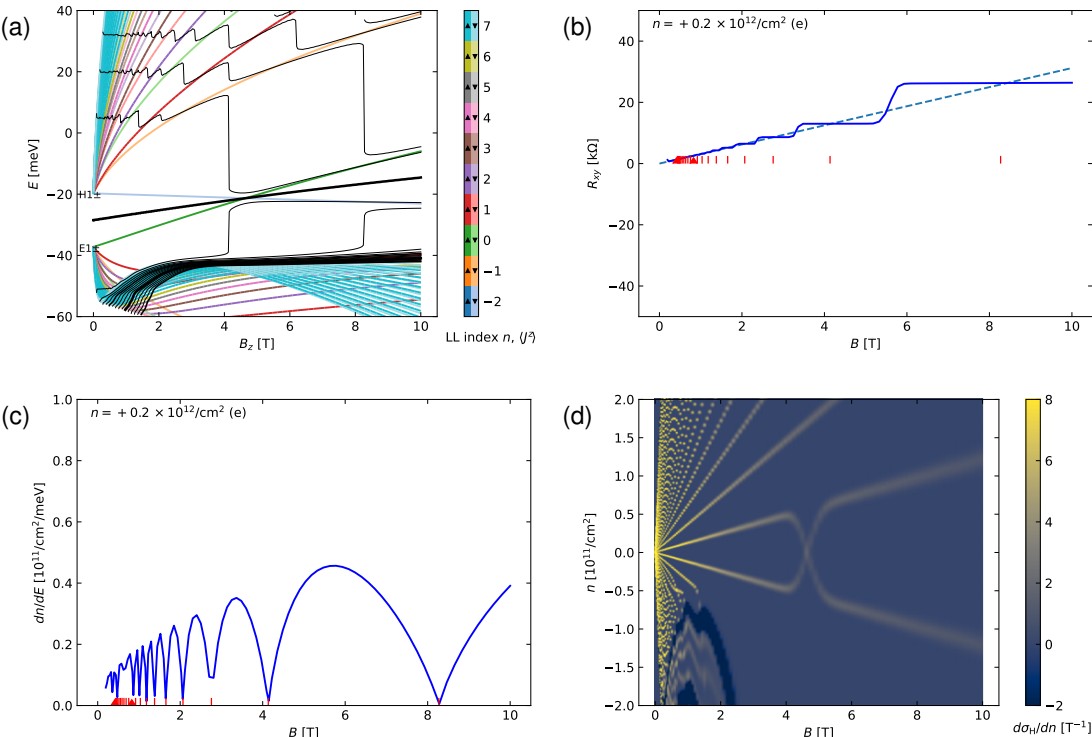

Figure 21: (a) In the file `bdependence-density.pdf`, equal-density contours are over-laid on top of the magnetic field depdendence plot, cf. Fig. 20(a). The thick line near **−20 meV** is the charge neutral point **$n = 0$**. Contours are drawn for each multiple of **$1 \times 10^{11}\,\mathrm{cm}^{-2}$** for **$|n| \leq 15 \times 10^{11}\,\mathrm{cm}^{-2}$**. The medium-thick lines are at odd multiples of **$5 \times 10^{11}\,\mathrm{cm}^{-2}$** and the very thick lines are at multiples of **$10 \times 10^{11}\,\mathrm{cm}^{-2}$**. (b) The plot in `rxy-constdens-7nm-hall.pdf` shows the simulated Hall resistance **$R_{xy} \equiv 1/\sigma_{\mathrm{H}}$** as function of magnetic field **$B$** for the constant density **$2 \times 10^{11}\,\mathrm{cm}^{-2}$**. The dashed line indicates the classical Hall slope given by **$R_{xy} = B/ne$**. (c) The file `dos-constdens-7nm-hall.pdf` visualizes the density of states as function of magnetic field for a constant carrier density. (d) In `dsigmah-dn-7nm-hall.pdf`, we plot the derivative **$d\sigma_{\mathrm{H}}/dn$** of the Hall conductance with respect to the carrier density **$n$**. This type of plot can easily be compared to plots obtained from experiments which measure Hall conductance as function of magnetic field and gate voltage. The figures (b) and (c) are single pages in a multipage PDF unless `cardens` has been used in order to specify a single density. For this figure, we have used the configuration settings `dos_unit=cm`, `dos_energy_points=1000`, and `berry_ll_simulate=false`.

- *Can we simulate a Landau fan such that it resembles experimental data?*— In magneto-transport experiments, Landau fans are obtained by measuring $R_{xy}$ while sweeping the gate voltage for many values of the magnetic field. This yields $R_{xy}$ as function of magnetic field $B$ and gate voltage $V_g$. In simulations, however, the natural quantity on the 'vertical axis' is energy. The DOS calculation in kdotpy bridges this gap: The relation of carrier density $n$ as function of energy $E$ can be reversed as to obtain a spectrum as function of density. The latter often correlates approximately linearly with gate voltage $V$, so that comparison between the two is physically meaningful. In Fig. 21(d), we visualize the plateau transitions by taking the derivative $d\sigma_H/dn$ in order to highlight the plateau transitions. This method is also customarily applied to experimental data.

## 5.5 Example calculation: Optical transitions

### 5.5.1 Introduction

Instead of extracting information on magneto-transport experiments in postprocessing, we can also simulate magneto-optical transition spectra in kdotpy. We will use an example simulation to answer the following questions:

- How large is the band gap of the system?

- What are the involved LL states for a specific transition?

- What information regarding band ordering can we extract from the transition spectrum?

### 5.5.2 Setting up the simulation step by step

Again, we take the the command line from Sec. 5.3 as a starting point:

```
kdotpy ll 8o msubst CdZnTe 4% mlayer HgCdTe 68% HgTe HgCdTe 68%
llayer 10 7 10 zres 0.25 b 0 10 // 100 split 0.01 erange -80 0 nll 20
neig 240 targetenergy 0 obs llindex.jz legend char out -7nm-landau
outdir data-landau
```

We modify and add command line arguments as follows:

1. As for the magneto-transport analysis, we slightly adjust some parameters. We choose erange -80 50 for a better view of the data. We increase the number of eigenvalues from neig 240 to neig 300 to get a sufficient number of states to view in this energy range. We also change the output filenames and output folder.

   ```
   kdotpy ll 8o msubst CdZnTe 4% mlayer HgCdTe 68% HgTe HgCdTe 68%
   llayer 10 7 10 zres 0.25 b 0 10 // 100 split 0.01 erange -80 50 nll 20
   neig 300 targetenergy 0 obs llindex.jz legend char
   out -7nm-optical-transitions outdir data-optical-transitions
   ```

2. In order to calculate all optical transitions, we add the option transitions.

   ```
   kdotpy ll 8o msubst CdZnTe 4% mlayer HgCdTe 68% HgTe HgCdTe 68%
   llayer 10 7 10 zres 0.25 b 0 10 // 100 split 0.01 erange -80 50 nll 20
   neig 300 targetenergy 0 obs llindex.jz legend char
   out -7nm-optical-transitions outdir data-optical-transitions transitions
   ```

3. It is also advisable to filter transitions by state occupancy, since a transition can only be observed if the initial state is (partially) occupied while the final state is (partially) unoccupied. Thus, we add the option cardens 0, assuming a charge neutral sample. (There is also the option to calculate filtered transitions for multiple carrier densities at once by using the input cardens # # / #, analogous to the range input for b.) Additionally, we also use the options dos and broadening 0.5. dos enables calculation of the (electro-)chemical potential, which is used to determine the occupation of states, while broadening applies a Gaussian square-root broadening to the

LL states (shortened, see Sec. 5.4). Omitting broadening often leads to abrupt jumps in the (electro-)chemical potential, leading to unphysical gaps along specific transition features. Note, that broadening only is applied to DOS-related quantities, transition spectra will always be delta peaks.

Per default, these options suppress the output of all possible transitions.

```
kdotpy ll 8o msubst CdZnTe 4% mlayer HgCdTe 68% HgTe HgCdTe 68%
llayer 10 7 10 zres 0.25b 0 10 // 100 split 0.01 erange -80 50 nll 20
neig 300 targetenergy 0 obs llindex.jz legend char
out -7nm-optical-transitions outdir data-optical-transitions transitions
cardens 0 dos broadening 0.5
```

4. Last, we change some configuration values using `config`. With `transitions_max_deltae=80` we set the upper limit of the energy axis in the transition plot to **80 meV**, while `transitions_min_amplitude=3e11` filters out all transitions with transition matrix elements smaller than the given threshold, keeping only the most prominent transitions.

```
kdotpy ll 8o msubst CdZnTe 4% mlayer HgCdTe 68% HgTe HgCdTe 68%
llayer 10 7 10 zres 0.25 b 0 10 // 100 split 0.01 erange -80 50 nll 20
neig 300 targetenergy 0 obs llindex.jz legend char
out -7nm-optical-transitions outdir data-optical-transitions transitions
cardens 0 dos broadening 0.5
config 'transitions_max_deltae=80;transitions_min_amplitude=3e11'
```

The quantity that will be plotted in the transitions spectrum can be changed by using the configuration option `plot_transitions_quantity`. We will use the default value `rate`, plotting the rate density as defined in Eq. (121). More information on this topic can be found in Sec. 3.8.4.

### 5.5.3 Results and interpretation

Using the commands above produces additional output files. This includes `bdependence-transitions-7nm-optical-transitions.pdf`, where all filtered transitions are drawn as vertical lines into the LL fan chart, and `transitions-filtered-7nm-optical-transitions.pdf`, together with its csv file, showing the filtered optical transitions spectrum. We will address the question given above by discussing the transitions spectrum in Fig. 22(a).

- *How large is the band gap of the system?*— The interband transition between the highest valence and the lowest conduction subband at low magnetic fields extrapolates to the energetic value of the direct band gap, see e.g. Fig. 21(a). We suspect, that the transition feature marked with the blue arrow in Fig. 22(a) is this specific transition. To confirm this, we use the `transitions-filtered-7nm-optical-transitions.csv` file and filter the initial state B1 by highest valence subband and the final state B2 by lowest conduction subband. Per definition, these subbands, respectively, have the band index **−1** and **1**. The blue and green columns in Fig. 22(b) show these indices, while the red column shows the transition energies. The lowest calculated magnetic field bx is **0.001 T** and the corresponding transition energies `deltaE` are **> 17.5 meV**, confirming that this feature extrapolates to the same band gap as discussed in Sec. 5.2. For zero magnetic field all LLs in a subband are degenerate, thus, all transition energies would be identical (up to the artificial offset added by `split`).

- *What are the involved LL states for a specific transition?*— We analyse the transition marked with the black arrow in Fig. 22(a) regarding which subbands and LL indices are involved. We use the same output file `transitions-filtered-7nm-optical-transitions.csv` and filter the magnetic field values bx to be **> 6 T**, see Fig. 22(c), so that we can clearly distinguish this transition from other transitions at higher energies than shown in the plot. The marked columns are the initial (blue) and the final (green) LL (light) and band indices (dark). As can be seen, this transition is an intraband transition inside the subband with band index **1**, which is the inverted subband H1. The respective LL indices of initial and final state are **−2** and **−1**, hence the polarization of this transition corresponds to $O_+$ [see Sec. 3.8].

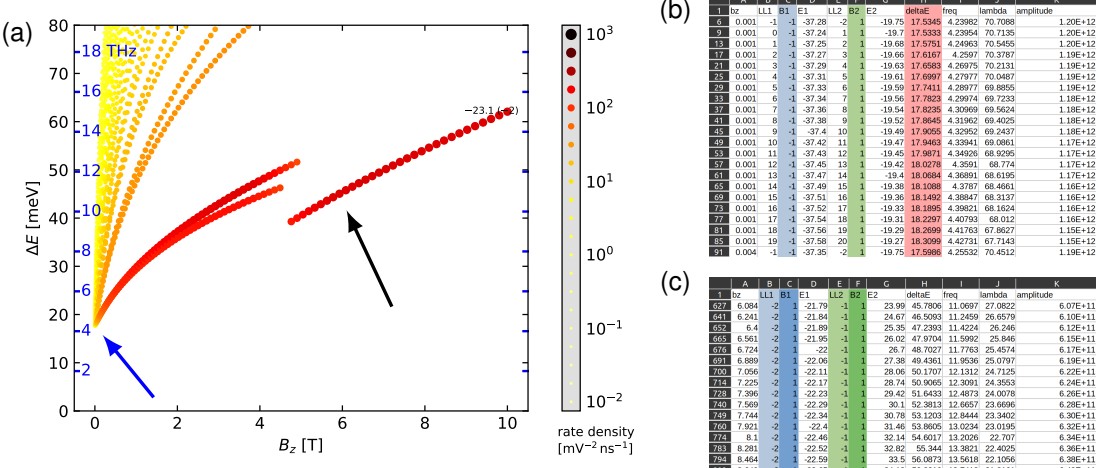

Figure 22: (a) The filtered transitions spectrum, `transitions-filtered-7nm-optical-transitions.pdf`. The strength of each individual transition is indicated by the size and colour of its respective symbol. The blue arrow points towards the transition feature that extrapolates to the band gap energy at small magnetic fields, while the black arrow indicates the feature for which we want to find the involved LL states. Both were added in post-production and are normally not include in the plot. The screenshots on the right are taken from the `transitions-filtered-7nm-optical-transitions.csv` file. In (b) the data is filtered by the band indices B1 and B2, taking the values **−1** and **1**, respectively. For (c) data was filtered for magnetic fields bx ≥ **6 T**. The Landau level spectrum corresponding to this figure is the one shown in Fig. 21.

- *What information regarding band ordering can we extract from the transition spectrum?*— Continuing the analysis for the same transition, we discuss why it can only be observed at magnetic fields ≳ **5 T**. For that, we use the LL fan chart shown in the previous tutorial section Fig. 21, where the same sample was simulated. The relevant (electro-)chemical potential for the current case is the thick line near **−20 meV**, the charge neutral point. At magnetic fields slightly below **5 T** the lowest LL band of H1 and E1 cross, reverting to a trivial band order, while simultaneously changing from completely unoccupied to fully occupied in the case of the H1 state (vice versa for the E1 state). This enables the corresponding intraband transition in the H1 subband, which we see in the transition spectrum. Consequently, we can indirectly identify the critical magnetic field, where the lowest LL states revert to trivial band order in the magneto-optical transition spectrum. The abruptness of the crossing is a consequence of the axial approximation, which we used here for illustrative purposes. In a more realistic picture, the non-axial and bulk-inversion asymmetric terms lead to an anticrossing instead.

## 5.6 Example calculation: Dispersions and wave functions of a 3D TI

### 5.6.1 Introduction

In thick layers of a band inverted material like HgTe, so called *three-dimensional topological insulators*, the transport properties are dominated by topological surface states. At an interface between materials with inverted and normally ordered bands, the band inversion induces eigenstates with a linear (two-dimensional Dirac) dispersion, confined to the interface [43]. The physics is markedly different from those of narrow quantum wells (see Sec. 5.2), where instead the strong confinement in the *z* direction leads to the inversion of subbands.

In this usage example, we investigate the key features of a 3D topological insulator by analyzing the dispersions and wave functions. We try to answer the following questions:

- Can we identify the Dirac point and the surface states in the dispersion?

- How well are the surface states confined?

- How much is the overlap between the surface states?

- What is the orbital content of the wave functions?

### 5.6.2 Setting it up step by step

1. Despite the system being called '3D topological insulator', the appropriate geometry is still 2D, because there is confinement in the $z$ direction. We take the previous example, Sec. 5.2, and adjust the layer stack parameters `msubst` and `llayer`. We also use a different momentum and energy range. We raise the number of eigenvectors with `neig 100`, because there are more subbands due to weaker confinement in the $z$ direction. We choose the observable $z$ in order to see whether we can locate the surface states at the top and bottom interface.

   ```
   kdotpy 2d 8o noax msubst CdTe mlayer HgCdTe 68% HgTe HgCdTe 68%
   llayer 10 70 10 zres 0.25 k -0.3 0.3 / 120 kphi 0 split 0.01 erange -100 40
   neig 100 obs z legend char out -70nm outdir data-3dti
   ```

2. We obtain the dispersion plot of Fig. 23(a). At $E \approx -90\,\text{meV}$, we observe the linear dispersions characteristic of a Dirac point. Between valence band and conduction bands (negative and positive energies, respectively), there are states bridging the gap. The gray colour indicates that the expectation values $\langle z \rangle \approx 0$ for all states, which seemingly suggests absence of surface character. In order to diagnose this counterintuitive result, let us plot plot the wave functions at $k = 0.14\,\text{nm}^{-1}$ using `plotwf separate 0.14`.

   ```
   kdotpy 2d 8o noax msubst CdTe mlayer HgCdTe 68% HgTe HgCdTe 68%
   llayer 10 70 10 zres 0.25 k -0.3 0.3 / 120 kphi 0 split 0.01 erange -100 40
   neig 100 obs z legend char out -70nm outdir data-3dti plotwf separate 0.14
   ```

3. We obtain two files with wave functions, i.e., at $k = -0.14\,\text{nm}^{-1}$ and at $k = 0.14\,\text{nm}^{-1}$. In the latter file, let us look for one of the states at $E \approx 13.5\,\text{meV}$, see Fig. 23(b). The wave function contains symmetric and antisymmetric components for different orbitals. In order to break the degeneracy due to the mirror symmetry in $z$ direction, let us add a weak potential difference between top and bottom surface with `vinner 0.1`. This argument leads to an electrostatic potential consistent with a constant displacement field $\mathbf{D} = \epsilon(z)\mathbf{E}(z)$, such that the potential difference between top and bottom surface is **0.1 meV**.

   ```
   kdotpy 2d 8o noax msubst CdTe mlayer HgCdTe 68% HgTe HgCdTe 68%
   llayer 10 70 10 zres 0.25 k -0.3 0.3 / 120 kphi 0 split 0.01 erange -100 40
   neig 100 obs z legend char out -70nm outdir data-3dti plotwf separate 0.14
   vinner 0.1
   ```

### 5.6.3 Results and interpretation

After applying `vinner 0.1` in order to break the mirror symmetry, we obtain `dispersion-70nm.pdf` and `wfs-70nm_0.140_0.pdf` as shown in Fig. 23(c) and (d). With these plots and the corresponding csv files, we can answer the questions above:

- *Can we identify the Dirac point and the surface states in the dispersion?*— In Fig. 23(c), we find two pairs of linearly dispersing states, crossing at $E \approx -89\,\text{meV}$ at zero momentum. The subband labels are **E1±** and **L1±**. The states emanating from the Dirac point have a distinct surface character is indicated by the red and blue colour (where the latter is poorly visible because the dispersions almost coincide).

  These four Dirac states can be described by a massless two-dimensional Dirac Hamiltonian. We note that the Dirac model for the surface states is distinct from the BHZ model [39], which has a different set of basis states, namely the **E1±** and **H1±** subbands; see also the example in Sec. 5.2.

  The surface states seem to disappear into the bulk valence band for energies $-70\,\text{meV} \le E \le -10\,\text{meV}$ approximately, due to hybridization with the bulk valence band. In the bulk gap ($0\,\text{meV} \le E \le 20\,\text{meV}$ approximately), the surface states cross the gap from valence to conduction band.

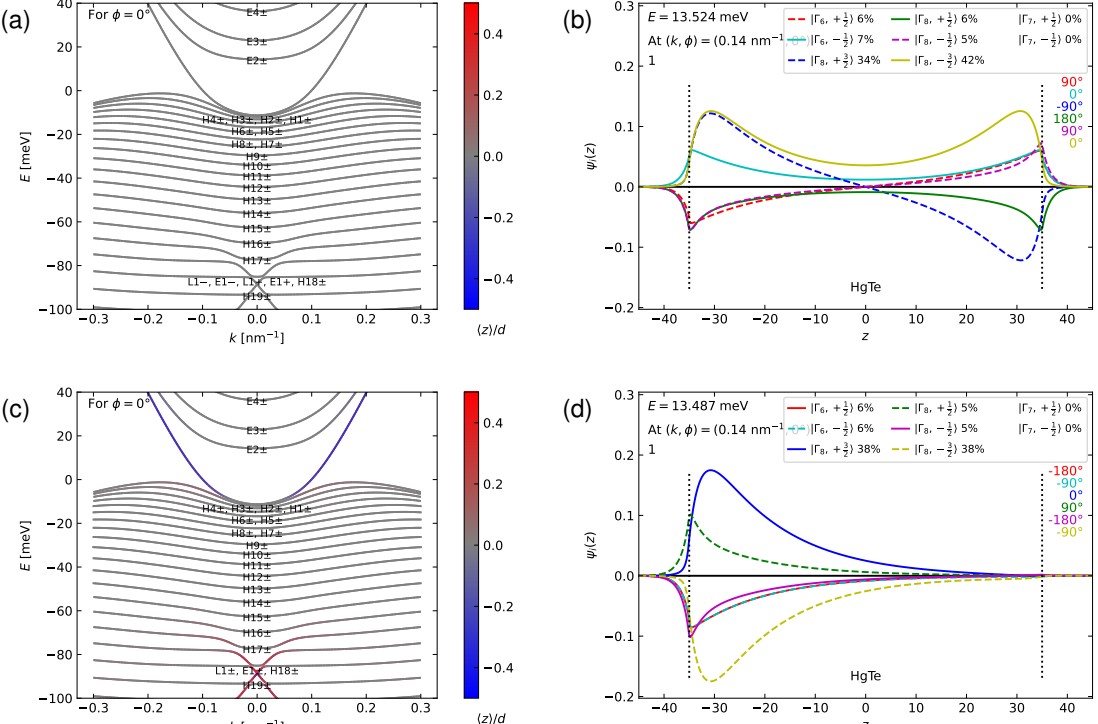

Figure 23: (a) Dispersion of a **70 nm** thick HgTe layer on a CdTe substrate, `dispersion-70nm.pdf`. The colours indicate the expectation value $\langle z \rangle$. Due to mirror symmetry $z \to -z$, $\langle z \rangle = 0$ for all states. (b) A wave function at $k = 0.14\,\text{nm}^{-1}$ and $E \approx 13.5\,\text{meV}$. The solid curves indicate the real parts, the dashed curves the imaginary parts for each orbital component. (c) The dispersion with a degeneracy breaking potential (argument `vinner 0.1`). The surface character of the states in the gap and near the Dirac point is indicated by red and blue colour. States with bulk character are gray. (d) One of the surface states at $k = 0.14\,\text{nm}^{-1}$ and $E \approx 13.5\,\text{meV}$. Clearly, the degeneracy is resolved. The angles on the right-hand side indicate the complex arguments in each orbital component.

- *How well are the surface states confined?*— The level of confinement can be extracted from the wave functions. In this example, we investigate one of the surface states in the bulk gap, at $k = 0.14\,\mathrm{nm}^{-1}$ and $E \approx 13.5\,\mathrm{meV}$. From the wave functions shown in Fig. 23(d), we estimate a characteristic 'size' of about **10–20 nm**.

  For a more rigorous value, we can take the file {wfs-70nm_0.140_0.1.csv (depending on the configuration settings, this might be inside a tar or zip archive) and calculate $|\psi|^2(z) = \sum_i |\psi_i|^2(z)$, where $i$ are the orbitals. We determine the characteristic length $\xi$ by fitting $\psi(z) \sim e^{-z/\xi}$. From approximating $\xi^{-1} \approx \partial_z \ln(|\psi|^2)$ by a discrete derivative around $z = 0$, we find $\xi \approx 7\,\mathrm{nm}$. Note that the apparent discrepancy with the estimate below comes from the difference between $\psi_i(z)$ and $|\psi|^2(z)$.

- *How much is the overlap between the surface states?*— In order to estimate the overlap between the surface states at top and bottom surface, we calculate the integral of $|\psi|(z)$ for $z > 0$, for one of the surface states. We find $\int_0^{z_{\max}} |\psi|^2(z) \approx 0.01$ by summing up the appropriate values in wfs-70nm_0.140_0.1.csv. We finally point out that the overlap and the confinement length are momentum dependent, so that for a complete picture of the physics, this analysis has to be repeated for other momenta.

- *What is the orbital content of the wave functions?*— The data in Fig. 23(d) visualizes the wave function as $\psi(z) \sum_i \psi_i(z)|i\rangle$, where $|i\rangle$ are the orbital basis functions (see Eq. 10). The curves being solid or dashed indicate the $\psi_i(z)$ being real or imaginary. In many cases, the functions $\psi_i(z)$ can be factorized as a complex constant $e^{i\phi_i}$ times a real function $f_i(z)$. The angles in degrees in the right hand side of Fig. 23(d) indicate the complex arguments $\psi_i$. Thus, the wave function for this state can be written as

$$\psi(z) = f_{1,2}(z)(-|1\rangle - i|2\rangle) + f_{3,6}(z)(|3\rangle - i|6\rangle) + f_{4,5}(z)(i|4\rangle - |5\rangle) + f_{7,8}(z)(i|7\rangle + |8\rangle) \tag{136}$$

  in terms of four real functions $f_{i,i'}(z)$, where components $i$, $i'$ with identical shapes of the envelope function have been taken together pairwise.

## 5.7 Auxiliary scripts

### 5.7.1 kdotpy merge

The subprogram `kdotpy merge` imports multiple XML files and merges the result into a single data set and plots the result in a single dispersion or magnetic-field dependence plot. (Further postprocessing is not supported in version 1.0.0 of kdotpy.) The merging is done 'horizontally', i.e., data sets with different values of $k$ or $B$) as well as 'vertically', i.e., with data at the same values of $k$ and $B$, but for example with respect to different target energies. The command-line syntax may be as simple as

```
kdotpy merge data/output.1.xml data/output.2.xml
```

Usually, additional options are provided, for example setting the output filenames and directory.

```
kdotpy merge bandalign -25 2 out .merged
outdir data-merged -- data/output.1.xml data/output.2.xml
```

The `--` (double hyphen) symbol separates the input file names from the other options. It may be omitted if there is no potential confusion between the other options and the input file names. The command line option `bandalign` can be used to do or redo band alignment on the combined data set. Manually assisted realignment as described in Sec. 3.6.4 is also supported.

For data integrity, `kdotpy merge` compares the physical parameters associated with the source data files (stored in the `` tag). If values differ, it will show a warning as to inform the user that the data sets might not fit together. There is also a check on whether the grids of the data files are compatible, i.e., whether the combination of the grids can also be represented as a `VectorGrid`.

One of the important use cases of `kdotpy merge` is the ability to split up computationally demanding computations into more manageable pieces. For example, strip calculation with `kdotpy 1d` (which has a high RAM footprint) can be split up across multiple nodes of a high-performance computing cluster. Another boon of `kdotpy merge` is that incomplete data sets can be completed with just the missing data and using the already existing data to save calculation time. This situation may occur,

for example, when one out of many cluster job fails, or when in hindsight the value of $n_{eig}$ was set too low to cover the desired certain energy interval. In Refs. [4] and [6], we have used this strategy for the strip calculations with kdotpy 1d, (with kdotpy versions v0.42 and v0.64, respectively [6]. For the dispersion figures, **3–10** data sets were merged.

### 5.7.2  kdotpy compare

The subprogram kdotpy compare is similar to kdotpy merge, but it can take an arbitrary number of data files and merge them into multiple data sets, which can then be compared in a single plot. The command-line syntax is similar to kdotpy merge, however data sets must be separated using vs, as in the following example,

```
kdotpy compare data/output.1a.xml data/output.1b.xml vs data/output.2.xml
```

In this example, the first data set is constructed by merging the data from data/output.1a.xml and data/output.1b.xml, and the second data set is the single file data/output.2.xml. The two data sets are visualized with different markers and/or colours. It is also possible to enter more than two data sets. The syntax for extra options is similar to that of kdotpy merge.

kdotpy compare only handles one-dimensional dispersions and magnetic field dependence. It is primarily used for quick comparisons between two related systems, for example to study the effect of enabling or disabling a term in the Hamiltonian. It is also useful as a debugging tool.

### 5.7.3  kdotpy batch

The auxiliary subprogram kdotpy batch is a tool for running any of the calculation subprograms in batch mode, where many calculations are run in sequence, but with a variation in one or more parameters. The philosophy is similar to a shell (bash) script, but kdotpy batch may have a slightly more intuitive way of iterating over the parameters and it can handle runs in parallel.

The behaviour is best illustrated from an example command line,

```
kdotpy batch [opts] @x 5.0 9.0 / 4 @y [CdTe, HgTe, HgCdTe 68%] do kdotpy 2d
8o noax llayer 10 @x 10 mater HgCdTe 68% @y HgCdTe 68% out -@0of@@ ...
```

This kdotpy batch command runs iteratively the kdotpy subprogram after the argument do. Here, kdotpy 2d is run fifteen times, with @x replaced by the values 5.0, 6.0, 7.0, 8.0, and 9.0, and @y replaced by CdTe, HgTe, and HgCdTe 68%. The command syntax for ranges is the same as for the grid components, see Sec. 3.2.3. The special notations @0 and @@ denote a counter and the total value respectively, i.e., @0of@@ yields 1of15, 2of15, etc. All '@ variables' defined before do may appear after do an arbitrary number of times. Curly brackets may be used (e.g., @{x}) if variables appear inside a longer string.

Multiple processes may run in parallel with the cpu and proc arguments, to be specified before do. These options also take into account the number of CPUs of each process, if it is given by the argument cpu after do. Another useful argument is dryrun, which prints the command lines that result from the '@ substitutions', but does not execute the tasks.

In a batch calculation, the console output of the calculation scripts is written to the files stdout.1.txt, stdout.2.txt, etc. and stderr.1.txt, stderr.2.txt, etc., where the extension can be adjusted with the configuration values batch_stdout_extension and batch_stderr_extension. The kdotpy batch subprogram itself writes to the standard stdout and stderr streams. At the end of the batch run, kdotpy batch provides a status message that summarizes how many tasks were completed successfully.

### 5.7.4  kdotpy config

Whereas the user may edit the configuration file ~/.kdotpy/kdotpyrc directly, we also provide a command line tool kdotpy config, which can do the following simple operations on the configuration:

- kdotpy config list or kdotpy config show: Show all non-default configuration values.

---

[6]The actual versions used were development versions preceding v0.42 (for Ref. [4]) and v0.64 (for Ref. [6]).

- `kdotpy config all`: Show all configuration values.

- `kdotpy config help key`: Show information on `key` (taken from the help file; multiple `key` arguments possible).

- `kdotpy config set key=value`: Set key to a new value (multiple `key=value` pairs possible).

- `kdotpy config reset key`: Set key to its default value (multiple key arguments possible).

- `kdotpy config file`: Print the full path of the configuration file.

- `kdotpy config edit`: Open the configuration file in an editor. (Use environment variable `VISUAL` or `EDITOR` to set your command line editor.)

- `kdotpy config fig_lmargin=10 fig_rmargin=10 fig_hsize` (no operation specified): Set the values for `fig_lmargin` and `fig_rmargin`, show all three values.

Multiple `key` and/or `key=value` can also be combined with a semicolon `;`, for example

`kdotpy config 'fig_lmargin=10;fig_rmargin=10;fig_hsize'`

(In bash one must use single quotes because of the special meaning of `;`.) The syntax is thus analogous to the `config` argument for the calculation subprograms, for example `kdotpy 2d`.

### 5.7.5 kdotpy test

The package includes a set of standardized tests that can be run with the command `kdotpy test`. The purpose of these tests is to catch errors during code development and to catch performance issues. Each standardized test is defined in `kdotpy-test.py` by a command line. Success or failure is determined by the exit code. The output (by default written into the subdirectory `test` relative to the current working directory) is not checked for errors and has to be inspected manually, if desired. After all tests have been attempted, a summary is printed to the terminal indicating success or failure and the time it took to run.

Currently, in kdotpy version 1.0, there are 19 standardized tests, labelled with the following test ids: `1d`, `2d_cartesian`, `2d_offset`, `2d_orient`, `2d_polar`, `2d_qw`, `2d_qw_2`, `2d_qw_bia`, `2d_selfcon`, `batch`, `bulk`, `bulk_3d`, `bulk_ll`, `ll_axial`, `ll_bia`, `ll_legacy`, `compare_2d`, `compare_ll`, and `merge`. The full test suite may be run by typing

`kdotpy test`

To run only a selection (of one or more tests), one may add the test ids as arguments, for example

`kdotpy test 2d_qw ll_axial`

to run the specified ones only. The test suite may be interrupted with Ctrl-C.

The test suite supports the following extra commands:

- `kdotpy test list`: Shows all test ids on the terminal.

- `kdotpy test showcmd testids`: Shows the command lines without running the tests. The command lines may be copied, modified, and run manually for the purpose of testing and debugging.

- `kdotpy test verbose testids`: Runs the tests in verbose mode, by adding the argument `verbose` to their command lines.

- `kdotpy test python3.9 testids`: Uses a specific Python command (in this example `python3.9`) for running the test commands. It is recommended to use a virtual environment for this purpose.

In these commands, `testids` may be any sequence of test ids, or it may be omitted to apply the command to all 19 tests.

The tests are defined in `kdotpy-test.py` in a way that is compatible with the `pytest` package for automated testing. We have included automated testing with `pytest` into the CI/CD workflow in our Gitlab project.

# 6 Conclusion and outlook

This publication marks the public release of kdotpy as an open source software project. We release it with the expectation that kdotpy is beneficial to the research community. Conversely, the open source nature encourages input from the community as to improve and extend the project even further. It is our hope that interaction with the community will lead to new insights and new directions for the future development of kdotpy. We encourage our users to actively participate in the discussions in our Gitlab repository [42] or via other channels.

At the time of writing, we are already in the process of designing new features for future releases. Improved strain handling is scheduled to be included in an upcoming release in the near future. The treatment of electrostatics is currently undergoing continuous improvement, where in particular we are reconsidering the boundary conditions appropriate for several experimental setups. We will also add the necessary infrastructure for treatment of dopants in the material. We also expect to gain more experience with materials other than the built-in ones. In general, the future directions of kdotpy will also be influenced by input from new experimental results from transport and spectroscopy and by feedback and suggestions from our users.

# Acknowledgements

We thank Domenico Di Sante, Giorgio Sangiovanni, Björn Trauzettel, Florian Goth, and Fakher Assaad for feedback and support at various stages of the project.

**Author contributions** W.B. designed the software package. W.B., F.B., C.B., and M.H. developed the source code, with W.B. acting as lead developer. W.B., F.B., C.B., L.B., M.H., and M.S. performed code review and benchmarks. W.B., C.F., S.S., L.-X.W., H.B., and L.W.M. provided input on the physics of quantum transport. F.B., C.B., L.B., M.H., M.S., and T.K. provided input on the physics of optical spectroscopy. W.B., J.B., and E.M.H. developed the theoretical model at the early stages of the project. W.B., T.K., and L.W.M. managed the project. The writing of the paper was led by W.B. with input from all authors.

**Funding information** We acknowledge financial support from the Deutsche Forschungsgemeinschaft (DFG, German Research Foundation) in the project SFB 1170 *ToCoTronics* (Project ID 258499086) and in the Würzburg-Dresden Cluster of Excellence on Complexity and Topology in Quantum Matter *ct.qmat* (EXC 2147, Project ID 39085490).

# A  Implementation details

## A.1  Parallelization

### A.1.1  Basic method

The Python programming language was not designed with parallelization in mind. The global interpreter lock (GIL) is often considered (see, e.g., Ref. [44]) as a major obstacle for running parallelized Python code. However, the `multiprocessing` module avoids this obstacle by running Python code in separate parallel subprocesses, such that each process is affected only by its own GIL.

In kdotpy, the `multiprocessing` module is used to parallelize iterative tasks of the form

```
data = [f(x, *f_args, **f_kwds) for x in vals]
```

where each call to the function `f` is an independent task. The main loop in kdotpy is the iterated diagonalization of the Hamiltonian over the momentum $k$ and magnetic field $B$ values of the grid. Due to the uniform nature of these tasks, this iteration lends itself well for parallelization.

In kdotpy, the function `parallel_apply()` in the `parallel` submodule facilitates parallelization over the diagonalization function by implementing the necessary boilerplate code around the `multiprocessing.Pool` object. The following code illustrates the recipe implemented in `parallel_apply()` (simplified compared to the actual code for the sake of illustration):

```
pool = multiprocessing.Pool(processes = num_processes)
output = [
    pool.apply_async(f, args=(x,) + f_args, kwds = f_kwds) for x in vals
]
while True:
    jobsdone = sum(1 for x in output if x.ready())
    if jobsdone >= n:
        break
    print(f"{jobsdone} / {n}")
    time.sleep(poll_interval)
data = [r.get() for r in output]
pool.close()
pool.join()
```

Here, the `while` loop, that runs in the parent process, is used for process monitoring by printing the number of completed tasks to standard output [7]. This loop runs one iteration every `poll_interval` seconds in the main process until all tasks are completed. The result `data` is collected before the pool is joined and closed.

Unfortunately, `multiprocessing` does not handle keyboard interrupts (pressing Ctrl-C to abort the program) and other signals properly out of the box. Thus, `parallel_apply()` contains additional code in order to handle these asynchronous events gracefully: Once a worker process is aborted or terminated, the parent process of kdotpy detects it and aborts the full calculation, including the remaining worker processes. If this is not done correctly, worker processes can become *orphaned* (i.e., detached from the parent process) and continue working even when the parent process no longer exists. These problems are mitigated by using a custom signal handler with `parallel_apply()`.

### A.1.2  Advanced method

The Task-Model framework is also built around the process pools of the `multiprocessing` module, but with a higher level of abstraction, that allows a more fine-grained control of how tasks are run. The kdotpy submodules `tasks` and `models` define the following classes that constitute this framework:

- `ModelX` class: This class stores the 'recipe', analogous to a diagonalization function. For the different 'models' (1D, 2D, LL, etc.), a separate class is derived from the `ModelBase` class. It stores the steps of the recipe as a list of functions

---

[7]In the actual code, an instance of the `Progress` class handles the progress counter. In the code example here, we have replaced this by a `print` statement for clarity.

```
self.steps = [
    self.load_ddp,
    self.construct_ham,
    self.solve_ham,
    self.post_solve
]
```

where `self.load_ddp` enables loading of a `DiagDataPoint` object, and `self.construct_ham`, `self.solve_ham`, and `self.post_solve` apply the construction of the Hamiltonian, application of the eigensolver, and the processing of eigenstates (i.e., the steps listed in Sec. 3.4.2).

- `Task` class: This class contains the functions `self.worker_func`, `self.callback`, and `self.error_callback` as well as references to the process or thread pool. The `Task.run()` method calls `pool.apply_async()`, where `pool` is either a process or a thread pool.

- `TaskManager` class: The single instance of this class is responsible for initializing, managing, and joining the process or thread pool, somewhat similar to `parallel_apply()`. The primary job of the `TaskManager` is to send the tasks in the queue to the available CPU or GPU workers and to join if all tasks are completed. The handling of asynchronous events is also done by the `TaskManager`. The `TaskManager` class is derived from `PriorityQueue` from the `queue` module of Python.

The Task-Model framework has several benefits compared to `parallel_apply()`. Firstly, the `ModelX` classes can be derived from. The derived class does not need to redefine all steps, but only those one where it differs from its parent class. Secondly, the data points are split into subtasks, so that each subtasks can be executed where this is done most optimally (CPU or GPU). The added flexibility of the Task-Model framework comes with a small price: As a result of more overhead, the Task-Model framework is marginally slower than `parallel_apply()`, but the difference is often barely noticeable.

## A.2   Density of states

### A.2.1   Triangular IDOS element

In Eq. (101), we have determined the fraction of the interval $[k_i, k_{i+1}]$ where a linear function is below a certain value $E$. For two dimensions, the analogous problem would be considering a linearly interpolated function on a triangle in the two dimensional plane. Let us label the vertices $1, 2, 3$, and assume the function values are $e_{1,2,3}$ with $e_1 \leq e_2 \leq e_3$ (without loss of generality). The momentum coordinates of the vertices are irrelevant. The interpolated function is given by

$$e(u, v) = e_1 + (e_2 - e_1)u + (e_3 - e_1)v = (1 - u - v)e_1 + u\, e_2 + v\, e_3 \tag{A.1}$$

where $(u, v)$ are scaled coordinates with $0 \leq u \leq 1$, $0 \leq v \leq 1$, and $u + v \leq 1$, such that the vertices of the triangle correspond to $(u, v) = (0, 0)$, $(1, 0)$, and $(0, 1)$. We are interested in the fraction $f(E)$ of the triangle where $e(u, v) < E$. This can be found by finding the intersection points of the line $e(u, v) = E$ with the sides of the triangle and calculating the area of the resulting polygon. The fraction $f(E)$ is this area divided by $\frac{1}{2}$ (the area of the full triangle), and is given by

$$f(E) = \begin{cases} 1 & \text{if } E \geq e_3 \\ 1 - \frac{(E - e_3)^2}{(e_3 - e_2)(e_3 - e_1)} & \text{if } e_2 \leq E < e_3 \\ \frac{(E - e_1)^2}{(e_2 - e_1)(e_3 - e_1)} & \text{if } e_1 \leq E < e_2 \\ 0 & \text{if } E < e_1 \end{cases} \tag{A.2}$$

The conditions are such that the denominators are nonzero; for example, if $e_2 = e_1$, then the condition $e_1 \leq E < e_2$ is never fulfilled.

The implementation in `triangle_idos_element()` takes as input the three-dimensional array $e_{i,j,v}$, where $i, j$ label the elementary squares in momentum space, and $v = 1, 2, 3$ labels the three vertices. It applies the following recipe.

- The momentum space indices $(i, j)$ are flattened to $I$. The result is a two-dimensional array $e_{I,v}$ of shape $(n_k, 3)$, where $n_k = n_{k_x} n_{k_y}$ is the number of elementary squares in momentum space.

- Sort along the last axis so that $e_{I,1} \leq e_{I,2} \leq e_{I,3}$ is satisfied for all $I$.

- Evaluate $f^{(1)} = (E - e_{I,1})^2 / (e_{I,2} - e_{I,1})(e_{I,3} - e_{I,1})$ and $f^{(2)} = 1 - (E - e_{I,3})^2 / (e_{I,3} - e_{I,2})(e_{I,3} - e_{I,1})$ (see Eq. (A.2)) for all $I$ and for all $E$ in the energy range. At the points where the denominator vanishes, substitute **0**. In both cases, the resulting arrays f1 and f2 are two-dimensional arrays of shape $(n_k, n_E)$, where $n_E$ is the number of energy values in the energy range.

- Evaluate the conditions $E < e_1$, $E < e_2$, and $E < e_3$ as the two-dimensional boolean arrays cond0, cond1, and cond2. Then evaluate $f_I(E)$ as

```
f = np.where(cond0, zeros,
    np.where(cond1, f1, np.where(cond2, f2, ones))
)
```

  where zeros = np.zeros_like(f1) and ones = np.ones_like(f1). The output is an array of shape $(n_k, n_E)$, like all input arrays.

- Subtract **1** if the band is hole-like.

- For all momenta indices $I$, where any $e_{I,v}$ ($v = 1, 2, 3$) is undefined (NaN), substitute $f_I(E) = 0$ for all $E$.

### A.2.2 Tetrahedral IDOS element

For three dimensions, we follow similar ideas to arrive at an expression for $f(E)$. Let the energy values at the vertices of an elementary tetrahedron be $e_{1,2,3,4}$, where $e_1 \leq e_2 \leq e_3 \leq e_4$. The interpolated function is given by $e(u, v, w) = e_1 + (e_2 - e_1)u + (e_3 - e_1)v + (e_4 - e_1)w$, with $0 \leq u, v, w \leq 1$ and $u + v + w \leq 1$. The fraction of the tetrahedra (with volume $\frac{1}{6}$ that satisfies $e(u, v, w) < E$ is

$$
f(E) = \begin{cases}
1 & \text{if } E \geq e_4 \\
1 - \dfrac{(E - e_4)^3}{(e_4 - e_3)(e_4 - e_2)(e_4 - e_1)} & \text{if } e_3 \leq E < e_4 \\
\dfrac{(E - e_1)^3}{(e_2 - e_1)(e_3 - e_1)(e_4 - e_1)} - \dfrac{(E - e_2)^3}{(e_2 - e_1)(e_3 - e_2)(e_4 - e_2)} & \text{if } e_2 \leq E < e_3 \text{ and } e_1 < e_2 \\
\dfrac{(E - e_1)^2}{(e_3 - e_1)(e_4 - e_1)} \left(1 + \dfrac{e_3 - E}{e_3 - e_1} + \dfrac{e_4 - E}{e_4 - e_1}\right) & \text{if } e_2 \leq E < e_3 \text{ and } e_1 = e_2 \\
\dfrac{(E - e_1)^3}{(e_2 - e_1)(e_3 - e_1)(e_4 - e_1)} & \text{if } e_1 \leq E < e_2 \\
0 & \text{if } E < e_1
\end{cases}
\tag{A.3}
$$

In `tetrahedral_idos_element()`, the arrays defining $f(E)$ are calculated analogous to those in `triangle_idos_element()` but with some additional intermediate steps to avoid recalculation of the products of $(e_w - e_v)$ in the denominators. The conditions $E < e_1$, $E < e_2$, $E < e_3$, and $E < e_4$ are applied as

```
f = np.ones_like(f1)
f[cond3] = f3[cond3]
f[cond2] = f2[cond2]
f[cond1] = f1[cond1]
f[cond0] = 0.0
```

which is equivalent to a sequence of calls to `np.where()`, but with a better performance for the larger arrays in three momentum dimensions.

NOTE: In the expression for $f(E)$ with $e_2 \leq E < e_3$ and $e_1 < e_2$, it is possible to factor out $e_2 - e_1$. Doing this yields

$$
f^{(2)}(E) = \frac{(e_{31}e_{41} + e_{41}(e_3 - E) + e_{31}(e_4 - E))(E - e_1)^2 + e_{21}(E - e_1)^3 - 3e_{21}e_{31}e_{41}(E - e_1) + e_{21}^2 e_{31}e_{41}}{e_{32}e_{42}e_{31}e_{41}},
\tag{A.4}
$$

using the short-hand notation $e_{vw} = e_v - e_w$. Unlike Eq. (A.3), there is no factor $e_2 - e_1$ in the denominator, hence it is valid for $e_1 = e_2$ as well. If we substitute $e_1 = e_2$, we retrieve the expression for $e_2 \leq E < e_3$ and $e_1 = e_2$ of Eq. (A.3).

## A.3   Charge neutrality point in symbolic Landau level mode

In dispersion mode and Landau level mode `full`, the charge neutrality point always lies between the bands with band indices $-1$ and $1$, by definition. In the Landau level mode `sym`, each state is characterized by a pair $(n, b)$ of Landau level index $n = -2, -1, 0, \ldots$ and band index $b$, so that finding the charge neutrality point requires an extra step: For each state, we determine a *universal band index* $u$, equivalent to ordinary band indices in full Landau level mode, with the property that the charge neutrality point lies between the states with $u = -1$ and $u = +1$.

The algorithm for finding the universal band indices is implemented in `DiagDataPoint.get_ubindex()` and proceeds as follows. The states in `DiagDataPoint` are sorted by eigenvalue. We define an array with ones for all states with band index $b > 0$ and an array with ones where $b < 0$,

```
pos = np.where(
    bindex_sort > 0,
    np.ones_like(bindex_sort),
    np.zeros_like(bindex_sort)
)
neg = 1 - pos
```

The arrays are summed cumulatively from below and above, respectively,

```
npos = np.cumsum(pos)
nneg = neg.sum() - np.cumsum(neg)
```

The array of universal band indices is then simply the difference between the two cumulative sums

```
ubindex = npos - nneg
ubindex[ubindex <= 0] -= 1
```

where the nonpositive values have to be decreased by one in order to make sure that the integers $u$ are nonzero. By definition, the resulting array is an increasing sequence as function of energy, and identical to that obtained from regular band indices in the Landau level mode `full`.

## A.4   Solving Poisson's equation

As already mentioned in Sec. 3.2.7, a way to incorporate electrostatic potentials into the Hamiltonian is by parsing a set of boundary conditions. This set is used to solve Poisson's equation within the function `solve_potential()` in `potential.py`, for 'static' as well as self-consistent potentials (see Secs. 3.2.7 and 3.11, respectively). For the remainder of this section we will refer to this potential as Hartree potential $V_{\text{H}}$. It should be emphasized that $V_{\text{H}}$ parametrizes the potential energy of the carriers in the electrostatic potential, not the electric potential itself [8] The Hartree potential $V_{\text{H}}$ satisfies Poisson's equation

$$\partial_z \left[ \varepsilon(z) \, \partial_z V_{\text{H}}(z) \right] = \frac{e}{\varepsilon_0} \rho(z) \tag{A.5}$$

[identical to Eq. (132)] with dielectric function/constant $\varepsilon(z)$, elementary charge $e$, vacuum permittivity $\varepsilon_0$ and spatial charge density $\rho(z)$. In the self-consistent case, as described in Secs. 3.11 and 3.7.8, $\rho(z)$ is extracted from all eigenvectors, whereas in the 'static' case $\rho(z) = 0$.

The generalized solution can be calculated by integrating twice over $z$, with generally different lower integration limits $z_a$ for the first integration and $z_b$ for the second. The first integral over Eq. (A.5) yields

$$\varepsilon(z) \, \partial_z V_{\text{H}}(z) - \varepsilon(z_a) \, \partial_z V_{\text{H}}(z_a) = \frac{e}{\varepsilon_0} \int_{z_a}^{z} dz' \rho(z') = \frac{e}{\varepsilon_0} \left[ \mathcal{I}_\rho(z) - \mathcal{I}_\rho(z_a) \right], \tag{A.6}$$

where we use the shorthand notation $\mathcal{I}_\rho(z) \equiv \int_0^z dz' \rho(z')$. As this equation is valid for all $z$, it follows that

$$\mathcal{C}(z) \equiv \varepsilon(z) \, \partial_z V_{\text{H}}(z) - \frac{e}{\varepsilon_0} \mathcal{I}_\rho(z) \tag{A.7}$$

---

[8]These two quantities differ by a factor of $-e$, which is $-1$ in the chosen system of units.

is constant. We evaluate this constant at $z = z_a$, divide by $\varepsilon(z)$ and integrate a second time to find the Hartree potential,

$$V_{\mathrm{H}}(z) = V_{\mathrm{H}}(z_b) + \frac{e}{\varepsilon_0} \left( \int_0^z dz' \frac{\mathcal{I}_\rho(z')}{\varepsilon(z')} - \int_0^{z_b} dz' \frac{\mathcal{I}_\rho(z')}{\varepsilon(z')} \right) + \mathcal{C}(z_a) \left( \int_0^z dz' \frac{1}{\varepsilon(z')} - \int_0^{z_b} dz' \frac{1}{\varepsilon(z')} \right)$$

$$(A.8)$$

We have written the integrals $\int_{z_b}^z$ explicitly as $\int_0^z - \int_0^{z_b}$, which reflects the implementation in the code.

In order to find a unique solution, two boundary conditions need to be given, which fix $V_{\mathrm{H}}$ or $\partial_z V_{\mathrm{H}}$ (which can be interpreted as electric field) at specific $z$ coordinates. The function solve_potential() takes the boundary conditions as a set of keyword arguments, namely a subset of values v1, v2, v3, dv1, dv2, and v12 and coordinates z1, z2, and z3. Out of the former set, only certain combinations of values may be set to a numerical value (the other being None); any other combination will cause a ValueError exception to be raised. The following combinations are valid:

1. dv1 and v1, with z1 as coordinate: $\partial_z V_{\mathrm{H}}(z_1) = \partial V_1$ and $V_{\mathrm{H}}(z_1) = V_1$. The $z$ coordinate for both boundary conditions is the same, thus $z_a = z_b = z_1$.

$$V_{\mathrm{H}}(z) = V_1 + \frac{e}{\varepsilon_0} \left( \int_0^z dz' \frac{\mathcal{I}_\rho(z')}{\varepsilon(z')} - \int_0^{z_1} dz' \frac{\mathcal{I}_\rho(z')}{\varepsilon(z')} \right)$$
$$+ \left( \varepsilon(z_1) \partial V_1 - \frac{e}{\varepsilon_0} \mathcal{I}_\rho(z_1) \right) \left( \int_0^z dz' \frac{1}{\varepsilon(z')} - \int_0^{z_1} dz' \frac{1}{\varepsilon(z')} \right). \qquad (A.9)$$

2. dv2 and v2, with z2 as coordinate: Analogous to case 1, using $z_a = z_b = z_2$ instead.

3. dv1 and v2, with z1 and z2 as coordinates: $\partial_z V_{\mathrm{H}}(z_1) = \partial V_1$ and $V_{\mathrm{H}}(z_2) = V_2$. The $z$ coordinates for both boundary conditions are different, thus $z_a = z_1$ and $z_b = z_2$.

$$V_{\mathrm{H}}(z) = V_2 + \frac{e}{\varepsilon_0} \left( \int_0^z dz' \frac{\mathcal{I}_\rho(z')}{\varepsilon(z')} - \int_0^{z_2} dz' \frac{\mathcal{I}_\rho(z')}{\varepsilon(z')} \right)$$
$$+ \left( \varepsilon(z_1) \partial V_1 - \frac{e}{\varepsilon_0} \mathcal{I}_\rho(z_1) \right) \left( \int_0^z dz' \frac{1}{\varepsilon(z')} - \int_0^{z_2} dz' \frac{1}{\varepsilon(z')} \right). \qquad (A.10)$$

4. v1 and dv2, with z1 and z2 as coordinates: Analogous to case 3, using $z_a = z_2$ and $z_b = z_1$ instead.

5. v1 and v2, with z1 and z2 as coordinates: $V_{\mathrm{H}}(z_1) = V_1$ and $V_{\mathrm{H}}(z_2) = V_2$. Only potential values are given. The solution is given by Eq. (A.8), but we cannot evaluate the constant $\mathcal{C}(z_a)$ directly, because the derivative $\partial_z V_{\mathrm{H}}(z_a)$ is unknown. By setting $z_b = z_1$ and $z = z_2$ (or vice versa), we find that

$$\mathcal{C}(z_a) = \frac{V_{\mathrm{H}}(z_2) - V_{\mathrm{H}}(z_1) - \frac{e}{\varepsilon_0} \left( \int_0^{z_1} dz' \frac{\mathcal{I}_\rho(z')}{\varepsilon(z')} - \int_0^{z_1} dz' \frac{\mathcal{I}_\rho(z')}{\varepsilon(z')} \right)}{\int_0^{z_2} dz' \frac{1}{\varepsilon(z')} - \int_0^{z_1} dz' \frac{1}{\varepsilon(z')}}. \qquad (A.11)$$

By substitution of $\mathcal{C}(z_a)$ into Eq. (A.8), we thus find

$$V_{\mathrm{H}}(z) = V_1 + \frac{e}{\varepsilon_0} \left( \int_0^z dz' \frac{\mathcal{I}_\rho(z')}{\varepsilon(z')} - \int_0^{z_1} dz' \frac{\mathcal{I}_\rho(z')}{\varepsilon(z')} \right)$$
$$+ \frac{V_2 - V_1 - \frac{e}{\varepsilon_0} \int_{z_1}^{z_2} dz' \frac{\mathcal{I}_\rho(z')}{\varepsilon(z')}}{\int_{z_1}^{z_2} dz' \frac{1}{\varepsilon(z')}} \left( \int_0^z dz' \frac{1}{\varepsilon(z')} - \int_0^{z_1} dz' \frac{1}{\varepsilon(z')} \right). \qquad (A.12)$$

6. v12 and v3, with z1, z2, and z3 as coordinates: $V_{\mathrm{H}}(z_2) - V_{\mathrm{H}}(z_1) = V_{12}$ and $V_{\mathrm{H}}(z_3) = V_3$. We

substitute $\mathcal{C}(z_a)$ from Eq. (A.11) into Eq. (A.8) with $z_b = z_3$, and obtain

$$
V_{\mathrm{H}}(z) = V_3 + \frac{e}{\varepsilon_0} \left( \int_0^z dz' \frac{\mathcal{I}_\rho(z')}{\varepsilon(z')} - \int_0^{z_3} dz' \frac{\mathcal{I}_\rho(z')}{\varepsilon(z')} \right)
$$
$$
+ \frac{V_{12} - \frac{e}{\varepsilon_0} \int_{z_1}^{z_2} dz' \frac{\mathcal{I}_\rho(z')}{\varepsilon(z')}}{\int_{z_1}^{z_2} dz' \frac{1}{\varepsilon(z')}} \left( \int_0^z dz' \frac{1}{\varepsilon(z')} - \int_0^{z_3} dz' \frac{1}{\varepsilon(z')} \right). \tag{A.13}
$$

In `solve_potential()` the NumPy arrays `densz` $[\rho(z)]$ and `epsilonz` $[\varepsilon(z)]$ first are integrated incrementally, yielding the arrays

```
int_densz = integrate_arr(densz) * dz
int_invepsilonz = integrate_arr(1. / epsilonz) * dz
```

which represent the integrals $\mathcal{I}_\rho(z)$ and $\int_0^z dz' 1/\varepsilon(z')$ for each individual value of $z$. Subsequently, $\int_0^z dz' \mathcal{I}_\rho(z')/\varepsilon(z')$ is calculated as

```
int_dens_over_epsz = integrate_arr(int_densz / epsilonz) * dz
```

The value of the integrals at the chosen $z$ coordinates $(z_1, z_2, z_3)$ are evaluated by using `np.interp()`, e.g.,

```
int_dens_over_eps_z1, int_dens_over_eps_z2, int_dens_over_eps_z3 = \
    np.interp([z1, z2, z3], zval, int_dens_over_epsz)
```

where `zval` is the array of $z$ coordinates where the other arrays are defined. A similar evaluation is done for `epsilonz`, `int_densz`, `int_invepsilon`, followed by

```
int_dens_over_epsz_1_2 = int_dens_over_eps_z2 - int_dens_over_eps_z1
int_invepsilonz_1_2 = int_invepsilon_z2 - int_invepsilon_z1
```

Depending on the combination of boundary conditions, the corresponding solution is selected from the six cases given above. Taking case 1 (`dv1` and `v1`) as example, the Hartree potential is then evaluated as

```
int_const = epsilon_z1 * dv1 - eovereps0 * int_dens_z1
vz = v1 + eovereps0 * (int_dens_over_epsz - int_dens_over_eps_z1)
        + int_const * (int_invepsilonz - int_invepsilon_z1)
```

where `int_const` is the integration constant $\mathcal{C}(z_a)$ from Eq. (A.7). The implementations for the other cases are analogous.

The boundary conditions are chosen automatically by parsing command line arguments like `vinner`, `vouter`, `vsurf` or `efield`. In self-consistent calculations the automatically determined boundary conditions can be manually overwritten by using the command line argument `potentialbc`.

On a side note, one could also imagine solving the Poisson equation as a matrix equation involving the vector $V_{\mathrm{H}}(z_i)$ on the grid of coordinates $z_i$, obtained by replacing the derivative operator by the appropriate matrix (cf. Sec. 2.2.1). This results in a matrix-vector equation, which can be solved by matrix inversion. In the continuum limit (grid resolution $\Delta z \to 0$), this method converges to the same solution as the numerical integration method detailed above. Our benchmarks show that the numerical integration method outperforms the matrix method within the context of `kdotpy`, with smaller numerical errors at finite resolutions $\Delta z$. We thus decided to include the numerical integration approach only.

## A.5   Extrema

### A.5.1   Nine-point extremum solver

The function `nine_point_extremum_solver()` takes a $3 \times 3$ grid of momentum values $(k_{x,i'}, k_{y,j'})$ values and a $3 \times 3$ grid of energy values $e_{i',j'}$ as inputs. The indices $i'$ and $j'$ are taken as $i' = i-1, i, i+1$

and $j' = j-1, j, j+1$, where $i, j$ is a location where an extremum is detected: We say there is a minimum (maximum) at $i, j$ if the values

$$e_{i+1,j}, e_{i-1,j}, e_{i,j+1}, e_{i,j-1}, e_{i+1,j+1}, e_{i-1,j+1}, e_{i+1,j-1}, e_{i-1,j-1} \tag{A.14}$$

are all $> e_{i,j}$ ($< e_{i,j}$). If this is the case, the `nine_point_extremum_solver()` locates the extremum more precisely by fitting Eq. (125) to the input data. Assume that the momenta are aligned on an equally spaced cartesian grid, and define $\Delta x = k_{x,i+1} - k_{x,i} = k_{x,i} - k_{x,i-1}$ and $\Delta y = k_{y,j+1} - k_{y,j} = k_{y,j} - k_{y,j-1}$. First calculate the coefficients of the Hessian matrix,

$$a = \frac{e_{i+1,j} - 2e_{i,j} + e_{i-1,j}}{2(\Delta x)^2}, \qquad c = \frac{e_{i+1,j+1} - e_{i-1,j+1} - e_{i+1,j-1} + e_{i-1,j-1}}{4\,\Delta x\,\Delta y},$$
$$b = \frac{e_{i,j+1} - 2e_{i,j} + e_{i,j-1}}{2(\Delta y)^2}. \tag{A.15}$$

We note that $c$ is the only coefficient where the $e_{i\pm1,j\pm1}$ appear; the other linear combinations of $e_{i\pm1,j\pm1}$ are not considered. We calculate the auxiliary variables

$$X = \frac{e_{i+1,j} - e_{i-1,j}}{2\Delta x}, \qquad Y = \frac{e_{i,j+1} - e_{i,j-1}}{2\Delta y} \tag{A.16}$$

and use them to find the location $(k_{x,0}, k_{y,0})$ of the extremum in momentum space,

$$k_{x,0} = k_{x,i} + \frac{cY - 2bX}{4ab - c^2}, \qquad k_{y,0} = k_{y,i} + \frac{cX - 2aY}{4ab - c^2}. \tag{A.17}$$

The denominator $4ab - c^2$ is the determinant of the Hessian matrix. If the location defined by Eq. (A.17) lies outside of the rectangle $[k_{x,i-1}, k_{x,i+1}] \times [k_{y,j-1}, k_{y,j+1}]$, raise the warning *"Poorly defined extremum found"* and use the location $(k_{x,i} - X/2a, k_{y,j} - Y/2b)$ instead, i.e., effectively setting $c = 0$. The energy value $f_0$ of the extremum is

$$f_0 = e_{i,j} - a(k_{x,i} - k_{x,0})^2 - b(k_{y,j} - k_{y,0})^2 - c(k_{x,i} - k_{x,0})(k_{y,j} - k_{y,0}). \tag{A.18}$$

The function `nine_point_extremum_solver()` returns the result as $f_0$, $(k_{x,0}, k_{y,0})$, $(a, b, c)$.

The band masses are calculated from the eigenvalues of the Hessian matrix, constructed from $(a, b, c)$. For cartesian coordinates, the eigenvalues are equal to $\lambda_\pm = \frac{1}{2}(a + b \pm \sqrt{(a-b)^2 + c^2})$. For polar coordinates, $k_x$ and $k_y$ should be interpreted as $k_r = |k|$ and $k_\phi$ respectively. In order to obtain band masses for cartesian coordinates, the elements of the Hessian matrix must be rescaled as

$$h_{\text{polar}} = \begin{pmatrix} 2a & c/k_r \\ c/k_r & 2b/k_r^2 \end{pmatrix}, \tag{A.19}$$

where the factor $1/k_r$ comes from the Jacobian of the coordinate transformation between polar and cartesian coordinates (assuming the angular coordinate is in radians). The band masses are then obtained from the eigenvalues of $h_{\text{polar}}$.

### A.5.2 Nineteen-point extremum solver

The function `nineteen_point_extremum_solver()` for three dimensions effectively applies the method of the nine-point extremum solver in the $k_x k_y$, $k_x k_z$, and $k_y k_z$ planes separately. The elements of the Hessian matrix $h$ [the $3 \times 3$ matrix in Eq. (127)] are calculated as

$$a = \frac{e_{i+1,j,l} - 2e_{i,j,l} + e_{i-1,j,l}}{2(\Delta x)^2}, \qquad d = \frac{e_{i+1,j+1,l} - e_{i-1,j+1,l} - e_{i+1,j-1,l} + e_{i-1,j-1,l}}{4\,\Delta x\,\Delta y},$$
$$b = \frac{e_{i,j+1,l} - 2e_{i,j,l} + e_{i,j-1,l}}{2(\Delta y)^2}, \qquad e = \frac{e_{i+1,j,l+1} - e_{i-1,j,l+1} - e_{i+1,j,l-1} + e_{i-1,j,l-1}}{4\,\Delta x\,\Delta z}, \tag{A.20}$$
$$c = \frac{e_{i,j,l+1} - 2e_{i,j,l} + e_{i,j,l-1}}{2(\Delta z)^2}, \qquad f = \frac{e_{i,j+1,l+1} - e_{i,j-1,l+1} - e_{i,j+1,l-1} + e_{i,j-1,l-1}}{4\,\Delta y\,\Delta z}.$$

The coefficients $d$, $e$, and $f$ involve the twelve pink-coloured points in Fig. 9(c), but for each group of four, only one linear combination is considered. We note that the corner points $e_{i\pm1,j\pm1,l\pm1}$ [coloured black in Fig. 9(c)] are included in the input arguments, but not considered.

If the Hessian matrix is non-singular (in `nineteen_point_extremum_solver()`, we use the condition $|\det h| > 10^{-6}$), we find the momentum location as

$$\begin{pmatrix} k_{x,0} \\ k_{y,0} \\ k_{z,0} \end{pmatrix} = \begin{pmatrix} k_{x,i} \\ k_{y,j} \\ k_{z,l} \end{pmatrix} + h^{-1} \begin{pmatrix} -X \\ -Y \\ -Z \end{pmatrix} \tag{A.21}$$

where $h^{-1}$ is the inverse of the Hessian matrix and $X = (e_{i+1,j,l} - e_{i-1,j,l})/2\Delta x$, $Y = (e_{i,j+1,l} - e_{i,j-1,l})/2\Delta y$, and $Z = (e_{i,j,l+1} - e_{i,j,l-1})/2\Delta z$ [analogous to Eq. (A.16)]. If $|\det h| \leq 10^{-6}$ or the point lies outside of the box $[k_{x,i-1}, k_{x,i+1}] \times [k_{y,j-1}, k_{y,j+1}] \times [k_{z,l-1}, k_{z,l+1}]$, the *"Poorly defined extremum found"* warning is raised and the location replaced by $(k_{x,i} - X/2a, k_{y,j} - Y/2b, k_{z,l} - Z/2c)$. The energy value $f_0$ of the extremum is given by

$$f_0 = e_{i,j,l} - \begin{pmatrix} \tilde{k}_x & \tilde{k}_y & \tilde{k}_z \end{pmatrix} \begin{pmatrix} 2a & d & e \\ d & 2b & f \\ e & f & 2c \end{pmatrix} \begin{pmatrix} \tilde{k}_x \\ \tilde{k}_y \\ \tilde{k}_z \end{pmatrix}, \tag{A.22}$$

with $(\tilde{k}_x, \tilde{k}_y, \tilde{k}_z) = (k_{x,i}, k_{y,j}, k_{z,l}) - (k_{x,0}, k_{y,0}, k_{z,0})$. The `nineteen_point_extremum_solver()` function returns the result as $f_0$, $(k_{x,0}, k_{y,0}, k_{z,0})$, $(a, b, c, d, e, f)$. The band masses are calculated from the eigenvalues of the Hessian matrix. If the coordinate system is cylindrical or spherical, the appropriate coordinate transformation is applied prior to finding the eigenvalues.

# B Reference

## B.1 Units and constants

### B.1.1 Units

In `kdotpy`, physical quantities are represented by numerical data types without the explicit specification of units. For this reason, it is necessary to agree on a system of units such that calculations that physical quantities can be done intuitively, possibly without the need of conversion factors. Whereas a system where units are treated explicitly would be more fail-safe, it would create a lot of overhead which may reduce performance.

We choose a system of units such that physical quantities in context of solid-state physics have reasonable values. The basic units are:

- Length: **nm**

- Time: **ns**

- Energy: **meV** (*not* **eV**)

- Voltage: **mV** (*not* **V**)

- Temperature: **K**

- Electric charge: **e**

- Magnetic field: **T**

With *magnetic field* we mean *magnetic flux density*, more accurately speaking. The unit for magnetic flux density in this unit system is defined as Tesla, $\mathbf{T = V\,s/m^2}$. This unit is 'incompatible' with the combination of units for voltage, length, and time, which would be $\mathbf{mV\,ns/nm^2}$. The conversion, given by $\mathbf{1\,T = 10^{-6}\,mV\,ns/nm^2}$ is done internally by `kdotpy` when it interprets magnetic field values.

From the basic units, we derive units for other physical quantities, for example,

- Momentum (more appropriately wave vector) $\mathbf{k}$: $\mathbf{nm^{-1}}$

- Density (such as particle/carrier density): $\mathbf{nm^{-1}}$, $\mathbf{nm^{-2}}$, or $\mathbf{nm^{-3}}$ depending on dimensionality

- Density of states: $\mathbf{nm^{-1}\,meV^{-1}}$, $\mathbf{nm^{-2}\,meV^{-1}}$, or $\mathbf{nm^{-3}\,meV^{-1}}$

- Charge density: $\mathbf{e\,nm^{-1}}$, $\mathbf{e\,nm^{-2}}$, or $\mathbf{e\,nm^{-3}}$

- Electric field: $\mathbf{mV/nm}$

- Velocity: $\mathbf{nm/ns = m/s}$

Inputs and outputs use the basic and derived units if not explicitly stated otherwise.

### B.1.2 Physical constants

The following physical constants are defined in `physconst.py`.

- `m_e = 0.510998910e9` in **meV**: Electron mass in energy equivalents ($E = m_e c^2$).

- `e_el = 1.6021766208e-19` in **C**: Elementary charge $e$. We define this value to be positive.

- `cLight = 299792458.` in $\mathbf{nm/ns = m/s}$: Speed of light $c$.

- `hbar = 6.582119514e-4` in **meV ns**: Reduced Planck constant $\hbar$.

- `hbarm0 = hbar**2 * cLight**2 / m_e / 2` in $\mathbf{meV\,nm^2}$:
  $\hbar^2/2m_e = \mathbf{38.09982350(23)\,meV\,nm^2}$. The approximate value $\mathbf{38\,meV\,nm^2}$ is useful for making estimates.

- eoverhbar = 1e-6 / hbar in $\mathbf{1/(T\,nm^2)}$: $e/\hbar$ defined such, that eoverhbar times magnetic field in T yields a density in $\mathbf{nm^{-2}}$. This takes into account the conversion factor $\mathbf{10^{-6}}$ in $\mathbf{T = 10^{-6}\,mV\,ns/nm^2}$. The product eoverhbar * A, with the vector potential **A** in $\mathbf{T\,nm}$, yields a value in $\mathbf{nm^{-1}}$, as appropriate for a momentum quantity. This appears in the Peierls substitution, for example. The product eoverhbar * B, with the magnetic field $B$ in **T**, yields a quantity in units of $\mathbf{nm^{-2}}$, cf. $l_B^2 = \hbar/eB$

- muB = 5.7883818012e-2 in **meV/T**: Bohr magneton $\mu_B = e\hbar/2m_e$.

- kB = 8.6173303e-2 in **meV/K**: Boltzmann constant $k_B$.

- eovereps0 = 1.80951280207e4 in **mV nm**: $e/\varepsilon_0$, electron charge divided by permittivity constant (also called vacuum permittivity).

- gg = 2 (dimensionless): Gyromagnetic ratio.

- r_vonklitzing = 25812.8074555 in $\mathbf{\Omega}$ (ohm): Von Klitzing constant $R_K$, resistance value corresponding to one quantum of conductance.

These values have been extracted from the NIST Reference on Constants, Units, and Uncertainty, revision 2014, see Ref. [45] and online at https://physics.nist.gov/cuu/Constants/index.html. The revised NIST values from 2018 may deviate slightly (but negligibly) from the values listed here and in physconst.py.

## B.2 Material parameters

### B.2.1 Chemistry

- compound: The chemical formula of the compound, for example HgTe.

- elements: Comma-separated list of the elements, for example Hg, Te. If elements is not given, it is determined automatically from compound (and vice versa).

- composition: Numbers for each of the elements in the chemical formula, that indicate their (stoichiometric) proportions in the compound. The numbers must be separated by commas and may be a function of the variables x, y, and/or z. For example, for a crystal with molecular formula $Hg_{1-x}Cd_xTe$, use compound = HgCdTe and composition = 1-x, x, 1.

### B.2.2 Special commands

- copy: Copy all valid parameters of another material into the present one. The value is the id of the source material. For example

  ```
  [mat2]
  copy = mat1
  ```

  copies all parameters from material mat1 into material mat2. Each parameter may be subsequently overwritten manually.

- linearmix: Define a material as a linear combination of two others, where the parameters are linearly interpolated. The value is a 3-tuple of the form mat1, mat2, var, where mat1 and mat2 are the source materials and var is the interpolation variable (x, y, or z). For example

  ```
  [HgCdTe]
  linearmix = HgTe, CdTe, x
  ```

  makes material HgCdTe a linear combination of HgTe and CdTe. Each parameter $p$ is interpolated as $p_{\text{HgCdTe}} = (1-x)\,p_{\text{HgTe}} + x\,p_{\text{CdTe}}$. Each parameter may be subsequently overwritten manually.

### B.2.3  Band energies

- Ev: 'Valence' band energy in meV. This is the energy of the $\Gamma_8$ orbitals at $k = 0$ for the *unstrained bulk* material. For inverted materials, Ev indicates the energy of the $\Gamma_8$ orbitals, not that of the actual valence band.

- Ec: 'Conduction' band energy in meV. This is the energy of the $\Gamma_6$ orbitals at $k = 0$ for the *unstrained bulk* material. For inverted materials, Ec indicates the energy of the $\Gamma_6$ orbitals, not that of the actual conductance band.

- delta_so: 'Split-off' energy difference $\Delta_{SO}$ in meV between the $\Gamma_8$ and $\Gamma_7$ orbitals in the unstrained bulk material. To put it more precisely, the energy of the $\Gamma_8$ orbital at $k = 0$ is Ev - delta_so.

### B.2.4  Quadratic terms

- P: Kane matrix element $P = (\hbar/m_0)\langle S|p_x|X\rangle$ in meV nm. NOTE: The definition of $P$ differs between references by a sign and/or a factor of $i$. Here, we take a positive real value for P.

- gamma1: Luttinger parameter $\gamma_1$, dimensionless value. This Luttinger parameter describes the spherically isotropic component of the (bare) band mass of the $\Gamma_8$ orbitals.

- gamma2: Luttinger parameter $\gamma_2$, dimensionless value. This Luttinger parameter describes the component proportional to $k_x^2 + k_y^2 - 2k_z^2$ for the $\Gamma_8$ orbitals. This component is axially, but not spherically symmetric.

- gamma3: Luttinger parameter $\gamma_3$, dimensionless value. This Luttinger parameter describes the component proportional to $k_x^2 - k_y^2$ for the $\Gamma_8$ orbitals. This component breaks axial symmetry.

- F: Band mass parameter for the $\Gamma_6$ orbitals. The bare band mass term is $\frac{\hbar^2}{2m_0}(2F+1)(k_x^2+k_y^2+k_z^2)$. In $2F + 1$, the $1$ is the contribution of free electrons and $2F$ is the contribution from perturbative corrections from remote bands in $\mathbf{k} \cdot \mathbf{p}$ theory.

- kappa: See Section *Magnetic couplings* below.

### B.2.5  Bulk-inversion asymmetry

- bia_c: Linear bulk-inversion asymmetry coefficient $C$ in meV nm$^{-1}$.

- bia_b8p: Quadratic bulk-inversion asymmetry coefficient $B_{8v}^+$ in meV nm$^{-2}$.

- bia_b8m: Quadratic bulk-inversion asymmetry coefficient $B_{8v}^-$ in meV nm$^{-2}$.

- bia_b7: Quadratic bulk-inversion asymmetry coefficient $B_{7v}$ in meV nm$^{-2}$.

### B.2.6  Magnetic couplings

- ge: Gyromagnetic factor $g_e$ for the $\Gamma_6$ orbitals. The magnetic coupling is also known as the Zeeman term and is equal to $g_e\mu_B\mathbf{B} \cdot \mathbf{S}$. The value of $g_e$ contains contributions from the free electron (equal to $2$) and perturbative corrections from remote bands.

- kappa: Gyromagnetic factor $\kappa$ for the $\Gamma_8$ and $\Gamma_7$ orbitals. The magnetic coupling is of the form $-2\mu_B\kappa\mathbf{J}\cdot\mathbf{B}$ in the $(\Gamma_8, \Gamma_8)$ block of the Hamiltonian and analogous couplings in the $(\Gamma_8, \Gamma_7)$ and $(\Gamma_7, \Gamma_7)$ blocks. The coefficient kappa also appears as a non-magnetic term containing a commutator $[\kappa, k_z]$, which contributes only at material interfaces.

- q: Coefficient $q$ of the $\Gamma_8$ magnetic coupling $-2\mu_B q\,\mathcal{J} \cdot \mathbf{B}$, where $\mathcal{J} = (J_x^3, J_y^3, J_z^3)$.

- exch_yNalpha: Exchange energy parameter in meV for the $\Gamma_6$ orbitals. The paramagnetic exchange interaction is relevant for paramagnetic materials like (Hg,Mn)Te. The interaction strength typically depends linearly on the Mn concentration ($y$ in Ref. [3], hence the y in the parameter name). In the materials file, this dependence must be given explicitly, e.g., by setting exch_yNalpha = y * 400. The given dependence may be chosen non-linear if desired. See Ref. [3] for more details on the physics of this paramagnetic coupling in (Hg,Mn)Te.

- `exch_yNbeta`: Exchange energy parameter in meV for the $\Gamma_8$ and $\Gamma_7$ orbitals.

- `exch_g`: The gyromagnetic factor $g_{\mathrm{ex}}$ in the argument of the Brillouin function. For (Hg,Mn)Te, this quantity is usually written as $g_{\mathrm{Mn}}$ see Eq. (58) and Ref. [3]. The default value is 2.

- `exch_TK0`: The offset temperature $T_0$ in K that appears in the denominator of the argument of the Brillouin function. The default value is $10^{-6}$. See Sec. 2.5.2.

### B.2.7   Electrostatic properties

- `diel_epsilon`: Dielectric constant $\varepsilon_{\mathrm{r}}$, also known as relative permittivity. The value is dimensionless. The present version of kdotpy implements a constant value only, i.e., no frequency dependence is considered.

- `piezo_e14`: Piezo-electric constant $e_{14}$ in $e$ nm$^{-2}$. Note that this is the only independent component of the piezo-electric tensor for the zincblende crystal structure. To convert from units of C m$^{-2}$, multiply the value by $10^{18}e$, for example `piezo_e14 = 0.035 * 1e18 * e_el`.

### B.2.8   Lattice constant and strain

- `a`: Lattice constant in nm of the unstrained material.

- `strain_C1`: Strain parameter (deformation potential) $C_1$. The corresponding strain term is $C_1 \operatorname{tr} \epsilon$ acting on the $\Gamma_6$ orbitals. In Refs. [2,3], the notation $C$ is used.

- `strain_Dd`: Strain parameter (deformation potential) $D_d$. The corresponding strain term is $D_d \operatorname{tr} \epsilon$ acting on the $\Gamma_8$ and $\Gamma_7$ orbitals. In Refs. [2,3], the notation $a$ is used.

- `strain_Du`: Strain parameter (deformation potential) $D_u$. The corresponding strain term is a linear combination of the diagonal entries $\epsilon_{ii}$ of the strain tensor, and acts on the $\Gamma_8$ and $\Gamma_7$ orbitals. In Refs. [2,3], the notation $b = -\frac{2}{3}D_u$ is used.

- `strain_Duprime`: Strain parameter (deformation potential) $D'_u$. The corresponding strain term is a linear combination of the off-diagonal entries $\epsilon_{ij}$ (shear components) of the strain tensor, and acts on the $\Gamma_8$ and $\Gamma_7$ orbitals. In Refs. [2,3], $d = -\frac{2}{\sqrt{3}}D'_u$ is used.

## B.3   Lattice orientation

The **SO(3)** rotation matrix $R$ can be parametrized in different ways, for example directly in terms of the device coordinate axis in terms of the lattice direction, or in terms of Euler angles. The command line option `orientation` is the most generic way to input a rotation. By giving up to three angles or directions, the crystal lattice can be rotated to any possible orientation with respect to the device coordinate system:

- `orientation #ang`: Rotation around $z$, equivalent to `stripangle`.

- `orientation #ang #ang`: Tilt the $z$ axis, then rotate around the $z$ axis.

- `orientation #ang #ang #ang`: Euler rotation $z, x, z$, i.e., rotate around the $c$ crystal axis, tilt the $z$ axis, and rotate around the $z$ axis.

- `orientation #dir`: Longitudinal direction $x$, equivalent to `stripdir`.

- `orientation - #dir`: Growth direction $z$.

- `orientation #dir #dir`: Longitudinal and growth direction $x, z$.

- `orientation - #dir #dir`: Transversal and growth direction $y, z$.

- `orientation #dir #dir #dir`: Longitudinal, transversal, and growth direction $x, y, z$.

- `orientation #dir #ang` or `orientation #ang #dir`: Growth direction $z$ and rotation around $z$.

Angles `#ang` are entered an as explicit floating point number containing a decimal sign `.` or with the degree symbol (° or `d`). Directions `#dir` are triplets either of digits without separators and possibly with minus signs (e.g., 100, 111, 11-2, -110) or of numbers separated by commas without spaces (e.g., `1,1,0` or `10,-10,3`). If multiple directions are given as arguments, they must be pairwise orthogonal, which is tested by calculating the inner products between them.

The command line arguments `stripangle` and `stripdir` are simplified options for rotation of a strip geometry around the `z` axis.

## B.4   Custom eigensolvers

As discussed in Sec. 3.4.1, the diagonalization involves two components: The actual eigensolver and an efficient LU solver for the shift-and-invert step. `kdotpy`can be configured to use custom solvers provided by external packages for both steps. In this section, we list them and discuss their strengths and weaknesses.

### B.4.1   Eigensolvers

- `eigsh` from SciPy: The default implementation that ships with the SciPy module. It interfaces to the external ARPACK package [38] that implements the Lanczos/Arnoldi algorithm.
  PRO: It is stable and there are no convergence problems. It comes with SciPy, so it does not require any steps to set up.
  CONTRA: A single calculation can only make use of one CPU core (multi-core usage possible by parallelizing over $k$ or $B$ points). The memory bandwidth may be a bottleneck for large problems, due to `gemm` operations (matrix-matrix products with general matrices in BLAS [46]) during eigenvector orthonormalization, which is the slowest operation per Lanzcos/Arnoldi iteration.

- `eigsh` from CuPy [47]: Also a Lanzcos/Arnoldi algorithm, written in pure Python. The CuPy library mimics, extends and replaces the NumPy/SciPy packages. Where possible and efficient, the computational workload is done on a CUDA capable GPU. Performance is best on larger problem sizes. Depending on the availability of single and double precision GPU cores, using single precision floating-point arithmetic may be faster, at the expense of lower precision. Using CuPy requires a CUDA capable GPU and a compatible version of the CUDA library [9].
  PRO: It is much faster for large problems compared to the CPU `eigsh` solver, for example about 6 times in non-axial LL mode, depending on setup and problem parameters. The GPUs typically have higher memory bandwidth. TensorCores can drastically speed up the `gemm` operation.
  CONTRA: The RAM limits on GPU are generally tighter than those on CPU. The solver can fail to converge, especially if single precision floating-point arithmetic is used. This can be mitigated by switching to higher precision or falling back to the CPU solver (done automatically by default), but this leads to a reduction in speed. Setting up and configuring this solver requires more effort.

- FEAST solver: Either provided via Intel MKL (version 2.1) or by manual compilation of a shared library (version 4.0).
  PRO: Can be more efficient than default `eigsh`; no additional LU solver is required.
  CONTRA: This solver can be unreliable with densely clustered eigenstates, which is frequently the case for `kdotpy`.

- `eigh` from jax: Converts the sparse matrix to a dense matrix, then uses `eigh` from the Google JAX library to solve on accelerators like GPUs using highly optimized instructions. Scales across multiple accelerators. Currently implemented as fixed double precision.
  PRO: Extremely fast when a large fraction of eigenvalues are requested (in fact, the algorithm does a full diagonalization internally).
  CONTRA: Very memory intensive on GPUs. The entire Hamiltonian (including solutions) needs to fit into GPU memory.

### B.4.2   LU solvers

- SuperLU: The default implementation shipped with current SciPy module.
  PRO: It performs well and does not require additional effort to set up.

---

[9]The CUDA API, provided by Nvidia, is an extension of C/C++ that facilitates computation on GPUs that support it.

CONTRA: Can fail with a memory allocation error for very large matrix sizes.

- UMFPACK: Uses external library shipped with scikit/UMFPACK. Previously, this was used as default by SciPy, but this was changed due to license issues. If UMFPACK is installed (scikit libraries and Python packages), SciPy chooses it automatically for use with `eigsh`.
  PRO: It is stable and efficient, and usually fast.
  CONTRA: In some cases, it may rather be slower than SuperLU.

- PARDISO: Provided by the Intel MKL library. It requires the Intel MKL libraries to be installed along with a suitable Python package that links to it.
  PRO: It is fast, for example about 3 to 4 times with `kdotpy 1d`, efficient and stable.
  CONTRA: The official Python package is no longer maintained. It may require more work to set up.

### B.4.3   Performance considerations

Optimizing solver speed is connected to the problem to be solved, as well as to the computation environment used for the calculation. For the very different workloads created by various `kdotpy` problem sets and the large range of hardware that `kdotpy` could be run on (from small dual core, low RAM PCs over powerful GPU accelerated workstations up to multiple HPC cluster nodes), it is not possible to identify a single eigensolver and a set of optimal parameters that satisfies every scenario.

Automatic settings in `kdotpy` always try to use settings for highest stability, maximum speed and minimal RAM requirements as a rough guideline (in order of priority). However in most scenarios, performance data is not available and has to be found by user experience. The following hardware parameters are important aspects to that affect eigensolver performance:

- CPU/GPU core clock speed: This parameter is mostly fixed. For some systems one might be able to increase the speed by overclocking without affecting other limiting factors, but this typically gives minor speed boosts on the order of few percent.

- CPU instruction sets: Vector and matrix BLAS operations profit much from modern CPU instructions such as AVX. Intel MKL downthrottles AMD CPUs by not making use of some of those features, but this behaviour can be disabled. An older version of MKL might perform best in such cases, but should be benchmarked against current versions.

  On CPU cores: While Python itself has many pitfalls in terms of efficient threading and multi core usage, prominently due to its global interpreter lock (GIL), we can still make good use of additional CPU cores by solving multiple matrices in parallel processes by parallelization over $k$ or $B$ points. As long as we do not run into other limits listed below, this yields almost linear performance improvements. Some of the external libraries (all ones mentioned above, except ARPACK) can also use more cores through multithreading, however this higher core usage is not always rewarded with speed gains. A high degree of parallelization can be energy-inefficient due to the extra overhead. On hardware with inadequate cooling, pushing the CPU to its limits for an extensive time may also lead to performance degradation from thermal throttling.

- RAM size: The maximum amount of RAM in the system sets a hard limit on the number of matrices that can be solved in parallel. Each independent problem requires a similar amount of RAM space during calculation, that can not be shared. Once the threshold is exceeded, the whole process is likely to crash.

- CPU cache size and RAM bandwidth: This is the amount of data the CPU can request from RAM (per second). For some calculations, it is possible, that data for CPU operations can not be stored efficiently enough in the CPU cache (cache miss) and a lot of data has to fetched from RAM. This is a limitation for large matrix-matrix-multiplications (BLAS function `gemm`), as the operation's data cannot fit in cache. In non-axial LL mode, this can be observed when requesting many eigenvalues (large $n_{eig}$ for thick layers, as the orthogonalization of eigenvectors (as part of the Lanczos algorithm) is basically done using two `gemm` calls. Running more simultaneous processes does not increase total solution speed, as it slows down each process due to shared RAM access.

## B.5 Observables

### B.5.1 Spatial observables

The following observables are defined as integrals over spatial coordinates. In the discrete coordinates used in `kdotpy`, they are implemented as summations, e.g., $\sum_i \psi_i^* z_i \psi_i$ for the expectation value $\langle z \rangle$.

- y: Expectation value $\langle y \rangle$, where $y$ is the coordinate transverse to the strip.

- y2: Expectation value $\langle y^2 \rangle$.

- `sigmay`: Standard deviation of y, $\sigma_y = \sqrt{\langle y^2 \rangle - \langle y \rangle^2}$.

- z: Expectation value $\langle z \rangle$, where $z$ is the coordinate parallel to the growth direction.

- z2: Expectation value $\langle z^2 \rangle$.

- `sigmaz`: Standard deviation of $z$, $\sigma_y = \sqrt{\langle z^2 \rangle - \langle z \rangle^2}$.

- zif: Expectation value $\langle z_{\mathrm{if}} \rangle$, where $z_{\mathrm{if}}$ is the (signed) distance to the nearest interface.

- zif2: Expectation value $\langle z_{\mathrm{if}}^2 \rangle$.

- `sigmazif`: Standard deviation of $z_{\mathrm{if}}$, $\sigma_{z_{\mathrm{if}}} = \sqrt{\langle z_{\mathrm{if}}^2 \rangle - \langle z_{\mathrm{if}} \rangle^2}$.

- well: Probability density inside the well.

- wellext: Probability density inside the well and the adjacent 2 nm of barriers.

- interface: Probability density less than 1 nm from each interface. This quantity cannot be used to compare samples of different size.

- interface10nm: Probability density less than 10 nm from each interface. This quantity cannot be used to compare samples of different size.

- custominterface[]: Probability density up to a length in nm, set by the command-line argument custominterfacelength.

- interfacechar: 'Interface character'. Probability density less than 1 nm from each interface divided by what this quantity would be for a uniform probability density, i.e., for a normalized wave function with constant magnitude given by $|\psi|^2 \equiv 1/V$, where $V$ is the volume (size) of the complete sample. If $\Omega$ denotes the domain near the interfaces (e.g., less than 1 nm away) and $V_{\mathrm{if}} = \mathrm{vol}(\Omega)$ its size, then the interface character is $(V/V_{\mathrm{if}}) \int_\Omega |\psi(r)|^2 dr$. Regardless of sample size, values larger than 1 indicate strong interface character.

- interfacechar10nm: Interface character, like interfacechar, but with an interface region of 10 nm instead of 1 nm.

- custominterfacechar[]: Interface character, like interfacechar, with the size of the interface region set by the command-line argument custominterfacelength.

### B.5.2 Inverse participation ratio

Generically, an inverse participation ratio (IPR) for a probability distribution is defined in terms of the fourth moment (kurtosis) scaled by the square of the second moment (variance). For a wave function $\psi(r)$ in terms of spatial coordinate(s) $r$, we define

$$\mathrm{IPR} = \frac{\left( \int |\psi(r)|^2 dr \right)^2}{\int |\psi(r)|^4 dr} = \left( \int |\psi(r)|^4 dr \right)^{-1}, \tag{B.1}$$

where the second equality assumes normalization of the wave function. The resulting IPR has dimensions of length (for a one-dimensional spatial coordinate). `kdotpy` provides IPR observables for three choices of coordinates:

- `ipry` : IPR over the coordinate $y$. If necessary, we first integrate over the coordinate z in order to obtain $|\psi(y)|^2$. The dimensionful variety has units of length. For the dimensionless variety, divide by the width $w$ in $y$ direction.

- `iprz` : IPR over the coordinate $z$. If necessary, we first integrate over the coordinate y in order to obtain $|\psi(z)|^2$. The dimensionful variety has units of length. For the dimensionless variety, divide by the thickness $d$ in $z$ direction.

- `ipryz` : IPR over the coordinates $(y, z)$. The dimensionful variety has units of area (length squared). For the dimensionless variety, divide by $w\,d$.

### B.5.3 Internal degrees of freedom

The following observables are defined in terms of internal degrees of freedom, i.e., (combinations of) orbitals. This includes spin operators.

- `sx`, `sy`, `sz`: Spin expectation value $\langle S_i \rangle$ ($i = x, y, z$). This is proper spin in units of $\hbar$, so that the range of possible values is $[-\frac{1}{2}, \frac{1}{2}]$.

- `jx`, `jy`, `jz`: Expectation values $\langle J_i \rangle$ ($i = x, y, z$) of the total angular momentum. The range of possible values is $[-\frac{3}{2}, \frac{3}{2}]$.

- `yjz`: Expectation values $\langle y J_z \rangle$, i.e., the product of the coordinate $y$ and the angular momentum $J_z$. This is roughly chirality for edge states.

- `gamma6`, `gamma7`, `gamma8`. Expectation values $\langle P_{\Gamma_i} \rangle$ ($i = 6, 7, 8$) of the projection to the $\Gamma_i$ orbitals. This is the probability density in these orbitals.

- `gamma8h`: Expectation values $\langle P_{\Gamma_{8H}} \rangle$ of the projection to the orbital $|\Gamma_8, j_z = \pm 3/2\rangle$. 'H' stands for heavy hole.

- `gamma8l`: Expectation values $\langle P_{\Gamma_{8L}} \rangle$ of the projection to the orbital $|\Gamma_8, j_z = \pm 1/2\rangle$. 'L' stands for light hole.

- `orbital`: Difference of `gamma6` and `gamma8`. The value ranges between $-1$ for purely $\Gamma_8$ states and $1$ for purely $\Gamma_6$ states.

- `jz6`, `jz7`, `jz8`. Expectation values $\langle J_z P_{\Gamma_i} \rangle$ ($i = 6, 7, 8$) of the angular momentum in $z$ direction in the $\Gamma_i$ orbitals.

- `split`: Expectation value of the 'artificial split' Hamiltonian, which can be encoded as $\mathrm{sgn}(J_z)$.

- `orbital[j]`: The squared overlaps of the eigenstates within orbital number $j$, where $j$ runs from 1 to $n_{\mathrm{orb}}$ (the number of orbitals, i.e., 6 or 8). Available for `kdotpy` 2d only, if the command line option `orbitalobs` is given.

### B.5.4 Plain parity and 'isoparity' operators

The following observables implement several plain parity operators (a purely spatial operation) as well as 'isoparity' operators, which combine a spatial and a spin component [48]. The latter are given as actions of the appropriate representations of the relevant point group, namely the double group of $T_d$ or one of its subgroups.

- `pz`: Expectation value of the parity operator in the $z$ coordinate, i.e., $\langle \mathcal{P}_z \rangle = \int \psi^*(z)\psi(-z)\,dz$.

- `px`, `py`: Expectation value of the parity operator in the coordinate $x$ or $y$, i.e., $\langle \mathcal{P}_x \rangle$, $\langle \mathcal{P}_y \rangle$

- `pzy`: Expectation value of the combined parity operator $\langle \mathcal{P}_z \mathcal{P}_y \rangle$

- `isopz`: Expectation value of isoparity $\langle \tilde{\mathcal{P}}_z \rangle$: this is the parity operator combined with the diagonal matrix $Q$ in orbital space, given by $Q = \mathrm{diag}(1, -1, 1, -1, 1, -1, -1, 1)$ in the basis given by Eq. 10. This is a conserved quantum number under a few assumptions [48].

- `isopx`, `isopy`: Expectation value of in-plane isoparity $\langle \tilde{\mathcal{P}}_x \rangle$, $\langle \tilde{\mathcal{P}}_y \rangle$. This is the parity operator in x or y, combined with the appropriate matrix $Q_x$, $Q_y$ in orbital space, which can be found from the appropriate representations of the point group [48].

- `isopzy`: Expectation value of the combined isoparity operator $\langle \tilde{\mathcal{P}}_z \tilde{\mathcal{P}}_y \rangle$

- `isopzw`: Expectation value of a modified isoparity operator like `isopz`, that acts only in the quantum well layer. Formally speaking, if the well ranges from $z = z_{\min}$ to $z_{\max}$, then it maps $z' = z - c$ ($c = (z_{\min} + z_{\max})/2$ being the center of the well) to $-z'$. While `isopz` is not a conserved quantity for asymmetric geometries (e.g., a well layer and two barriers with unequal thickness), `isopzw` can remain almost conserved in that case. Due to incomplete confinement in the well region, the eigenvalues may deviate significantly from $\pm 1$.

- `isopzs`: Expectation value of a modified isoparity operator like `isopz`, that acts only in a symmetric region around the centre of the quantum well layer. It is like `isopzw`, but $z_{\min}$ and $z_{\max}$ are chosen such that the range fits inside the stack, while keeping $c = (z_{\min} + z_{\max})/2$ at the centre of the well. It tends to have eigenvalues closer to $\pm 1$, because it generally covers the probability density in (a part of) the barrier layers as well.

### B.5.5 Hamiltonian terms

The following observables are expectation values of the individual terms in the Hamiltonian.

- `hex`: Exchange energy (expectation value of the exchange Hamiltonian).

- `hex1t`: Exchange energy at $1\,\mathrm{T}$. Regardless of the actual value of the $B$ field, give the expectation value of the exchange Hamiltonian evaluated at $B_z = 1\,\mathrm{T}$.

- `hexinf`: Exchange energy in the large-field limit. Regardless of the actual value of the $B$ field, give the expectation value of the exchange Hamiltonian evaluated for $B_z \to \infty$. In this limit, the Brillouin function is saturated at its maximum absolute value.

- `hz`: Zeeman energy (expectation value of the exchange Hamiltonian).

- `hz1t`: Zeeman energy at $1\,\mathrm{T}$. Regardless of the actual value of the $B$ field, give the expectation value of the Zeeman Hamiltonian evaluated at $B_z = 1\,\mathrm{T}$.

- `hstrain`: Expectation value of the strain Hamiltonian.

### B.5.6 Landau levels

The following are observables based on the Landau level index $n$.

- `llindex`: In the LL mode `sym` or `legacy`, this is the (conserved) LL index $n$. The minimal value is $-2$. The lowest indices $n = -2, -1, 0$ are incomplete, i.e., for these indices, not all orbitals contribute.

- `llavg`: In the LL mode `full`, the Hamiltonian is not diagonal in the basis of LL indexed by $n$. This observable returns the expectation value $\langle n \rangle$ in this basis.

- `llbymax`: In the LL mode `full`, write the wave function in the basis of LL indexed by $n$, and provide the value $n$ with the highest probability density.

- `llmod4`: In the LL mode `full`, analogous to `llavg`, returns the expectation value $\langle n \bmod 4 \rangle$. While `llavg` is not a conserved quantity if axial symmetry is broken, this observable is usually conserved if bulk inversion symmetry is not broken.

- `llmod2`: In the LL mode `full`, analogous to `llavg`, returns the expectation value $\langle n \bmod 2 \rangle$. While `llavg` is not a conserved quantity if axial symmetry is broken, this observable is usually conserved even if bulk inversion symmetry is broken.

- `ll[j]`: The squared overlaps of the eigenstates within Landau level $j$, where $j$ runs from $-2$ to $n_{\max}$, the largest LL index. This observable is available for `kdotpy ll` in full LL mode only, if the command line option `llobs` is given.

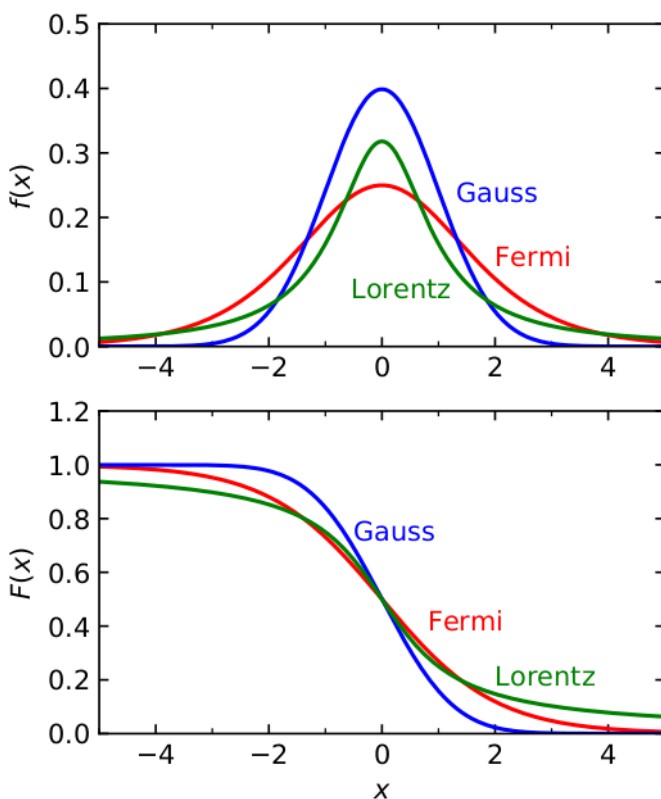

Figure 24: Broadening function kernels $f(x)$ and occupation functions $F(x)$ for the Fermi, Gaussian, and Lorentzian shapes. The argument $x$ is scaled by the width parameter $k_B T$, $\sigma$, or $\gamma$, respectively.

## B.6   Broadening

In `kdotpy`, the following types of broadening are implemented. The application of broadening is done by means of the broadening kernel $f(E)$. These are scaled versions of the probability density functions $\mathbf{PDF}(x)$ of the corresponding distributions. The occupation functions $F(E)$ are also implemented in the code. These are defined as the complementary cumulative density functions $1-\mathbf{CDF}(x)$ of the probability density functions $\mathbf{PDF}(x)$ above. Since the PDFs are symmetric, the CDFs satisfy $1-\mathbf{CDF}(x) = \mathbf{CDF}(-x)$. The kernels $f(x)$ and occupation functions $F(x)$ are plotted in Fig. 24.

- *Thermal (Fermi function)*:
  The thermal broadening function models the broadening of the occupation (function) at finite temperature. The shape of the broadening is given by the Fermi function,

$$f(E) = \frac{1}{k_B T}\frac{e^{-x}}{(1+e^{-x})^2} = \frac{1}{k_B T}\frac{1}{4\cosh^2(x/2)}, \tag{B.2}$$

  where we define $x = E/k_B T$. The characteristic width is $k_B T$, where $T$ is the temperature. The input argument is `broadening T thermal` with the temperature value $T$. The occupation function is

$$F(x) = \tfrac{1}{2}(1 + \tanh(-x/2)) = \frac{e^{-x}}{1+e^{-x}}. \tag{B.3}$$

  The corresponding distribution is also known as the *logistic distribution*.

- *Disorder (Gaussian)*:
  Random disorder is modelled by smearing the density of states with a Gaussian shape,

$$f(E) = \frac{1}{\sqrt{2\pi\sigma^2}}e^{-x^2/2\sigma^2}, \tag{B.4}$$

where $x = E/\sigma$. The broadening parameter $\sigma$ is the standard deviation of the Gaussian distribution. Note that subtly different definitions may be found in literature, for example [3] uses the parameter $\Gamma = \sqrt{2}\sigma$. The input argument is `broadening` $\sigma$ `gauss` with the standard deviation $\sigma$. The occupation function is

$$F(x) = \tfrac{1}{2}\,\mathrm{erfc}(x/\sqrt{2}), \tag{B.5}$$

where $\mathrm{erfc}(z) = 1 - \mathrm{erf}(z)$ is the complementary error function. In statistics, the distribution is usually called the *normal distribution*.

- *Lorentzian*:
  Alternatively, we can use a Lorentzian broadening function,

$$f(E) = \frac{1}{\pi\gamma(1 + x^2)}, \tag{B.6}$$

where $x = E/\gamma$. The width parameter is $\gamma$. Lorentzian line shapes can be used to account for life-time limited broadening in optical transitions experiments. In that context, $\gamma$ is proportional to the inverse of the lifetime of excited states. The input argument is `broadening` $\gamma$ `lorentz` with the width $\gamma$. The occupation function is

$$F(x) = \tfrac{1}{2} + \tfrac{1}{\pi}\,\mathrm{arctan}(-x). \tag{B.7}$$

The Lorentzian line shape is equivalent to the *Cauchy distribution*.

- *Delta function*:
  In the limit of the three functions above, the occupation functions converge to a step function when the broadening parameter approaches $0$,

$$F(x) = \tfrac{1}{2}(1 - \mathrm{sgn}(x)) = \begin{cases} 1 & \text{for } x < 0 \\ 0 & \text{for } x > 0 \end{cases} \tag{B.8}$$

where $x = E$. This function is implemented separately because the other functions are ill-defined in this limit. The corresponding broadening kernel is $f(x) = \delta(x)$, the Dirac delta function (or *delta distribution*).

## B.7 Configuration Options

### B.7.1 Solvers and tasks

- `diag_solver`: The implementation of the function that does the matrix diagonalization; see Appendix B.4 for detailed information. Possible values:

  - `feast`: Use FEAST algorithm (Intel MKL). If this package is not available, fall back to eigsh. Can be tried as an alternative if eigsh fails, e.g. for very large matrices (dim > 5e6).
  - `eigsh`: Use eigsh from SciPy sparse matrix library.
  - `superlu_eigsh`: Same as `eigsh`, but SuperLU is requested explicitly. Enables detailed timing statistics, as with other custom eigsh solvers.
  - `umfpack_eigsh`: Like `eigsh`, but uses umfpack instead of SuperLU for matrix inversion. Recommended for large matrices. Falls back to SuperLU if Scikit UMFPACK is not available. **REQUIRES** available scikit-umfpack and a suitable scipy version.
  - `pardiso_eigsh`: Like `umfpack_eigsh`, but uses Intel MKL PARDISO instead. **REQUIRES** pyMKL package.
  - `cupy_eigsh`: Alternative implementation of the eigsh solver in python. Uses CUDA libraries for Lanczos iteration (on GPU) and PARDISO to SuperLU for matrix inversion, depending on availability. **REQUIRES** CUDA libraries and the CuPy package.
  - `jax_eigh`: Uses the JAX `eigh` solver. First converts sparse matrices to dense. Extremely memory inefficient! Fails if not enough VRAM can be allocated. Use the `gpus` option to reduce the number of workers on the GPU if this happens. This solver is best suited for a large number of `neig`. **REQUIRES** jax package.

- **auto**: Decision based on subprogram. Uses `pardiso_eigsh` for `kdotpy 1d` if available, otherwise uses `eigsh` for all scripts. Suggests alternative solvers if they could be faster. (*default*; *alias*: `automatic`)

- `diag_solver_worker_type`: Sets the parallelization strategy for solver workers. Options:

  - **process**: Use a process pool for solve workers. Recommended strategy for most solvers.
  - **thread**: Use a thread pool in the main process for solve workers. Recommended for CUDA based solver `cupy_eigsh` for optimal GPU workload.
  - **none**: No parallel execution of the solve step. Every solve task is executed serially in the main thread. Recommended for debugging.
  - **auto**: Decision based on `diag_solver`. (*default*; *alias*: `automatic`)

- `diag_solver_cupy_dtype`: Sets the data type for the CuPy solver. Options:

  - **single**: Uses complex numbers with single float precision. This leads to a large speed boost on GPUs with TensorCores. Precision of eigenvalues is worse (on the order of 10 μeV).
  - **double**: Uses complex numbers with double float precision. Solution speed and precision of eigenvalues comparable to other solvers. Medium speed boost expected on GPUs with modern FP64-TensorCores (e.g. Nvidia A100). (*default*)

- `diag_solver_cupy_iterations`: Maximum number of Lanczos iteration steps for both precision options. If number of iterations is exceeded, fall back to better precision first or CPU based solver next. (*default*: 5)

- `diag_solver_cupy_gemm_dim_thr`: Maximum dimension for matrix matrix multiplication in single precision mode. If problem size exceeds this value, the solution is split into multiple smaller problem sets. Smaller values can lead to worse solution speeds, larger values can lead to more numerical problems and fallback to slower double precision solver. (*default*: 4e-6)

- `task_retries`: Number of times a task is restarted after any exception was raised. (*default*: 2)

- `tasks_grouped`: If set to true, all steps for a single `DiagDataPoint` are executed within the same worker/thread with the settings for the `solve_ham` step. Compared to the default mode, this involves less inter-worker data transfers (via pickling), which can give rise to issues with very large eigenvectors. As such, the worker communication behaves similar to kdotpy versions < v0.72. (*default*: false)

### B.7.2 Band Alignment and Character

- `band_align_exp`: Value of the exponent in the minimization function of the 'error' in the band alignment algorithm. A numerical value equal to e means that $\sum(|\Delta E|^e)$ is minimized. Alternatively, if the special value `max` is used, then the minimization function is $\max(|\Delta E|)$. (*default*: 4)

- `band_align_ndelta_weight`: Coefficient of the penalty for reduction of the number of bands in the band alignment algorithm. The higher this value, the more the algorithm 'resists' changes in the number of bands. The value may not be negative (however, 0 is allowed), and too high values should be avoided. It is recommended to use the default value unless the band alignment algorithm does not proceed correctly. (*default*: 20.0)

- `band_char_node_threshold`: In the band character algorithm, this value times the resolution (`zres`) is the minimum value the wave function should reach such that a node (zero) is counted. (*default*: 1e-6)

- `band_char_orbital_threshold`: In the band character algorithm, the maximum value for the probability density ($|\psi|^2$) in an orbital for the probability density to be considered zero. In that case, the orbital content of that orbital is ignored. (*default*: 5e-3)

- `band_char_use_minmax`: In the band character algorithm, whether to use the 'new' node counting method, that counts flips between local extrema. If set to false, use the legacy method. (boolean value; *default*: true)

- `band_char_make_real`: In the band character algorithm, whether to divide the orbital component by the complex phase factor at its maximum, so that the function becomes effectively real, prior to counting the nodes. If set to false, consider both real and imaginary part as is. (boolean value; *default*: false)

- `bandindices_adiabatic_debug`: Whether to write the intermediate result for adiabatic band index initialization to a csv file. This is useful for debugging this algorithm, for example if the charge neutrality point ends up at an incorrect position. (boolean value; *default*: false)

### B.7.3 kdotpy batch

- `batch_float_format`: Format string for representation of float values being replaced in the command string. This is a standard %-style conversion, with the following addition: If a . (period) is added to the end, for example %f ., apply the smart decimal option, i.e., strip superfluous zeros at the end, but keep the decimal point if the value is integer. Useful examples are, among others: %s, %f, %f ., %.3f, %g. (*default*: %s)

- `batch_stderr_extension`: Extension for the file, that kdotpy batch writes stderr to (*default*: txt)

- `batch_stdout_extension`: Extension for the file, that kdotpy batch writes stdout to (*default*: txt)

### B.7.4 BHZ Calculation

- `bhz_allow_intermediate_bands`: Whether to allow a non-contiguous set of A bands. By default (`false`), do not allow B bands in between the A bands. If set to true, relax this restriction. This only takes effect if the input are band labels, e.g., bhz E1 H1 L1. It does not apply to numeric input (e.g., bhz 2 2), which is a contiguous set by definition. NOTE: Setting `true` is experimental, it may cause unexpected errors. (boolean value; *default*: false)

- `bhz_points`: Number of horizontal data points for the BHZ dispersion plot. (*default*: 200)

- `bhz_gfactor`: Whether to output dimensionless g factors in the BHZ output (tex file). If set to false (default), output dimensionful quantities 'G' in meV / T. (boolean value; *default*: false)

- `bhz_abcdm`: Whether to output (tex file) a four-band BHZ model in 'standard form', using coefficients A, B, C, D, M. If this cannot be done, use the generic form instead. (boolean value; *default*: false)

- `bhz_ktilde`: If BHZ is done at a nonzero momentum value $k_0$, whether to express the Hamiltonian in the TeX output as shifted momentum $\tilde{k} = k - k_0$. If set to false, express it in terms of unshifted momentum $k$. This option has no effect for BHZ done at $k_0 = 0$. (boolean value; *default*: true)

- `bhz_plotcolor`: Colour of the BHZ dispersion in the BHZ output file. It may be a single matplotlib colour, a pair separated by a comma (separate colours for each block), or a triplet separated by commas (one block, other block, states without specific block). (*default*: red,blue,black, legacy value: red)

- `bhz_plotstyle`: Style of the BHZ dispersion in the BHZ output file. It may be a single matplotlib line style, a pair separated by a comma (separate styles for each block), or a triplet separated by commas (one block, other block, states without specific block). Examples are solid, dashed, dotted.

### B.7.5   Density of states

- `dos_interpolation_points`: The minimal number of points on the horizontal axis for some DOS and Berry curvature (Hall conductivity) plots. When the calculated number of (k or b) points is smaller, then perform interpolation to at least this number. Must be an integer; if equal to 0, do not interpolate. (*default*: 100)

- `dos_energy_points`: The minimal number of points on the energy axis for some DOS and Berry curvature (Hall conductivity) plots. An energy resolution will be chosen so that the energy interval spans at least this many values. (*default*: 1000)

- `dos_convolution_points`: The minimal number of points in the energy variable taken when applying a broadening function (a convolution operation) to an integrated DOS. If the energy range contains fewer points than this value, the integrated DOS is interpolated. Note that this value affects the 'dummy variable' of the convolution integral only, i.e., the internal accuracy of the (numerical) integration. The broadened integrated DOS (the result) will always be with respect to the same energies as the input. (*default*: 200)

- `dos_print_validity_range`: Print the lower and upper bound of the validity range for DOS and IDOS. If the lower bound (first value) is larger than the upper bound (second value), then the DOS and IDOS are invalid for all energies. See also `plot_dos_validity_range`. (boolean value; *default*: true)

- `dos_print_momentum_multiplier`: The momentum range can be extended by using a multiplier that takes into account the part of momentum space not explicitly calculated. Examples: If only positive momenta are calculated, simulate the negative values by multiplying by 2; or, if in polar coordinates the calculation was done from 0 to 90 degrees angle, multiply by 4 for a full circle. This setting determines whether this multiplicative factor should be printed to the standard output. (boolean value; *default*: false)

- `dos_quantity`: The quantity in which to express density of states. Prior to version v0.95, this was done using the command line arguments `densitypnm`, `densityecm`, etc. Possible values:

  - k: Occupied volume in momentum space; units $1/\mathrm{nm}^d$ (*alias*: `momentum`)
  - p: Density of particles/carriers ($n$ or $dn/dE$); units $1/\mathrm{nm}^d$ (*default*; *alias*: `n`, `particles`, `carriers`, `cardens`)
  - s: Density of states (IDOS or DOS); units $1/\mathrm{nm}^d$. The only difference with p is the way the quantities are labelled. (*alias*: `dos`, `states`)
  - e: Density of charge ($\sigma$ or $d\sigma/dE$); units $e/\mathrm{nm}^d$ (*alias*: `charge`) (The exponent d in the unit is adjusted according to the dimensionality.)

- `dos_unit`: The length units used for density of states. Prior to version v0.95, this was done using the command line arguments `densitypnm`, `densityecm`, etc. Possible values:

  - nm: Units of $1/\mathrm{nm}^d$, $e/\mathrm{nm}^d$ (*default*)
  - cm: Units of $1/\mathrm{cm}^d$, $e/\mathrm{cm}^d$
  - m: Units of $1/\mathrm{m}^d$, $e/\mathrm{m}^d$ In the output, the density values are also scaled to a suitable 'power of ten'. The exponent $d$ in the unit is adjusted according to the dimensionality.

- `dos_strategy_no_e0`: The strategy to follow when trying to extract DOS or IDOS from the band structure, when the zero energy $E_0$ is not well defined. Possible values:

  - strict: Neither DOS nor IDOS can be extracted.
  - dos: DOS can be extracted, but IDOS cannot (*default*).
  - ignore: Both DOS and IDOS can be extracted, ignoring the fact that $E_0$ may lie at an arbitrary energy value. When $E_0$ is defined (either manually or automatically), the extraction of DOS and IDOS is always possible, regardless of this setting.

### B.7.6  Self-consistent Hartree

- `selfcon_acceptable_status`: Maximum status level for the result of the self-consistent Hartree calculation to be considered valid. Possible values:

  - 0: Successful
  - 1: Calculation skipped or aborted (*default*)
  - 2: Did not converge, but convergence is likely after more iterations
  - 3: Did not converge, convergence cannot be estimated or is unlikely
  - 4: Failed

- `selfcon_check_chaos_steps`: Number of previous iterations used for the detection of chaotic behaviour. If this value is set to $n$, we say chaos occurs at iteration $i$ if the previous $V^{(j)}$ closest to $V^{(i)}$ are more than $n$ iterations ago, i.e., $i - j > n$. When chaos is detected, adjust the time step if `selfcon_dynamic_time_step` is set to true. (*default*: 4)

- `selfcon_check_orbit_steps`: Number of previous iterations used for the detection of periodic orbits. We say a periodic orbit occurs at iteration $i$ if the previous $V^{(j)}$ closest to $V^{(i)}$ show a regular pattern like $j - i = 2, 4, 6, 8$; the value $n$ set here is the minimum length of the regular pattern. When a periodic orbit is detected, adjust the time step if `selfcon_dynamic_time_step` is set to true. (*default*: 4)

- `selfcon_convergent_steps`: Number of consecutive convergent steps (iteration steps where the convergence condition is met) required for the self-consistent Hartree calculation to be considered successful. This prevents accidental convergence which could lead to a spurious solution. (*default*: 5)

- `selfcon_debug`: Whether to enable debug mode for the self-consistent Hartree calculation. In debug mode, write temporary files and provide traceback for all exceptions (including `KeyboardInterrupt`) within the iteration loop, which is useful for debugging. If debug mode is disabled, then do not write temporary files and continue on `SelfConError` and `KeyboardInterrupt` exceptions. (boolean value; *default*: false)

- `selfcon_diff_norm`: Method that defines a measure of convergence for the self-consistent calculation. This method is essentially a function applied to the difference of potentials of the last two iteration steps. The result, a nonnegative value, is compared to the convergence criterion. Possible values:

  - `max`: The maximum of the difference. Also known as supremum norm or $L^\infty$ (L-infinity) norm.
  - `rms`: The root-mean-square of the difference. This is the $L^2$ norm. (*default*)

- `selfcon_dynamic_time_step`: Whether the "time" step for the self-consistent calculation is adapted automatically between iterations. If set to false, the time step stays the same between iterations. (boolean value; *default*: false)

- `selfcon_erange_from_eivals`: Whether to use the eigenvalues from first diagonalization result to determine the energy range used for calculating the density of states for the self-consistent calculation. If false, the energy range given in the command line is used instead. (boolean value; *default*: false).

- `selfcon_full_diag`: Whether to use the full-diagonalization approach for the self-consistent Hartree calculation. If true, use the full-diagonalization approach that calculates all conduction band states to determine density as function of $z$. If false, use the standard mode that calculates bands around the charge neutrality point (CNP). The latter is significantly faster, but the results are based on an implausible assumption on the density at the CNP. (boolean value; *default*: true)

- `selfcon_ll_use_broadening`: Whether to enable broadening during self-consistent calculation in LL mode. This can lead to bad convergence behaviour (or no convergence at all, depending on selected broadening), but results in more accurate Hartree potentials for the given broadening. This does not affect the broadening applied to the main diagonalization/postprocessing after the self-consistent calculation has finished. (boolean value; *default*: false)

- `selfcon_energy_points`: The minimal number of points on the energy axis for the self-consistent calculation. An energy resolution will be chosen so that the energy interval spans at least this many values. This number may be fairly high without performance penalty. (*default*: 1000)

- `selfcon_min_time_step`: The minimal value for the "time" step (or "weight") for the self-consistent calculation. If `selfcon_dynamic_time_step` is set to true, the time step can never get lower than this value. Allowed values are between 0 and 1. (*default*: 0.001)

- `selfcon_potential_average_zero`: Shift the potential such that its average will be zero at each iteration of the self-consistent calculation. Enabling this option is recommended for reasons of stability and for consistency of the output. (boolean value; *default*: true)

- `selfcon_symmetrization_constraint`: Constraint on how the symmetry is checked and symmetrization performed on multiple quantities when solving the Poisson equation. When symmetry norm is below threshold the quantity is always fully symmetrized over whole layer stack (except for `never`). Possible values:

  - `never`: Symmetry will not be checked. No symmetrization is performed.
  - `strict`: Symmetry is checked over whole layer stack. (*default*)
  - `loose`: Symmetry is checked over the well region only. This method is preferred for asymmetric layer stacks.

- `selfcon_use_init_density`: Whether a uniform density profile (consistent with the total carrier density) is applied in the initialization of the self-consistent Hartree calculation. If enabled, calculate the potential and apply it to the Hamiltonian in the first iteration. If disabled, use the Hamiltonian with zero potential, unless an initial potential is loaded from a file. (boolean value; *default*: false)

### B.7.7 Optical transitions

- `transitions_min_amplitude`: Minimum amplitude to consider for transitions. The lower this number, the larger the number of data points and the larger the data files and plots. (*default*: 0.01)

- `transitions_min_deltae`: Minimum energy difference in meV to consider for transitions. This value is proportional to a minimal frequency. The smaller this number, the larger the number of data points and the larger the data files and plots. (*default*: 0.1)

- `transitions_max_deltae`: Maximum energy difference in meV of transitions, i.e., upper limit of the vertical axis (filtered transitions plot only). If set to 0, determine the vertical scale automatically. (*default*: 0)

- `transitions_dispersion_num`: Number of transitions to include in the dispersion or B dependence (LL fan) plot. If set to n, the transitions with n highest transitions rates will be shown. If set to 0, show an unlimited number of transitions. (*default*: 4)

- `transitions_broadening_type`: Shape of the broadening function used for broadening the transitions in the absorption plot. Possible choices:

  - `step`: A step function (*alias*: `delta`)
  - `lorentzian`: Lorentzian function (Cauchy distribution), scale parameter gamma, which is the half-width at half-maximum. (*default*; *alias*: `lorentz`)

- – `gaussian`: Gaussian function, scale parameter sigma, which is the standard deviation. (*alias*: `gauss`, `normal`)

- – `fermi`: Fermi function (thermal distribution), scale parameter is energy. (*alias*: `logistic`, `sech`)

- – `thermal`: Fermi function (thermal distribution), scale parameter is temperature.

  The broadening functions for the absorption are the probability density functions for all of these choices; see Appendix B.6 for further information.

- `transitions_broadening_scale`: Scale parameter of the broadening function. This may be an energy (in meV) or a temperature (in K) that determines the amount of broadening (i.e., its 'width'). (*default*: 2.5)

- `transitions_spectra`: (*experimental*) Output spectral plots and tables if a carrier density is set. If false, skip (time consuming) spectra calculation. (boolean value, *default*: false)

- `transitions_plot`: Output transition plot. If false, do not create and save a a transitions plot. (boolean value, *default*: true)

### B.7.8  Colours and Colormaps

- `color_bindex` Colormap for band index. For band index only, the range of the observable is adjusted to the number of colours in the colormap. (*default*: `tab21posneg`; NOTE: for the 'old' set of colours, use `tab20alt`)

- `color_dos`: Colormap for density of states (*default*: `Blues`)

- `color_energy`: Colormap for energy plot (2D dispersion) (*default*: `jet`)

- `color_idos`: Colormap for integrated density of states (*default*: `RdBu_r`)

- `color_indexed`: Colormap for indexed (discrete) observables (*default*: `tab20alt,tab20`)

- `color_indexedpm`: Colormap for indexed (discrete) observables using a 'dual' colour scale, such as `llindex.sz` (*default*: `tab20`)

- `color_ipr`: Colormap for IPR observables (*default*: `inferno_r`)

- `color_localdos`: Colormap for local density of states (*default*: `cividis,jet`)

- `color_posobs`: Colormap for observables with positive values (*default*: `grayred`)

- `color_shadedpm`: Colormap for continuous positive observables using a 'dual' colour scale, such as `y2.isopz` (*default*: `bluereddual`)

- `color_sigma`: Colormap for 'sigma observables' (standard deviation) (*default*: `inferno_r`)

- `color_symmobs`: Colormap for observables with a symmetric (positive and negative) range of values (*default*: `bluered`)

- `color_threehalves`: Colormap for observables with range [-3/2, 3/2] (*default*: `yrbc`)

- `color_trans`: Colormap for transition plots (*default*: `hot_r`)

- `color_wf_zy`: Colormap for wavefunction plot $|\psi(z, y)|^2$ (*default*: `Blues`)

### B.7.9 Figures

- `fig_matplotlib_style`: Matplotlib style file for changing the properties of plot elements. This may be a file in the configuration directory `~/.kdotpy` or in the working directory, or a built-in matplotlib style. (*default*: `kdotpy.mplstyle`)

- `fig_hsize`, `fig_vsize`: Horizontal and vertical size of the figures, in mm. (*default*: 150, 100, respectively)

- `fig_lmargin`, `fig_rmargin`, `fig_bmargin`, `fig_tmargin`: Figure margins (left, right, bottom, top), i.e., the space in mm between the figure edge and the plot area. (*default*: 20, 4, 12, 3, respectively)

- `fig_charlabel_space`: Vertical space for the character labels in the dispersion plot, in units of the font size. To avoid overlapping labels, use a value of approximately 0.8 or larger. (*default*: 0.8)

- `fig_colorbar_space`: Space reserved for the colour bar legend in mm (*default*: 30) In other words, this is the distance between the right-hand edges of the figure and the plot if a colour bar is present. It 'replaces' the right margin.

- `fig_colorbar_margin`: Space between the right-hand edge of the plot and the colour bar legend, in mm. This space is taken from the colour bar space (set by `fig_colorbar_width`), so it does not affect the right-hand edge of the plot. (*default*: 7.5)

- `fig_colorbar_size`: Width of the actual colour bar in mm (*default*: 4)

- `fig_colorbar_method`: Way to place the colour bar; one of the following options:

    - `insert`: Take space inside the existing plot; keep the figure size, but decrease the plot size. (*default*)
    - `extend`: Add extra space; keep the plot size but increase the figure size.
    - `file`: Save into a separate file. The original figure is not changed.

- `fig_colorbar_labelpos`: Method to determine the position of the label of the colour bar. One of the following options:

    - `legacy`: The 'old' method, using `colorbar.set_label` plus a manual shift.
    - `xaxis`: As label for the 'x axis', directly below the the colour bar.
    - `yaxis`: As label for the 'y axis', vertically up on the right-hand side.
    - `center`: Centred in the whole space allocated for the colour bar, including margins; very similar to 'legacy' for default settings of the colour bar size and margins. (*default*)
    - `left`: Left aligned with the left border of the colour bar.

- `fig_colorbar_abstwosided`: Whether a shaded dual colour bar, where the vertical value is the absolute value of the observable, should show the observable itself, with values running from -max to max (if set to true; default). Otherwise show the absolute value, running from 0 to max. (boolean value; *default*: true)

- `fig_extend_xaxis`: Relative extension of the horizontal plot range for dispersion and magnetic field dependence. For example, a value of 0.05 means 5% of the range is added left and right of the minimum and maximum x value (where x is k or B), respectively. This does not affect the range if the command line argument `xrange` is used. Use 0 to not extend the plot range. The value may not be negative. (*default*: 0.05)

- `fig_inset_size`: Size (width and height) of the inset legend in mm. Values smaller than 30 are not recommended. (*default*: 30)

- `fig_inset_margin`: Space between inset edge and plot edge in mm. (*default*: 3)

- `fig_inset_color_resolution`: Number of color gradations along each axis for the RGB (inset) legend. Do not change unless file size is an issue. (*default*: 20)

- `fig_legend_fontsize`: Specify the font size of the legend. May also be set to 'auto' for automatic; this yields a font size of 8 for RGB (inset) legend, 10 for other legend or colorbars (may be subject to settings in matplotlibrc and/or style files). (*default*: `auto`)

- `fig_spin_arrow_length`: Arrow length in spin plots. The value is the length in mm for arrows representing spin value 0.5 or a direction. (*default*: 5)

- `fig_max_arrows`: Maximum number of arrows in a vector plot in each dimension. The value 0 means no limit. (*default*: 20)

- `fig_arrow_color_2d`: Color of the arrows in a 2D vector plot. This must be a valid matplotlib color. (*default*: #c0c0c0)

- `fig_ticks_major`: Strategy to determine the major ticks in the plots. Possible choices:

    - `none`: No major ticks
    - `auto`: Determine number of ticks automatically (based on plot size). (*default*)
    - `fewer`: A few ticks per axis (typically 3)
    - `normal`: A moderate amount of ticks per axis (typically 6)
    - `more`: Many ticks per axis (typically 12)

  One can use different choices for the horizontal and vertical axis, as follows:
  `fig_ticks_major=normal,fewer`

- `fig_ticks_minor`: Strategy to determine the minor ticks in the plots. Possible choices:

    - `none`: No minor ticks (*default*)
    - `auto`: Determine automatically (matplotlib's algorithm)
    - `fewer`: Few minor ticks (major interval divided by 2)
    - `normal`: Moderately many ticks (major interval divided by 4 or 5).
    - `more`: Many minor ticks (major interval divided by 10)

  One can use different choices for the horizontal and vertical axis, as follows:
  `fig_ticks_minor=fewer,none`

- `fig_unit_format`: Opening and closing bracket of the units in axis and legend labels. (*default*: [])

### B.7.10   Plot output

- `plot_constdens_color`: The colour of the curves in the 'constdens' plots. The value must be a valid matplotlib colour. (*default*: blue)

- `plot_dispersion_default_color`: The uniform colour of the dispersion curves, if there is no colour scale set for the given observable (or if no observable is set). The value must be a valid matplotlib colour. (*default*: blue)

- `plot_dispersion_energies`: Plot special energies, e.g., charge-neutrality point, Fermi energy/chemical potential at zero and finite density in dispersion plots. (boolean value; *default*: true)

- `plot_dispersion_energies_color`: The line colour for special energies. The value must be a valid matplotlib colour. If left empty, take `lines.color` from matplotlibrc or a style file. (*default*: black)

- `plot_dispersion_parameter_text`: Write an indication in the plot for constant parameter values, e.g., when plotting along $k_x$ for a nonzero ky value, write "For $k_y = $". (boolean value; *default*: true)

- `plot_dispersion_stack_by_index`: If enabled, make sure the data with the lowest band or Landau-level index is shown on top, to make sure the 'most interesting data' (low-index states) is not obscured by 'less interesting data' (high-index states). Otherwise, the plot function uses the default plot stacking order: the data is then drawn simply in the order by which it is processed. (boolean value; *default*: false)

- `plot_dos_color`: Colour of the curves in the (integrated) density of states (IDOS/DOS) plots. The value must be a valid matplotlib colour. (*default*: `blue`)

- `plot_dos_energies`: Plot special energies, e.g., charge-neutrality point, Fermi energy/chemical potential at zero and finite density in density (DOS) plots. (boolean value; *default*: true)

- `plot_dos_fill`: Fill the area between the curve and zero in the DOS plot (not integrated DOS). (boolean value; *default*: false)

- `plot_idos_fill`: Fill the area between the curve and zero in the integrated DOS plot. (boolean value; *default*: false)

- `plot_dos_units_negexp`: Use negative exponents in DOS units in density plots. If set to true, write $\mathbf{nm^{-2}}$ instead of $\mathbf{1/nm^2}$, for example. (boolean value; *default*: false)

- `plot_dos_validity_range`: Shade the area in the (integrated) DOS plot where the value is expected to be incorrect due to missing data (due to momentum cutoff). (boolean value; *default*: true)

- `plot_dos_vertical`: Plot the (integrated) DOS sideways, so that energy is plotted on the vertical axis. The vertical scale will match the dispersion plot, so that these figures can be put side-by-side with a common axis. (boolean value; *default*: true)

- `plot_ecnp`: Plot the charge neutral energy as function of k or B. This is the boundary between "electron" and "hole" states (positive and negative band indices, respectively). (boolean value; *default*: false)

- `plot_rasterize_pcolormesh`: Whether to rasterize plot elements created with `pcolormesh` from matplotlib. This is used primarily for two-dimensional color plots with `kdotpy ll` when one uses quadratic stepping for the magnetic field values. Rasterization leads to improved performance both in creating the plots as well as in rendering them with a pdf viewer. The resolution can be controlled with the matplotlibrc parameters `figure.dpi` and `savefig.dpi`. If the old behaviour is desired, i.e., that the data is rendered as vector graphics, set the value of `plot_rasterize_pcolormesh` to `false`. (boolean value; *default*: true)

- `plot_rxy_hall_slope`: Plot the Hall slope $R_{xy} = B/(ne)$, where $B$ is magnetic field, $n$ is density and $e$ is electron charge, in the plots for $R_{xy}$ (`rxy-constdens.pdf`) as a dashed line. (boolean value; *default*: true)

- `plot_sdh_markers`: Whether to show markers for the period of the Shubnikov-de Haas (SdH) oscillations in the 'constdens' plot (both 'normal' and 'SdH' versions). The markers are placed at the values for which 1 / B is a multiple of $e/(2\pi\hbar n)$, where n is the density. (boolean value; *default*: true)

- `plot_sdh_markers_color`: Colour of the SdH markers in the 'constdens' plots. The value must be a valid matplotlib colour. (*default*: `red`)

- `plot_sdh_scale_amount`: The maximum number of SdH oscillations to be shown in the SdH plot. If set to a nonzero value, the scale on the horizontal axis is magnified to this amount of SdH oscillations. The scale is never shrunk, so there may be fewer SdH oscillations on the axis. The 'constdens' plot linear in B is unaffected. If set to 0 (default), do not scale the axis. A typical useful nonzero value is 20.

- `plot_transitions_labels`: Show some labels in transitions plot. (boolean value; *default*: true)

- `plot_transitions_quantity`: Which quantity to use for colouring in the transitions plot. Possible choices:

  - `amplitude`: 'Raw' amplitude gamma from Fermi's golden rule
  - `rate`: Transition rate density, $n\Gamma(f_2 - f_1)$ (*default*; *alias*: `rate_density`)
  - `occupancy`: Occupancy difference $f_2 - f_1$
  - `deltae`: Energy difference $|E_2 - E_1|$ in meV
  - `freq` : Corresponding frequency in THz (*alias*: `freq_thz`)
  - `lambda`: Corresponding wave length in μm (*alias*: `lambda_μm`, `lambda_um`)
  - `absorption`: Absorption (relative attenuation of intensity) A

- `plot_transitions_frequency_ticks`: Plot frequency ticks at the left and right energy axis for transitions plots. (boolean value; *default*: true)

- `plot_transitions_max_absorption`: Upper limit of the colour scale in the transitions absorption plot. For the relative absorption, use `[-value, value]` as the colour range. (*default*: 0.03)

- `plot_wf_orbitals_realshift`: Phase-shift the orbital functions to purely real values before plotting. This results in a single line plot per orbital with consistent amplitudes and signs. The actual phases are still given at the right side of the figure. Uses straight/dashed lines for +/- angular momentum orbitals. (boolean value; *default*: false)

- `plot_wf_orbitals_order`: Order of the orbitals in the legend, for wave function plot style 'separate'. Possible choices:

  - `standard` (*default*):
    $$\begin{array}{ccc} \Gamma_6, +1/2 & \Gamma_8, +1/2 & \Gamma_7, +1/2 \\ \Gamma_6, -1/2 & \Gamma_8, -1/2 & \Gamma_7, -1/2 \\ \Gamma_8, +3/2 & \Gamma_8, -3/2 & \end{array}$$

  - `paired`:
    $$\begin{array}{ccc} \Gamma_6, +1/2 & \Gamma_6, -1/2 & \Gamma_7, +1/2 \\ \Gamma_8, +1/2 & \Gamma_8, -1/2 & \Gamma_7, -1/2 \\ \Gamma_8, +3/2 & \Gamma_8, -3/2 & \end{array}$$

  - `table`:
    $$\begin{array}{ccc} & \Gamma_8, +3/2 & \\ \Gamma_6, +1/2 & \Gamma_8, +1/2 & \Gamma_7, +1/2 \\ \Gamma_6, -1/2 & \Gamma_8, -1/2 & \Gamma_7, -1/2 \\ & \Gamma_8, -3/2 & \end{array}$$

  For the six-orbital basis, the $\Gamma_7$ states are omitted.

- `plot_wf_zy_format`: File format for wavefunction plots $|\psi(z, y)|^2$. Possible choices:

  - `pdf`: Multi-page PDF if possible, otherwise separate PDF files. (*default*)
  - `png`: Separate PNG files.
  - `pngtopdf`: Separate PNG files are converted and merged into a multi-page PDF. Requires the 'convert' command to be available. (*alias*: `png_to_pdf`)

- `plot_wf_mat_label_rot`: For wave function plots, the rotation (in degrees) of material labels inside the layers. Can be used to fit long labels in thin layers. (*default*: 0)

- `plot_wf_zy_bandcolors`: Colour model for the wavefunction plots $|\psi(z, y)|^2$ separated by bands. Possible choices:

- **hsl**: Hue-saturation-lightness. The colour (hue) is determined by the relative content of the bands, the saturation and lightness by the density.
- **hsv**: Hue-saturation-value. Like **hsl**, the colour (hue) is determined by the relative content of the bands, the saturation by the density, and the value is equal to 1.
- **rgb**: Red-green-blue. The red, green, and blue channels are determined by the contents of the bands.

NOTE: This is not a colormap! For the absolute value without band content, use the colormap set by **color_wf_zy**.

- **plot_wf_zy_scale**: Scaling method (colour scale normalization) for wavefunction plots $|\psi(z,y)|^2$. Possible choices:

  - **separate**: Normalize the colour scale for each wavefunction individually. (*default*)
  - **together**: Normalize the colour scale for all wavefunctions collectively.

- **plot_wf_y_scale**: Scaling method for the vertical axis for wave function plots $|\psi(y)|^2$. Possible choices:

  - **size**: Determine scale from sample size (width in y direction. (*default*; *alias*: **width**)
  - **magn**: Determine scale from magnetic field. For small fields, use the sample size.
  - **separate**: Determine scale from the maximum of each wave function individually.
  - **together**: Determine scale from the maximum of all wave functions collectively.

- **plot_wf_delete_png**: If the wavefunction plots are saved in PNG format and subsequently converted to a single multi-page PDF, delete the PNG files if the conversion is successful. (boolean value; *default*: true)

- **plot_wf_together_num**: For the wavefunction plot in **together** style, plot this many wave functions. Must be a positive integer. (*default*: 12)

### B.7.11 CSV output

- **csv_style**: Formatting for csv output. Possible values:

  - **csvpandas**: Comma separated values using pandas module
  - **csvinternal**: Comma separated values using internal function
  - **csv**: Choose **csvpandas** if pandas is available, otherwise choose **csvinternal** (*default*)
  - **align**: Align values in columns in the text file

- **csv_multi_index**: Determines how a multi-index (LL index, band index) is formatted in csv output. Possible values:

  - **tuple**: As a tuple (##, ##) (*default*)
  - **llindex**: Only the LL index
  - **bindex**: Only the band index
  - **split**: LL index on first row, band index on second row (*alias*: **tworow**)
  - **short**: Short version of tuple ##,## (space and parentheses are omitted)

- **csv_bandlabel_position**: Location of the band labels in the 'by-band' CSV output. Possible values:

  - **top**: At the very top, above the other column headings. (*default*; *alias*: **above**)
  - **second**: Between the data and the other column headings. (*alias*: **between**)
  - **bottom**: At the bottom, below the data. (*alias*: **below**)

### B.7.12 Table output

- `table_berry_precision`: Precision (number of decimals) for floating point numbers, for the Berry curvature csv files. (*default*: 4)

- `table_data_label_style`: Style for expressing data labels in generic two-dimensional csv output, such as density of states and Berry curvature. The label is positioned at the end of the first row with data. Possible choices: `none` (*alias*: `false`), `raw`, `plain`, `unicode`, `tex` (for details, see `table_dispersion_unit_style` below). If `none`, do not write a label. (*default*: `plain`)

- `table_data_unit_style`: Style for expressing the unit in generic two-dimensional csv output. Possible choices: `none` (*alias*: `false`), `raw`, `plain`, `unicode`, `tex` (for details, see `table_dispersion_unit_style` below). If `none`, do not write a unit. Also, if `table_data_label_style` is set to `none`, this option is ignored and no unit is written. (*default*: `plain`)

- `table_dos_precision`: Precision (number of decimals) for floating point numbers, for the density of states csv files. (*default*: 8)

- `table_dos_scaling`: Whether to apply density scaling for csv output of densities. If false, use the native units ($nm^{-2}$ in two dimensions). Otherwise, use the same scaling as for plots. (boolean value; *default*: false)

- `table_dos_units_negexp`: Use negative exponents in DOS units for csv output. If set to true, write $nm^{-2}$ instead of $1/nm^2$, for example. (boolean value; *default*: false)

- `table_dispersion_precision`: Precision (number of decimals) for floating point numbers, for the dispersion csv files. Energy and momentum values may use a different number of decimals. (minimum: 2, *default*: 5)

- `table_dispersion_data_label`: Whether to include the observable at the end of the first data row in a multi-dimensional dispersion csv table (e.g., with two or three momentum variables). (boolean value; *default*: true)

- `table_dispersion_units`: Whether to include units of the variables and observables in dispersion csv files. For a one-dimensional dispersion, these are included as second header row. For a multi-dimensional dispersion, the unit is added at the end of the first data row. (boolean value; *default*: true)

- `table_dispersion_unit_style`: Style for expressing units. Possible choices:

  - `raw`: 'Without' formatting
  - `plain`: Plain-text formatting using common symbols (e.g., square is ^2 and Greek letters are spelled out)
  - `unicode`: Formatting using 'fancy' Unicode symbols (e.g., square is the superscript-2 symbol and Greek letters use their corresponding Unicode symbol).
  - `tex`: LaTeX formatting

  (*default*: `plain`)
  NOTE: Even with `raw` or `plain`, there may still be some non-ASCII symbols, for example µ.

- `table_dispersion_obs_style`: Style for expressing observables/quantities. Possible choices: `raw`, `plain`, `unicode`, `tex` (see above). (*default*: `raw`)

- `table_qz_precision`: Precision (number of decimals) for floating point numbers, for the 'Q(z)' (z-dependent quantity) csv files. (*default*: 5)

- `table_extrema_precision`: Precision (number of decimals) for floating point numbers, for the extrema csv files. (*default*: 5)

- `table_transitions_precision`: Precision (number of decimals) for floating point numbers, for the transitions csv files. (*default*: 3)

- `table_absorption_precision`: Precision (number of decimals) for floating point numbers, for the absorption csv files (associated with transitions). (*default*: 5)

- `table_transitions_ratecoeff_unit`: Unit for the rate coefficient for optical transitions. (*default*: nm^2/mV/ns)

- `table_wf_files`: Which type of files should be written for the wave function data. Possible choices:

    - `none`: No files are written.
    - `csv`: Write csv files only. (*default*)
    - `tar`: Write csv files, pack them into a tar file.
    - `targz`: Write csv files, pack them into a gzipped tar file (compression level 6). (*alias*: gz, gzip, tar.gz)
    - `zip`: Write csv files, pack them into a zip file with 'deflate' compression.
    - `zipnozip`: Write csv files, pack them into a zip file without compression.

  For the archive options (`tar`, `zip`, etc.), the csv files are deleted if the archive has been written successfully; otherwise they are kept. Some options may be unavailable depending on the installed Python modules.

- `table_wf_precision`: Precision (number of decimals) for floating point numbers, for the wave function csv files. (*default*: 5)

### B.7.13 XML output

- `xml_omit_default_config_values`: If set to true, do not save all configuration values to the XML output file, but only the ones that are set to a value different than the default value. Otherwise save all values (*default*); this is recommended for reproducibility. (boolean value; *default*: false)

- `xml_shorten_command`: If set to true, replace the script path in the `<cmdargs>` tag by `kdotpy xx` (where xx = 1d, 2d, etc.) if typing `kdotpy` on the command line refers to the kdotpy main script. For this, the main script (or a link to it) must be in the `PATH` environment variable; this is generally the case if kdotpy has been installed with pip. (boolean value; *default*: false)

### B.7.14 Miscellaneous

- `berry_dk`: Momentum step size (in $\mathbf{nm^{-1}}$) for calculating the derivative of the Hamiltonian in the calculation of the Berry curvature as function of momentum. It does not apply to the Berry curvature calculation in Landau-level mode. The value must be positive. (*default*: 1e-3)

- `berry_ll_simulate`: Whether to use simulated Berry curvature (more accurately: Chern numbers) for Berry / Hall output, for `kdotpy ll`, instead of the calculated one. The calculated value may sometimes show artifacts that cannot be easily resolved by increasing number of eigenstates for example. The simulated Berry curvature (observable `berrysim`) is set to exactly 1 for all states at nonzero magnetic field. (boolean value; *default*: false)
  **Hint:** One may do a comparison by doing the calculation twice with settings `true` and `false`, respectively. The output is written to different file names as to ease the comparison.

- `diag_save_binary_ddp`: Whether and how to save intermediate binary files for each `DiagDataPoint` (diagonalization data point). Possible choices:

    - `npz`: The NumPy (compressed) binary file format[10] (*alias*: numpy)

---

[10]See https://numpy.org/doc/stable/reference/generated/numpy.lib.format.html.

- h5: HDF5 data format. This requires the Python module h5py to be installed[11]. (*alias*: hdf5)

- false: Do not save intermediate files (*default*)

NOTE: This configuration value is independent from the command line option tempout. The npz and hdf5 formats are meant for permanent data storage, the tempout files are only safe for immediate re-use and should not be used for long-term storage.

- job_monitor_limit: If the number of data points is smaller than this value, show the full job monitor with information about the intermediate steps. Otherwise, show the simple in-line progress indicator. For the value 0, always show the simple progress indicator. (*default*: 101)

- lattice_regularization: Enables or disables lattice regularization. The settings true and false correspond to the obsolete command-line arguments latticereg and nolatticereg, respectively. The recommended value and default value is false. Note that for older kdotpy versions (kdotpy v0.xx), the default value was true for compatibility reasons. (boolean value; *default*: false)

- lattice_zres_strict: Enables or disable strict check of commensurability of z resolution with thickness of the layers, i.e., whether the thicknesses are integer multiples of the z resolution. If they are incommensurate, quit with an error if strict checking is enabled. If disabled, change the thicknesses to match the z resolution and raise a warning. (boolean value; *default*: true)

- magn_epsilon: Numeric value that determines whether small values near zero need to be inserted if the grid contains magnetic fields. The value zero means disabling this feature. Otherwise, $+/-$ the absolute value of magn_epsilon is inserted at either side of $B = 0$, whichever side (positive or negative) is included in the range. If negative, insert the values only if the range is two-sided. The motivation for including this option is to reduce some plot artifacts for ranges that contain positive and negative magnetic fields. For this option to be effective, it might also be necessary to set the split parameter to a small value. (*default*: -1e-4)

- numpy_linewidth: Sets the (approximate) line width for NumPy array output. (This output is used in verbose mode mostly.) The value is passed to numpy.set_printoptions(). The value has to be an integer $\geq 0$. The output is always at least one column, so small values may be exceeded. (*default*: 200)

- numpy_printprecision: The number of digits of precision for NumPy array floating point output. (This output is used in verbose mode mostly.) The value is passed to numpy.set_printoptions(). The value has to be an integer $\geq 0$. The number of digits shown does not exceed the number needed to uniquely define the values, e.g., 17 digits for 64-bit floating point numbers. (*default*: 6)

- wf_locations_exact_match: If set to true (*default*), the wave function locations should match the momentum/magnetic field values exactly. If no exact match is found, skip the location ('old behaviour'). If set to false, find the nearest value to each location. (boolean value; *default*: true)

- wf_locations_filename: Whether to label the wave function files using the position (momentum/magnetic field). If set to false, label with numbers. (boolean value; *default*: true)

NOTE: Boolean configuration options may have the following values (not case sensitive):

- For "True": yes, y, true, t, 1, enabled, on

- For "False": no, n, false, f, 0, disabled, off

---

[11]See https://docs.h5py.org.

# C  Command-line arguments

This command line reference lists the commands for the present version, kdotpy v1.0.0. We note that commands may change (sometimes only subtly) between versions. Thus, we recommend the user to refer to the wiki [34] and/or the built-in help for up-to-date information if a newer version is used.

In the list below, we use the symbol # to indicate additional arguments. The bracketed [#] stands for an optional argument. The list is ordered thematically. Some commands have aliases which may be used instead of the listed command, with identical functionality. The input of the listed commands is case insensitive and ignores underscores, so that for example DOS may be used instead of dos and z_res instead of zres. The additional arguments are usually case sensitive. For example, in out –HgTe, the distinction between uppercase and lowercase is respected.

## C.1  Options affecting computation

### C.1.1  Modelling

Determine the model, i.e., the type of Hamiltonian that needs to be constructed.

- norb #: Number of orbitals in the Kane model. The argument can be either 6 or 8, which means exclusion or inclusion, respectively, of the $\Gamma_7$ orbitals. (*Alias*: orbitals, orb).
  Shorthand for norb 6: 6o, 6orb, 6orbital, 6band, sixband
  Shorthand for norb 8: 8o, 8orb, 8orbital, 8band, eightband
  NOTE: Omission of this input is not permitted.

- noren: Do not renormalize the parameters if using anything else than then eight-orbital Kane model. (*Alias*: norenorm, norenormalization, norenormalisation)

- lllegacy, llfull: Force Landau level mode to be 'legacy' or 'full'. By default, the Landau level calculation uses either the symbolic mode 'sym' if possible or the full mode if necessary. The legacy mode may not be used if the full mode were required. By giving llfull, one may also use the full mode if the automatically chosen 'sym' mode does not give the desired results. Beware that full mode is much heavier on resources. (kdotpy ll and kdotpy bulk-ll)

- llmax: Maximum Landau level index. This has to be an integer $\geq 0$. If omitted, 30 is used. Larger values yield a more complete result, but require more computation time and memory. (*Alias*: nll)

### C.1.2  Regularizations, degeneracy lifting, etc.

These options fine-tune the model.

- noax: Include non-axial terms, i.e., break the axial symmetry. (*Alias*: noaxial, nonaxial)

- ax: Use axial symmetry, i.e., the axial approximation. This is the default for Landau level mode (kdotpy ll and kdotpy bulk-ll). For dispersions, it is mandatory to provide either ax or noax. (*Alias*: axial)

- split #: Splitting (in meV) to lift the degeneracies. It is recommended to keep it small, e.g., 0.01.

- splittype #: Type of degeneracy splitting. One of the following choices:

  - automatic: Choose sgnjz if BIA is disabled, bia if BIA is enabled. (*Alias*: auto; *default*)

  - sgnjz: Use the operator $\mathbf{sgn}(J_z)$, i.e., the sign of the total angular momentum. Despite the fact that this quantity is not a conserved quantum number, it works remarkably well. This is also the default for calculations without BIA.

  - sgnjz0: Use the operator $\mathbf{sgn}(J_z)$ at $\mathbf{k} = \mathbf{0}$ only. This type can be useful if the degeneracy is broken for $\mathbf{k} \neq \mathbf{0}$ as a result of some other term, for example an electric field.

- isopz: Use isoparity ('isopz') $\tilde{\mathcal{P}}_z$. This observable distinguishes the two blocks and is a conserved quantum number for symmetric geometries in many circumstances. Sometimes gives cleaner results than sgnjz. See Ref. [48] for more information about isoparity.

- isopzw: Use isoparity applied to the well layer only, like observable isopzw. While isopz is not a conserved quantity for asymmetric geometries (e.g., a well layer and two barriers with unequal thickness), isopzw can remain almost conserved in that case. Due to incomplete confinement in the well region, the eigenvalues may deviate significantly from ±1.

- isopzs: Use isoparity applied to a region symmetric around the centre of the well layer, like observable isopzs. Like isopzw, the observable isopzs is also an almost conserved quantity for asymmetric geometries and tends to have eigenvalues closer to ±1, because it generally takes into account the decaying wave function in (a part of) the barriers.

- bia: Modified form of sgnjz, that works better if bulk inversion asymmetry (BIA) is present.

- helical: Momentum dependent splitting, with the quantization axis along the momentum direction. The splitting is proportional to $\mathbf{k} \cdot \mathbf{S}/|\mathbf{k}|$ for $\mathbf{k} \neq \mathbf{0}$. It is set to zero at $\mathbf{k} = \mathbf{0}$.

- helical0: Same as helical, but with $\mathbf{sgn}(J_z)$ at $\mathbf{k} = \mathbf{0}$. This option may be useful to prevent issues caused by degeneracies at $\mathbf{k} = \mathbf{0}$ for the option helical.

- cross: Momentum dependent splitting, with the quantization axis perpendicular to the in-plane momentum direction, i.e., $(k_x S_y - k_y S_x)/|\mathbf{k}|$ for $\mathbf{k} \neq \mathbf{0}$. It is set to zero at $\mathbf{k} = \mathbf{0}$.

- cross0: Same as cross, but with $\mathbf{sgn}(J_z)$ at $\mathbf{k} = \mathbf{0}$. This option may be useful to prevent issues caused by degeneracies at $\mathbf{k} = \mathbf{0}$ for the option cross.

For the relevant observables, see Appendix B.5.

- bia: Include bulk inversion asymmetry. Note that combination of BIA with 'split' may cause unwanted asymmetries, for example under $k_z \rightarrow -k_z$. (For kdotpy 2d, kdotpy ll, and kdotpy bulk)

- ignoremagnxy: Ignore the in-plane components of the magnetic field in the gauge field (i.e., the 'orbital field'). The in-plane components still have an effect through the Zeeman and exchange couplings even if this option is enabled. Enabling this option 'simulates' the calculation before the in-plane orbital fields were implemented, as of version v0.58 (kdotpy 1d) or v0.74 (kdotpy 2d), respectively. (*Alias*: ignoreorbxy, ignorebxy)

- gaugezero #: Set the *y* position where the magnetic gauge potential is zero. The position coordinates are relative: -1.0 and +1.0 for the bottom and top edges of the sample, 0.0 for the centre (*default*). (*Alias*: gauge0)

- yconfinement #: Set a confinement in the *y* direction; local potential on the outermost sites, in meV. A large value (such as the *default*) suppresses the wave function at the edges, which effectively imposes Dirichlet boundary conditions (wave functions = 0). If the value is set to zero, the boundary conditions are effectively of Neumann type (derivative of wave functions = 0). *Default*: 100000 (meV). (*Alias*: yconf, confinement)

### C.1.3   Diagonalization options

Options that affect the diagonalization, i.e., which energy eigenvalues are calculated from the Hamiltonian.

- neig #: Number of eigenvalues and -states to be asked from the Lanczos method. (*Alias*: neigs)

- targetenergy # [# ...]: Energy (meV) at which the shift-and-invert Lanczos is targeted. If multiple values are given, then apply Lanczos at each of these energies (experimental feature). If large numbers of eigenvalues are to be calculated (e.g., 500), it may be faster to calculate multiple sets with a smaller number of eigenvalues (e.g., 5 sets of 150). Note that the values need to be chosen carefully. If there is no overlap between the intervals where eigenvalues are found, the calculation is aborted. For smaller numbers of eigenvalues, it is recommended to use a single value for targetenergy. (*Alias*: e0)

- `energyshift #`: Shift energies afterwards by this amount (in meV). Other energy values may still refer to the unshifted energies. This is an experimental feature that should be used with care. In case one intends to merge data (e.g., using `kdotpy merge`), then one should avoid using this option for the individual runs. Afterwards, this option may be used with `kdotpy merge`. (*Alias*: `eshift`)

- `zeroenergy`: Try to align the charge-neutral gap with $E = 0\,\mathrm{meV}$. In combination with `energyshift`, align at that energy instead of $0\,\mathrm{meV}$. See also the warnings under `energyshift`.

- `bandalign [# [#]]`: Try to (re)connect the data points, by reassigning the band indices. The first optional argument determines the 'anchor energy', i.e., the energy at $\mathbf{k} = \mathbf{0}$ (or $\mathbf{B} = \mathbf{0}$) that separates bands with positive and negative indices. If the second argument is given, treat the gap at the given energy as having this index. Omission of the first argument causes the anchor energy to be determined automatically. Explicit specification of this energy is necessary only if the automatic method appears to fail, or if the correct assignment of the band indices is important (e.g., for calculation density of states). When the second argument is omitted, use the default gap index 0. Alternatively, `bandalign filename.csv` may be used to use the energies in the csv file in order to do assign the band indices. The format is that of `dispersion.byband.csv`, i.e., energies of each of the bands in columns. This option may be used to manually 'correct' an incorrect band connection result. If the data is not sorted (as function of $\mathbf{k}$ or $\mathbf{B}$), then try to sort the data automatically before applying the band alignment algorithm. See also Sec. 3.6 (*Alias*: `reconnect`) (For `kdotpy merge`, `kdotpy 2d`, and `kdotpy ll`)

### C.1.4 System options

- `cpus #`: Number of parallel processes to be used. Note that the processes do not share memory, so it should be chosen such that the total memory requirement does not exceed the available memory. Can also be set to value `max`, `auto` or `automatic`, for using all available cores; this is the *default*. For a single-core run, `cpus 1` must be given explicitly. (*Alias*: `cpu`, `ncpu`)

- `threads #`: Number of threads used per process in external libraries like Intel MKL (PARDISO), FEAST, LU decomposition. (Defaults to 1 if omitted; *Alias*: `nthreads`)

- `gpus #`: Number of parallel workers using the GPU when running a CUDA capable solver. (Defaults to `cpus` if omitted; *Alias*: `gpu`, `ngpu`)

- `showetf`: Show estimated completion time in the progress monitor. When this option is omitted, the 'estimated time left' (ETL) is shown. This option is particularly convenient for longer jobs. (*Alias*: `monitoretf`)

- `verbose`: Show more information on screen (written to stdout). Useful for debugging purposes.

### C.1.5 Intermediate results

Arguments that allow saving intermediate result from partially completed diagonalization runs, for example for saving calculation time in debugging. It saves the data container for each data point (called `DiagDataPoint`), into a binary file that may be reloaded later. Do this only at own risk; read the warnings below.

- `tempout`: Create a timestamped subdirectory in the output directory. After each step that updates a `DiagDataPoint` instance, it is 'pickled' (using Python library 'pickle' and saved to disk as temporary binary file. This output can be loaded with the `resume` argument. See also the notes below.

- `keepeivecs`: Keep eigenvectors in memory for all `DiagDataPoints` and also for temporary output files (see `tempout`). Warning: This can drastically increase the RAM usage.

- `resume #1 [#2]`: Path to folder (#1) created by argument `tempout` during a previous kdotpy run. If a matching DiagDataPoint is found in this folder, it is restored into RAM and already processed calculation steps are skipped.
  Optionally, an integer step index (#2) may be specified to overwrite the step from which the

process is resumed. This can be used, e.g. to redo the postprocessing for each DiagDataPoint, if eigenvectors have been saved (see `keepeivecs`).

NOTE: Some command line arguments may be changed between runs (e.g., cpu and/or threads configuration) without affecting the validity of older `DiagDataPoints` for new runs. Apart from matching **k** and **B** values, there is no further automatic validation.

NOTE: The binary file format is not a suitable research data format. Compatibility between different versions of kdotpy is *not* guaranteed.

NOTE: These files should be used for temporary storage and immediate re-use only. This is not a suitable data format for long-time storage. Compatibility between different versions of kdotpy is *not* guaranteed. For permanent storage of eigenvectors, enable the configuration option `diag_save_binary_ddp`. See also the security warnings for the pickle module.

HINT: Usage suggestions are resuming a preemptively cancelled job (e.g., due to walltime limit, out of resources, etc), and testing or debugging restartable from partial solutions in order to save calculation time.

See also system options related to output, Appendix C.7.

## C.2 Density of states, electrostatics, etc.

### C.2.1 Post-processing functions

- `dos`: Plot density of states and integrated density of states. The Fermi energy and chemical potential are also indicated in the plots and printed to stdout, if their calculation has been successful. The range of validity is shown in red: In the shaded regions, additional states (typically at larger momentum **k**) are not taken into account in the present calculation and may cause the actual DOS to be higher than indicated. The validity range typically grows upon increasing the momentum range (argument k). For kdotpy ll, dos will generate equal-DOS contours and put them in the LL plot. This is done either at a number of predefined densities, or at the density given by `cardens`. For this script, also plot the total DOS, and the 'numeric DOS' (roughly the number of filled LL).

- `localdos`: For kdotpy 2d, plot the 'local DOS', the momentum-dependent density of states. For kdotpy ll, plot the equivalent quantity, DOS depending on magnetic field. For kdotpy ll, additionally plot the 'differential DOS', the integrated DOS differentiated in the magnetic field direction.

- `banddos`: For kdotpy 2d and kdotpy bulk, output the DOS by band. One obtains two csv files, for DOS and IDOS, respectively. Each column represents one band. (*Alias*: `dosbyband`)

- `byblock`: For kdotpy 2d and kdotpy ll, in combination with dos. Give density of states where all states are separated by isoparity value ($\tilde{\mathcal{P}}_z = \pm 1$). Note: By nature, this function does not take into account spectral asymmetry of the individual blocks. (*Alias*: `byisopz`)

- `densityz`: For kdotpy ll, plot density as function of $z$ at the Fermi level, for all values of the magnetic field **B**. The output is a multipage pdf file and a csv file with $z$ and **B** values over the rows and columns, respectively. The output is for one carrier density only.

### C.2.2 Self-consistent Hartree

- `selfcon [# [#]]`: Do a selfconsistent calculation of the electrostatic potential ("selfconsistent Hartree"). This method solves the Poisson equation iteratively, taking into account the occupied states in the well. This option also provides plots of the density as function of $z$ and of the potential. Two optional numerical arguments: maximum number of iterations (*default*: 10) and accuracy in meV (*default*: 0.01)

- `selfconweight #`: Use this fractional amount to calculate the new potential in each iteration of the self-consistent Hartree method. This has to be a number between 0 and 1. The *default* value is 0.9. It may be set to a smaller value in case the iteration goes back and forth between two configurations, without really converging. A small number also slows down convergence, so the number of iterations may need to be increased. (*Alias*: `scweight`, `scw`)

### C.2.3 Potentials

The following options define a background potential that affects the calculation of the dispersion and/or the self-consistent Hartree calculation.

- `vtotal #`: Add a potential difference between top and bottom of the whole layer stack. The value is in meV and may be positive as well as negative. (*Alias*: `v_outer`, `vouter`)

- `vwell #`: Add a potential difference between top and bottom of the 'well region'. The value is in meV and may be positive as well as negative. (*Alias*: `v_inner`, `vinner`)

- `vsurf # [# [#]]`: Add a surface/interface potential. The first argument is the value of the potential at the interfaces (barrier-well) in meV. The second parameter determines the distance (in nm) for which the potential decrease to 0 (*default*: 2.0). If the latter argument is q (*alias*: `quadr`, `quadratic`), then the potential has a parabolic shape. Otherwise, the decrease to 0 is linear. (*Alias*: `vif`)

- `potential ## [# ...]`: Read potential from a file. The file must be in CSV format, i.e., with commas between the data values. The columns must have appropriate headings; only `z` and `potential` are read, whereas other columns are ignored. If the `z` coordinates of the file do not align with those of the current calculation, then values are found by linear interpolation or extrapolation. If extrapolation is performed, a warning is given.
  The first argument must be a valid file name. The following arguments may be further filenames, and each filename may be followed by a number, interpreted as multiplier. For example, `potential v1.csv -0.5 v2.csv` will yield the potential given by $V(z) = -0.5V_1(z) + V_2(z)$. Multiple arguments `potential` are also allowed; the results are added. Thus, the sequence of arguments `... potential v1.csv -0.5 ... potential v2.csv ...` is equivalent to the previous example.

- `cardens # [# / #]`: Carrier density in the well in units of $e/\mathrm{nm}^2$. This value sets the chemical potential, i.e., "filling" of the states in the well. The sign is positive for electrons and negative for holes. In combination with `kdotpy ll ... dos`, specify the density at which the equal-density contour should be drawn. (See dos above.) The argument may also be a range (`# # / #`; this affects most (but not all) postprocessing output functions. If omitted, the default range is equivalent to `-0.015 0.015 / 30` (i.e., $-1.5 \times 10^{12}$ to $1.5 \times 10^{12}\, e/\mathrm{cm}^2$ in steps of $10^{11}\, e/\mathrm{cm}^2$). (*Alias*: `carrdens`, `carrierdensity`, `ncarr`, `ncar`, `ncarrier`)

- `ndepletion # [#]`: Density of the depletion layer(s) in the barrier, in units of $e/\mathrm{nm}^2$. The sign is positive for holes and negative for electrons. The whole sample is neutral if the arguments `cardens` and `ndepletion` come with the same value. If one value is specified, the charge is equally divided between top and bottom barrier. If two values are specified, they refer to bottom and top layer, consecutively. (*Alias*: `ndepl`, `ndep`)

- `ldepletion # [#]`: Length (thickness) of the depletion layers in nm. The values may be numbers $> 0$ or `inf` or – for infinite (which means zero charge volume density). The numbers refer to the bottom and top barrier, respectively. If a single value is given, use the same value for both bottom and top barrier. The default (if the argument is omitted) is infinity. (*Alias*: `ldepl`, `ldep`)

- `efield # #`: Electric field at the bottom and top of the sample in **mV/nm**. Alternatively, one may enter a single value for either the top or the bottom electric field:

  - `efield -- #`, `efield top #`, `efield t #`, `efield # top`, etc. (top)
  - `efield # --`, `efield btm #`, `efield b #`, `efield # btm`, etc. (bottom)

  If the variant with two values is used, the carrier density is calculated automatically. In that case, the explicit input of the carrier density (option `cardens`) is not permitted.
  NOTE: `efield 0 #` is not the same as `efield -- #`.
  NOTE: A positive electric field at the top boundary corresponds to a negative gate voltage, and vice versa.

- `potentialbc #`: Apply custom boundary conditions for solving Poisson's equation in selfconsistent calculations. (*Alias*: `potbc`)
  The argument must be a string, which can be one of three different formats:

  1. Input like a Python dict instance without any spaces:
     `"{'v1':5,'z1':-10.,'v2':7,'z2':10.}"`
     All boundary names must be given explicitly, the order is irrelevant.

  2. Input single quantities as string separated with semicolon without any spaces:
     `"v1=5;z1=-10.;v2=7;z2=10."`
     All boundary names must be given explicitly, the order is irrelevant.

  3. Input quantity pairs as string separated with semicolon without any spaces:
     Either explicit: `'v1[-10.]=5;v2[10.]=7'`
     Or implicit: `'v[-10.]=5;v[10.]=7'`
     When using the explicit format, the order is irrelevant. When using the implicit format there is an internal counter, which applies an index to the quantity name, thus, the order does matter.

  Here, all given examples will result in the same boundary condition dictionary:
  `{'v1':5,'z1':-10.,'v2':7,'z2':10.}`
  The $z$ values must be given as coordinate in nm, or as one of the following labels:

  - `bottom`: Bottom end of the layer stack
  - `bottom_if`: Bottom interface of the "well" layer
  - `mid`: Center of the "well" layer
  - `top_if`: Top interface of the "well" layer
  - `top`: Top end of the layer stack

  If less than 4 key-value pairs are given, only the corresponding values of the automatically determined boundary conditions are overwritten. The ones that do not appear in the automatic determined boundary conditions, are ignored. This also means, you can decide to only overwrite the $z$-coordinates but keep the automatic determined values for v1, v2, etc. If two full boundary conditions are given (4 or 5 key-value pairs), automatic boundary conditions are always fully overwritten.
  NOTE: A special case for the implicit type 3 input is v12. The input `v[-10.,10.]=0;v[0.]=7` for example, yields `{'v12':0,z1':-10.,'z2':10.,'v3':7,'z3':0.}`. Another special case for the same input type is the combination of dv1 and v1 (or dv2 and v2). Here you can use `dv[-10.]=0;v[-10.]=7` (or `v[-10.]=7;dv[-10.]=0`); note the same $z$ coordinates.

### C.2.4 Broadening options

The following options affect the broadening applied to the density of states. A combination of options is possible, in particular also using multiple instances of `broadening` and `berrybroadening`. See Sec. 3.7.7 and Appendix B.6 for details and some physical background.

- `broadening [#1 #2 ...]`: Broadening parameter for the density of the eigenstates (for `kdotpy ll`: Landau levels). The broadening is determined by a numerical width parameter $w_1$ which may be supplemented by additional parameters, the broadening shape, the scaling function, and a parameter for the Berry broadening (the latter for `kdotpy ll` only). The broadening types are:

  - `thermal`: Fermi distribution, width parameter is temperature; if the width parameter is omitted, use the temperature set by `temp`.
  - `fermi`: Fermi distribution, width parameter is energy (*alias*: `logistic`, `sech`)
  - `gauss`: Gaussian distribution, width parameter is energy (*alias*: `gaussian`, `normal`)
  - `lorentz`: Lorentzian distribution, width parameter is energy (*alias*: `lorentzian`)

  – `step`: Dirac-delta or Heaviside-step function, if width parameter is given, it is ignored (*alias*: `delta`)

If omitted, use the default `auto` which selects `thermal` for dispersion mode and `gauss` for LL mode. The scaling function determines how the width $w$ scales as function of $x$ (momentum $k$ in $\mathbf{nm^{-1}}$ or field $B$ in $\mathbf{T}$) ($w_1$ is the input width parameter):

  – `auto`: Use `const` for dispersion mode and `sqrt` for LL mode (*alias*: `automatic`)
  – `const`: Use constant width, $w = w_1$
  – `lin`: The width scales as $w = w_1 x$ (*alias*: `linear`)
  – `sqrt`: The width scales as $w = w_1 \sqrt{x}$
  – `cbrt`: The width scales as $w = w_1 \sqrt[3]{x}$
  – `^n`: Where n is a number (integer, float, or fraction like 1/2). The width scales as $w = w_1 x^n$.

The final optional value (`kdotpy ll` only) is numeric (floating point number or a percentage like `10%`) that defines a different broadening width for the Berry curvature/Hall conductivity. Floating point input is interpreted as the broadening width itself, a percentage defines this broadening width as percentage of the density broadening width. The Berry/Hall broadening inherits the shape and scaling function from the density broadening.

Multiple `broadening` arguments may be combined; these will then be iteratively applied to the (integrated) DOS, in the given order. See Sec. 3.7.7 and Appendix B.6 for details.
NOTE: Due to limitations of the numerical integration (convolution operation), combining multiple broadening functions may lead to larger numerical errors than a single broadening function. The convolution operation is commutative only up to numerical errors, so changing the order may lead to slight differences in the result.

*Examples*:

  – `broadening 2 thermal const`: A thermal broadening with width of $T = 2$ K, constant in momentum $k$. This is the default shape and scaling for dispersion mode.
  – `broadening 2 gauss sqrt`: A Gaussian broadening of width **2 meV** at **1 T** scaling proportionally to $\sqrt{B}$. This is the default shape and scaling for LL mode.
  – `broadening 2`: For dispersion mode, equivalent to `broadening 2 thermal const`. For LL mode, equivalent to `broadening 2 gauss sqrt`.
  – `broadening 2 10%`: In LL mode, set the Berry/Hall broadening width to 10% of that of the density broadening. That is, for the Berry/Hall broadening, the parameters are effectively `0.2 gauss sqrt`.

• `berrybroadening [#] [#] [#]`: Broadening parameter for the Berry curvature/Hall conductivity. The syntax is the same as the ordinary `broadening` parameter. Also multiple ones can be combined. Note that it is not permitted to combine `berrybroadening` with a `broadening` with argument with an extra numerical argument (for example `broadening 0.5 gauss 10%`). For `kdotpy ll` only. (*Alias*: `hallbroadening`, `chernbroadening`)

• `dostemp #`: Temperature used for thermal broadening of the DOS. This argument is equivalent to the setting `broadening # thermal const` (but only one of these may be used at a time). This temperature may be different than the temperature set by `temp` on the command line (which controls the temperature in the exchange coupling, for example). If neither `dostemp` nor `broadening` is given, no broadening is applied. If both `dostemp` and `broadening` are given, the setting for `broadening` takes priority. This option is especially useful for calculating the DOS with `kdotpy merge` and `kdotpy compare`: In that case, `temp` has no effect, because the value is read from the data files, whereas `dostemp` *can* be used to set the thermal broadening.

### C.2.5 Additional options

The following options affect calculations of the density of states, self-consistent Hartree, etc.

- `ecnp #`: Set charge-neutral energy (zero density) to this energy. The value determines the point where the density is zero. This affects integrated density of states in dispersion mode only. In order to manipulate band indices by determining the zero gap, use the `bandalign` argument. (*Alias*: `cnp, efermi, ef0`)

- `densoffset #`: Set a density offset. Basically, this number is added to the integrated DOS / carrier density in the selfconsistent calculation. The value is in units of charge density and can be interpreted as free carriers inside the quantum well. (*Alias*: `noffset, ncnp`)

- `cardensbg #`: Set a background density. Calculates a rectangular carrier distribution for this number, which is then added to the carrier distribution used in solving Poisson's equation. The value is in units of charge density and can be interpreted as immobile background charge.

- `idosoffset #`: Set an offset to the density of states, in appropriate DOS units. This option is identical to `densoffset` up to a factor of $4\pi^2$. (*Alias*: `dosoffset`)

### C.2.6 Output options

- `dosrange [#1] #2`: Plot range of integrated density plots. If just one value $n_{\max} = $ #2 is given, use $[0, n_{\max}]$ for densities and $[-n_{\max}, n_{\max}]$ for integrated densities. Omission means that the plot range is determined automatically. If a density unit is given, e.g., `densityenm`, the values are interpreted in the quantity being plotted. Here, large numbers ($> 1000$) are interpreted as having units of $\mathrm{cm}^{-1}$, $\mathrm{cm}^{-2}$, or $\mathrm{cm}^{-3}$ and small numbers as $\mathrm{nm}^{-1}$, $\mathrm{nm}^{-2}$, or $\mathrm{cm}^{-3}$ (with the appropriate dimension). (*Alias*: `densityrange, dosmax, densitymax`)

## C.3 Extra functions

NOTE: Most of these functions are available only for a limited number of kdotpy scripts.

### C.3.1 Pre-diagonalization

- `plotfz`: Plot several parameters as a function of $z$:

  - $E_c$ and $E_v$ (conduction and valence band edges)
  - $F$, $\gamma_{1,2,3}$, $\kappa$ (Luttinger and miscellaneous band parameters)
  - $yN_0\alpha$ and $yN_0\beta$ (Mn exchange energies)

  The option `legend` will include a legend in these plots (recommended). (*Alias*: `plotqz`)

### C.3.2 Using wave functions or eigenvectors

Functions that derive extra data based on the eigenvectors.

- `overlaps`: Calculate overlaps between the eigenstates with those at zero momentum. By default, the overlaps are calculated with $|\mathbf{E1\pm}\rangle$, $|\mathbf{H1\pm}\rangle$, $|\mathbf{H2\pm}\rangle$, and $|\mathbf{L1\pm}\rangle$. A nice visualization can be obtained with `obs subbandrgb`, which assigns colours depending on the overlaps with E1, H1, and H2. A visualization with different bands can be obtained by using `obs subbandh1e1e2`, for example, where the observable id ends with $\geq 3$ pairs of subband identifiers. Each subband identifier is a band character (e, l, or h followed by a number) denoting a pair of subbands, a single subband (the previous followed by + or -), or a band index (a signed integer preceded by b or parenthesized, e.g., b+2, (-25)). See also observables, Appendix B.5 (`kdotpy 2d` and `kdotpy ll`)

- `transitions [#1 #2] [#3]`: Calculate and plot transitions between levels. There can be up to 3 optional numerical arguments: The first pair is the energy range where transitions are calculated. If omitted, calculate transitions between all calculated states (which may be controlled with `neig` and `targetenergy`). The last argument is the square-amplitude threshold above which the transitions are taken into account. If omitted, the program uses the *default* value 0.05.

- `berry`: Calculate and plot the Berry curvature for the states close to the neutral gap. Also plot the integrated Berry curvature as function of energy. If combined with the option `dos`, then also plot the integrated Berry curvature as function of density. For LL mode (`kdotpy ll`), the Berry curvature is implicitly integrated, and the resulting output values are the Chern numbers of the eigenstates instead.

- `hall`: Shortcut that activates all options for calculation of Hall conductivity with `kdotpy ll`. It is equivalent to the combination `berry dos localdos broadening 0.5 10%`. The default value of the broadening can be overridden with the explicit option `broadening # [#]` combined with `hall`. See also options for density of states and broadening, Appendix C.2.

- `plotwf [# ...]`: Plot wave functions. The extra arguments are the plot style for the wave function plot and the locations (momenta) for which the plots are made. See also Sec. 3.9.2.

### C.3.3 Post-diagonalization (postprocessing)

The postprocessing functions rely predominantly on the eigenvalues (dispersion).

- `dos`: See Appendix C.2.1.

- `localdos`: See Appendix C.2.1.

- `banddos`: See Appendix C.2.1.

- `minmax`: Output the minimum, maximum, and zero-momentum energy of each subband. (`kdotpy 2d` and `kdotpy bulk`)

- `extrema`: Output the local extrema of each subband. The output contains the type (min or max), the momentum, the energy, and an estimate for the effective inertial mass along the momentum direction. (`kdotpy 2d` and `kdotpy bulk`; *Alias*: `localminmax`, `mimaxlocal`)

- `symmetrytest`: Analyze the symmetries of the eigenvalues and observables under various transformations in momentum space. This results in a list of compatible representations of the maximal point group $O_\mathrm{h}$, from which the program tries to determine the actual symmetry group (point group at the $\Gamma$ point).
  NOTE: For a reliable result, the momentum grid must be compatible with the symmetries; a cartesian grid should be used for cubic symmetry, a polar or cylindrical grid otherwise.
  For `kdotpy 2d`, `kdotpy bulk`: full analysis. For `kdotpy 1d`, `kdotpy merge`: partial analysis (full analysis to be implemented).

- `symmetrize`: Extend the data in the momentum space by symmetrization. For example, a 1D range for positive $k$ can be extended to negative $k$, or a 2D range defined in the first quadrant can be extended to four quadrants. The extension is done by taking the known eigenvalues and observables and transforming them appropriately.
  NOTE: The algorithm relies on some pre-defined transformation properties of the observables, and should be used with care. A cross-check with a symmetric range and `symmetrytest` is advised.
  (`kdotpy 1d`, `kdotpy 2d`, and `kdotpy bulk`, `kdotpy merge`)

- `bhz [# ...]`: Do a Löwdin expansion around zero momentum in order to derive a simplified Hamiltonian in the subband basis. This is the generalization of the BHZ model. For information on the argument pattern and further information, see Sec. 3.9.4.

- `kbhz #`: Set the reference momentum for the BHZ (Löwdin) expansion. The argument refers to a momentum value on the $k_x$ axis. This experimental option can be used for expansions around nonzero momentum. Please consider the results with care, as they are not always meaningful. See also Sec. 3.9.4. (*Alias*: `bhzk`, `bhzat`)

## C.4 Definition of the layer stack

- `lwell #`: Thickness of the active layer (e.g., quantum well) in nm. (*Alias*: `lqw`, `qw`)

- `lbarr #1 [#2]`: Thickness of the barrier layers in nm. If one thickness is given, assume this value for both barrier layers (bottom and top). If two thicknesses are given, the first and second argument refer to the bottom and top layer, respectively. The input `lbarr #1 lbarr #2` is equivalent to `lbarr #1 #2`. (*Alias*: `lbar`, `lbarrier`, `bar`, `barr`, `barrier`)

- `llayer #1 [#2 ...]`: Thicknesses of the layers in nm. The number of layers may be arbitrary, but the number of thicknesses must always be equal to the number of materials. This argument may not be combined with `lwell` and `lbarr`. (*Alias*: `llayers`, `layer`, `layers`, `thickness`, `thicknesses`, `thicknesses`, `thick`)

- `mwell #1 [#2]`: Material for the well layer. See below for instructions on how to input a material. (*Alias*: `mqw`)

- `mbarr #1 [#2]`: Material for the barrier layers. See below for instructions on how to input a material. (*Alias*: `mbarrier`, `mbar`)

- `mlayer #1 [#2 ...]`: Material specification for an arbitrary number of layers. See below for instructions on how to input a material. The number of specified materials must match the number of thicknesses (`llayer`; `lwell` and `lbarr`). (*Alias*: `mater`, `material`)

- `msubst #1 [#2]`: Material for the substrate. This only sets the lattice constant which is used to calculate strain. If this argument is omitted, the strain is taken from the `strain` or the `alattice` argument. (*Alias*: `msub`, `substrate`, `msubstrate`)

- `ltypes #1`: Define the type (purpose of each layer). The argument must be a string of the following letters whose length must match the number of layers in the stack:

    - `b`: barrier
    - `c`: cap
    - `d`: doping
    - `q` OR `w`: well
    - `s`: spacer

  (*Alias*: `ltype`, `lstack`)
  NOTE: Some functions will work properly only if there is exactly one 'well' layer.

- `ldens #1 [#2 ...]`: For each layer, the 'background density' of charge, for example doping. There need to be as many values as there are layers. The values are expressed in $e/\text{nm}^2$. (*Alias*: `layerdens`, `layerdensity`)

### C.4.1 Material input

Each material instance is a material id or compound (e.g., `HgMnTe`, `HgCdTe`), optionally followed by extra numerical arguments that define the composition. The composition can either be specified as par of the compound (chemical formula) or as these extra arguments. Fractions and percentages are both accepted. Thus, all of the following are equivalent: `HgMnTe 2%`, `HgMnTe 0.02`, `HgMn0.02Te`, `Hg0.98Mn0.02Te`, `HgMn2%Te`, `HgMn_{0.02}Te`, etc.
The chemical formulas (or material ids) are case sensitive, which eliminates ambiguity.

### C.4.2 Material parameters

- `matparam #`: Modify the material parameters. The argument can either be a materials file or a sequence of `parameter=value` pairs. For the latter, multiple parameters must be separated by semicolons (`;`) and must be preceded by the material identifier, like so:
  `matparam 'HgTe:gamma1=4.1;gamma2=0.7;CdTe:gamma1=1.6'`

Spaces are ignored and the colon (`:`) after the material may be replaced by period (`.`) or underscore (`_`). The argument must be quoted in the shell if it contains spaces. The material need not be repeated for subsequent parameters, so that in the example, gamma2 refers to the material HgTe. The values may be Python expressions, but restrictions apply (see Appendix B.2 for information for material parameter files). Note that all expressions must resolve to numerical values in order for kdotpy to run successfully. Multiple `matparam` arguments will be processed in order of appearance on the command line. (*Alias*: `materialparam`)

## C.5   Other geometrical parameters

- `zres #`: Resolution in the $z$ direction in nm. (*Alias*: `lres`)

- `width #`: Width of the sample (in the $y$ direction. If a single number is given, this determines the width in nm. If the argument is given as #1*#2 or #1 * #2, where #1 is an integer, then the sample has #1 sites in the $y$ direction spaced by a distance of #2 nm each. If the argument is given as #1/#2 or #1 / #2, then the total width is #1 and the resolution #2. (*Alias*: `W`)

- `yres #`: Resolution in the $y$ direction in nm. (*Alias*: `wres`)

- `linterface #`: Smoothing width of the interface in nm. (*Alias*: `interface`)

- `periodicy`: Enables periodic boundary conditions in the $y$ direction. (Only applies to 1D geometry.)

- `stripangle #`: Angle in degrees between the translationally invariant direction of the strip (or ribbon) and the (100) lattice vector (`kdotpy 1d` only). *Default*: 0 (*Alias*: `ribbonangle`)

- `stripdir #`: Direction of the translationally invariant direction of the strip/ribbon in lattice coordinates. The argument may be a lattice vector, e.g., 130 for (1,3,0) or any of x, y, xy, and -xy (equivalent to 0, 90, 45, and -45 degrees). Only one argument `stripangle` or `stripdir` should be given. (*Alias*: `ribbondir`)

- `radians`: Use radians for angular coordinate values. If omitted, use degrees (*default*).

- `orientation # [#] [#]`: Orientation of the lattice, see also Sec. 3.3.6 and Appendix B.3. (*Alias*: `orient`)
  Possible patterns (#ang and #dir denote angles and direction triplets, respectively):

  - #ang: Rotation around $z$ (like `stripangle`)
  - #ang #ang: Tilt $z$ axis, then rotate around $z$ axis
  - #ang #ang #ang: Euler rotation $z, x, z$. Rotate around $c$ axis, tilt $z$ axis, rotate around $z$ axis.
  - #dir: Longitudinal direction $x$ (like `stripdir`)
  - – #dir: Growth direction $z$
  - #dir #dir: Longitudinal and growth direction $x, z$
  - – #dir #dir: Transversal and growth direction $y, z$
  - #dir #dir #dir: Longitudinal, transversal, and growth direction $x, y, z$.
  - #dir #ang *or* #ang #dir: Growth direction $z$ and rotation around $z$.

  Format for the inputs: For angles #ang, enter an explicit floating point number containing a decimal sign (period .). Integer values (for example 45 degrees) can be entered as 45., 45.0, 45d, or 45°. Direction triplets (#dir) are a triplet of digits without separators and possibly with minus signs (e.g., 100, 111, 11-2, -110) or numbers separated by commas without spaces (e.g., 1,1,0 or 10,-10,3).

  If `orientation` is combined with `stripangle` or `stripdir`, the latter are ignored.

  NOTE: If multiple #dir inputs are given, they must be orthogonal directions. If the inner product between any pair of them is nonzero, an error is raised.

NOTE: With the option `orientation`, the program uses an alternative construction method for the Hamiltonian, which may cause the time consumption by this step to increase by a factor of approximately 4. There is no exception for trivial orientations, like `orientation - 001`, which still invokes the alternative construction method.

## C.6 Other physical parameters

### C.6.1 External parameters

- `b #`: External magnetic field in T. Ranges may be input using the same syntax as the momenta **k**. For information on vectors and ranges, see Sec. 3.2.3.

- `temp #`: Temperature in K. The temperature affects the gap size (band edges) and the Mn exchange coupling. Optionally, it sets the thermal broadening of the density of states if the argument 'broadening thermal' (without value) is given, see Appendix C.2.

  NOTE: Thermal broadening is *not implied* by `temp`. In order to apply thermal broadening, specifying `broadening thermal` or `dostemp` is required, see Appendix C.2 for more information.

### C.6.2 Specification of strain

- `ignorestrain`: Ignore the strain terms in the Hamiltonian. (*Alias*: `nostrain`)

- `strain # [# #]`: Set strain value. The value may be set as a number or percentage (e.g., -0.002 or -0.2%). The value is interpreted as the 'relative strain' $\epsilon = (a_{\text{strained}}/a_{\text{unstrained}}) - 1$, where $a_{\text{unstrained}}$ refers to the well material. (In layer stacks with more than three layers, the well may not be identified, and then this option cannot be used. Setting `strain none` is equivalent to `ignorestrain`. It is also possible to specify more than one argument; then the values are interpreted as $\epsilon_{xx}, \epsilon_{yy}, \epsilon_{zz}$. It is possible to enter – for one or two values; then the strain values corresponding to these components are determined from the other one(s). If `strain` is used together with `ignorestrain`, the latter has priority, i.e., no strain is applied.

- `alattice #1`: Set the lattice constant of the strained materials. (*Alias*: `alatt`, `latticeconst`)

  NOTE: Exactly one of the three options `msubst`, `alattice`, and `strain` must be used at once.

## C.7 Options affecting plots

### C.7.1 Observables

- `obs #`: Use observable # (see Appendix B.5 for the colouring of the plot. It must be one of the available observables in the data files. There is a special case `orbitalrgb`, which colours the states with RGB colours determined by the `gamma6,gamma8l,gamma8h` expectation values. In `kdotpy compare`, using `obs` will leave only the markers to distinguish the data sets; without `obs`, distinct markers and colours are used.

- `obsrange [#] #`: Minimum and maximum value of the observable that determines the colour scale. If one value is given, it is the maximum and the minimum is either 0.0 or the minus the maximum, which is determined by whether the standard scale is symmetric or not. If this option is omitted, use the predefined setting for the colour scale (recommended). (*Alias*: `orange`, `colorrange`, `colourrange`)

- `dimful`: Use dimensionful observables. Some observables, for example `z` and `y`, are dimensionless by default, and this option changes them to observables with a dimension (for example length in nm). This option affects output data (xml and csv) and graphics. (*Alias*: `dimfull`)

- `orbitalobs`: Calculate the observables `orbital[j]`, that is the squared overlaps of the eigenstates within orbital number j, where $j$ runs from 1 to $n_{\text{orb}}$ (the number of orbitals). (For `kdotpy` 2d only.) (*Alias*: `orbitaloverlaps`, `orbobs`, `orboverlaps`)

- `llobs`: Calculate the observables `ll[j]`, that is the squared overlaps of the eigenstates within Landau level `j`, where `j` runs from `-2` to `llmax` (the largest LL index). This option is available for `kdotpy ll` in full LL mode only. (*Alias*: `lloverlaps`)

- `custominterfacelengthnm #`: When given, calculate additional 'interface (character)' observables, but within a custom length interval given by `#` (integer value in nm).

### C.7.2 Data and plot range

- `erange #1 #2`: Energy range, minimum and maximum value in meV. The energy range determines the vertical plot range in plots. It is also used as range for density of states calculations. For Landau level calculations, states outside the energy range are not saved in the B dependence data file.

- `xrange [#1] #2`: Horizontal range to display in the plot. If just one value is given, the range runs from 0 to the specified value. (*Alias*: `krange`, `brange`)

- `dosrange [#1] #2`: Vertical range for density of states plots.

- `plotvar #`: Plot against the given variable, instead of the default variable (coordinate component).

- `xoffset`: Offsets the data points slightly in horizontal direction, so that (almost) degenerate points can be resolved. The direction (left or right) is determined by the sign of the requested observable.

### C.7.3 Plot style

- `plotstyle #`: Choose the plot style. (*Alias*: `plotmode`). The second argument is one of the following plot styles:

    - `normal`: Unconnected data points
    - `curves`: Connect the data points horizontally, i.e., by band index. This option replaces the old `join` option. (*Alias*: `join`)
    - `horizontal`: Group the data points 'horizontally', but plot them as separate data points.
    - `auto`: Use `curves` if possible; otherwise use `normal`. (*Alias*: `automatic`)
    - `spin`: Use different markers based on the `jz` observable value. (NOTE: $J_z$ is the total angular momentum, not the actual 'proper' spin)
    - `spinxy`, `spinxz`, `spinyz`: Like the 'normal' plot, but add arrows to indicate the spin components $(S_x, S_y)$, $S_x, S_z)$ or $(S_y, S_z)$, respectively.
    - `spinxy1`, `spinxz1`, `spinyz1`: Like `spinxy`, `spinxz`, and `spinyz`, but rather plot directions (unit vectors) that indicate the spin direction in the given plane.
    - `berryxy`, `berryxz`, `berryyz`, `berryxy1`, `berryxz1`, `berryyz1` Arrows indicating Berry curvature, analogous to the above spin arrow modes
    - `isopz`: Use different markers based on the `isopz` observable value.

    Upon omission, the *default* value is `auto`.

- `spin`: Indicate spin expectation value (up, down) with different plot markers/symbols.

### C.7.4 Other plot elements, style, etc.

- `labels`: Display band characters or band labels at $\mathbf{k} = \mathbf{0}$ ($\mathbf{B} = \mathbf{0}$ if the horizontal axis is magnetic field) and Landau level indices, if applicable. (*Alias*: `plotlabels`, `char`)

- `title #`: Assign plot title. One may use `{var}` to substitute the variable named `var`. In order to find out which are the available variable names (keys), use `title ?` to get a list. The format syntax follows Python's string format function, including the format specification "Mini-Language" [12]. Here, only named variables can be specified. Positional ones, like `{0}` or `{1}` are not permitted. Some special variable names are:

  - `llayer(#)`: For layer properties, append parenthesized integer index (n), e.g., `llayer(1)`, for the property of the *n*'th layer.
  - `b_x, b_y, b_z`: Cartesian vector components
  - `b_phi, b_theta`: Angular coordinates of a vector in degrees
  - `b_len, b_abs`: Vector length (`len` and `abs` are equivalent)

  (*Alias*: `plottitle`)

- `titlepos #`: Position of the plot title. This may be any of the following:

  - `l, r, t, b`;
  - `left, right, top, bottom, center`;
  - `top-center, bottom-center`;
  - `tl, tr, bl, br`;
  - `top-left, top-right, bottom-left, bottom-right`
  - `n, s, ne, nw, se, sw`;
  - `north, south, north-east, north-west, south-east, south-west`.

  NOTE: `left` and `right` are synonyms to `top left` and `top right`
  NOTE: Double words can be with hyphen (`top-center`), underscore (`top_center`), space (`"top center"`; quotes are usually needed) or be joined (`topcenter`).
  NOTE: `e, east, w, west` are not legal values
  (*Alias*: `plottitlepos, titleposition, plottitleposition`)

- `legend`: Include a legend in the plot. For coloured observables, this is a colour bar plus the indication of the observable. (*Alias*: `filelegend`)

- `legend label # [label # ...]`: If the argument `legend` is directly followed by `label` followed by the label text, use this text in the legend instead of the file names. The label text must be quoted on the command line if it contains spaces. (For `kdotpy compare` only.)

### C.7.5 System options

- `out #`: Determines the names of the output files. For example, if the argument is 1, the program produces `output1.xml`, `plot1.pdf`, etc. This option also uses variable substitution using Python's string format function; see command `plottitle` above. (*Alias*: `outfile, outid, outputid, outputname`)

- `outdir #`: Name of the output directory. If the directory does not exist, try to create it. If omitted, try to write to the subdirectory `data` if it exists, otherwise in the current directory. (*Alias*: `dir, outputdir`)

See also Appendix C.1 for system options related to calculations.

---

[12]See the Python documentation at https://docs.python.org/3/library/string.html#formatspec.

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
