# Peer review of "kdotpy: k·p theory on a lattice for simulating semiconductor band structures"

_SciPost Physics Codebases, doi:SciPost Phys. Codebases 47-r1.0 (2025) , SciPost Phys. Codebases 47 (2025)_

## Round 1 · Referee Report · Anonymous (Referee 1) · 2024-9-8

Report

This is an excellent manuscript ready for publication in this journal.
It contains interesting (and important) physical background and a very detailed description of the developed software package.
As of now I had no time to play with the software but it will be my priority in a month or so.

Recommendation

Publish (surpasses expectations and criteria for this Journal; among top 10%)

---

## Round 1 · Referee Report · Christoph Groth (Referee 2) · 2024-10-18

Strengths

  1. The article is clearly written and very complete.
  2. It includes a thorough discussion of the relevant physics,
  3. a discussion of the code design, and a detailed walk-through of the implementation,
  4. detailed usage examples,
  5. as well as additional information like installation instructions, or a complete reference of configuration parameters.
  6. The source code is organized clearly and documented well, its version control history is very well kept.
  7. The design of the program interface ensures that code runs are self-documenting. Together with the open nature of this code this is very useful for long-term reproducibility.
  8. It is evident that this is the fruit of years of in-house development and usage by one of the leading experimental groups in the field.

Weaknesses

  1. While the command line interface (CLI) is useful, it would have been preferable if the code was organized as a CLI layer around a real software library. Such a design would have greatly facilitated non-interactive use, increased the potential for reuse, and last but not least helped to better structure the code itself.
  2. The custom CLI is neither very readable, nor robust. When using the CLI, errors are not easy to debug. (See below.)

Report

Sharing this code in a well-documented way is a great service to the community. I think that it will be of great use to other research groups. There is no need to comment further on the very complete and well-written article as well as the usefulness of the code. I enthusiastically recommend publication.

In the following I list a number of observations. I do not think that these must be addressed before publication, but hope that they will be useful to the authors.

  1. Band alignment The authors may find the following relevant: https://gitlab.kwant-project.org/kwant/kwantspectrum In particular, if it was possible to obtain not only E(k) but also its derivative(s), this could be used for better band alignment and reconstruction.

  2. Command line interface There is a lot of merit in providing a CLI as a main way of using this code. However, I find that the actual realization could be improved. Each kdotpy subcommand takes a large number of options which have the form "option_name [param_1 [param_2 [...]]]". The parameters can be either numerical or textual. The options and their parameters follow one after another without any structure. If the user makes a mistake and provides a wrong number or type of parameters, there is an error message, but the user is left alone to guess where. For example if the last parameter is omitted in the option "llayer 10 7 10" the error message is: "ERROR: The number of specified materials and of specified thicknesses must match." I think that better CLI could be provided. For example the user could be obliged to separate options on the command line by comma characters (which have no special meaning in the shell). In this way, the long command lines would be somewhat structured visually, parsing could be made more robust, and error messages improved.

  3. The config options have long names. Perhaps some form of namespaces could be used to group related options, e.g. in the config file:

[fig.colorbar] margin=7.5 method=insert

Ideally, this scheme could be also used to compactify configuration on the command line. e.g. "config fig.colorbar margin=7.5 method=insert," (observe the use of a comma to signal the end of the config command)

  1. Custom test runner It is possible to use "python3 -m kdotpy ..." even without the kdotpy executable in $PATH (but with the kdotpy Python module available). However, "python3 -m kdotpy test" produces the following error:

------ kdotpy test suite ------ Starting test 1d... /bin/sh: 1: kdotpy: not found Test 1d: Failed (...)

This showcases a lack of modularity: kdotpy is usable as an executable python module, but needlessly assumes that it may call itself as the kdotpy shell command.

Requested changes

none

Recommendation

Publish (easily meets expectations and criteria for this Journal; among top 50%)

---

## Editorial Decision

published